# Multi-head Transformers Provably Learn Symbolic Multi-step Reasoning via Gradient Descent

**Tong Yang***
CMU

**Huang Yu**[†]
UPenn

**Yingbin Liang**[‡]
OSU

**Yuejie Chi**[§]
Yale

## Abstract

Transformers have demonstrated remarkable capabilities in multi-step reasoning tasks. However, understandings of the underlying mechanisms by which they acquire these abilities through training remain limited, particularly from a theoretical standpoint. This work investigates how transformers learn to solve symbolic multi-step reasoning problems through chain-of-thought processes, focusing on path-finding in trees. We analyze two intertwined tasks: a backward reasoning task, where the model outputs a path from a goal node to the root, and a more complex forward reasoning task, where the model implements two-stage reasoning by first identifying the goal-to-root path and then reversing it to produce the root-to-goal path. Our theoretical analysis, grounded in the dynamics of gradient descent, shows that trained one-layer transformers can provably solve both tasks with generalization guarantees to unseen trees. In particular, our multi-phase training dynamics for forward reasoning elucidate how different attention heads learn to specialize and coordinate autonomously to solve the two subtasks in a single autoregressive path. These results provide a mechanistic explanation of how trained transformers can implement sequential algorithmic procedures. Moreover, they offer insights into the emergence of reasoning abilities, suggesting that when tasks are structured to take intermediate chain-of-thought steps, even shallow multi-head transformers can effectively solve problems that would otherwise require deeper architectures.

## 1 Introduction

Transformers [30]—the building blocks of large language models (LLMs)—have demonstrated impressive capabilities in tasks that require *multi-step reasoning* [32], where LLMs start to excel in solving hard problems via taking simpler step-by-step actions, and executing different subtasks within a single response. Many of these capabilities are further fueled by the remarkable phenomenon called *emergence of reasoning ability*, where simply extending the length of intermediate reasoning steps can dramatically boost accuracy [10].

Spurred by their remarkable success, understanding how transformers enable Chain-of-Thought (CoT) and multi-step reasoning has attracted considerable research attention recently. Most of the theoretical studies on the multi-step reasoning mechanism focus on the expressive power [9, 23, 25, 5], statistical properties [12, 28, 22] or learnability [1, 11, 34, 29, 19, 2] of the CoT mechanism. Notwithstanding, the training theory of transformers, which examines the optimization and generalization properties of transformers trained via gradient-based methods, is another important direction for understanding

---

*Department of Electrical and Computer Engineering, Carnegie Mellon University; email: tongyang@andrew.cmu.edu

[†]Department of Statistics and Data Science, Wharton School, University of Pennsylvania; email: yuh42@wharton.upenn.edu

[‡]Department of Electrical and Computer Engineering, The Ohio State University; email: liang.889@osu.edu

[§]Department of Statistics and Data Science, Yale University; email: yuejie.chi@yale.edu

39th Conference on Neural Information Processing Systems (NeurIPS 2025).

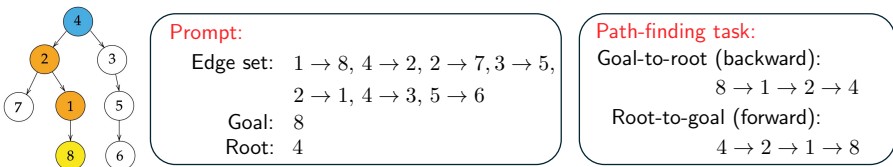

Figure 1: An illustration of the path-finding reasoning task in a tree. Solving the forward task (finding the root-to-goal path) requires solving the backward task (finding the goal-to-root path) first.

the CoT mechanism. This line of work is more closely aligned with practical implementations, as it investigates how training enables transformers to develop multi-step reasoning capabilities through CoT. A small but growing number of studies have recently explored this direction, focusing on tasks such as in-context learning, algorithmic emulation, and parity checking (e.g., [21, 33, 19, 13, 14]).

However, existing theoretical studies do not fully capture the complexity present in real-world applications that transformers are expected to handle through CoT reasoning: (i) *Graph-based structural complexity:* Reasoning over structured relational dependencies introduces a richer level of complexity. It requires transformers to perform multi-hop traversal and extract rule-based generalizations, making it more representative of realistic symbolic reasoning scenarios. (ii) *Multi-stage reasoning and autonomous stage transition:* In real-world CoT reasoning, achieving a complex reasoning objective often involves performing multiple consecutive reasoning tasks. Crucially, the model must learn to autonomously determine when to switch between stages, reflecting a hierarchical and self-regulated reasoning process.

## 1.1 Symbolic multi-step reasoning: path finding on trees

In response to the above gap, this paper resorts to a canonical symbolic multi-step reasoning problem of *path-finding on trees*, motivated by the empirical investigation in [4]; see Figure 1 for an illustration. The input prompt is the list of edges in the tree, the root node, and the goal node (which is a leaf node). We are interested in two fundamental reasoning tasks: *(i) backward reasoning*, which involves finding the path from the goal node to the root node; and *(ii) forward reasoning*, which aims to find the path from the root node to the goal node. While the backward reasoning task can be solved by sequentially chaining edges connected to the child node recursively, tackling the forward reasoning task requires solving the backward reasoning task (i.e., finding the goal-to-root path) first and then reversing the path, as there can be multiple nodes connected to a parent node. Studying these two intertwined tasks has the potential to elucidate deeper understandings of CoT on the following front.

- *How CoT implements chaining over graphs.* Both tasks require *step-by-step reasoning*, as one needs to trace the edges connecting the goal node and the root node in a recursive manner.
- *How CoT implements subtask control.* The harder, forward reasoning task requires solving two *subtasks in a single CoT path*, where the transformer needs to automatically identify the *turning point* between the subtasks, when it traverses the root node.

The empirical study in [4] revealed that deep transformers, when tasked with forward reasoning (i.e., outputting the root-to-goal path), internally implement a backward chaining algorithm across their layers that effectively from the goal node, "climbs" the tree layer by layer towards the root node. The discrepancy between the internal reasoning direction (i.e., backward) and the required output format (i.e., forward) suggests that architectural depth is leveraged by these models to manage intermediate representations before producing the final answer. While inspiring, it is intriguing to ask if one can *provably train* shallow, one-layer, transformers to solving the path-finding task in trees via CoT, and if extending the reasoning length—rather than using deeper architectures—can be leveraged to solve the harder forward reasoning task. Answering these questions will lead to new insights into the multi-head attention mechanism that are unavailable in today's literature.

## 1.2 Our contributions

This paper investigates how a *one-layer* transformer can learn symbolic multi-step reasoning—path finding in a tree—by employing the CoT mechanism to make sufficient intermediate reasoning steps. We provide comprehensive analyses covering transformer constructions, training dynamics, and generalization guarantees, demonstrating that when equipped with CoT, even one-layer multi-head transformers are capable to solving complex tasks that require multi-stage and multi-step reasoning. Specifically, our contributions are as follows.

- *Construction.* We provide explicit constructions detailing how a one-layer transformer can perform both tasks using CoT. While a single attention head is sufficient for backward reasoning, it takes two to implement forward reasoning. Our construction elucidates how different attention heads, through coordination, can specialize for distinct reasoning subtasks.

- *Optimization.* We prove that gradient descent successfully trains the transformers to acquire multi-step reasoning capability for solving both tasks via CoT. Our multi-phase training dynamics analysis demonstrates that the model parameters indeed converge to those specified in our construction, explaining how attention heads learn their respective roles in a non-asymptotic manner.

- *Generalization.* We demonstrate that the learned reasoning mechanisms, including the specialized head functions, generalize effectively to correctly solve path-finding problems on unseen tree structures. This demonstrates that transformers learn the generalizable rule for path-finding rather than memorizing paths in training graphs. We also conduct experiments that empirically validate our theoretical predictions.

Our finding offers a mechanistic explanation for how CoT can empower even shallow models to tackle tasks that seemingly require the capabilities of deeper architectures, which is particularly illuminating in the study of forward reasoning. In this setup, the one-layer transformer, utilizing two attention heads, is trained to perform two sequential subtasks: first, it must generate the path in reverse order (goal-to-root) as an explicit CoT sequence; then, upon identifying the root, it must sequentially output the path in the correct forward order (root-to-goal). Our optimization-based analysis reveals how these two heads learn to coordinate: one head is responsible for identifying the next node in the current path-finding stage, while the other head acts as a stage controller, monitoring the current reasoning phase (e.g., backward chaining) and triggering the switch to forward chaining once the root node is detected. This learned specialization allows the one-layer, multi-head transformer to perform this entire two-stage process within a single autoregressive pass.

Importantly, by explicitly generating the backward path as a "scratchpad" in CoT outputs, the one-layer model can then leverage this intermediate information to produce the forward path, rather than relying on depth like in [4]. This resonates with the emergent reasoning abilities in LLMs through test-time scaling [10], where generating longer intermediate steps via CoT often unlocks more sophisticated problem-solving capabilities. Our two-stage setup illustrates how extending the reasoning trace allows a one-layer model to effectively perform a complex task that, without such explicit intermediate steps, might necessitate a deeper model.

## 1.3 Related work

**Theoretical understanding of transformers with CoT.** A growing body of research seeks to theoretically demystify the success of CoT reasoning in transformer models. Most existing studies examine how CoT enhances transformer expressiveness [9, 23, 25, 5], showing that polynomial-length CoT enables transformers to transcend $\mathsf{TC}^0$, a class efficiently handled by constant-depth transformers without CoT [24]. Other works identify intrinsic limitations, establishing lower bounds on necessary intermediate steps for certain problems [27, 3, 2]. Another line of research addresses the learnability [1, 11, 34, 29, 18] and statistical properties [12, 28, 22] of CoT-enabled transformers.

**Optimization theory for CoT.** Recent analyses explore optimization aspects of transformers trained with CoT [21, 13, 33, 18, 13, 17]. The most closely related works are [33, 18, 14], which analyze the optimization dynamics of one-layer transformers trained with CoT to solve parity-check tasks. However, these studies are limited to the single-head setting. In contrast, our work considers multi-head transformers solving a more challenging task involving reasoning over graphs as well as auto-transition over two subtasks, involving specializations and coordination of different heads.

**Training dynamics of transformers.** Many existing works have investigated the training dynamics of transformers across a variety of tasks, including in-context learning [15, 35, 36, 20, 6], induction heads [26, 7], sparse token selection [31], self-supervised learning [16]. Due to the rapid growth of this area, we are unable to reference all related studies. Particularly relevant to our work are recent analyses of CoT reasoning [21, 13, 33, 18, 14]. However, the majority of these studies are limited to the single-head setting. While a few works have considered the multi-head case, they typically address only simple tasks such as linear regression, leaving open the question of how multi-head architectures can be trained to solve more complex reasoning problems.

**Notation.** Let $[N] = \{1, \ldots, N\}$. For a matrix $A$, denote $A_{:,-1}$ or $A(:,-1)$ as its last column. $I_n$ denotes the identity matrix of size $n \times n$. Let $\|\cdot\|_p$ represent the $\ell_p$ norm of a vector.

## 2 Formulation

We consider a path-finding task in randomly generated trees [4], defined formally as below.

**Tree.** We consider the trees with *distinct* nodes, i.e., the nodes of the trees are generated by random sampling from set $[S]$ without replacement. Let $\mathcal{T} = (\mathcal{V}(\mathcal{T}), \mathcal{E}(\mathcal{T}), g(\mathcal{T}))$ denote a tree with node set $\mathcal{V}(\mathcal{T})$, edge set $\mathcal{E}(\mathcal{T})$, and a goal node $g(\mathcal{T})$, which is a leaf node in the tree. Denote the root node as $r(\mathcal{T})$. For each non-root node $i$, i.e., $i \in \mathcal{V}(\mathcal{T}) \backslash r(\mathcal{T})$, let $p(i) = p(i|\mathcal{T})$ denote the parent node of node $i$. Then the edge set is given by $\mathcal{E}(\mathcal{T}) = \{(p(i), i)\}_{i \in \mathcal{V}(\mathcal{T}) \backslash r(\mathcal{T})}$. For $k \geqslant 2$, we recursively define $p^k(i) = p(p^{k-1}(i))$, i.e., $p^k(i)$ is the $k$-th ancestor of $i$.

**Path finding.** Given a tree $\mathcal{T}$, the *backward* path from the goal node to the root node is given by $P_{\mathsf{g2r}}(\mathcal{T}) = g(\mathcal{T}) \rightarrow p(g(\mathcal{T})) \rightarrow p^2(g(\mathcal{T})) \rightarrow \ldots \rightarrow r(\mathcal{T}) = p^{m(\mathcal{T})}(g(\mathcal{T}))$, where $m(\mathcal{T}) \geqslant 2$ is the length of the path. Accordingly, the *forward* path from the root node to the goal node can be obtained by reversing the backward path, yielding, $P_{\mathsf{r2g}}(\mathcal{T}) = r(\mathcal{T}) \rightarrow \cdots \rightarrow p^2(g(\mathcal{T})) \rightarrow p(g(\mathcal{T})) \rightarrow g(\mathcal{T})$. We aim to understand whether and how transformers can be trained to discover the underlying rule for identifying both backward and forward paths in a tree $\mathcal{T}$ through multi-step reasoning. Since a parent node may have multiple children, determining the forward path requires the transformer to autonomously perform two-stage reasoning: first, identifying the backward path, and then reversing it. Therefore, we consider two path-finding tasks:

- *Backward* reasoning: find the goal-to-root path.

- *Forward* reasoning: find the goal-to-root path and then reverse it to produce the root-to-goal path.

**Input embedding.** For each node $i \in [S]$, we let $a_i \in \mathbb{R}^{d_1}$ denote its token embedding. We embed each edge $e = (p(i), i)$ in $\mathcal{E}(\mathcal{T})$ as $(x^\top, y^\top)^\top = (a_{p(i)}^\top, a_i^\top)^\top \in \mathbb{R}^{2d_1}$ where $x = a_{p(i)}$ and $y = a_i$ denote the parent and child node embeddings, respectively. We let $l(\mathcal{T}) = |\mathcal{E}(\mathcal{T})|$ denote the number of edges in $\mathcal{T}$, and let $e_1, \ldots, e_{l(\mathcal{T})}$ denote the edges in $\mathcal{E}(\mathcal{T})$ in a random order, and let $(x_j^\top, y_j^\top)^\top$ be the embedding of edge $e_j$.

**Transformer architecture and multi-step reasoning.** For both reasoning tasks, we consider a one-layer transformer $f$ with $H \geqslant 1$ attention heads. Given an input matrix $E$, the transformer with parameter $\theta$ computes the output as

$$f(E; \theta) = W^O \begin{pmatrix} \mathrm{head}_1(E) \\ \vdots \\ \mathrm{head}_H(E) \end{pmatrix}, \quad \mathrm{head}_h(E) = W_h^V E \cdot \mathsf{softmax}\left(E^\top W_h^{K\top} W_h^Q E\right), \forall h \in [H], \quad (1)$$

where $\mathsf{softmax}(\cdot)$ is the column-wise softmax function and $\theta$ denotes the trainable parameters of the transformer. For simplicity, we follow standard practice [15, 13, 26] to combine $W_h^K$ and $W_h^Q$ into a single matrix $W_h^{KQ}$. We consider *step-by-step* reasoning in an autoregressive manner. At each reasoning step $k \geqslant 0$, given the input embedding $E^{(k)}(\mathcal{T})$, the output vector is taken as the last column of the output matrix, which is then concatenated with the input embedding to form the new input to the next reasoning step, i.e.,

$$\widehat{o}^{(k+1)}(\mathcal{T}; \theta) = f(E^{(k)}(\mathcal{T}); \theta)_{:,-1}, \quad E^{(k+1)}(\mathcal{T}; ; \theta) = (E^{(k)}(\mathcal{T}; ; \theta), \widehat{o}^{(k+1)}(\mathcal{T}; \theta)). \quad (2)$$

Let the number of reasoning steps be $K(\mathcal{T})$, and then $\widehat{O}(\mathcal{T}; \theta) = \left(\widehat{o}^{(1)}(\mathcal{T}; \theta), \ldots, \widehat{o}^{(K(\mathcal{T}))}(\mathcal{T}; \theta)\right)$ denote the output matrix of the model given $\mathcal{T}$.

## 3 Construction of Transformers

In this section, we show that there exists parameter setting $\theta$ such that the one-layer transformer can solve both forward and backward reasoning problems through multi-step chaining and reasoning. Whenever it is clear from the context, we drop the dependency on $\mathcal{T}$ and $\theta$ in the notation.

## 3.1 Construction for backward reasoning

For the backward reasoning task, we are interested in outputting the path from the goal node to the root node (g2r). Let the input embedding of the transformer be

$$E = E_{\text{g2r}}(\mathcal{T}) = \begin{pmatrix} X(\mathcal{T}) \\ Y(\mathcal{T}) \end{pmatrix} = \begin{pmatrix} x_1 & \cdots & x_{l(\mathcal{T})} & a_0 & a_{g(\mathcal{T})} \\ y_1 & \cdots & y_{l(\mathcal{T})} & a_{r(\mathcal{T})} & a_0 \end{pmatrix} \in \mathbb{R}^{2d_1 \times (l(\mathcal{T})+2)}, \quad (3)$$

where $a_0 \in \mathbb{R}^{d_1}$ is used to fill the empty slots. We set the ground label of $\mathcal{T}$ as the embedding of the reversed path:

$$O_{\text{g2r}}(\mathcal{T}) = \begin{pmatrix} a_{p(g(\mathcal{T}))} & a_{p^2(g(\mathcal{T}))} & \cdots & a_{r(\mathcal{T})} \\ a_{g(\mathcal{T})} & a_{p(g(\mathcal{T}))} & \cdots & a_{p^{m(\mathcal{T})-1}(g(\mathcal{T}))} \end{pmatrix} \in \mathbb{R}^{2d_1 \times m(\mathcal{T})}. \quad (4)$$

We consider a one-layer single-head transformer, where $H = 1$. We impose the following parameters: let $W^O = W_1^V = I_{2d_1}$, and set

$$W_1^{KQ} = \begin{pmatrix} 0_{d_1 \times d_1} & 0_{d_1 \times d_1} \\ B & 0_{d_1 \times d_1} \end{pmatrix}$$

where $B \in \mathbb{R}^{d_1 \times d_1}$ is trainable. In this case, $\theta = \{B\}$.

**Multi-step reasoning.** We set the first step $E_{\text{g2r}}^{(0)} = E_{\text{g2r}}(\mathcal{T})$, and let $X^{(k-1)} := E_{\text{g2r}}^{(k-1)}(1 : d_1, :) \in \mathbb{R}^{d_1 \times (l+1+k)}$ and $Y^{(k-1)} := E_{\text{g2r}}^{(k-1)}(d_1 + 1 : 2d_1, :) \in \mathbb{R}^{d_1 \times (l+1+k)}$ obtained recursively following (2) for $k \geqslant 1$. Then according to the transformer architecture described above, we have the output $\hat{o}^{(k)}$ at reasoning step $k$ as

$$\hat{o}^{(k)} = \begin{pmatrix} X^{(k-1)} \\ Y^{(k-1)} \end{pmatrix} \cdot \text{softmax}\left(Y^{(k-1)\top} B x_{-1}^{(k-1)}\right), \quad (5)$$

where $x_{-1}^{(k-1)}$ is the last column of $X^{(k-1)}$. Reasoning recursively for $m(\mathcal{T})$ steps according to (2), we obtain the output $\hat{O}_{\text{g2r}}(\mathcal{T}; \theta) = \left(\hat{o}^{(1)}, \ldots, \hat{o}^{(m(\mathcal{T}))}\right)$.

We make the following assumption on the node embeddings for the backward reasoning case.

**Assumption 1** (linear independent embeddings (backward reasoning)). *Suppose $a_0 = 0_{d_1}$, and $\{a_i\}_{i \in [S]}$ are linearly independent.*

Let $A := (a_1, \ldots, a_S) \in \mathbb{R}^{d_1 \times S}$ be the embedding matrix. The following theorem provides a construction of the transformer that solves backward reasoning.

**Theorem 1** (Construction for backward reasoning). *Under Assumption 1, for any $\alpha \in \mathbb{R}$, there exists $B = B_\alpha \in \mathbb{R}^{d_1 \times d_1}$ such that*

$$A^\top B A = \alpha I_S. \quad (6)$$

*Let $\theta = \{B_\alpha\}$, then for any tree $\mathcal{T}$, we have $\hat{O}_{\text{g2r}}(\mathcal{T}; \theta) \to O_{\text{g2r}}(\mathcal{T})$ as $\alpha \to +\infty$.*

Theorem 1 suggests that it is possible to learn a sharp attention pattern—when applying softmax to $A^\top B A$—between the node embeddings, where the query node attends to only itself among all nodes. Figure 2 (b) provides an illustration of the chain of input-output pairs for each backward reasoning step. This also highlights the path-finding strategy that the transformer learns during training. At each step, the transformer focuses on the current node on the path (starting from the goal node) and uses its attention mechanism to assign high weight to the next node (i.e., its parent) in the tree. The identified node is then selected as the next step in the path and subsequently encoded into the input matrix for the following iteration, enabling the model to reason sequentially through the structure.

## 3.2 Construction for forward reasoning

The forward reasoning task is more challenging than backward reasoning, as it requires first identifying the path from goal to root, and then reversing it to obtain the root-to-goal (r2g) path. Crucially, the transformer must learn to autonomously determine when to switch to the second stage of reasoning.

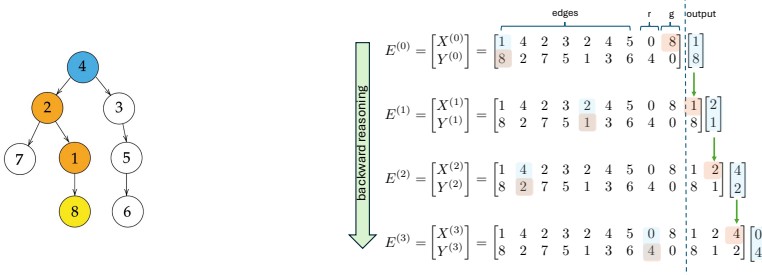

(a) task illustration | (b) multi-step reasoning for finding the goal-to-root path

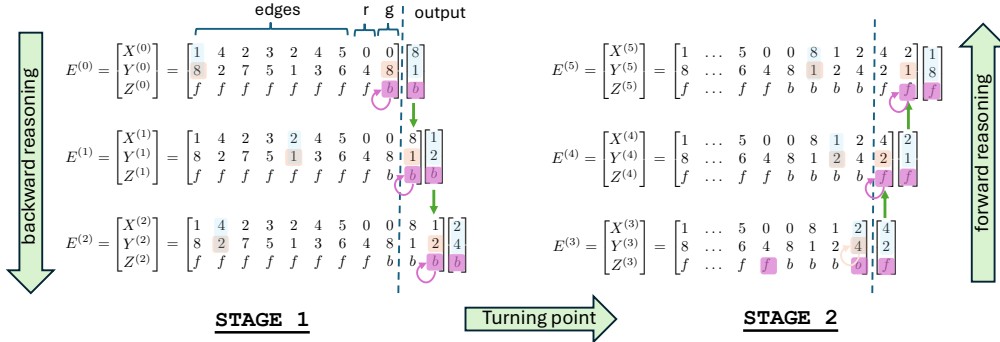

(c) two-stage multi-step reasoning for finding the root-to-goal path

Figure 2: The multi-step reasoning process of the constructed transformers for the backward and forward reasoning tasks. Color indicates the attention association and output in each step.

To this end, we introduce two stage token embeddings $s_f, s_b \in \mathbb{R}^{d_2}$ used to distinguish the two reasoning stages. We include the stage token embeddings in the input as follows:

$$E_{\text{r2g}}(\mathcal{T}) = \begin{pmatrix} X(\mathcal{T}) \\ Y(\mathcal{T}) \\ Z(\mathcal{T}) \end{pmatrix} = \begin{pmatrix} x_1 & \cdots & x_{l(\mathcal{T})} & a_0 & a_0 \\ y_1 & \cdots & y_{l(\mathcal{T})} & a_{r(\mathcal{T})} & a_{g(\mathcal{T})} \\ s_f & \cdots & s_f & s_f & s_b \end{pmatrix} \in \mathbb{R}^{(2d_1+d_2) \times (l(\mathcal{T})+2)}, \quad (7)$$

where $s_b$ marks the start of the first reasoning stage. We require the model to output the path first in the backward order, then in the forward order, along with a stage indicator at each step. This results in a total of $K(\mathcal{T}) = 2m(\mathcal{T})$ steps. We set the ground truth label $O_{\text{r2g}}(\mathcal{T}) \in \mathbb{R}^{(2d_1+d_2) \times 2m(\mathcal{T})}$ of $\mathcal{T}$ as the embedding of both the reverse path and the forward path along with their stage indicators:

$$O_{\text{r2g}}(\mathcal{T}) = \begin{pmatrix} a_{g(\mathcal{T})} & a_{p(g(\mathcal{T}))} & \cdots & a_{r(\mathcal{T})} & a_{p^{m(\mathcal{T})-1}(g(\mathcal{T}))} & \cdots & a_{p(g(\mathcal{T}))} \\ a_{p(g(\mathcal{T}))} & a_{p^2(g(\mathcal{T}))} & \cdots & a_{p^{m(\mathcal{T})-1}(g(\mathcal{T}))} & a_{p^{m(\mathcal{T})-2}(g(\mathcal{T}))} & \cdots & a_{g(\mathcal{T})} \\ s_b & s_b & \cdots & s_f & s_f & \cdots & s_f \end{pmatrix}. \quad (8)$$

We consider a one-layer two-head transformer, where $H = 2$. We impose the following parameters: let $W^O = I_{2d_1+d_2}$, $W_1^V = \begin{pmatrix} 0_{d_1 \times d_1} & I_{d_1} & 0_{d_2} \\ I_{d_1} & 0_{d_1 \times d_1} & 0_{d_2} \end{pmatrix}$, $W_2^V = (0_{d_2 \times 2d_1}, I_{d_2})$, and set

$$W_1^{KQ} = \begin{pmatrix} 0_{d_1 \times d_1} & B_1 & 0_{d_1 \times d_2} \\ 0_{d_1 \times d_1} & B_2 & 0_{d_1 \times d_2} \\ 0_{d_2 \times d_1} & 0_{d_2 \times d_1} & B_3 \end{pmatrix} \quad \text{and} \quad W_2^{KQ} = \begin{pmatrix} 0_{d_1 \times d_1} & C_1 & 0_{d_1 \times d_2} \\ 0_{d_1 \times d_1} & C_2 & 0_{d_1 \times d_2} \\ 0_{d_2 \times d_1} & 0_{d_2 \times d_1} & C_3 \end{pmatrix}$$

with trainable matrices $B_1, B_2, B_3$ and $C_1, C_2, C_3$, respectively. In this case, $\theta = \{B_1, B_2, B_3, C_1, C_2, C_3\}$.

**Multi-step reasoning.** We set the first step $E_{\text{r2g}}^{(0)} = E_{\text{r2g}}(\mathcal{T})$. At the reasoning step $k \geq 1$, let $X^{(k-1)} := E_{\text{r2g}}^{(k-1)}(1 : d_1, :) \in \mathbb{R}^{d_1 \times (l+1+k)}$, $Y^{(k-1)} := E_{\text{r2g}}^{(k-1)}(d_1 + 1 : 2d_1, :) \in \mathbb{R}^{d_1 \times (l+1+k)}$, and $Z^{(k-1)} := E_{\text{r2g}}^{(k-1)}(2d_1 + 1 : 2d_1 + d_2, :) \in \mathbb{R}^{d_2 \times (l+1+k)}$, where $E_{\text{r2g}}^{(k-1)}$ is obtained recursively

from (2). Then we have

$$\widehat{o}^{(k)} = \left( \begin{array}{c} \left( \begin{array}{c} Y^{(k-1)} \\ X^{(k-1)} \end{array} \right) \cdot \mathsf{softmax} \left( X^{(k-1)\top} B_1 y_{-1}^{(k-1)} + Y^{(k-1)\top} B_2 y_{-1}^{(k-1)} + Z^{(k-1)\top} B_3 z_{-1}^{(k-1)} \right) \\ Z^{(k-1)} \cdot \mathsf{softmax} \left( X^{(k-1)\top} C_1 y_{-1}^{(k-1)} + Y^{(k-1)\top} C_2 y_{-1}^{(k-1)} + Z^{(k-1)\top} C_3 z_{-1}^{(k-1)} \right) \end{array} \right). \tag{9}$$

Here, the upper portion $\widehat{o}^{(k)}(1 : 2d_1, :)$ is the output of the first attention head, predicting the next node based on the learned path encoded in $X^{(k-1)}$ and $Y^{(k-1)}$, and the lower portion $\widehat{o}^{(k)}(2d_1 + 1 : 2d_1 + d_2, :)$ is the output of the second attention head, indicating whether the second stage begins.

The process of predicting $\{\widehat{o}^{(k)}\}_{k=1}^{(m(\mathcal{T}))}$ is called *STAGE 1*, where the model outputs the reverse path from goal to root similarly to backward reasoning. The prediction of $\widehat{o}^{(m(\mathcal{T})+1)}$ is the *turning point*, because in this step the model detects the root and switches to *STAGE 2* (indicated by $s_f$), where the model outputs the path in forward order from root to goal. The entire process includes reasoning recursively for $2m(\mathcal{T})$ steps, and the final output is given by $\widehat{O}_{r2g}(\mathcal{T}; \theta) = (\widehat{o}^{(1)}, \dots, \widehat{o}^{(2m(\mathcal{T}))})$.

We make the following mild assumption on the embeddings for forward reasoning.

**Assumption 2** (linear independent embeddings (forward reasoning)). *Suppose (i) $a_0, a_1, \dots, a_S$ are linearly independent; (ii) $s_f, s_b$ are linearly independent.*

Denote $\widetilde{A} = (a_0, a_1, \dots, a_S) \in \mathbb{R}^{d_1 \times (S+1)}$ and $\widetilde{S} = (s_f, s_b)$. The following theorem provides a construction of a one-layer two-head transformer that solves forward reasoning.

**Theorem 2** (Construction for forward reasoning). *Under Assumption 2, there exist $\theta = \{B_i, C_i\}_{i \in [3]}$ that satisfies*

$$\widetilde{A}^\top B_1 A = -a\alpha_1 \begin{pmatrix} 1 & 1 & \cdots & 1 \\ 0 & 0 & \cdots & 0 \\ \vdots & \vdots & & \vdots \\ 0 & 0 & \cdots & 0 \end{pmatrix}, \quad A^\top B_2 A = \alpha_1 I_S, \quad \widetilde{S}^\top B_3 \widetilde{S} = \alpha_1 \begin{pmatrix} -b_1 & b_2 \\ b_1 & -b_2 \end{pmatrix}, \tag{10a}$$

$$\widetilde{A}^\top C_1 A = \alpha_2 \begin{pmatrix} 1 & 1 & \cdots & 1 \\ 0 & 0 & \cdots & 0 \\ \vdots & \vdots & & \vdots \\ 0 & 0 & \cdots & 0 \end{pmatrix}, \quad A^\top C_2 A = \alpha_2 I_S, \quad \widetilde{S}^\top C_3 \widetilde{S} = \alpha_2 \begin{pmatrix} c_1 & -c_2 \\ -c_1 & c_2 \end{pmatrix}. \tag{10b}$$

*When $a \in (0, 1]$, $b_1 > 0$, $b_2 \in (0, a/2)$ and $c_1 > 0$, $c_2 \in (0, 1/2)$, we have $\widehat{O}_{r2g}(\mathcal{T}; \theta) \to O_{r2g}(\mathcal{T})$ as $\alpha_1, \alpha_2 \to +\infty$.*

Specifically, let $m := m(\mathcal{T})$ denote the path length and $n^{(k-1)} \in [S]$ denote the node with embedding $y_{-1}^{(k-1)}$ for any $k \in [2m]$. Then we can show that when $\alpha_1, \alpha_2 \to +\infty$, our construction ensures that at each reasoning step $k$, the model performs the following (see Figure 2 (c) for an illustration):

1. when $k \leqslant m$, i.e., the model is at *STAGE 1*. Our construction ensures the first attention head attends to the embedding of $(p(n^{(k-1)}), n^{(k-1)})$, switches the parent-child order, and outputs the embedding of $(n^{(k-1)}, p(n^{(k-1)}))$, and the second head attends to and outputs the stage signal $s_b$;

2. when $k = m + 1$, i.e., the model is at the *turning point*. Our construction ensures that the first attention head attends to $(y_{-1}^{(k-1),\top}, x_{-1}^{(k-1),\top})^\top$ and outputs $(x_{-1}^{(k-1),\top}, y_{-1}^{(k-1),\top})^\top$, and the second attention head attends to and outputs the stage signal $s_f$, changing the stage signal.

3. when $k > m + 1$, i.e., the model is at *STAGE 2*. The first attention head attends to the embedding of $(n^{(k-1)}, p(n^{(k-1)}))$ generated in *STAGE 1*, switches the parent-child order, and outputs the embedding of $(p(n^{(k-1)}), n^{(k-1)})$, and the second attention head attends to and outputs $s_f$.

This process demonstrates that the first attention head is responsible for traversing the reasoning path, while the second attention head manages the stage signal, which dictates the direction of path-finding.

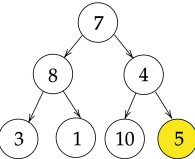

Figure 3: An example of a perfect binary tree of depth $m = 2$ and distinct nodes.

The importance of Theorems 1 and 2 lies in demonstrating that even a single-layer transformer suffices to carry out the above logical steps with enough CoT steps. This highlights the surprising expressive power of even shallow transformer architectures for multi-step reasoning, in contrast to leveraging multi-layer transformers for such a multi-step reasoning task in **(author?)** [4].

## 4   Training Dynamics and Generalization of Transformers

In Section 3, we showed the existence of a one-layer transformer capable of solving both backward and forward reasoning tasks. However, those results do not address whether such transformers can be learned in practice. In this section, we analyze the training dynamics of the transformer in both settings and demonstrate that gradient descent (GD) can successfully minimize the loss to 0. Upon convergence, the transformer performs the desired backward and forward reasoning. This establishes that transformers can, in fact, be trained via GD to acquire multi-step reasoning capabilities. We further show that the trained model generalizes well to unseen trees.

**Training.** We train the transformer using standard autoregressive supervised learning. We define $o^{(k)}(\mathcal{T})$ as the $k$-th column of $O(\mathcal{T}) := O_{\mathsf{g2r}}(\mathcal{T})$ or $O_{\mathsf{r2g}}(\mathcal{T})$, the ground truth label for either case. For each tree $\mathcal{T} \sim P_{\mathsf{train}}$, where $P_{\mathsf{train}}$ is the training distribution to be described later, at each reasoning step $k \geqslant 2$, the input $E_{\mathsf{train}}^{(k-1)}(\mathcal{T}) = (E_{\mathsf{train}}^{(k-2)}(\mathcal{T}), o^{(k-1)}(\mathcal{T}))$ is the concatenation of the input at step $k-1$ and the $(k-1)$-th column of the ground truth output matrix $O(\mathcal{T})$. We then obtain the output $\hat{o}_{\mathsf{train}}^{(k)}(\mathcal{T}; \theta) = f(E_{\mathsf{train}}^{(k-1)}(\mathcal{T}); \theta)_{:,-1}$ at step $k$. Let $\hat{O}_{\mathsf{train}}(\mathcal{T}; \theta) = (\hat{o}^{(1)}(\mathcal{T}; \theta), \ldots, \hat{o}^{(K(\mathcal{T}))}(\mathcal{T}; \theta))$ denote the output matrix of the model given $\mathcal{T}$.

We train the model by minimizing the squared error loss between the output and the label. The training loss is given by

$$\mathcal{L}_{\mathsf{train}}(\theta) = \frac{1}{2}\mathbb{E}_{\mathcal{T} \sim P_{\mathsf{train}}} \left\| O(\mathcal{T}) - \hat{O}_{\mathsf{train}}(\mathcal{T}; \theta) \right\|_F^2. \tag{11}$$

For both cases, we initialize all parameters to be zero, and we train the model using gradient descent with a learning rate $\eta > 0$, i.e.,

$$\theta^{(t+1)} = \theta^{(t)} - \eta \nabla_\theta \mathcal{L}_{\mathsf{train}}(\theta^{(t)}), \quad \forall t \geqslant 0. \tag{12}$$

Here, $\theta^{(t)}$ is the parameter at the $t$-th iteration.

**Training distribution.** We make the following assumption on the training set $P_{\mathsf{train}}$.

**Assumption 3** (training distribution). *Let $m \geqslant 3$ and $S \geqslant 2^{m+1} - 1 := N$. The training distribution $P_{\text{train}}$ is the uniform distribution over all perfect binary trees $\mathcal{T} = (\mathcal{V}(\mathcal{T}), \mathcal{E}(\mathcal{T}), g(\mathcal{T}))$ of depth[5] $m$ with distinct nodes chosen from $[S]$ and a goal node $g(\mathcal{T})$ being one of the leaf nodes.*

We note that we fix the tree to be a perfect binary tree of depth $m$ during training only for simplicity of analysis and presentation, and our results can be easily extended to other types of trees. Also note that under Assumption 3, the number of nodes in any tree sampled from $P_{\mathsf{train}}$ is $N = 2^{m+1} - 1$. An example of a perfect binary tree generated from $P_{\mathsf{train}}$ is shown in Figure 3, where we set $m = 2$.

**Testing.** At test time, given any tree $\mathcal{T}$ (that may not be in the training set), we recursively input

$$E^{(k-1)}(\mathcal{T}; \theta) = (E(\mathcal{T}), \hat{O}^{(k-1)}(\mathcal{T}; \theta))$$

---

[5]The depth (or height) of a tree is the length of the longest path from the root down to any leaf, measured in the number of edges.

into the model at the $k$-th reasoning step and obtain the output $\widehat{o}^{(k)}(\mathcal{T};\theta) = f(E^{(k-1)}(\mathcal{T};\theta);\theta)_{:,-1}$. The test loss on $\mathcal{T}$ is given by

$$\mathcal{L}_{\text{test}}(\mathcal{T};\theta) = \frac{1}{2}\left\|O(\mathcal{T}) - \widehat{O}(\mathcal{T};\theta)\right\|_F^2. \tag{13}$$

## 4.1 Optimization for Backward Reasoning

For ease of analysis, we make the following assumption on the node embeddings for backward reasoning, which is a slightly stronger version of Assumption 1.

**Assumption 4** (Orthonormal embeddings (backward reasoning)). *Suppose (i) $a_0 = 0_{d_1}$, and (ii) $\{a_i\}_{i\in[S]}$ are orthonormal vectors in $\mathbb{R}^{d_1}$.*

Note that commonly used one-hot embeddings are orthonormal [8], and orthogonality of token embeddings is also assumed in **(author?)** [15, 26, 6, 21] for analyzing the training dynamics of transformers.

**Training dynamics.** We have the following theorem regarding the convergence of the training error for backward reasoning.

**Theorem 3** (Convergence of backward reasoning). *We initialize $\theta^{(0)} = B^{(0)} = 0_{d_1 \times d_1}$. Under Assumptions 3 and 4, for any learning rate $\eta \in (0,1]$ and any $\epsilon \in (0,1]$, there exists $T = \mathcal{O}\left(\frac{mS}{\eta\epsilon}\log\left(\frac{Nm}{\epsilon}\right)\right)$ such that for any $t \geqslant T$, we have $\mathcal{L}_{train}(\theta^{(t)}) \leqslant \epsilon$.*

Theorem 3 establishes that to achieve $\epsilon$-accuracy, it takes at most $\widetilde{\mathcal{O}}(1/\epsilon)$ iterations, omitting the dependency with other salient parameters. Recall in Theorem 1 (cf. Section 3.1), we construct $B$ such that $H = A^\top BA = \alpha I_S$, ensuring the test loss approaches zero as $\alpha \to +\infty$. In our convergence analysis of GD, we track the dynamic of $H^{(t)} = A^\top B^{(t)}A$. Specifically, during training, the diagonal entries of $H^{(t)}$ continue to grow while the off-diagonal entries remain small, indicating $B^{(t)}$ indeed converge toward our constructed solution.

**Generalization.** We further provide the following generalization result, which shows that the model can perform well on unseen trees.

**Theorem 4** (Generalization of backward reasoning). *Under the same setting as in Theorem 3, given any tree $\mathcal{T}$ with $\widetilde{N}$ distinct nodes chosen from $[S]$ and a path length $\widetilde{m}$, there exists small enough $\epsilon_0 = \epsilon_0(m, \widetilde{N}, \widetilde{m}) > 0$ such that for any $\epsilon \in (0, \epsilon_0]$, there exists $T = \widetilde{\mathcal{O}}\left(\frac{mS}{\eta\epsilon}\right)$ such that after training the model for $t \geqslant T$ steps on the training loss, we have*

$$\mathcal{L}_{test}(\mathcal{T};\theta^{(t)}) \leqslant 4\max\left\{1, \left(\frac{\widetilde{N}+\widetilde{m}-1}{N+m-2}\right)^2\right\} \cdot \frac{\widetilde{m}}{m}\epsilon.$$

Theorem 4 implies that for any tree $\mathcal{T}$ (including those not seen during training), the test loss converges to zero, provided the model is sufficiently trained on $P_{\text{train}}$. This reveals an appealing insight: the transformer *learns a rule* for path finding, rather than memorizing training examples.

## 4.2 Optimization for Forward Reasoning

Similar to Assumption 2, we make the following assumption for analyzing the training dynamics of forward reasoning.

**Assumption 5** (Orthonormal embeddings (forward reasoning)). *Suppose (i) $a_0, a_1, \ldots, a_S$ are orthonormal vectors in $\mathbb{R}^{d_1}$; and (ii) $s_f, s_b$ are orthonormal vectors in $\mathbb{R}^{d_2}$.*

**Training dynamics.** The following theorem guarantees the convergence of GD on the forward reasoning task.

**Theorem 5** (Training dynamics of forward reasoning). *We initialize $\theta^{(0)} = \{B_1^{(0)}, B_2^{(0)}, B_3^{(0)}, C_1^{(0)}, C_2^{(0)}, C_3^{(0)}\}$ all by zero. Suppose Assumptions 3 and 5 hold, and $N$ is sufficiently large, and learning rate $\eta \lesssim \frac{1}{mN}$. Then for any $\eta \in (0, \eta_0]$ and $\epsilon \in (0, \epsilon_0]$, where*

$\eta_0 > 0$ *and* $0 < \epsilon_0 < 1/\boldsymbol{poly}(N)$ *are sufficiently small, there exists* $T = \mathcal{O}\left(\frac{S}{\eta}\left(\frac{m}{\epsilon}\right)^{3/2}\log\left(\frac{N}{\epsilon}\right)\right)$ *such that for any* $t \geqslant T$*, we have* $\mathcal{L}_{train}(\theta^{(t)}) \leqslant \epsilon$.

Theorem 5 establishes that to achieve $\epsilon$-accuracy, it takes at most $\widetilde{\mathcal{O}}\left(1/\epsilon^{3/2}\right)$ iterations, omitting the dependency with other salient parameters. Define the following trainable matrices, which can be viewed as counterparts to those constructed in (10a) and (10b):

$$U_1^{(t)} := \widetilde{A}^\top B_1^{(t)} A \in \mathbb{R}^{(S+1)\times S}, \ U_2^{(t)} := A^\top B_2^{(t)} A \in \mathbb{R}^{S\times S}, \ U_3^{(t)} := \widetilde{S}^\top B_3^{(t)} \widetilde{S} \in \mathbb{R}^{2\times 2}, \quad \text{(14a)}$$

$$V_1^{(t)} := \widetilde{A}^\top C_1^{(t)} A \in \mathbb{R}^{(S+1)\times 2}, \ V_2^{(t)} := A^\top C_2^{(t)} A \in \mathbb{R}^{S\times 2}, \ V_3^{(t)} := \widetilde{S}^\top C_3^{(t)} \widetilde{S} \in \mathbb{R}^{2\times 2}. \quad \text{(14b)}$$

Our convergence analysis reveals that $U_l^{(t)}$ and $V_l^{(t)}$, $l = 1, 2, 3$, converge to the matrices given in the construction (10a) and (10b), respectively. In particular, we show that

- For $U_1^{(t)}$ (resp. $V_1^{(t)}$), the first row decreases (resp. increases), and the other entries stay small.

- For both $U_2^{(t)}$ and $V_2^{(t)}$, the diagonal entries are strictly increasing, and the other entries stay small.

- For $U_3^{(t)}$ and $V_3^{(t)}$, we have

$$U_{3,0,0}^{(t)} = -U_{3,1,0}^{(t)}, \quad U_{3,0,1}^{(t)} = -U_{3,1,1}^{(t)}, \quad V_{3,0,0}^{(t)} = -V_{3,1,0}^{(t)}, \quad V_{3,0,1}^{(t)} = -V_{3,1,1}^{(t)}. \quad \text{(15)}$$

  In addition, $U_{3,0,1}^{(t)}$ first decreases and then increases, $U_{3,0,0}^{(t)}$ (resp. $V_{3,0,0}^{(t)}$) is strictly decreasing (resp. increasing), and $V_{3,1,1}^{(t)}$ also grows large, but not monotonically.

- At convergence, $U_k^{(t)}, V_k^{(t)}$ are close to the matrices in (10a) and (10b) with large enough $\alpha_1, \alpha_2, a$ close to 1, $b_1 \geqslant b_2 \approx 1/4$, and $c_1 \geqslant c_2 \approx 1/4$.

**Generalization.** We further provide the generalization guarantee for forward reasoning. We show that the model performs well on unseen trees at test time, similarly to the generalization result in Theorem 4 established for backward reasoning.

**Theorem 6** (Generalization of forward reasoning)**.** *Under the same setting as in Theorem 5, given any tree* $\mathcal{T}$ *at test time with* $\widetilde{N}$ *distinct nodes chosen from* $[S]$ *and a path length* $\widetilde{m}$*, there exists small enough* $\epsilon_0 = \epsilon_0(m, \widetilde{N}, \widetilde{m}) > 0$ *such that for any* $\epsilon \in (0, \epsilon_0]$*, there exists* $T = \widetilde{\mathcal{O}}\left(\frac{m^{3/2}S}{\eta\epsilon^{3/2}}\right)$ *such that after training the model for* $t \geqslant T$ *steps on the training loss, we have*

$$\mathcal{L}_{test}(\mathcal{T}; \theta) \leqslant 4 \max\left\{\left(\frac{\widetilde{N} + 2\widetilde{m} - 1}{N + 2m - 1}\right)^2, \left(\frac{\widetilde{N}}{N}\right)^2, 1\right\}\epsilon. \quad \text{(16)}$$

We also conduct experiments to validate the theoretical results. Due to the space limit, we refer the reader to Appendix A.

# 5  Conclusion

This work provides a theoretical investigation into how one-layer transformers learn symbolic multi-step reasoning via gradient descent, focusing on path-finding in trees. We demonstrate through explicit constructions that transformers can provably solve both backward (goal-to-root) and the more complex, two-stage forward (root-to-goal) reasoning tasks by employing CoT mechanisms. Our optimization analysis reveals that gradient descent successfully guides the model parameters to these effective configurations, with multi-head attention mechanisms learning to specialize and autonomously coordinate for distinct subtasks—such as path traversal and stage control—within a single autoregressive pass. We also show that these learned abilities generalize to unseen tree structures, indicating the transformer acquires underlying algorithmic rules, rather than memorizing instances. These findings offer a mechanistic interpretation of how shallow transformers leverage CoT for sequential, multi-stage reasoning procedures.

## Acknowledgments and Disclosure of Funding

The work of T. Yang and Y. Chi is supported in part by the grants NSF CCF-2007911, DMS-2134080 and ONR N00014-19-1-2404. T. Yang is also supported by the Wei Shen and Xuehong Zhang Presidential Fellowship at Carnegie Mellon University. The work of Y. Liang was supported in part by the U.S. National Science Foundation under the grants ECCS-2515482 and DMS-2134145.

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

# A Numerical Experiments

In this section, we provide experimental validation of our theoretical results.

**Setup.** We construct the transformer model as in (5) and (9) for the backward and forward reasoning task, respectively. We use one-hot embedding: we embed $a_i = e_i$ for all $i \in [S]$, where $e_i$ is the one-hot vector with the $i$-th entry being 1. For training, since the expected loss is not available, for both backward and forward reasoning, we use stochastic gradient descent with a batch size 256 on randomly generated perfect binary trees. The tree generation process is the same as described in Section 4 with $m = 4$, $S = 31$ for backward reasoning, and $m = 3$, $S = 25$ for forward reasoning. We use learning rates 1 and 0.2 for backward and forward reasoning, respectively. To validate the generalization performance, we randomly generate 1024 trees with various depths and number of nodes as the test set. Different nodes of every tree in the test set are assigned with a unique number chosen from $[S]$, and each node can have different numbers of children (range from 0 to 3).

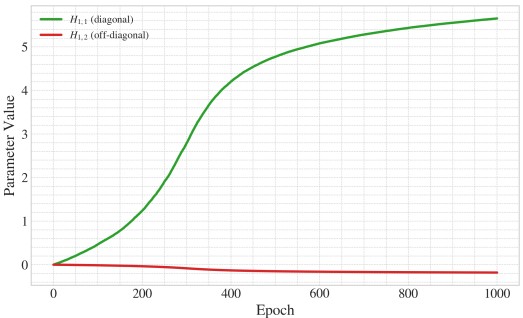

Figure 4: Training dynamics of selected entries of $H$.

**Experimental results for backward reasoning.** Figure 5 shows the training and test loss curves for backward reasoning, which (i) validates the convergence of the training process, and (ii) shows the generalization error also converges to 0. Moreover, same as our construction and training dynamics

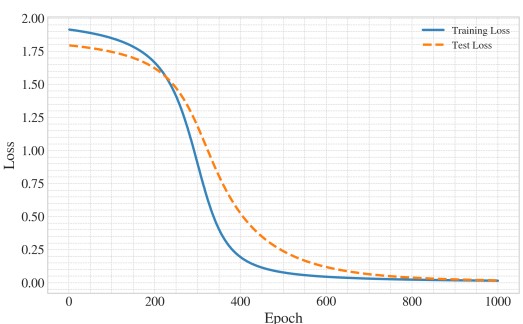

Figure 5: Training and test loss curves for backward reasoning.

analysis, during the training process, the diagonal entries of $H^{(t)} = A^\top B^{(t)} A$ (cf. (6)) increases while the off-diagonal entries stay small. See Figure 4, where we plot the training dynamics of $H_{1,1}$ and $H_{1,2}$.

**Experimental results for forward reasoning.** Figure 6 shows the training and test loss curves for forward reasoning, which (i) validates the convergence of the training process, and (ii) shows the generalization error also converges to 0. Moreover, by tracking $U_l^{(t)}, V_l^{(t)}$ ($l = 1, 2, 3$) defined in (14), we can see that they converge to our construction, and the true training dynamics also matches our theoretical analysis, see Figure 7. For example, in the right top figure, we plot the curves of the four entries of $U_3$. Identical to our analysis (c.f. Appendix C.2.3), $U_{3,0,0} = -U_{3,1,0}$, $U_{3,0,1} = -U_{3,1,1}$, $U_{3,1,0}$ keeps increasing while $U_{3,0,1}$ first decreases and then increases.

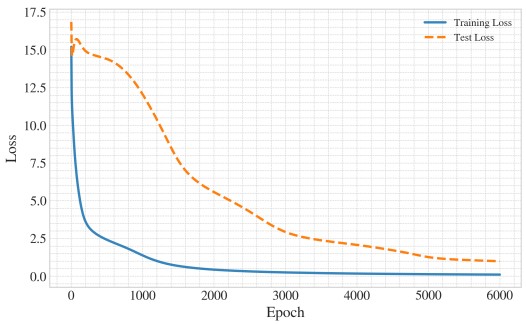

Figure 6: Training and test loss curves for forward reasoning.

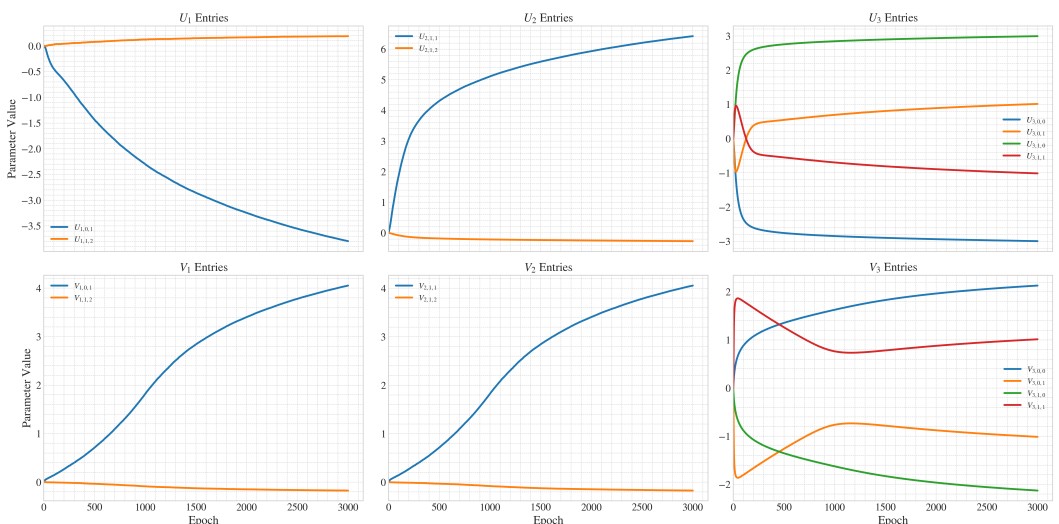

Figure 7: Training dynamics of selected entries of $U_l, V_l$ for $l = 1, 2, 3$.

## B  Construction Analysis

We first present the following lemma, which presents the existence of parameters for the desired attention patterns.

**Lemma 1.** *Given $\{u_i\}_{i \in [N_1]} \subset \mathbb{R}^{m_1}$, $\{v_j\}_{j \in [N_2]} \subset \mathbb{R}^{m_2}$, where $u_1, \ldots, u_{N_1}$ are linearly independent, and $v_1, \ldots, v_{N_2}$ are linearly independent. Then for any fixed matrix $A \in \mathbb{R}^{N_1 \times N_2}$, there exists a matrix $B \in \mathbb{R}^{m_1 \times m_2}$ such that*

$$u_i^\top B v_j = A_{ij}, \ \forall i \in [N_1], j \in [N_2].$$

*Proof.* We let

$$U := (u_1, \ldots, u_{N_1}) \in \mathbb{R}^{m_1 \times N_1}, \quad V := (v_1, \ldots, v_{N_2}) \in \mathbb{R}^{m_2 \times N_2}, \tag{17}$$

and use $\otimes$ to denote the Kronecker product. We have for any $B \in \mathbb{R}^{m_1 \times m_2}$,

$$U^\top B V = A \ \Leftrightarrow \ \left(V^\top \otimes U^\top\right) \mathsf{vec}(B) = \mathsf{vec}(A). \tag{18}$$

Note that since $u_1, \ldots, u_{N_1}$ are linearly independent, $v_1, \ldots, v_{N_2}$ are linearly independent, we have $\mathrm{rank}(U) = N_1$ and $\mathrm{rank}(V) = N_2$, which implies

$$\mathrm{rank}(V^\top \otimes U^\top) = \mathrm{rank}(U^\top)\mathrm{rank}(V^\top) = N_1 N_2.$$

Therefore, the linear system

$$\left(V^\top \otimes U^\top\right) b = \mathsf{vec}(A) \tag{19}$$

has a solution $b \in \mathbb{R}^{m_1 m_2}$. (18) implies reshaping $b$ to a matrix $B \in \mathbb{R}^{m_1 \times m_2}$ gives the desired $B$. □

## B.1 Proof of Theorem 1

When Assumption 1 holds, the existence of $B$ satisfying (6) follows from Lemma 1. Letting $\alpha \to +\infty$, it follows that

$$\mathsf{softmax}(A^\top B A) = I_S$$

converges to an identity matrix. By (4), we know that at the $k$-th reasoning step, $\mathsf{softmax}\left(Y^{(k-1)\top} B x_{-1}^{(k-1)}\right)$ becomes a one-hot vector with the $j$-th entry being 1 if $y_j^{(k-1)} = x_{-1}^{(k-1)}$ and 0 otherwise (note that $x_{-1}^{(k-1)}$ will appear exactly once in $Y^{(k-1)}$), and thus the model will output $\widehat{o}^{(k)} = (x_j^{(k-1)\top}, y_j^{(k-1)\top})^\top$, which is the embedding of the edge that connects the current node $x_{-1}^{(k-1)}$ with its parent $x_j^{(k-1)\top}$. Thus by induction $\widehat{o}^{(k)} = o^{(k)}$ for all $k \in [m]$.

## B.2 Proof of Theorem 2

By Assumption 2, the existence of $\theta$ satisfying (10) follows from Lemma 1.

For the forward reasoning task, there are $2m$ reasoning steps in total, where $m$ represents the path length. For any $k \in [2m]$, we let

$$\mu^{(k-1)} := X^{(k-1)\top} B_1 y_{-1}^{(k-1)} + Y^{(k-1)\top} B_2 y_{-1}^{(k-1)} + Z^{(k-1)\top} B_3 z_{-1}^{(k-1)}, \tag{20a}$$

$$\nu^{(k-1)} := X^{(k-1)\top} C_1 y_{-1}^{(k-1)} + Y^{(k-1)\top} C_2 y_{-1}^{(k-1)} + Z^{(k-1)\top} C_3 z_{-1}^{(k-1)}. \tag{20b}$$

Let $\mu_i^{(k-1)}$ (resp. $\nu_i^{(k-1)}$) denote the $i$-th entry of $\mu^{(k-1)}$ (resp. $\nu^{(k-1)}$). We also denote $E^{(0)}(:, i) = E_{\mathsf{r2g}}(:, i) = (x_i^\top, y_i^\top, z_i^\top)^\top$ for $i \in [l+2]$, where $x_i \in \mathbb{R}^{d_1}$, $y_i \in \mathbb{R}^{d_1}$, and $z_i \in \mathbb{R}^{d_2}$, and $l$ stands for the number of edges in the tree $\mathcal{T}$.

In the following, we will prove by induction that under the assumptions in Theorem 2, for any $k \in [2m]$, we have

$$\widehat{o}^{(k)} = o^{(k)}, \quad \text{as } \alpha_1, \alpha_2 \to +\infty. \tag{21}$$

**Base case ($k = 1$).** Without loss of generality, we can assume

$$E^{(0)}(1 : 2d_1, 1) := (x_1^\top, y_1^\top)^\top = \left(a_{p(g)}^\top, a_g^\top\right)^\top,$$

where $a_g$ is the embedding of the goal node. We compute the attention pattern by checking the entries in (20).

- $i = 1$. Since $y_1 = a_g = y_{-1}^{(0)}$, $z_1 = s_f$, and $z_{-1}^{(0)} = s_b$, by our construction (10), we have

$$\forall k \in [m]: \quad \mu_1^{(0)} = x_1^\top B_1 y_{-1}^{(0)} + y_1^\top B_2 y_{-1}^{(0)} + z_1^\top B_3 z_{-1}^{(0)} = (1 + b_2)\alpha_1, \tag{22a}$$

$$\nu_1^{(0)} = x_1^\top C_1 y_{-1}^{(0)} + y_1^\top C_2 y_{-1}^{(0)} + z_1^\top C_3 z_{-1}^{(0)} = (1 - c_2)\alpha_2. \tag{22b}$$

- $i = 2, 3, \ldots, l$. We have

$$x_i \neq a_0, \quad y_i \neq y_{-1}^{(0)}, \quad \text{and} \quad z_i = s_f,$$

where the second relation holds because each node embedding appears exactly once in $Y^{(0)}(:, 1 : l+1)$. Thus we have

$$\forall k \in [m]: \quad \mu_i^{(0)} = x_i^\top B_1 y_{-1}^{(0)} + y_i^\top B_2 y_{-1}^{(0)} + z_i^\top B_3 z_{-1}^{(0)} = b_2 \alpha_1, \tag{23a}$$

$$\nu_i^{(0)} = x_i^\top C_1 y_{-1}^{(0)} + y_i^\top C_2 y_{-1}^{(0)} + z_i^\top C_3 z_{-1}^{(0)} = -c_2 \alpha_2, \quad \forall i = 2, 3, \ldots, l. \tag{23b}$$

- $i = l + 1$. Since we have

$$E^{(0)}(:, l+1) = (a_0^\top, a_r^\top, s_f^\top)^\top$$

where $a_r$ is the embedding of the root node, and $a_r \neq y_{-1}^{(0)}$. We deduce

$$\mu_{l+1}^{(0)} = a_0^\top B_1 y_{-1}^{(0)} + a_r^\top B_2 y_{-1}^{(0)} + s_f^\top B_3 z_{-1}^{(0)} = (-a + b_2)\alpha_1, \tag{24a}$$

$$\nu_{l+1}^{(0)} = a_0^\top C_1 y_{-1}^{(0)} + a_r^\top C_2 y_{-1}^{(0)} + s_f^\top C_3 z_{-1}^{(0)} = (1 - c_2)\alpha_2. \tag{24b}$$

- $i = l + 2$. Since

$$E^{(0)}(:, l+2) = \left(x_{-1}^{(0)\top}, y_{-1}^{(0)\top}, z_{-1}^{(0)\top}\right)^\top = \left(a_0^\top, a_g^\top, s_b^\top\right)^\top,$$

we have

$$\mu_{l+2}^{(0)} = (-a + 1 - b_2)\alpha_1, \quad \nu_{l+2}^{(0)} = (2 + c_2)\alpha_2. \tag{25}$$

Combining the above relation, we see that

$$\mathsf{softmax}(\mu^{(0)}) \to e_1, \quad \mathsf{softmax}(\nu^{(0)}) \to e_{l+2}, \quad \text{as } \alpha_1, \alpha_2 \to +\infty. \tag{26}$$

Therefore, by (9) we have

$$\hat{o}^{(1)} = \left( \begin{pmatrix} Y^{(0)} \\ X^{(0)} \end{pmatrix} \cdot \mathsf{softmax}\left(\mu^{(0)}\right) \\ Z^{(0)} \cdot \mathsf{softmax}\left(\nu(0)\right) \right) \to \begin{pmatrix} a_g \\ a_{p(g)} \\ s_b \end{pmatrix} \overset{(8)}{=} o^{(1)}, \quad \text{as } \alpha_1, \alpha_2 \to +\infty, \tag{27}$$

i.e., (21) holds for $k = 1$.

**Induction hypothesis.** Suppose (21) is true for $k = 2, \ldots, p - 1$ ($2 \leqslant p \leqslant 2m$). Below we will prove that (21) is also true for $k = p$. By the induction hypothesis, the input at the $p$-th reasoning step satisfies

$$E^{(p-1)} = (E, \hat{O}^{(p-1)}) \to (E, O^{(p-1)}) := \overline{E}^{(p-1)} := \begin{pmatrix} \overline{X}^{(p-1)} \\ \overline{Y}^{(p-1)} \\ \overline{Z}^{(p-1)} \end{pmatrix} \quad \text{as } \alpha_1, \alpha_2 \to +\infty, \tag{28}$$

where $\overline{X}^{(p-1)} := E^{(p-1)}(1 : d_1, :) \in \mathbb{R}^{d_1 \times (l+p+1)}$, $\overline{Y}^{(p-1)} := E^{(p-1)}(d_1 + 1 : 2d_1, :) \in \mathbb{R}^{d_1 \times (l+p+1)}$, and $\overline{Z}^{(p-1)} := E^{(p-1)}(2d_1 + 1 : 2d_1 + d_2, :) \in \mathbb{R}^{d_2 \times (l+p+1)}$. For notation simplicity, we drop the superscript $(p - 1)$ and let $x_i$, $y_i$, and $z_i$ denote the $i$-th column of $X^{(p-1)}$, $Y^{(p-1)}$, and $Z^{(p-1)}$, respectively, and let $\overline{x}_i$, $\overline{y}_i$, and $\overline{z}_i$ denote the $i$-th column of $\overline{X}^{(p-1)}$, $\overline{Y}^{(p-1)}$, and $\overline{Z}^{(p-1)}$, respectively for all $i \in [l + p + 1]$. We have

$$(\overline{x}_i^\top, \overline{y}_i^\top, \overline{z}_i^\top)^\top = (x_i^\top, y_i^\top, z_i^\top)^\top, \quad \forall i \in [l + 2], \tag{29a}$$

$$(x_i^\top, y_i^\top, z_i^\top)^\top \to (\overline{x}_i^\top, \overline{y}_i^\top, \overline{z}_i^\top)^\top \quad \text{as } \alpha_1, \alpha_2 \to +\infty, \quad \forall i = l + 3, \ldots, l + p + 1. \tag{29b}$$

We also let $\overline{y}_{-1}^{(p-1)} := \overline{y}_{l+p+1}, \overline{z}_{-1}^{(p-1)} := \overline{z}_{l+p+1}$, and define

$$\overline{\mu}^{(p-1)} := \overline{X}^{(p-1)\top} B_1 \overline{y}_{-1}^{(p-1)} + \overline{Y}^{(p-1)\top} B_2 \overline{y}_{-1}^{(p-1)} + \overline{Z}^{(p-1)\top} B_3 \overline{z}_{-1}^{(p-1)}, \tag{30a}$$

$$\overline{\nu}^{(p-1)} := \overline{X}^{(p-1)\top} C_1 \overline{y}_{-1}^{(p-1)} + \overline{Y}^{(p-1)\top} C_2 \overline{y}_{-1}^{(p-1)} + \overline{Z}^{(p-1)\top} C_3 \overline{z}_{-1}^{(p-1)}. \tag{30b}$$

We let $n \in [S]$ denote the node with embedding $\overline{y}_{-1}^{(p-1)}$, i.e., $\overline{y}_{-1}^{(p-1)} = a_n$.

**Case 1:** $p \in \{2, \ldots, m\}$ **(at Stage 1).** In this case, we have $\overline{x}_{-1}^{(p-1)} = a_{c(n)}$, $\overline{y}_{-1}^{(p-1)} = a_n$, and $\overline{z}_{-1}^{(p-1)} = s_b$, where we define $c(n)$ as the child of node $n$ that's on the path. Same as the backward reasoning, without loss of generality, we can assume

$$(x_1^\top, y_1^\top)^\top = \left(a_{p(n)}^\top, a_n^\top\right)^\top.$$

By a similar argument as the backward reasoning, we can compute that for any $p \in \{2, \dots, m\}$, we have

$$\overline{\mu}_i^{(p-1)} = \begin{cases} (1+b_2)\alpha_1, & \text{if } i = 1, \\ b_2\alpha_1, & \text{if } i = 2, \dots, l, \\ (-a+b_2)\alpha_1, & \text{if } i = l+1, \\ (-a-b_2)\alpha_1, & \text{if } i = l+2, \\ -b_2\alpha_1, & \text{if } i = l+3, \dots, l+p, \\ (1-b_2)\alpha_1, & \text{if } i = l+p+1, \end{cases} \quad \text{and} \quad \overline{\nu}_i^{(p-1)} = \begin{cases} (1-c_2)\alpha_2, & \text{if } i = 1, \\ -c_2\alpha_2, & \text{if } i = 2, \dots, l, \\ (1-c_2)\alpha_2, & \text{if } i = l+1, \\ (1+c_2)\alpha_2, & \text{if } i = l+2, \\ c_2\alpha_2, & \text{if } i = l+3, \dots, l+p, \\ (1+c_2)\alpha_2, & \text{if } i = l+p+1, \end{cases}$$

$$(31)$$

which suggests

$$\begin{pmatrix} \overline{Y}^{(p-1)} \\ \overline{X}^{(p-1)} \end{pmatrix} \cdot \mathsf{softmax}\left(\overline{\mu}^{(p-1)}\right) \to \begin{pmatrix} a_n \\ a_{p(n)} \end{pmatrix}, \quad \overline{Z}^{(p-1)} \cdot \mathsf{softmax}\left(\overline{\nu}^{(p-1)}\right) \to s_b, \quad \text{as } \alpha_1, \alpha_2 \to +\infty. \tag{32}$$

Thus by (29) and the above relations, we have

$$\widehat{o}^{(p)} = \left( \begin{pmatrix} Y^{(p-1)} \\ X^{(p-1)} \end{pmatrix} \cdot \mathsf{softmax}\left(\mu^{(p-1)}\right) \\ Z^{(p-1)} \cdot \mathsf{softmax}\left(\nu^{(p-1)}\right) \right) \to \begin{pmatrix} a_n \\ a_{p(n)} \\ s_b \end{pmatrix} \overset{(8)}{=} o^{(p)}, \quad \text{as } \alpha_1, \alpha_2 \to +\infty. \tag{33}$$

**Case 2:** $p = m + 1$ **(at the turning point).** In this case $n = n_r$, where $n_r$ is the root node, and we have $\overline{x}_{-1}^{(p-1)} = a_{c(n_r)}$, $\overline{y}_{-1}^{(p-1)} = a_{n_r}$, and $\overline{z}_{-1}^{(p-1)} = s_b$, where we define $c(n_r)$ as the child of node $n_r$ that's on the path. Then analogous to the above argument, we can compute that

$$\overline{\mu}_i^{(p-1)} = \begin{cases} b_2\alpha_1, & \text{if } i \in [l] \\ (-a+1+b_2)\alpha_1, & \text{if } i = l+1, \\ (-a-b_2)\alpha_1, & \text{if } i = l+2, \\ -b_2\alpha_1, & \text{if } i = l+3, \dots, l+p, \\ (1-b_2)\alpha_1, & \text{if } i = l+p+1, \end{cases} \quad \text{and} \quad \overline{\nu}_i^{(p-1)} = \begin{cases} -c_2\alpha_2, & \text{if } i \in [l] \\ (2-c_2)\alpha_2, & \text{if } i = l+1, \\ (1+c_2)\alpha_2, & \text{if } i = l+2, \\ c_2\alpha_2, & \text{if } i = l+3, \dots, l+p, \\ (1+c_2)\alpha_2, & \text{if } i = l+p+1. \end{cases}$$

$$(34)$$

Recall that we require $a \in (0, 1]$, $b_2 < a/2$ and $c_2 < 1/2$, which guarantees that

$$\overline{\mu}_{l+p+1} > \overline{\mu}_i \; \forall i \neq l+p+1, \quad \text{and} \quad \overline{\nu}_{l+1} > \overline{\nu}_i, \; \forall i \neq l+1, \tag{35}$$

and thus

$$\begin{pmatrix} \overline{Y}^{(p-1)} \\ \overline{X}^{(p-1)} \end{pmatrix} \cdot \mathsf{softmax}\left(\overline{\mu}^{(p-1)}\right) \to \begin{pmatrix} a_{n_r} \\ a_{c(n_r)} \end{pmatrix}, \quad \overline{Z}^{(p-1)} \cdot \mathsf{softmax}\left(\overline{\nu}^{(p-1)}\right) \to s_f, \quad \text{as } \alpha_1, \alpha_2 \to +\infty. \tag{36}$$

Thus by (29) and the above relations, we have

$$\widehat{o}^{(p)} = \left( \begin{pmatrix} Y^{(p-1)} \\ X^{(p-1)} \end{pmatrix} \cdot \mathsf{softmax}\left(\mu^{(p-1)}\right) \\ Z^{(p-1)} \cdot \mathsf{softmax}\left(\nu^{(p-1)}\right) \right) \to \begin{pmatrix} a_{n_r} \\ a_{c(n_r)} \\ s_f \end{pmatrix} \overset{(8)}{=} o^{(p)}, \quad \text{as } \alpha_1, \alpha_2 \to +\infty. \tag{37}$$

**Case 3:** $p \in \{m+2, \dots, 2m\}$ **(at Stage 2).** In this case, by (8), we have $\overline{x}_{-1}^{(p-1)} = a_{p(n)}$, $\overline{y}_{-1}^{(p-1)} = a_n$, and $\overline{z}_{-1}^{(p-1)} = s_f$, and $\overline{x}_{l+3+2m-p}^\top = a_{c(n)}^\top$, $\overline{y}_{l+3+2m-p}^\top = a_n^\top$, $\overline{z}_{l+3+2m-p}^\top = s_b^\top$. Then by a similar argument as the backward reasoning, we can compute that for any $p \in \{m+2, \dots, 2m\}$, we have

$$\overline{\mu}_i^{(p-1)} = \begin{cases} (1-b_1)\alpha_1, & \text{if } i = 1, \\ -b_1\alpha_1, & \text{if } i = 2, \dots, l, \\ (-a-b_1)\alpha_1, & \text{if } i = l+1, \\ (-a+b_1)\alpha_1, & \text{if } i = l+2, \\ b_1\alpha_1, & \text{if } i = l+3, \dots, l+m+2, i \neq l+3+2m-p, \\ (1+b_1)\alpha_1, & \text{if } i = l+3+2m-p, \\ -b_1\alpha_1, & \text{if } i = l+4+2m-p, \dots, l+p, \\ (1-b_1)\alpha_1, & \text{if } i = l+p+1, \end{cases}$$

$$(38)$$

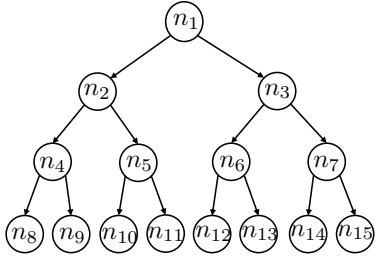

Figure 8: An illustration of the node ordering of a perfect binary tree with depth $m = 3$.

$$\overline{\nu}_i^{(p-1)} = \begin{cases} (1+c_1)\alpha_2, & \text{if } i = 1, \\ c_1\alpha_2, & \text{if } i = 2, \ldots, l, \\ (1+c_1)\alpha_2, & \text{if } i = l+1, \\ (1-c_1)\alpha_2, & \text{if } i = l+2, \\ -c_1\alpha_2, & \text{if } i = l+3, \ldots, l+m+2, i \neq l+3+2m-p, \\ (1-c_1)\alpha_2, & \text{if } i = l+3+2m-p, \\ -c_1\alpha_2, & \text{if } i = l+4+2m-p, \ldots, l+p, \\ (1+c_1)\alpha_2, & \text{if } i = l+p+1. \end{cases} \tag{39}$$

This indicates

$$\begin{pmatrix} \overline{Y}^{(p-1)} \\ \overline{X}^{(p-1)} \end{pmatrix} \cdot \mathsf{softmax}\left(\overline{\mu}^{(p-1)}\right) \to \begin{pmatrix} a_n \\ a_{c(n)} \end{pmatrix}, \quad \overline{Z}^{(p-1)} \cdot \mathsf{softmax}\left(\overline{\nu}^{(p-1)}\right) \to s_f, \quad \text{as } \alpha_1, \alpha_2 \to +\infty. \tag{40}$$

Thus by (29) and the above relations, we have

$$\widehat{o}^{(p)} = \left( \begin{pmatrix} Y^{(p-1)} \\ X^{(p-1)} \end{pmatrix} \cdot \mathsf{softmax}\left(\mu^{(p-1)}\right) \\ Z^{(p-1)} \cdot \mathsf{softmax}\left(\nu^{(p-1)}\right) \right) \to \begin{pmatrix} a_n \\ a_{c(n)} \\ s_f \end{pmatrix} \overset{(8)}{=} o^{(p)}, \quad \text{as } \alpha_1, \alpha_2 \to +\infty. \tag{41}$$

## C Convergence Analysis

We use a multi-phase analysis to track the training dynamics of both backward reasoning and forward reasoning tasks, where the details can be found below.

**Tree node ordering.** We let $N := 2^{m+1} - 1$ denote the number of nodes in the tree $\mathcal{T} \sim P_{\mathsf{train}}$, and let $n_1, n_2, \cdots, n_N$ denote the nodes in the tree $\mathcal{T}$. In particular, we let $n_1$ denote the root node, and for each $i \in [2^m - 1]$, we let $n_{2i}$ and $n_{2i+1}$ denote the left child and right child of node $n_i$, respectively. For $i \in [N]$, we let $s(n_i)$ to denote the sibling node of $n_i$. Without loss of generality, we assume the goal node is $n_{2^m}$, and the sibling of node $n_{2i}$ is $n_{2i+1}$ for $i \in [2^m - 1]$, and thus the path from the root node to the goal node is

$$n_1 \to n_2 \to n_4 \to \cdots \to n_{2^m}. \tag{42}$$

See Figure 8 for an illustration of our ordering.

Let $x^{(k)} := o^{(k)}(1 : d_1), y^{(k)} := o^{(k)}(d_1 + 1 : 2d_1), \widehat{x}^{(k)} := \widehat{o}^{(k)}(1 : d_1), \widehat{y}^{(k)} := \widehat{o}^{(k)}(d_1 + 1 : 2d_1)$. Under this ordering, we have

$$x_{-1}^{(k)} = a_{n_{2^{m-k}}}. \tag{43}$$

Throughout, let $n_{-1}^{(k)}$ denote the node index of the embedding $x_{-1}^{(k-1)}$.

### C.1 Proof of Theorem 3

We assume Assumption 3 and Assumption 4 hold throughout the proof.

**Loss simplification and gradient computation.** We first simplify the training loss and compute the gradients needed for the proof, where we drop the dependency with the parameter $\theta = \{B\}$ whenever it is clear from the context. Define

$$\Delta_x^{(k)} := \frac{1}{2}\mathbb{E}_{\mathcal{T}\sim P_{\text{train}}}\left\|x^{(k)} - \widehat{x}^{(k)}\right\|_2^2, \quad \Delta_y^{(k)} := \frac{1}{2}\mathbb{E}_{\mathcal{T}\sim P_{\text{train}}}\left\|y^{(k)} - \widehat{y}^{(k)}\right\|_2^2. \tag{44}$$

It follows that

$$\mathcal{L}_{\text{train}} := \sum_{k=1}^{m}\left(\Delta_x^{(k)} + \Delta_y^{(k)}\right). \tag{45}$$

Define $H_{0,j} = 0$ for all $j \in \{0, 1, \ldots, S\}$,

$$H := A^\top B A, \tag{46}$$

and $\delta x^{(k)}, \delta y^{(k)} \in \mathbb{R}^{(S+1)\times(S+1)}$ be as follows:

$$\delta x_{i,j}^{(k)} := \frac{\partial \Delta_x^{(k)}}{\partial H_{i,j}}, \quad \delta y_{i,j}^{(k)} := \frac{\partial \Delta_y^{(k)}}{\partial H_{i,j}}, \quad \forall i, j = 0, 1, \cdots, S. \tag{47}$$

Given a tree $\mathcal{T}$, we let $N_i^{(k)} = N_i^{(k)}(\mathcal{T})$ denote the number of times embedding $a_i$ appears in the columns of $Y^{(k)}$, for all $i = 0, 1, \cdots, S$. We define the *attention matrix* $\alpha^{(k)} = \alpha^{(k)}(\mathcal{T}) \in \mathbb{R}^{(S+1)\times(S+1)}$ as follows:

$$\forall k \in [m]: \quad \alpha_{i,j}^{(k)} = \frac{N_i^{(k-1)}\exp(a_i^\top B a_j)}{\sum_{l=0}^{S} N_l^{(k-1)}\exp(a_l^\top B a_j)}, \quad \forall i, j = 0, 1, \cdots, S. \tag{48}$$

For any $i, j \in [S]$, if $s(j)$ does not exist, we let $\alpha_{s(j),i}^{(k)} = 0$. Assumption 3 and Assummption 4 allow us to simplify the training loss, as in Lemma 2, whose proof is deferred to Appendix C.1.2.

**Lemma 2** (Loss simplification). *Under Assumption 3, Assumption 4, we have for all $k \in [m]$,*

$$\Delta_x^{(k)} = \frac{1}{2}\sum_{j\in[S]}\mathbb{E}\left[\mathbb{1}\left\{n_{-1}^{(k)} = j\right\}\left(\frac{1}{2}\sum_{l\in[S]\backslash\{n_1,j,s(j)\}}\left(\alpha_{l,j}^{(k)} + \alpha_{s(l),j}^{(k)}\right)^2 + \left(1 - \alpha_{j,j}^{(k)} - \alpha_{s(j),j}^{(k)}\right)^2 + \left(\alpha_{0,j}^{(k)}\right)^2\right)\right], \tag{49a}$$

$$\Delta_y^{(k)} = \frac{1}{2}\sum_{j\in[S]}\mathbb{E}\left[\mathbb{1}\{n_{-1}^{(k)} = j\}\left(\sum_{l\in[S]\backslash\{j\}}\left(\alpha_{l,j}^{(k)}\right)^2 + \left(1 - \alpha_{j,j}^{(k)}\right)^2\right)\right], \tag{49b}$$

*where $n_{-1}^{(k)}$ denotes the node index of the embedding $x_{-1}^{(k-1)}$.*

The following lemma computes the gradients $\delta x^{(k)}$ and $\delta y^{(k)}$, whose proof is also deferred to Appendix C.1.3.

**Lemma 3** (Gradient computation). *Under Assumption 3, Assumption 4, we have for all $k \in [m]$, $\delta x_{0,j}^{(k)} = 0$, $\delta y_{0,j}^{(k)} = 0$ for all $j = \{0, 1, \ldots, S\}$, and*

$$\forall j \in [S]: \quad \delta x_{j,j}^{(k)} = -\frac{1}{S}\mathbb{E}\left[\left(\alpha_{0,j}^{(k)}\right)^2 \alpha_{j,j}^{(k)}\Big| n_{-1}^{(k)} = j\right]$$

$$-\frac{S-3}{2S(S-1)}\mathbb{E}\left[\sum_{p\in[S]\backslash\{j,s(j),n_1\}}\left(\alpha_{p,j}^{(k)} + \alpha_{s(p),j}^{(k)}\right)^2 \alpha_{j,j}^{(k)}\Big| n_{-1}^{(k)} = j\right]$$

$$-\frac{1}{S}\mathbb{E}\left[\left(1 - \alpha_{j,j}^{(k)} - \alpha_{s(j),j}^{(k)}\right)^2 \alpha_{j,j}^{(k)}\Big| n_{-1}^{(k)} = j\right], \tag{50a}$$

$$\forall i \in [S]\backslash\{j\}: \quad \delta x_{i,j}^{(k)} = -\frac{1}{S}\mathbb{E}\left[\left(\alpha_{0,j}^{(k)}\right)^2 \alpha_{i,j}^{(k)}\Big| n_{-1}^{(k)} = j\right]$$

$$+\frac{1}{S}\mathbb{E}\left[\left(1 - \alpha_{j,j}^{(k)} - \alpha_{s(j),j}^{(k)}\right)\alpha_{i,j}^{(k)}\left(\alpha_{j,j}^{(k)} + \alpha_{s(j),j}^{(k)}\right)\Big| n_{-1}^{(k)} = j\right]$$

$$-\frac{S-3}{2S(S-1)}\mathbb{E}\left[\sum_{\substack{p\in[S]\\p\notin\{j,s(j),n_1\}}}\left(\alpha_{p,j}^{(k)}+\alpha_{s(p),j}^{(k)}\right)^2\alpha_{i,j}^{(k)}\Big|n_{-1}^{(k)}=j\right]$$

$$-\frac{1}{S(S-1)}\mathbb{E}\left[\left(1-\alpha_{j,j}^{(k)}-\alpha_{i,j}^{(k)}\right)\alpha_{i,j}^{(k)}\Big|n_{-1}^{(k)}=j,s(j)=i\right]$$

$$+\frac{S-3}{S(S-1)}\mathbb{E}\left[\left(\alpha_{i,j}^{(k)}+\alpha_{s(i),j}^{(k)}\right)\alpha_{i,j}^{(k)}\Big|n_{-1}^{(k)}=j,i\notin\{s(j),n_1\}\right],\quad\text{(50b)}$$

*and*

$$\forall j\in[S]:\quad \delta y_{j,j}^{(k)}=-\frac{1}{S}\mathbb{E}\left[\alpha_{j,j}^{(k)}\left(\sum_{p\in[S]\setminus\{j\}}\left(\alpha_{p,j}^{(k)}\right)^2+\left(1-\alpha_{j,j}^{(k)}\right)^2\right)\Big|n_{-1}^{(k)}=j\right],\quad\text{(51a)}$$

$$\forall i\in[S]\setminus\{j\}:\quad \delta y_{i,j}^{(k)}=-\frac{1}{S}\mathbb{E}\left[\alpha_{i,j}^{(k)}\left(\sum_{p\in[S]\setminus\{j\}}\left(\alpha_{p,j}^{(k)}\right)^2-\left(1-\alpha_{j,j}^{(k)}\right)\alpha_{j,j}^{(k)}-\alpha_{i,j}^{(k)}\right)\Big|n_{-1}^{(k)}=j\right].$$
$$\text{(51b)}$$

Here, $n_{-1}^{(k)}$ denotes the node index of the embedding $x_{-1}^{(k-1)}$.

We'll repeatedly use the gradient inequality given in the following lemma, whose proof is provided in Appendix C.1.4.

**Lemma 4** (Key gradient inequality). *Under Assumption 3, Assumption 4, we have for all $k\in[m]$,*

$$\delta x_{i,j}^{(k)}+\delta y_{i,j}^{(k)}\geqslant\frac{1}{S}\mathbb{E}\left[\left(1-\alpha_{j,j}^{(k)}-\alpha_{s(j),j}^{(k)}\right)\alpha_{i,j}^{(k)}\left(\frac{N-2}{N-1}\alpha_{j,j}^{(k)}-\max_{p\in[S]\setminus\{j\}}\alpha_{p,j}^{(k)}-\frac{1}{N-1}\alpha_{0,j}^{(k)}\right)\Big|n_{-1}^{(k)}=j\right]$$

$$+\frac{1}{S}\mathbb{E}\left[\left(1-\alpha_{j,j}^{(k)}\right)\alpha_{i,j}^{(k)}\left(\alpha_{j,j}^{(k)}-\max_{\substack{0\leqslant p\leqslant S\\p\neq j}}\alpha_{p,j}^{(k)}\right)\Big|n_{-1}^{(k)}=j\right],\quad\text{(52)}$$

*where $n_{-1}^{(k)}$ denotes the node index of the embedding $x_{-1}^{(k-1)}$.*

### C.1.1 Proof outline

Letting $H^{(t)}:=A^\top B^{(t)}A$, by the update rule (12), we have

$$H_{i,j}^{(t+1)}=H_{i,j}^{(t)}-\eta\sum_{k=1}^m\left(\delta x_{i,j}^{(t,k)}+\delta y_{i,j}^{(t,k)}\right),\quad\forall i,j=0,1,\cdots,S,\quad\text{(53)}$$

where $x^{(t,k)},y^{(t,k)},\widehat{x}^{(t,k)},\widehat{y}^{(t,k)}$ denote the value of $x^{(k)},y^{(k)},\widehat{x}^{(k)},\widehat{y}^{(k)}$, respectively. Further, let $\alpha_{i,j}^{(t,k)}$ be the value of $\alpha_{i,j}^{(k)}$ at the $t$-th iteration when replacing $B$ by $B^{(t)}$, which can be written in terms of $H_{i,j}^{(t)}$ as

$$\forall k\in[m]:\quad \alpha_{i,j}^{(t,k)}=\frac{N_i^{(k-1)}\exp(H_{i,j}^{(t)})}{\sum_{l=0}^S N_l^{(k-1)}\exp(H_{l,j}^{(t)})}$$

satisfying $\sum_i\alpha_{i,j}^{(t,k)}=1$. Note that by symmetry, we have for any $j\in[S]$, $H_{i,j}^{(t)}$ are equal for all $i\in[S]\setminus\{j\}$, and $H_{j,j}^{(t)}$ are equal for all $j\in[S]$.

In what follows, we divide the training dynamics of $H^{(t)}$ into two phases to prove the convergence of the backward reasoning task. We show there exists $T_1=\widetilde{\mathcal{O}}\left(\frac{SN}{m\eta}\right)$ and $T_2=T_1+\widetilde{\mathcal{O}}\left(\frac{mS}{\eta\epsilon}\right)$ such that

- At Phase I ($t=\{0,1,\cdots,T_1\}$): the diagonal elements $H_{j,j}^{(t)}$ are strictly increasing for all $j\in[S]$, and the off-diagonal elements

$$\left|H_{i,j}^{(t)}\right|\lesssim\frac{\log N}{N}$$

for all $i\in[S]\setminus\{j\}$.

- At Phase II ($t = \{T_1 + 1, \cdots, T_2\}$): the diagonal elements $H_{j,j}^{(t)}$ keep strictly increasing, and the off-diagonal elements $H_{i,j}^{(t)}$ are strictly decreasing, and satisfy

$$-\frac{H_{j,j}^{(t)}}{S} \lesssim H_{i,j}^{(t)} \lesssim \frac{\log N}{N}.$$

- After Phase II ($t \geq T_2 + 1$): $H_{j,j}^{(t)}$ (resp. $H_{i,j}^{(t)}$) keeps increasing (resp. decreasing), and

$$H_{j,j}^{(t)} \gtrsim \log\left(\frac{m}{\epsilon}\right),$$

and the training loss $\mathcal{L}_{\text{train}}(\theta^{(t)}) \leq \epsilon$.

**Training dynamics of Phase I.** Let $T_1 \in \mathbb{N} \cup \{+\infty\}$ be the number of iterations of Phase I, defined as

$$T_1 := \max\left\{t \in \mathbb{N} : \widehat{\alpha}^{(t)} \leq \frac{1}{2}\right\}, \tag{54}$$

where

$$\widehat{\alpha}^{(t)} := \frac{\exp(H_{1,1}^{(t)})}{(N-1)\exp(H_{2,1}^{(t)}) + \exp(H_{1,1}^{(t)}) + 1}, \tag{55}$$

which satisfies

$$\alpha_{j,j}^{(t,1)} = \widehat{\alpha}^{(t)}, \quad \forall j \in [S]. \tag{56}$$

In the following, we show by induction the following claim holds for all $t \in [T_1 - 1]$:

Induction Hypothesis I:   $\forall i \in \{0\} \cup [S], j \in [S], i \neq j :$   $H_{j,j}^{(t)} \geq 0, \quad H_{i,j}^{(t)} \leq \frac{9}{N-1} H_{j,j}^{(t)}.$
$$\tag{57}$$

We start with the base case when $t = 0$. Since $H_{i,j} = 0$ for all $i, j$, (57) holds true trivially. We next assume (57) holds up to time $t$. We'll make use of the following lemma, whose proof is postponed to Appendix C.1.5.

**Lemma 5** (Gradient proportions at Phase I). *Under Assumption 3, Assumption 4, Induction Hypothesis I (57), we have for all $k \in [m]$,*

$$-\delta x_{i,j}^{(t,k)} - \delta y_{i,j}^{(t,k)} \leq \frac{9}{N-1}\left(-\delta x_{j,j}^{(t,k)} - \delta y_{j,j}^{(t,k)}\right). \tag{58}$$

We now establish (57) holds for $t+1$. From the expression of $\delta x_{j,j}^{(t,k)}$ and $\delta y_{j,j}^{(t,k)}$ (c.f. (50a) and (51a)), we can see that $H_{j,j}^{(t,k)}$ is increasing, and thus $H_{j,j}^{(t+1,k)} \geq 0$. Moreover, By Lemma 5 and (53), we have

$$H_{i,j}^{(t+1)} = H_{i,j}^{(t)} - \eta\left(\sum_{k=1}^{m}\left(\delta x_{i,j}^{(t,k)} + \delta y_{i,j}^{(t,k)}\right)\right)$$

$$\leq \frac{9}{N-1}H_{j,j}^{(t)} - \frac{9}{N-1}\eta\left(\sum_{k=1}^{m}\left(\delta x_{j,j}^{(t,k)} + \delta y_{j,j}^{(t,k)}\right)\right) = \frac{9}{N-1}H_{j,j}^{(t+1)}, \tag{59}$$

where the first inequality follows from the induction hypothesis (57). Therefore, (57) holds for $t+1$, and the induction is complete.

The following lemma guarantees that Phase I terminates within $T_1 = \widetilde{\mathcal{O}}\left(\frac{SN}{m\eta}\right)$ iterations, see Appendix C.1.6 for the proof.

**Lemma 6** (Upper bound of $T_1$). *Under Assumption 3, Assumption 4, we have*

$$T_1 \leqslant \frac{6S(N+m)}{m\eta} \log\left(\frac{N+1}{2}\right). \tag{60}$$

*In addition, we have*

$$H_{j,j}^{(t)} \leqslant 3\log\left(\frac{N+1}{2}\right). \tag{61}$$

The above lemma, together with (57), implies that for all $i \in [S]\backslash\{j\}$,

$$\forall t \in \{0, 1, \ldots, T_1\}: \quad H_{i,j}^{(t)} \lesssim \frac{\log(N)}{N}. \tag{62}$$

Finally, we give a lower bound on $H_{i,j}^{(t)}$ in the following lemma, whose proof is postponed to Appendix C.1.7.

**Lemma 7** (Lower bound on $H_{i,j}^{(t)}$). *Under Assumption 3, Assumption 4, we have*

$$\forall t \in \{0, 1, \ldots, T_1\}: \quad H_{i,j}^{(t)} \geqslant -\frac{19}{S-1} H_{j,j}^{(t)}. \tag{63}$$

Combining (61) and Lemma 7, we have

$$\forall t \in \{0, 1, \ldots, T_1\}: \quad -H_{i,j}^{(t)} \lesssim \frac{\log(N)}{S}. \tag{64}$$

**Training dynamics of Phase II.** Let $T_2 \in \mathbb{N} \cup \{+\infty\}$ be the number of iterations of Phase II, defined as

$$T_2 := \max\left\{t \in \mathbb{N}: \breve{\alpha}^{(t)} \leqslant 1 - \sqrt{\frac{\epsilon}{2m}}\right\}, \tag{65}$$

where

$$\breve{\alpha}^{(t)} := \frac{\exp(H_{1,1}^{(t)})}{(N+m-2)\exp(H_{2,1}^{(t)}) + \exp(H_{1,1}^{(t)}) + 1}. \tag{66}$$

Note that by definition (48) and symmetry, we have

$$\alpha_{j,j}^{(t,k)} = \frac{\exp(H_{1,1}^{(t)})}{(N+k-2)\exp(H_{2,1}^{(t)}) + \exp(H_{1,1}^{(t)}) + 1} \geqslant \breve{\alpha}^{(t)} \tag{67}$$

for all $j \in [S]$ and $k \in [m]$. Since $\epsilon \in (0, 1]$, we have $T_2 > T_1$.

We'll show by induction the following claim holds for all $t \in \{T_1 + 1, \ldots, T_2\}$:

**Induction Hypothesis II:** $\quad \forall i \in \{0\} \cup [S], j \in [S], i \neq j: \quad -\sum_{k=1}^{m}\left(\delta x_{j,j}^{(t,k)} + \delta y_{j,j}^{(t,k)}\right) \geqslant \frac{\epsilon}{4mS},$

$$\tag{68a}$$

$$-\sum_{k=1}^{m}\left(\delta x_{i,j}^{(t,k)} + \delta y_{i,j}^{(t,k)}\right) < 0.$$

$$\tag{68b}$$

We start with the base case when $t = T_1 + 1$. By the definition of $T_1$, we have $\alpha_{j,j}^{(T_1+1,1)} \geqslant \frac{1}{2}$ for all $j \in [S]$. For any $j \in [S]$, when $n_{-1}^{(k)} = j$, we have

$$\alpha_{j,j}^{(T_1+1,1)} = \widehat{\alpha}^{(T_1+1)} \geqslant \frac{1}{2} \quad \text{which implies} \quad \forall i \in [S]\backslash\{j\}: \quad \alpha_{i,j}^{(T_1+1,k)} \leqslant \frac{1}{N-1}$$

for all $k \in [m]$, where we use the fact that $N_i^{(k)} \leqslant 2$ for any $i$, and

$$\alpha_{j,j}^{(T_1+1,k)} \geqslant \frac{N-1}{N+m-2} \cdot \frac{1}{2} \geqslant \frac{1}{3}, \tag{69}$$

where we use the fact that

$$\forall t \in \mathbb{N}: \quad \frac{\alpha_{j,j}^{(t,k)}}{\alpha_{j,j}^{(t,1)}} \geqslant \frac{(N-1)\exp(H_{2,1}^{(t)}) + \exp(H_{1,1}^{(t)}) + 1}{(N+m-2)\exp(H_{2,1}^{(t)}) + \exp(H_{1,1}^{(t)}) + 1} \geqslant \frac{N-1}{N+m-2}. \tag{70}$$

Since $H_{j,j}^{(t)} \geqslant 0 = H_{0,j}^{(t)}$, we also have

$$\frac{\alpha_{0,j}^{(T_1+1,k)}}{\alpha_{j,j}^{(T_1+1,k)}} = \frac{1}{\exp(H_{j,j}^{(T_1+1)})} \leqslant 1, \quad \forall k \in [m],$$

Thus we have

$$\frac{N-2}{N-1}\alpha_{j,j}^{(T_1+1,k)} - \max_{p \in [S]\setminus\{j\}} \alpha_{p,j}^{(T_1+1,k)} - \frac{1}{N-1}\alpha_{0,j}^{(T_1+1,k)} \geqslant \frac{N-2}{N-1} \cdot \frac{1}{3} - \frac{1}{N-1} - \frac{1}{N-1} \cdot \frac{1}{2} > 0,$$

$$\alpha_{j,j}^{(T_1+1,k)} - \max_{\substack{0 \leqslant p \leqslant S \\ p \neq j}} \alpha_{p,j}^{(T_1+1,k)} \geqslant 0. \tag{71}$$

By Lemma 4 and the above relations, we have (68b) holds for $t = T_1 + 1$.

Moreover, since $\delta x_{j,j}^{(T_1+1,k)} \leqslant 0$, we have

$$\sum_{k=1}^{m} \left(-\delta x_{j,j}^{(T_1+1,k)} - \delta y_{j,j}^{(T_1+1,k)}\right) \geqslant -\delta y_{j,j}^{(T_1+1,k)} \overset{(51a)}{\geqslant} \frac{1}{S}\mathbb{E}\left[\alpha_{j,j}^{(T_1+1,m)}\left(1 - \alpha_{j,j}^{(T_1+1,m)}\right)^2 \bigg| n_{-1}^{(k)} = j\right] \overset{(65)}{\geqslant} \frac{\epsilon}{4mS}. \tag{72}$$

Thus we have (68a) holds for $t = T_1 + 1$.

We now continue with the induction, by assuming (68) holds for all iterations up to $t \geqslant T_1 + 1$ and all $j \in [S], i \in \{0, 1, \ldots, S\}, i \neq j$, and aim to establish it continues to hold at iteration $t + 1$. By the induction hypothesis, we have $H_{i,j}^{(t)}$ ($i \neq j$) decreases after iteration $T_1 + 1$ while $H_{j,j}^{(t)}$ keeps increasing. Thus we have $\alpha_{j,j}^{(t,k)}$ monotonically increases after iteration $T_1 + 1$, and $\alpha_{i,j}^{(t,k)}$ ($i \neq j$) monotonically decreases after iteration $T_1 + 1$. Therefore, by (71), we have

$$\frac{N-2}{N-1}\alpha_{j,j}^{(t+1,k)} - \max_{p \in [S]\setminus\{j\}} \alpha_{p,j}^{(t+1,k)} - \frac{1}{N-1}\alpha_{0,j}^{(t+1,k)}$$

$$\geqslant \frac{N-2}{N-1}\alpha_{j,j}^{(T_1+1,k)} - \max_{p \in [S]\setminus\{j\}} \alpha_{p,j}^{(T_1+1,k)} - \frac{1}{N-1}\alpha_{0,j}^{(T_1+1,k)} > 0,$$

$$\alpha_{j,j}^{(t+1,k)} - \max_{\substack{0 \leqslant p \leqslant S \\ p \neq j}} \alpha_{p,j}^{(t+1,k)} \geqslant \alpha_{j,j}^{(T_1+1,k)} - \max_{\substack{0 \leqslant p \leqslant S \\ p \neq j}} \alpha_{p,j}^{(T_1+1,k)} \geqslant 0$$

for all $k \in [m]$. This suggests that (68b) hold at iteration $t + 1$.

Moreover, similar to (72), we have

$$\sum_{k=1}^{m} \left(-\delta x_{j,j}^{(t+1,k)} - \delta y_{j,j}^{(t+1,k)}\right) \geqslant -\delta y_{j,j}^{(t+1,k)} \overset{(51a)}{\geqslant} \frac{1}{S}\mathbb{E}\left[\alpha_{j,j}^{(t+1,m)}\left(1 - \alpha_{j,j}^{(t+1,m)}\right)^2 \bigg| n_{-1} = j\right] \overset{(65)}{\geqslant} \frac{\epsilon}{4mS}. \tag{73}$$

Thus we have (68a) holds at iteration $t + 1$. The induction is complete.

Similar as Lemma 7, we give a lower bound on $H_{i,j}^{(t)}$ after iteration $T_1$, see Appendix C.1.8 for its proof.

**Lemma 8.** *For any $i, j \in [S]$, $i \neq j$, we have*

$$\forall t \geqslant T_1 + 1: \quad -\left(H_{i,j}^{(t)} - H_{i,j}^{(T_1+1)}\right) \leqslant \frac{4}{S-1}\left(H_{j,j}^{(t)} - H_{j,j}^{(T_1+1)}\right).$$

Combining (64) and Lemma 8, we have

$$\forall t \geqslant T_1 + 1: \quad -H_{i,j}^{(t)} \leqslant \frac{4}{S-1} H_{j,j}^{(t)} + C \cdot \frac{\log(N)}{S}, \tag{74}$$

where $C$ is an absolute constant.

The next lemma states that the duration of Phase II, $T_2 - T_1$, is bounded by $\tilde{\mathcal{O}}\left(\frac{mS}{\eta\epsilon}\right)$. The proof is deferred to Appendix C.1.9.

**Lemma 9.** *We have*

$$T_2 - T_1 \leqslant \frac{12mS}{\eta\epsilon} \log\left((N + m - 1)\left(\sqrt{\frac{2m}{\epsilon}} - 1\right)\right). \tag{75}$$

**After Phase II.** By (50a) and (51a) we know that $H_{j,j}^{(t)}$ is strictly increasing after iteration $T_2$. Also, we can easily see that (68b) holds after iteration $T_2$, which guarantees $H_{i,j}^{(t)}$ $(i \in [S]\backslash\{j\})$ is strictly decreasing after iteration $T_2$. The following lemma guarantees that the training loss is less than $\epsilon$ after iteration $T_2$, and gives a lower bound on $H_{j,j}^{(t)}$ after iteration $T_2$; see Appendix C.1.10 for its proof.

**Lemma 10.** *After $t > T_2$, the training loss $\mathcal{L}_{train}(\theta^{(t)}) \leqslant \epsilon$. Moreover, for any $j \in [S]$, we have*

$$\forall t > T_2: \quad H_{j,j}^{(t)} \geqslant \log\left(\sqrt{\frac{2m}{\epsilon}} - 1\right). \tag{76}$$

### C.1.2 Proof of Lemma 2

For notation simplicity, we let $n_{-1} = n_{-1}^{(k)}$. For any $k \in [m]$, let (note that $n_{2^m}$ is the goal node)

$$w_i^{(k)} := \begin{cases} \alpha_{n_{2i},n_{-1}}^{(k)} + \alpha_{n_{2i+1},n_{-1}}^{(k)}, & \text{if } i \in [2^m - 1], \\ \alpha_{0,n_{-1}}^{(k)}, & \text{if } i = 2^m. \end{cases} \tag{77}$$

We then have

$$\begin{aligned}
\Delta_x^{(k)} &= \frac{1}{2}\mathbb{E}\left\|\sum_{i=1}^{2^m} w_i^{(k)} a_{n_i} - a_{n_{2^{m-k-1}}}\right\|_2^2 \\
&= \frac{1}{2}\mathbb{E}\left[\sum_{\substack{i=1, \\ i \neq 2^{m-k-1}}}^{2^m} (w_i^{(k)})^2 + (1 - w_{2^{m-k-1}}^{(k)})^2\right] \\
&= \frac{1}{2}\mathbb{E}\left[\sum_{\substack{i=1, \\ i \neq 2^{m-k-1}}}^{2^m-1} (\alpha_{n_{2i},n_{-1}}^{(k)} + \alpha_{n_{2i+1},n_{-1}}^{(k)})^2 + (1 - \alpha_{n_{-1},n_{-1}}^{(k)} - \alpha_{n_{s(n_{-1})},n_{-1}}^{(k)})^2 + (\alpha_{0,n_{-1}}^{(k)})^2\right] \\
&= \frac{1}{2}\sum_{j\in[S]}\mathbb{E}\left[\mathbb{1}\{n_{-1} = j\}\left(\frac{1}{2}\sum_{\substack{l\in[S] \\ l\notin\{n_1,j,s(j)\}}} (\alpha_{l,j}^{(k)} + \alpha_{s(l),j}^{(k)})^2 + (1 - \alpha_{j,j}^{(k)} - \alpha_{s(j),j}^{(k)})^2 + (\alpha_{0,j}^{(k)})^2\right)\right],
\end{aligned}$$

which gives (49a), where the second equality follows from our assumption that $\{a_i\}_{i\in[S]}$ are orthonormal, and in the third equality we use the fact that $n_{2^{m-k-1}}$ is the parent of $n_{-1}$. Similarly,

$$\begin{aligned}
\Delta_y^{(k)} &= \frac{1}{2}\mathbb{E}\left\|\sum_{i=1}^{N} \alpha_{n_i,n_{-1}}^{(k)} a_{n_i} - a_{n_{-1}}\right\|_2^2 \\
&= \frac{1}{2}\mathbb{E}\left[\sum_{\substack{i=1, \\ i \neq 2^{m-k}}}^{N} (\alpha_{n_i,n_{-1}}^{(k)})^2 + (1 - \alpha_{n_{-1},n_{-1}}^{(k)})^2\right] \\
&= \frac{1}{2}\sum_{j\in[S]}\mathbb{E}\left[\mathbb{1}\{n_{-1} = j\}\left(\sum_{\substack{l\in[S], \\ l\neq j}} (\alpha_{l,j}^{(k)})^2 + (1 - \alpha_{j,j}^{(k)})^2\right)\right].
\end{aligned}$$

### C.1.3 Proof of Lemma 3

For notation simplicity, we let $n_{-1} = n_{-1}^{(k)}$.

**Computation of $\delta x^{(k)}$.** The fact that $\delta x_{0,j}^{(k)} = 0$ follows from our assumption that $a_0 = 0$. Below we consider the case when $j \in [S]$. By the chain rule, we have

$$\delta x_{i,j}^{(k)} := \frac{\partial \Delta_x^{(k)}}{\partial H_{i,j}} = \mathbb{E}\left[\sum_{p=0}^{S} \frac{\partial \Delta_x^{(k)}}{\partial \alpha_{p,j}^{(k)}} \frac{\partial \alpha_{p,j}^{(k)}}{\partial H_{i,j}}\right]. \tag{78}$$

We compute the terms in the summand separately.

- When $p = j$, by (49a), we have

$$\frac{\partial \Delta_x^{(k)}}{\partial \alpha_{p,j}^{(k)}} = -\frac{1}{S}\mathbb{E}\left[1 - \alpha_{j,j}^{(k)} - \alpha_{s(j),j}^{(k)}\Big|n_{-1} = j\right]. \tag{79}$$

- When $p \in [S]\setminus\{j\}$, by (49a), we have

$\Delta_x^{(k)}$

$$= \frac{1}{2}\mathbb{E}\left[\mathbb{1}\{n_{-1} = j, s(j) = p\}\left(\frac{1}{2}\sum_{\substack{l\in[S]\\l\notin\{n_1,j,i\}}} (\alpha_{l,j}^{(k)} + \alpha_{s(l),j}^{(k)})^2 + (1 - \alpha_{j,j}^{(k)} - \alpha_{p,j}^{(k)})^2 + (\alpha_{0,j}^{(k)})^2\right)\right]$$

$$+ \frac{1}{2}\mathbb{E}\left[\mathbb{1}\{n_{-1} = j, s(j) \neq p, n_1 \neq p\}\left(\frac{1}{2}\sum_{\substack{l\in[S]\\l\notin\{n_1,j,s(j)\}}} (\alpha_{l,j}^{(k)} + \alpha_{s(l),j}^{(k)})^2 + (1 - \alpha_{j,j}^{(k)} - \alpha_{s(j),j}^{(k)})^2 + (\alpha_{0,j}^{(k)})^2\right)\right]$$

$$+ \frac{1}{2}\mathbb{E}\left[\mathbb{1}\{n_{-1} = j, n_1 = p\}\left(\frac{1}{2}\sum_{\substack{l\in[S]\\l\notin\{p,j,s(j)\}}} (\alpha_{l,j}^{(k)} + \alpha_{s(l),j}^{(k)})^2 + (1 - \alpha_{j,j}^{(k)} - \alpha_{s(j),j}^{(k)})^2 + (\alpha_{0,j}^{(k)})^2\right)\right],$$

from which we can compute that when $p \in [S]\setminus\{j\}$,

$$\frac{\partial \Delta_x^{(k)}}{\partial \alpha_{p,j}^{(k)}} = -\frac{1}{S(S-1)}\mathbb{E}\left[1 - \alpha_{j,j}^{(k)} - \alpha_{p,j}^{(k)}\Big|n_{-1} = j, s(j) = p\right]$$

$$+ \frac{S-3}{S(S-1)}\mathbb{E}\left[\alpha_{p,j}^{(k)} + \alpha_{s(p),j}^{(k)}\Big|n_{-1} = j, s(j) \neq p, n_1 \neq p\right]. \tag{80}$$

- When $p = 0$, we have

$$\frac{\partial \Delta_x^{(k)}}{\partial \alpha_{0,j}^{(k)}} = \frac{1}{S}\mathbb{E}\left[\alpha_{0,j}^{(k)}\Big|n_{-1} = j\right]. \tag{81}$$

In addition, by (48), we have for all $i, j \in \{0, 1, \ldots, S\}$,

$$\frac{\partial \alpha_{i,j}^{(k)}}{\partial B} = \alpha_{i,j}^{(k)} a_i a_j^\top - \alpha_{i,j}^{(k)} \sum_{l=0}^{S} \alpha_{l,j}^{(k)} a_l a_j^\top, \tag{82}$$

which gives

$$\frac{\partial \alpha_{i,j}^{(k)}}{\partial H_{p,q}} = \begin{cases} \alpha_{i,j}^{(k)}\left(1 - \alpha_{i,j}^{(k)}\right), & \text{when } p = i, q = j, \\ -\alpha_{i,j}^{(k)}\alpha_{p,j}^{(k)}, & \text{when } p \in [S]\setminus\{i\}, q = j, \\ 0, & \text{otherwise.} \end{cases} \tag{83}$$

Plugging (83) into (78), we have

$$\delta x_{i,j}^{(k)} = \mathbb{E}\left[\sum_{\substack{p=0\\p\neq i}}^{S} \frac{\partial \Delta_x^{(k)}}{\partial \alpha_{p,j}^{(k)}}\left(-\alpha_{i,j}^{(k)}\alpha_{p,j}^{(k)}\right) + \frac{\partial \Delta_x^{(k)}}{\partial \alpha_{i,j}^{(k)}}\alpha_{i,j}^{(k)}\left(1 - \alpha_{i,j}^{(k)}\right)\right]. \tag{84}$$

By (84), (79), (80) and (81), we can compute that

$$\delta x_{j,j}^{(k)} = -\frac{1}{S}\mathbb{E}\left[\left(\alpha_{0,j}^{(k)}\right)^2 \alpha_{j,j}^{(k)}\Big| n_{-1} = j\right]$$

$$+ \frac{1}{S(S-1)} \sum_{\substack{p\in[S]\\p\neq j}} \mathbb{E}\left[\left(1 - \alpha_{j,j}^{(k)} - \alpha_{p,j}^{(k)}\right)\alpha_{j,j}^{(k)}\alpha_{p,j}^{(k)}\Big| n_{-1} = j, s(j) = p\right]$$

$$- \frac{S-3}{S(S-1)} \sum_{\substack{p\in[S]\\p\neq j}} \mathbb{E}\left[\left(\alpha_{p,j}^{(k)} + \alpha_{s(p),j}^{(k)}\right)\alpha_{j,j}^{(k)}\alpha_{p,j}^{(k)}\Big| n_{-1} = j, s(j) \neq p, n_1 \neq p\right]$$

$$- \frac{1}{S}\mathbb{E}\left[\left(1 - \alpha_{j,j}^{(k)} - \alpha_{s(j),j}^{(k)}\right)\alpha_{j,j}^{(k)}\left(1 - \alpha_{j,j}^{(k)}\right)\Big| n_{-1} = j\right]. \tag{85}$$

Note that the second term can be simplified as follows:

$$\frac{1}{S(S-1)} \sum_{\substack{p\in[S]\\p\neq j}} \mathbb{E}\left[\left(1 - \alpha_{j,j}^{(k)} - \alpha_{p,j}^{(k)}\right)\alpha_{j,j}^{(k)}\alpha_{p,j}^{(k)}\Big| n_{-1} = j, s(j) = p\right]$$

$$= \frac{1}{S}\mathbb{E}\left[\left(1 - \alpha_{j,j}^{(k)} - \alpha_{s(j),j}^{(k)}\right)\alpha_{j,j}^{(k)}\alpha_{s(j),j}^{(k)}\Big| n_{-1} = j\right], \tag{86}$$

and we can rewrite the third term in (85) as

$$-\frac{S-3}{S(S-1)} \sum_{\substack{p\in[S]\\p\neq j}} \mathbb{E}\left[\left(\alpha_{p,j}^{(k)} + \alpha_{s(p),j}^{(k)}\right)\alpha_{j,j}^{(k)}\alpha_{p,j}^{(k)}\Big| n_{-1} = j, s(j) \neq p, n_1 \neq p\right]$$

$$= -\frac{S-3}{S(S-1)} \sum_{\substack{p\in[S]\\p\neq j}} \mathbb{E}\left[\left(\alpha_{p,j}^{(k)} + \alpha_{s(p),j}^{(k)}\right)\alpha_{j,j}^{(k)}\alpha_{p,j}^{(k)}\Big| n_{-1} = j, s(j) \notin \{p, s(p)\}, n_1 \notin \{p, s(p)\}\right]$$

$$= -\frac{S-3}{2S(S-1)} \sum_{\substack{p\in[S]\\p\neq j}} \mathbb{E}\left[\left(\alpha_{p,j}^{(k)} + \alpha_{s(p),j}^{(k)}\right)\alpha_{j,j}^{(k)}\alpha_{p,j}^{(k)}\Big| n_{-1} = j, s(j) \notin \{p, s(p)\}, n_1 \notin \{p, s(p)\}\right]$$

$$- \frac{S-3}{2S(S-1)} \sum_{\substack{p\in[S]\\p\neq j}} \mathbb{E}\left[\left(\alpha_{p,j}^{(k)} + \alpha_{s(p),j}^{(k)}\right)\alpha_{j,j}^{(k)}\alpha_{s(p),j}^{(k)}\Big| n_{-1} = j, s(j) \notin \{p, s(p)\}, n_1 \notin \{p, s(p)\}\right]$$

$$= -\frac{S-3}{2S(S-1)} \sum_{\substack{p\in[S]\\p\neq j}} \mathbb{E}\left[\left(\alpha_{p,j}^{(k)} + \alpha_{s(p),j}^{(k)}\right)^2 \alpha_{j,j}^{(k)}\Big| n_{-1} = j, s(j) \notin \{p, s(p)\}, n_1 \notin \{p, s(p)\}\right]$$

$$= -\frac{S-3}{2S(S-1)}\mathbb{E}\left[\sum_{\substack{p\in[S]\\p\neq j}} \left(\alpha_{p,j}^{(k)} + \alpha_{s(p),j}^{(k)}\right)^2 \alpha_{j,j}^{(k)}\Big| n_{-1} = j, p \notin \{s(j), n_1\}\right]$$

$$= -\frac{S-3}{2S(S-1)}\mathbb{E}\left[\sum_{\substack{p\in[S]\\p\notin\{j,s(j),n_1\}}} \left(\alpha_{p,j}^{(k)} + \alpha_{s(p),j}^{(k)}\right)^2 \alpha_{j,j}^{(k)}\Big| n_{-1} = j\right], \tag{87}$$

where the second equality follows from the symmetry of the tree structure (a node $p$ and its sibling $s(p)$ are symmetric). Thus we can rewrite (85) as

$$\delta x_{j,j}^{(k)} = -\frac{1}{S}\mathbb{E}\left[\left(\alpha_{0,j}^{(k)}\right)^2 \alpha_{j,j}^{(k)}\Big| n_{-1} = j\right]$$

$$- \frac{S-3}{2S(S-1)}\mathbb{E}\left[\sum_{\substack{p\in[S]\\p\notin\{j,s(j),n_1\}}} \left(\alpha_{p,j}^{(k)} + \alpha_{s(p),j}^{(k)}\right)^2 \alpha_{j,j}^{(k)}\Big| n_{-1} = j\right]$$

$$- \frac{1}{S}\mathbb{E}\left[\left(1 - \alpha_{j,j}^{(k)} - \alpha_{s(j),j}^{(k)}\right)^2 \alpha_{j,j}^{(k)}\Big| n_{-1} = j\right], \tag{88}$$

which gives (50a).

Similarly, for $i \in [S]\setminus\{j\}$, we have

$$\delta x_{i,j}^{(k)} = -\frac{1}{S}\mathbb{E}\left[\left(\alpha_{0,j}^{(k)}\right)^2 \alpha_{i,j}^{(k)}\Big| n_{-1} = j\right] + \frac{1}{S}\mathbb{E}\left[\left(1 - \alpha_{j,j}^{(k)} - \alpha_{s(j),j}^{(k)}\right)\alpha_{i,j}^{(k)}\alpha_{j,j}^{(k)}\Big| n_{-1} = j\right]$$

$$+ \frac{1}{S(S-1)} \sum_{p\in[S]\setminus\{i,j\}} \mathbb{E}\left[\left(1 - \alpha_{j,j}^{(k)} - \alpha_{p,j}^{(k)}\right)\alpha_{i,j}^{(k)}\alpha_{p,j}^{(k)}\Big| n_{-1} = j, s(j) = p\right]$$

$$- \frac{S-3}{S(S-1)} \sum_{p\in[S]\setminus\{i,j\}} \mathbb{E}\left[\left(\alpha_{p,j}^{(k)} + \alpha_{s(p),j}^{(k)}\right)\alpha_{i,j}^{(k)}\alpha_{p,j}^{(k)}\Big| n_{-1} = j, s(j) \neq p, n_1 \neq p\right]$$

$$- \frac{1}{S(S-1)}\mathbb{E}\left[\left(1 - \alpha_{j,j}^{(k)} - \alpha_{i,j}^{(k)}\right)\alpha_{i,j}^{(k)}\left(1 - \alpha_{i,j}^{(k)}\right)\Big| n_{-1} = j, s(j) = i\right]$$

$$+ \frac{S-3}{S(S-1)}\mathbb{E}\left[\left(\alpha_{i,j}^{(k)} + \alpha_{s(i),j}^{(k)}\right)\alpha_{i,j}^{(k)}\left(1 - \alpha_{i,j}^{(k)}\right)\Big| n_{-1} = j, s(j) \neq i, n_1 \neq i\right]$$

$$= -\frac{1}{S}\mathbb{E}\left[\left(\alpha_{0,j}^{(k)}\right)^2 \alpha_{i,j}^{(k)}\Big| n_{-1} = j\right] + \frac{1}{S}\mathbb{E}\left[\left(1 - \alpha_{j,j}^{(k)} - \alpha_{s(j),j}^{(k)}\right)\alpha_{i,j}^{(k)}\alpha_{j,j}^{(k)}\Big| n_{-1} = j\right]$$

$$+ \frac{1}{S(S-1)} \sum_{p\in[S]\setminus\{j\}} \mathbb{E}\left[\left(1 - \alpha_{j,j}^{(k)} - \alpha_{p,j}^{(k)}\right)\alpha_{i,j}^{(k)}\alpha_{p,j}^{(k)}\Big| n_{-1} = j, s(j) = p\right]$$

$$- \frac{S-3}{S(S-1)} \sum_{p\in[S]\setminus\{j\}} \mathbb{E}\left[\left(\alpha_{p,j}^{(k)} + \alpha_{s(p),j}^{(k)}\right)\alpha_{i,j}^{(k)}\alpha_{p,j}^{(k)}\Big| n_{-1} = j, s(j) \neq p, n_1 \neq p\right]$$

$$- \frac{1}{S(S-1)}\mathbb{E}\left[\left(1 - \alpha_{j,j}^{(k)} - \alpha_{i,j}^{(k)}\right)\alpha_{i,j}^{(k)}\Big| n_{-1} = j, s(j) = i\right]$$

$$+ \frac{S-3}{S(S-1)}\mathbb{E}\left[\left(\alpha_{i,j}^{(k)} + \alpha_{s(i),j}^{(k)}\right)\alpha_{i,j}^{(k)}\Big| n_{-1} = j, s(j) \neq i, n_1 \neq i\right]. \tag{89}$$

Similar as in (86), we can simplify the third term in (89) as

$$\frac{1}{S(S-1)} \sum_{p\in[S]\setminus\{j\}} \mathbb{E}\left[\left(1 - \alpha_{j,j}^{(k)} - \alpha_{p,j}^{(k)}\right)\alpha_{j,j}^{(k)}\alpha_{p,j}^{(k)}\Big| n_{-1} = j, s(j) = p\right]$$

$$= \frac{1}{S}\mathbb{E}\left[\left(1 - \alpha_{j,j}^{(k)} - \alpha_{s(j),j}^{(k)}\right)\alpha_{j,j}^{(k)}\alpha_{s(j),j}^{(k)}\Big| n_{-1} = j\right], \tag{90}$$

and similar to (87), we can rewrite the fourth term in (89) as

$$- \frac{S-3}{S(S-1)} \sum_{\substack{p\in[S] \\ p\neq j}} \mathbb{E}\left[\left(\alpha_{p,j}^{(k)} + \alpha_{s(p),j}^{(k)}\right)\alpha_{j,j}^{(k)}\alpha_{p,j}^{(k)}\Big| n_{-1} = j, s(j) \neq p, n_1 \neq p\right]$$

$$= -\frac{S-3}{2S(S-1)}\mathbb{E}\left[\sum_{\substack{p\in[S] \\ p\notin\{j,s(j),n_1\}}} \left(\alpha_{p,j}^{(k)} + \alpha_{s(p),j}^{(k)}\right)^2 \alpha_{i,j}^{(k)}\Big| n_{-1} = j\right] \tag{91}$$

Substituting (90) and (91) into (89) and reorganizing the terms, we have

$$\delta x_{i,j}^{(k)} = -\frac{1}{S}\mathbb{E}\left[\left(\alpha_{0,j}^{(k)}\right)^2 \alpha_{i,j}^{(k)}\Big| n_{-1} = j\right] + \frac{1}{S}\mathbb{E}\left[\left(1 - \alpha_{j,j}^{(k)} - \alpha_{s(j),j}^{(k)}\right)\alpha_{i,j}^{(k)}\left(\alpha_{j,j}^{(k)} + \alpha_{s(j),j}^{(k)}\right)\Big| n_{-1} = j\right]$$

$$- \frac{S-3}{2S(S-1)}\mathbb{E}\left[\sum_{\substack{p\in[S] \\ p\notin\{j,s(j),n_1\}}} \left(\alpha_{p,j}^{(k)} + \alpha_{s(p),j}^{(k)}\right)^2 \alpha_{i,j}^{(k)}\Big| n_{-1} = j\right]$$

$$- \frac{1}{S(S-1)}\mathbb{E}\left[\left(1 - \alpha_{j,j}^{(k)} - \alpha_{i,j}^{(k)}\right)\alpha_{i,j}^{(k)}\Big| n_{-1} = j, s(j) = i\right]$$

$$+ \frac{S-3}{S(S-1)}\mathbb{E}\left[\left(\alpha_{i,j}^{(k)} + \alpha_{s(i),j}^{(k)}\right)\alpha_{i,j}^{(k)}\Big| n_{-1} = j, i\notin\{s(j),n_1\}\right]. \tag{92}$$

This gives (50b).

**Computation of $\delta y^{(k)}$.** The fact that $\delta y_{0,j}^{(k)} = 0$ follows from our assumption that $a_0 = 0$, below we consider the case when $j \in [S]$. By the chain rule, we have

$$\delta y_{i,j}^{(k)} := \frac{\partial \Delta_y^{(k)}}{\partial H_{i,j}} = \mathbb{E}\left[\sum_{p=0}^{S} \frac{\partial \Delta_y^{(k)}}{\partial \alpha_{p,j}^{(k)}} \frac{\partial \alpha_{p,j}^{(k)}}{\partial H_{i,j}}\right]$$

$$\overset{(83)}{=} \mathbb{E}\left[\sum_{\substack{p=0\\p\neq i}}^{S} \frac{\partial \Delta_y^{(k)}}{\partial \alpha_{p,j}^{(k)}}\left(-\alpha_{i,j}^{(k)}\alpha_{p,j}^{(k)}\right) + \frac{\partial \Delta_y^{(k)}}{\partial \alpha_{i,j}^{(k)}}\alpha_{i,j}^{(k)}\left(1-\alpha_{i,j}^{(k)}\right)\right]. \tag{93}$$

By (49b), we have

$$\frac{\partial \Delta_y^{(k)}}{\partial \alpha_{p,j}^{(k)}} = \begin{cases} -\frac{1}{S}\mathbb{E}\left[1-\alpha_{j,j}^{(k)}\big|n_{-1}=j\right], & \text{if } p=j,\\ \frac{1}{S}\mathbb{E}\left[\alpha_{p,j}^{(k)}\big|n_{-1}=j\right], & \text{if } p\in[S]\backslash\{j\},\\ 0, & \text{if } p=0. \end{cases} \tag{94}$$

Plugging (94) into (93), we have

$$\delta y_{j,j}^{(k)} = -\frac{1}{S}\sum_{p\in[S]\backslash\{j\}}\mathbb{E}\left[\left(\alpha_{p,j}^{(k)}\right)^2\alpha_{j,j}^{(k)}\Big|n_{-1}=j\right] - \frac{1}{S}\mathbb{E}\left[\left(1-\alpha_{j,j}^{(k)}\right)^2\alpha_{j,j}^{(k)}\Big|n_{-1}=j\right]$$

$$= -\frac{1}{S}\mathbb{E}\left[\alpha_{j,j}^{(k)}\left(\sum_{p\in[S]\backslash\{j\}}\left(\alpha_{p,j}^{(k)}\right)^2 + \left(1-\alpha_{j,j}^{(k)}\right)^2\right)\Bigg|n_{-1}=j\right], \tag{95}$$

which gives (51a).

For $i\in[S]\backslash\{j\}$, we have

$$\delta y_{i,j}^{(k)} = -\frac{1}{S}\sum_{p\in[S]\backslash\{j\}}\mathbb{E}\left[\left(\alpha_{p,j}^{(k)}\right)^2\alpha_{i,j}^{(k)}\Big|n_{-1}=j\right] + \frac{1}{S}\mathbb{E}\left[\alpha_{i,j}^{(k)}\left(\alpha_{i,j}^{(k)}\left(1-\alpha_{i,j}^{(k)}\right)+\left(\alpha_{i,j}^{(k)}\right)^2\right)\Big|n_{-1}=j\right]$$

$$+ \frac{1}{S}\mathbb{E}\left[(1-\alpha_{j,j}^{(k)})\alpha_{i,j}^{(k)}\alpha_{j,j}^{(k)}\Big|n_{-1}=j\right]$$

$$= -\frac{1}{S}\mathbb{E}\left[\alpha_{i,j}^{(k)}\left(\sum_{p\in[S]\backslash\{j\}}\left(\alpha_{p,j}^{(k)}\right)^2 - \left(1-\alpha_{j,j}^{(k)}\right)\alpha_{j,j}^{(k)} - \alpha_{i,j}^{(k)}\right)\Bigg|n_{-1}=j\right], \tag{96}$$

which gives (51b).

### C.1.4 Proof of Lemma 4

For notation simplicity, we let $n_{-1}=n_{-1}^{(k)}$. By (50b), for all $j\in[S]$ and $i\in[S]\backslash\{j\}$,

$$\delta x_{i,j}^{(k)} + \frac{1}{S}\mathbb{E}\left[\left(\alpha_{0,j}^{(k)}\right)^2\alpha_{i,j}^{(k)}\Big|n_{-1}=j\right]$$

$$= \frac{1}{S}\mathbb{E}\left[\left(1-\alpha_{j,j}^{(k)}-\alpha_{s(j),j}^{(k)}\right)\alpha_{i,j}^{(k)}\left(\alpha_{j,j}^{(k)}+\alpha_{s(j),j}^{(k)}\right)\Big|n_{-1}=j\right]$$

$$- \frac{S-3}{2S(S-1)}\mathbb{E}\left[\sum_{\substack{p\in[S]\\p\notin\{j,s(j),n_1\}}}\left(\alpha_{p,j}^{(k)}+\alpha_{s(p),j}^{(k)}\right)^2\alpha_{i,j}^{(k)}\Big|n_{-1}=j\right]$$

$$- \frac{1}{S(S-1)}\mathbb{E}\left[\left(1-\alpha_{j,j}^{(k)}-\alpha_{i,j}^{(k)}\right)\alpha_{i,j}^{(k)}\Big|n_{-1}=j,s(j)=i\right]$$

$$+ \frac{S-3}{S(S-1)}\mathbb{E}\left[\left(\alpha_{i,j}^{(k)}+\alpha_{s(i),j}^{(k)}\right)\alpha_{i,j}^{(k)}\Big|n_{-1}=j,i\notin\{s(j),n_1\}\right]$$

$$\geqslant \underbrace{\frac{1}{S(S-1)}\mathbb{E}\left[\left(1-\alpha_{j,j}^{(k)}-\alpha_{s(j),j}^{(k)}\right)\alpha_{i,j}^{(k)}\left(\alpha_{j,j}^{(k)}+\alpha_{s(j),j}^{(k)}\right)\Big|n_{-1}=j,s(j)=i\right]}_{(a)}$$

$$+ \underbrace{\frac{S-2}{S(S-1)}\mathbb{E}\left[\left(1-\alpha_{j,j}^{(k)}-\alpha_{s(j),j}^{(k)}\right)\alpha_{i,j}^{(k)}\alpha_{j,j}^{(k)}\Big|n_{-1}=j,s(j)\neq i\right]}$$

$$\underbrace{+ \frac{S-2}{S(S-1)}\mathbb{E}\left[\left(1-\alpha_{j,j}^{(k)}-\alpha_{s(j),j}^{(k)}\right)\alpha_{i,j}^{(k)}\alpha_{s(j),j}^{(k)}\Big|n_{-1}=j,s(j)\neq i\right]}_{(b)}$$

$$-\frac{S-3}{2S(S-1)}\mathbb{E}\left[\sum_{\substack{p\in[S]\\p\notin\{j,s(j),n_1\}}}\left(\alpha_{p,j}^{(k)}+\alpha_{s(p),j}^{(k)}\right)^2\alpha_{i,j}^{(k)}\middle|n_{-1}=j\right]$$

$$\underbrace{-\frac{1}{S(S-1)}\mathbb{E}\left[\left(1-\alpha_{j,j}^{(k)}-\alpha_{i,j}^{(k)}\right)\alpha_{i,j}^{(k)}\middle|n_{-1}=j,s(j)=i\right]}_{(c)}. \tag{97}$$

Note that (a) and (c) add up to the following:

$$(a)+(c)=\frac{1}{S(S-1)}\mathbb{E}\left[\left(1-\alpha_{j,j}^{(k)}-\alpha_{s(j),j}^{(k)}\right)\alpha_{i,j}^{(k)}\left(\alpha_{j,j}^{(k)}+\alpha_{s(j),j}^{(k)}\right)\middle|n_{-1}=j,s(j)=i\right]$$

$$-\frac{1}{S(S-1)}\mathbb{E}\left[\left(1-\alpha_{j,j}^{(k)}-\alpha_{i,j}^{(k)}\right)\alpha_{i,j}^{(k)}\middle|n_{-1}=j,s(j)=i\right]$$

$$=-\frac{1}{S(S-1)}\mathbb{E}\left[\left(1-\alpha_{j,j}^{(k)}-\alpha_{s(j),j}^{(k)}\right)^2\alpha_{s(j),j}^{(k)}\middle|n_{-1}=j,s(j)=i\right]$$

$$=-\frac{1}{S(S-1)}\mathbb{E}\left[\left(1-\alpha_{j,j}^{(k)}-\alpha_{s(j),j}^{(k)}\right)^2\alpha_{s(j),j}^{(k)}\middle|n_{-1}=j\right]. \tag{98}$$

We could also rewrite (b) as follows:

$$(b)=\frac{S-2}{S(S-1)}\mathbb{E}\left[\left(1-\alpha_{j,j}^{(k)}-\alpha_{s(j),j}^{(k)}\right)\alpha_{i,j}^{(k)}\alpha_{s(j),j}^{(k)}\middle|n_{-1}=j,i\neq s(j)\right]$$

$$=\frac{1}{S(S-1)}\mathbb{E}\left[\left(1-\alpha_{j,j}^{(k)}-\alpha_{s(j),j}^{(k)}\right)\sum_{p\in[S]\setminus\{j,s(j)\}}\alpha_{p,j}^{(k)}\alpha_{s(j),j}^{(k)}\middle|n_{-1}=j\right]$$

$$=\frac{1}{S(S-1)}\mathbb{E}\left[\left(1-\alpha_{j,j}^{(k)}-\alpha_{s(j),j}^{(k)}\right)^2\alpha_{i,j}^{(k)}\alpha_{s(j),j}^{(k)}\middle|n_{-1}=j\right]$$

$$-\frac{1}{S(S-1)}\mathbb{E}\left[\left(1-\alpha_{j,j}^{(k)}-\alpha_{s(j),j}^{(k)}\right)\alpha_{0,j}^{(k)}\alpha_{s(j),j}^{(k)}\middle|n_{-1}=j\right]$$

$$=\frac{1}{S(S-1)}\mathbb{E}\left[\left(1-\alpha_{j,j}^{(k)}-\alpha_{s(j),j}^{(k)}\right)^2\alpha_{s(j),j}^{(k)}\middle|n_{-1}=j\right]$$

$$-\frac{1}{S(S-1)}\frac{S-1}{N-1}\mathbb{E}\left[\left(1-\alpha_{j,j}^{(k)}-\alpha_{s(j),j}^{(k)}\right)\alpha_{0,j}^{(k)}\mathbb{1}\{N_i^{(k-1)}\geq1\}\alpha_{s(j),j}^{(k)}\middle|n_{-1}=j\right]$$

$$\geq\underbrace{\frac{1}{S(S-1)}\mathbb{E}\left[\left(1-\alpha_{j,j}^{(k)}-\alpha_{s(j),j}^{(k)}\right)^2\alpha_{s(j),j}^{(k)}\middle|n_{-1}=j\right]}_{=-(a)-(c)\text{ by (98)}}$$

$$-\frac{1}{S(N-1)}\mathbb{E}\left[\left(1-\alpha_{j,j}^{(k)}-\alpha_{s(j),j}^{(k)}\right)\alpha_{0,j}^{(k)}\alpha_{i,j}^{(k)}\middle|n_{-1}=j\right], \tag{99}$$

where the second line follows from the symmetry of the training distribution, and the last inequality uses the fact that $\alpha_{i,j}^{(k)}\geq\mathbb{1}\{N_i^{(k-1)}\geq1\}\alpha_{s(j),j}^{(k)}$. This is because by symmetry we have $H_{s(j),j}=H_{i,j}$, and $N_{s(j)}^{(k-1)}=1$ because $s(j)$ is not on the path.

Plugging (98) and (99) into (97), we have

$$\delta x_{i,j}^{(k)}+\frac{1}{S}\mathbb{E}\left[\left(\alpha_{0,j}^{(k)}\right)^2\alpha_{i,j}^{(k)}\middle|n_{-1}=j\right]\geq\frac{S-2}{S(S-1)}\mathbb{E}\left[\left(1-\alpha_{j,j}^{(k)}-\alpha_{s(j),j}^{(k)}\right)\alpha_{i,j}^{(k)}\alpha_{j,j}^{(k)}\middle|n_{-1}=j,s(j)\neq i\right]$$

$$-\frac{1}{S(N-1)}\mathbb{E}\left[\left(1-\alpha_{j,j}^{(k)}-\alpha_{s(j),j}^{(k)}\right)\alpha_{0,j}^{(k)}\alpha_{i,j}^{(k)}\middle|n_{-1}=j\right]$$

$$-\frac{S-3}{2S(S-1)}\mathbb{E}\left[\sum_{\substack{p\in[S]\\p\notin\{j,s(j),n_1\}}}\left(\alpha_{p,j}^{(k)}+\alpha_{s(p),j}^{(k)}\right)^2\alpha_{i,j}^{(k)}\middle|n_{-1}=j\right]$$

$$=\underbrace{\frac{1}{S}\mathbb{E}\left[\left(1-\alpha_{j,j}^{(k)}-\alpha_{s(j),j}^{(k)}\right)\alpha_{i,j}^{(k)}\alpha_{j,j}^{(k)}\middle|n_{-1}=j\right]}_{(d)}$$

$$\underbrace{-\frac{1}{S(S-1)}\mathbb{E}\left[\left(1-\alpha_{j,j}^{(k)}-\alpha_{s(j),j}^{(k)}\right)\alpha_{i,j}^{(k)}\alpha_{j,j}^{(k)}\Big|n_{-1}=j,s(j)=i\right]}_{(e)}$$

$$\underbrace{-\frac{1}{S(N-1)}\mathbb{E}\left[\left(1-\alpha_{j,j}^{(k)}-\alpha_{s(j),j}^{(k)}\right)\alpha_{0,j}^{(k)}\alpha_{i,j}^{(k)}\Big|n_{-1}=j\right]}_{(f)}$$

$$\underbrace{-\frac{1}{2S}\mathbb{E}\left[\sum_{\substack{p\in[S]\\p\notin\{j,s(j),n_1\}}}\left(\alpha_{p,j}^{(k)}+\alpha_{s(p),j}^{(k)}\right)^2\alpha_{i,j}^{(k)}\Big|n_{-1}=j\right]}_{(g)}.$$

$$(100)$$

Following the same logic as in the proof of (99), we have that (e) can be expressed as

$$(e)=-\frac{1}{S(S-1)}\mathbb{E}\left[\left(1-\alpha_{j,j}^{(k)}-\alpha_{s(j),j}^{(k)}\right)\alpha_{s(j),j}^{(k)}\alpha_{j,j}^{(k)}\Big|n_{-1}=j\right]$$

$$=-\frac{1}{S(N-1)}\mathbb{E}\left[\left(1-\alpha_{j,j}^{(k)}-\alpha_{s(j),j}^{(k)}\right)\alpha_{s(j),j}^{(k)}\mathbb{1}\{N_i^{(k-1)}\geqslant 1\}\alpha_{j,j}^{(k)}\Big|n_{-1}=j\right]$$

$$\geqslant-\frac{1}{S(N-1)}\mathbb{E}\left[\left(1-\alpha_{j,j}^{(k)}-\alpha_{s(j),j}^{(k)}\right)\alpha_{i,j}^{(k)}\alpha_{j,j}^{(k)}\Big|n_{-1}=j\right]. \tag{101}$$

Thus we have

$$(d)+(e)\geqslant\frac{N-2}{S(N-1)}\mathbb{E}\left[\left(1-\alpha_{j,j}^{(k)}-\alpha_{s(j),j}^{(k)}\right)\alpha_{i,j}^{(k)}\alpha_{j,j}^{(k)}\Big|n_{-1}=j\right]. \tag{102}$$

Furthermore, (g) can be bounded as follows:

$$(g)\geqslant-\frac{1}{S}\mathbb{E}\left[\max_{p\in[S]\backslash\{j\}}\alpha_{p,j}^{(k)}\left(1-\alpha_{j,j}^{(k)}-\alpha_{s(j),j}^{(k)}\right)\alpha_{i,j}^{(k)}\Big|n_{-1}=j\right], \tag{103}$$

which gives

$$(f)+(g)\geqslant-\frac{1}{S}\mathbb{E}\left[\left(\max_{p\in[S]\backslash\{j\}}\alpha_{p,j}^{(k)}+\frac{1}{N-1}\alpha_{0,j}^{(k)}\right)\left(1-\alpha_{j,j}^{(k)}-\alpha_{s(j),j}^{(k)}\right)\alpha_{i,j}^{(k)}\Big|n_{-1}=j\right]. \tag{104}$$

Substituting (102) and (104) into (100), we have

$$\delta x_{i,j}^{(k)}+\frac{1}{S}\mathbb{E}\left[\left(\alpha_{0,j}^{(k)}\right)^2\alpha_{i,j}^{(k)}\Big|n_{-1}=j\right]$$

$$\geqslant\frac{N-2}{S(N-1)}\mathbb{E}\left[\left(1-\alpha_{j,j}^{(k)}-\alpha_{s(j),j}^{(k)}\right)\alpha_{i,j}^{(k)}\alpha_{j,j}^{(k)}\Big|n_{-1}=j\right]$$

$$-\frac{1}{S}\mathbb{E}\left[\left(\max_{p\in[S]\backslash\{j\}}\alpha_{p,j}^{(k)}+\frac{1}{N-1}\alpha_{0,j}^{(k)}\right)\left(1-\alpha_{j,j}^{(k)}-\alpha_{s(j),j}^{(k)}\right)\alpha_{i,j}^{(k)}\Big|n_{-1}=j\right]$$

$$=\frac{1}{S}\mathbb{E}\left[\left(1-\alpha_{j,j}^{(k)}-\alpha_{s(j),j}^{(k)}\right)\alpha_{i,j}^{(k)}\left(\frac{N-2}{N-1}\alpha_{j,j}^{(k)}-\max_{p\in[S]\backslash\{j\}}\alpha_{p,j}^{(k)}-\frac{1}{N-1}\alpha_{0,j}^{(k)}\right)\Big|n_{-1}=j\right]. \tag{105}$$

By (51b) we have that

$$\delta y_{i,j}^{(k)}-\frac{1}{S}\mathbb{E}\left[\left(\alpha_{0,j}^{(k)}\right)^2\alpha_{i,j}^{(k)}\Big|n_{-1}=j\right]$$

$$=-\frac{1}{S}\mathbb{E}\left[\alpha_{i,j}^{(k)}\left(\sum_{p\in[S]\backslash\{j\}}\left(\alpha_{p,j}^{(k)}\right)^2-\left(1-\alpha_{j,j}^{(k)}\right)\alpha_{j,j}^{(k)}-\alpha_{i,j}^{(k)}\right)\Big|n_{-1}=j\right]-\frac{1}{S}\mathbb{E}\left[\left(\alpha_{0,j}^{(k)}\right)^2\alpha_{i,j}^{(k)}\Big|n_{-1}=j\right]$$

$$\geqslant-\frac{1}{S}\mathbb{E}\left[\alpha_{i,j}^{(k)}\left(\sum_{\substack{0\leqslant p\leqslant S\\p\neq j}}\left(\alpha_{p,j}^{(k)}\right)^2-\left(1-\alpha_{j,j}^{(k)}\right)\alpha_{j,j}^{(k)}\right)\Big|n_{-1}=j\right]$$

$$\geqslant \frac{1}{S}\mathbb{E}\left[\left(1-\alpha_{j,j}^{(k)}\right)\alpha_{i,j}^{(k)}\left(\alpha_{j,j}^{(k)}-\max_{\substack{0\leqslant p\leqslant S\\p\neq j}}\alpha_{p,j}^{(k)}\right)\Big|n_{-1}=j\right]. \tag{106}$$

We have (52) follows from adding (105) and (106).

### C.1.5 Proof of Lemma 5

Recall in Assumption 3 we require $m \geqslant 3$, which implies $N \geqslant 15$ and (57) in turn implies $H_{j,j}^{(t)} \geqslant H_{i,j}^{(t)}$. In addition, when $n_{-1}^{(k)} = j$, we have $N_{i,j}^{(k-1)} \leqslant 2$, and $N_{j,j}^{(k-1)} = 1$. Thus the above facts together with the definition of $\alpha$ (c.f. (48)) lead to:

$$\alpha_{j,j}^{(t,k)} \geqslant \frac{1}{2}\alpha_{i,j}^{(t,k)}, \quad \forall 0 \leqslant i \leqslant S, i \neq j, \tag{107}$$

and

$$\forall t \in \mathbb{N}, p \in [S]\backslash\{j\} : \alpha_{p,j}^{(t,k)} \leqslant \frac{2\exp(H_{p,j}^{(t)})}{\sum_{q\in[S]}N_q^{(t,k-1)}\exp(H_{q,j}^{(t)})+1}$$

$$\leqslant \frac{2\exp(H_{p,j}^{(t)})}{\exp(H_{j,j}^{(t)})+(N-1)\exp(H_{p,j}^{(t)})}$$

$$\leqslant \frac{2}{\exp(H_{j,j}^{(t)}-H_{p,j}^{(t)})+N-1} \stackrel{(57)}{\leqslant} \frac{2}{N}, \tag{108}$$

where the second line follows from the symmetry of our training distribution (i.e., $H_{i,j}$'s are equal for any $i \in [S]\backslash\{j\}$). Thus by our choice of $T_1$ (see (54)) and the fact that $\alpha_{j,j}^{(t,k)} \leqslant \alpha_{j,j}^{(t,1)}$ for any $k \in [m]$, we have when $n_{-1} = j$:

$$\forall t \in [T_1] : 1 - \alpha_{j,j}^{(t,k)} \geqslant \frac{1}{2}, \quad \text{and} \quad 1 - \alpha_{j,j}^{(t,k)} - \alpha_{s(j),j}^{(t,k)} \geqslant \frac{1}{2} - \frac{2}{N} > \frac{1}{3}, \tag{109}$$

and

$$\alpha_{j,j}^{(t,k)} - \max_{\substack{0\leqslant p\leqslant S\\p\neq j}}\alpha_{p,j}^{(t,k)} \stackrel{(107)}{\geqslant} -\frac{1}{2}\max_{\substack{0\leqslant p\leqslant S\\p\neq j}}\alpha_{p,j}^{(t,k)} \stackrel{(108)}{\geqslant} -\frac{1}{N}. \tag{110}$$

Similarly, we have

$$\frac{N-2}{N-1}\alpha_{j,j}^{(t,k)} - \max_{p\in[S]\backslash\{j\}}\alpha_{p,j}^{(t,k)} - \frac{1}{N-1}\alpha_{0,j}^{(t,k)} \stackrel{(107)}{\geqslant} \left(\frac{N-2}{2(N-1)}-1\right)\max_{p\in[S]\backslash\{j\}}\alpha_{p,j}^{(t,k)} - \frac{1}{2(N-1)}$$

$$\stackrel{(108)}{\geqslant} \left(\frac{N-2}{2(N-1)}-1\right)\cdot\frac{2}{N} - \frac{1}{2(N-1)} = \frac{-3}{2(N-1)}, \tag{111}$$

where in the first inequality we also use the fact that $\alpha_{0,j}^{(t,k)} \leqslant \frac{1}{2}$, since by (57) we have $\alpha_{0,j}^{(t,k)} \leqslant \alpha_{j,j}^{(t,k)}$. Therefore, we have

$$\frac{1}{S}\mathbb{E}\left[\left(1-\alpha_{j,j}^{(t,k)}-\alpha_{s(j),j}^{(t,k)}\right)\alpha_{i,j}^{(t,k)}\left(\alpha_{j,j}^{(t,k)}-\frac{N+1}{N-1}\max_{\substack{0\leqslant p\leqslant S\\p\notin\{j\}}}\alpha_{p,j}^{(t,k)}\right)\Big|n_{-1}=j\right]$$

$$\stackrel{(111)}{\geqslant} -\frac{3}{2S(N-1)}\mathbb{E}\left[\left(1-\alpha_{j,j}^{(t,k)}-\alpha_{s(j),j}^{(t,k)}\right)\alpha_{i,j}^{(t,k)}\Big|n_{-1}=j\right]$$

$$\stackrel{(109)}{\geqslant} -\frac{9}{2S(N-1)}\mathbb{E}\left[\left(1-\alpha_{j,j}^{(t,k)}-\alpha_{s(j),j}^{(t,k)}\right)^2\alpha_{i,j}^{(t,k)}\Big|n_{-1}=j\right]$$

$$\stackrel{(107)}{\geqslant} -\frac{9}{S(N-1)}\mathbb{E}\left[\left(1-\alpha_{j,j}^{(t,k)}-\alpha_{s(j),j}^{(t,k)}\right)^2\alpha_{j,j}^{(t,k)}\Big|n_{-1}=j\right] \stackrel{(50a)}{\geqslant} -\frac{9}{N-1}\delta x_{j,j}^{(t,k)}. \tag{112}$$

Analogously, we have

$$\frac{1}{S}\mathbb{E}\left[\left(1-\alpha_{j,j}^{(t,k)}\right)\alpha_{i,j}^{(k)}\left(\alpha_{j,j}^{(t,k)}-\max_{\substack{0\leqslant p\leqslant S\\p\neq j}}\alpha_{p,j}^{(t,k)}\right)\Big|n_{-1}=j\right]$$

$$\overset{(110)}{\geqslant} -\frac{1}{SN}\mathbb{E}\left[\left(1-\alpha_{j,j}^{(t,k)}\right)\alpha_{i,j}^{(t,k)}\Big|n_{-1}=j\right]$$

$$\overset{(109)}{\geqslant} -\frac{2}{SN}\mathbb{E}\left[\left(1-\alpha_{j,j}^{(t,k)}\right)^2\alpha_{i,j}^{(t,k)}\Big|n_{-1}=j\right]$$

$$\overset{(107)}{\geqslant} -\frac{4}{SN}\mathbb{E}\left[\left(1-\alpha_{j,j}^{(t,k)}\right)^2\alpha_{j,j}^{(t,k)}\Big|n_{-1}=j\right]\overset{(51a)}{\geqslant} -\frac{9}{N-1}\delta y_{j,j}^{(t,k)}. \tag{113}$$

Combining the above two inequalities with (52), we have the desired bound

$$\delta x_{i,j}^{(t,k)}+\delta y_{i,j}^{(t,k)} \geqslant \frac{9}{N-1}\left(\delta x_{j,j}^{(t,k)}+\delta y_{j,j}^{(t,k)}\right).$$

### C.1.6 Proof of Lemma 6

By (50a) and (51a), we have

$$\forall j \in [S]: \quad -\delta x_{j,j}^{(t,k)}-\delta y_{j,j}^{(t,k)} \geqslant -\delta y_{j,j}^{(t,k)} \geqslant \frac{1}{S}\mathbb{E}\left[\alpha_{j,j}^{(t,k)}\left(1-\alpha_{j,j}^{(t,k)}\right)^2\Big|n_{-1}=j\right]. \tag{114}$$

Besides, when $n_{-1}^{(k)}=j$, we have $N_{j,j}^{(k-1)}=1$, and

$$\forall t \leqslant T_1, j \in [S]: \quad \alpha_{j,j}^{(t,k)}=\frac{\exp\left(H_{j,j}^{(t)}\right)}{\sum_{0\leqslant i\leqslant S}N_i^{(k-1)}\exp\left(H_{i,j}^{(t)}\right)} \overset{(57)}{\geqslant} \frac{\exp\left((1-\frac{9}{N-1})H_{j,j}^{(t)}\right)}{\sum_{0\leqslant i\leqslant S}N_i^{(k-1)}}, \tag{115}$$

which indicates

$$\forall t \leqslant T_1: \quad \frac{1}{N+m} \leqslant \alpha_{j,j}^{(t,k)} \leqslant \frac{1}{2}, \quad \forall j \in [S],$$

where we use the fact that $\alpha_{j,j}^{(t,k)}$ are equal for any $j \in [S]$ due to the symmetry of our training distribution. By (114) we have

$$-\delta x_{j,j}^{(t,k)}-\delta y_{j,j}^{(t,k)} \geqslant \frac{1}{S(N+m)}\left(\frac{N+m-1}{N+m}\right)^2 \geqslant \frac{1}{2S(N+m)}. \tag{116}$$

(115) also implies that

$$\forall t \leqslant T_1: \quad H_{j,j}^{(t)} \leqslant \frac{1}{1-\frac{9}{N-1}}\log\left((N+1)\alpha_{j,j}^{(t,1)}\right) \leqslant 3\log\left(\frac{N+1}{2}\right). \tag{117}$$

The above two expressions together with (53) imply that

$$\frac{m\eta}{2S(N+m)}T_1 \leqslant 3\log\left(\frac{N+1}{2}\right),$$

which gives the desired result.

### C.1.7 Proof of Lemma 7

For notation simplicity, we let $n_{-1}=n_{-1}^{(k)}$. By (50b), we know that

$$\delta x_{i,j}^{(t,k)} \leqslant \underbrace{\frac{1}{S}\mathbb{E}\left[\left(1-\alpha_{j,j}^{(t,k)}-\alpha_{s(j),j}^{(k)}\right)\alpha_{i,j}^{(t,k)}\left(\alpha_{j,j}^{(t,k)}+\alpha_{s(j),j}^{(k)}\right)\Big|n_{-1}=j\right]}_{(a)}$$

$$+ \underbrace{\frac{S-3}{S(S-1)}\mathbb{E}\left[\left(\alpha_{i,j}^{(t,k)}+\alpha_{s(i),j}^{(t,k)}\right)\alpha_{i,j}^{(t,k)}\Big|n_{-1}=j, i \notin \{s(j),n_1\}\right]}_{(b)}. \tag{118}$$

By symmetry and the fact that $N_i^{(k-1)} \leqslant 2$ for all $i \in [S]$, we have

$$\alpha_{i,j}^{(t,k)} \leqslant \mathbb{1}\left\{N_i^{(k-1)} \geqslant 1\right\}\frac{2\left(1-\alpha_{j,j}^{(t,k)}-\alpha_{s(j),j}^{(t,k)}\right)}{N-2} \quad \text{and} \quad \alpha_{s(j),j}^{(t,k)} \leqslant \alpha_{j,j}^{(t,k)}$$

since $N_{s(j)}^{(k-1)} = 1$. Thus we have

$$(a) \leqslant \frac{4}{S(N-2)} \mathbb{E}\left[\mathbb{1}\left\{N_i^{(k-1)} \geqslant 1\right\}\left(1 - \alpha_{j,j}^{(t,k)} - \alpha_{s(j),j}^{(t,k)}\right)^2 \alpha_{j,j}^{(t,k)}\Big|n_{-1} = j\right]$$

$$\leqslant \frac{4(N-1)}{S(S-1)(N-2)} \mathbb{E}\left[\left(1 - \alpha_{j,j}^{(t,k)} - \alpha_{s(j),j}^{(t,k)}\right)^2 \alpha_{j,j}^{(t,k)}\Big|n_{-1} = j\right]. \tag{119}$$

Similarly, we have

$$\alpha_{i,j}^{(t,k)} + \alpha_{s(i),j}^{(t,k)} \leqslant \mathbb{1}\left\{N_i^{(k-1)} \geqslant 1\right\} \frac{3\left(1 - \alpha_{j,j}^{(t,k)} - \alpha_{s(j),j}^{(t,k)}\right)}{N-2} \quad \text{and} \quad \alpha_{i,j}^{(t,k)} \leqslant 2\alpha_{j,j}^{(t,k)}.$$

Further, by (109) we have when $t \leqslant T_1$,

$$1 - \alpha_{j,j}^{(t,k)} - \alpha_{s(j),j}^{(t,k)} \geqslant \frac{1}{2} - \frac{2}{N}.$$

Thus we have

$$(b) \leqslant \frac{12(S-3)N}{S(S-1)(N-2)^2} \mathbb{E}\left[\mathbb{1}\left\{N_i^{(k-1)} \geqslant 1\right\}\left(1 - \alpha_{j,j}^{(t,k)} - \alpha_{s(j),j}^{(t,k)}\right)^2 \alpha_{j,j}^{(k)}\Big|n_{-1} = j, i \notin \{s(j), n_1\}\right]$$

$$\leqslant \frac{12N(N-3)}{S(S-1)(N-2)^2} \mathbb{E}\left[\left(1 - \alpha_{j,j}^{(t,k)} - \alpha_{s(j),j}^{(t,k)}\right)^2 \alpha_{j,j}^{(t,k)}\Big|n_{-1} = j, i \notin \{s(j), n_1\}\right]$$

$$\leqslant \frac{12N}{S(S-1)(N-2)} \mathbb{E}\left[\left(1 - \alpha_{j,j}^{(t,k)} - \alpha_{s(j),j}^{(t,k)}\right)^2 \alpha_{j,j}^{(t,k)}\Big|n_{-1} = j\right]. \tag{120}$$

Plugging (119) and (120) into (118), we have

$$\delta x_{i,j}^{(t,k)} \leqslant \frac{16N}{S(S-1)(N-2)} \mathbb{E}\left[\left(1 - \alpha_{j,j}^{(t,k)} - \alpha_{s(j),j}^{(t,k)}\right)^2 \alpha_{j,j}^{(t,k)}\Big|n_{-1} = j\right] \overset{(50a)}{\leqslant} -\frac{16N}{(S-1)(N-2)}\delta x_{j,j}^{(t,k)}. \tag{121}$$

Analogously, by (51b) we have

$$\delta y_{i,j}^{(t,k)} \leqslant \frac{1}{S} \mathbb{E}\left[\alpha_{i,j}^{(t,k)}\left(\left(1 - \alpha_{j,j}^{(t,k)}\right)\alpha_{j,j}^{(t,k)} + \alpha_{i,j}^{(t,k)}\right)\Big|n_{-1} = j\right]. \tag{122}$$

Note that

$$\alpha_{i,j}^{(t,k)}\left(1 - \alpha_{j,j}^{(t,k)}\right)\alpha_{j,j}^{(t,k)} \leqslant \mathbb{1}\left\{N_i^{(k-1)} \geqslant 1\right\} \cdot \frac{2}{N-1}\left(1 - \alpha_{j,j}^{(t,k)}\right)^2 \alpha_{j,j}^{(t,k)}, \tag{123}$$

and when $t \leqslant T_1$,

$$\left(\alpha_{i,j}^{(t,k)}\right)^2 \leqslant \mathbb{1}\left\{N_i^{(k-1)} \geqslant 1\right\} \cdot 4\alpha_{j,j}^{(t,k)}\frac{\left(1 - \alpha_{j,j}^{(t,k)}\right)}{N-1} \overset{(109)}{\leqslant} \mathbb{1}\left\{N_i^{(k-1)} \geqslant 1\right\} \cdot \frac{8}{N-1}\left(1 - \alpha_{j,j}^{(t,k)}\right)^2 \alpha_{j,j}^{(t,k)}. \tag{124}$$

Plugging the above two expressions into (122), we have

$$\delta y_{i,j}^{(t,k)} \leqslant \frac{1}{S} \mathbb{E}\left[\mathbb{1}\left\{N_i^{(k-1)} \geqslant 1\right\} \cdot \frac{10}{N-1}\left(1 - \alpha_{j,j}^{(t,k)}\right)^2 \alpha_{j,j}^{(t,k)}\Big|n_{-1} = j\right]$$

$$\leqslant \frac{10}{S(S-1)} \mathbb{E}\left[\left(1 - \alpha_{j,j}^{(t,k)}\right)^2 \alpha_{j,j}^{(t,k)}\Big|n_{-1} = j\right] \overset{(51a)}{\leqslant} -\frac{10}{S-1}\delta y_{j,j}^{(t,k)}. \tag{125}$$

Combining (121) and (125), we have

$$\delta x_{i,j}^{(t,k)} + \delta y_{i,j}^{(t,k)} \leqslant -\frac{16N}{(S-1)(N-2)}\left(\delta x_{j,j}^{(t,k)} + \delta y_{j,j}^{(t,k)}\right) \leqslant -\frac{19}{S-1}\left(\delta x_{j,j}^{(t,k)} + \delta y_{j,j}^{(t,k)}\right). \tag{126}$$

This indicates that for all $i, j \in [S]$, $i \neq j$,

$$\forall t \leqslant T_1: \quad H_{i,j}^{(t)} \geqslant -\frac{19}{S-1}H_{j,j}^{(t)}.$$

### C.1.8 Proof of Lemma 8

For notation simplicity, we let $n_{-1} = n_{-1}^{(k)}$. The proof is similar to that of Lemma 7, except that after iteration $T_1$, we have the following tighter bound:

$$\alpha_{s(j),j}^{(t,k)} \leqslant \frac{1}{2(N-1)} \overset{(69)}{\leqslant} \frac{3\alpha_{j,j}^{(t,k)}}{2(N-1)}, \quad \text{and} \quad \alpha_{i,j}^{(t,k)} \leqslant \frac{3\alpha_{j,j}^{(t,k)}}{N-1}, \tag{127}$$

and thus

$$(a) \leqslant \frac{2N+1}{S(S-1)(N-2)} \mathbb{E}\left[\left(1 - \alpha_{j,j}^{(t,k)} - \alpha_{s(j),j}^{(t,k)}\right)^2 \alpha_{j,j}^{(t,k)} \Big| n_{-1} = j\right], \tag{128}$$

where (a) is defined in (118). When $t \leqslant T_1 + 1$, we also have

$$\alpha_{i,j}^{(t,k)} \leqslant \frac{2\left(1 - \alpha_{j,j}^{(t,k)} - \alpha_{s(j),j}^{(t,k)}\right)}{N-1} \overset{(69)}{\leqslant} \frac{6\alpha_{j,j}^{(t,k)}\left(1 - \alpha_{j,j}^{(t,k)} - \alpha_{s(j),j}^{(t,k)}\right)}{N-1}, \tag{129}$$

where the second inequality also uses the fact that $\alpha_{j,j}^{(t,k)}$ increases after $t > T_1$. Therefore, we have

$$(b) \leqslant \frac{18(S-3)}{S(S-1)^2(N-2)} \mathbb{E}\left[\left(1 - \alpha_{j,j}^{(t,k)} - \alpha_{s(j),j}^{(t,k)}\right)^2 \alpha_{j,j}^{(t,k)} \Big| n_{-1} = j, i \notin \{s(j), n_1\}\right]$$

$$\leqslant \frac{18}{S(S-1)(N-2)} \mathbb{E}\left[\left(1 - \alpha_{j,j}^{(t,k)} - \alpha_{s(j),j}^{(t,k)}\right)^2 \alpha_{j,j}^{(t,k)} \Big| n_{-1} = j, i \notin \{s(j), n_1\}\right], \tag{130}$$

where (b) is also defined in (118).

Plugging (128) and (130) into (121), we have

$$\delta x_{i,j}^{(t,k)} \leqslant \frac{2N+19}{S(S-1)(N-2)} \mathbb{E}\left[\left(1 - \alpha_{j,j}^{(t,k)} - \alpha_{s(j),j}^{(t,k)}\right)^2 \alpha_{j,j}^{(t,k)} \Big| n_{-1} = j\right]$$

$$\leqslant \frac{4}{S(S-1)} \mathbb{E}\left[\left(1 - \alpha_{j,j}^{(t,k)} - \alpha_{s(j),j}^{(t,k)}\right)^2 \alpha_{j,j}^{(t,k)} \Big| n_{-1} = j\right]$$

$$\overset{(50a)}{\leqslant} -\frac{4}{S-1} \delta x_{j,j}^{(t,k)}. \tag{131}$$

Moreover, when $t \geqslant T_1 + 1$, (123) still holds, and we have

$$\left(\alpha_{i,j}^{(t,k)}\right)^2 \leqslant \mathbb{1}\left\{N_i^{(k-1)} \geqslant 1\right\} \cdot \frac{4\left(1 - \alpha_{j,j}^{(t,k)}\right)^2}{(N-1)^2} \overset{(69)}{\leqslant} \mathbb{1}\left\{N_i^{(k-1)} \geqslant 1\right\} \cdot \frac{12\left(1 - \alpha_{j,j}^{(t,k)}\right)^2}{(N-1)^2} \alpha_{j,j}^{(t,k)}, \tag{132}$$

where the second inequality also uses the fact that $\alpha_{j,j}^{(t,k)}$ increases after $t > T_1$. Plugging (123) and (132) into (122), we have

$$\delta y_{i,j}^{(t,k)} \leqslant \frac{2N+10}{S(S-1)(N-1)} \mathbb{E}\left[\left(1 - \alpha_{j,j}^{(t,k)}\right)^2 \alpha_{j,j}^{(t,k)} \Big| n_{-1} = j\right]$$

$$\leqslant \frac{4}{S(S-1)} \mathbb{E}\left[\left(1 - \alpha_{j,j}^{(t,k)}\right)^2 \alpha_{j,j}^{(t,k)} \Big| n_{-1} = j\right] \overset{(51a)}{\leqslant} -\frac{4}{S-1} \delta y_{j,j}^{(t,k)}. \tag{133}$$

Combining (131) and (133), we have

$$\delta x_{i,j}^{(t,k)} + \delta y_{i,j}^{(t,k)} \leqslant -\frac{4}{S-1}\left(\delta x_{j,j}^{(t,k)} + \delta y_{j,j}^{(t,k)}\right).$$

This combining with (63) indicates that for all $i, j \in [S]$, $i \neq j$,

$$\forall t \geqslant T_1 + 1: \quad -\left(H_{i,j}^{(t)} - H_{i,j}^{(T_1+1)}\right) \leqslant \frac{4}{S-1}\left(H_{j,j}^{(t)} - H_{j,j}^{(T_1+1)}\right).$$

### C.1.9 Proof of Lemma 9

By our choice of $T_2$ (see (65)), we have

$$\breve{\alpha}^{(T_2)} := \frac{\exp(H_{1,1}^{(T_2)})}{(N+m-2)\exp(H_{2,1}^{(T_2)}) + \exp(H_{1,1}^{(T_2)}) + 1} \leqslant 1 - \sqrt{\frac{\epsilon}{2m}}.$$

By Induction Hypothesis I and II we know that for all $j \in [S]$ and $i \in \{0, \ldots, S\} \backslash \{j\}$:

$$\forall t \in \mathbb{N}: \quad H_{i,j}^{(t)} \leqslant \frac{9}{N-1} H_{j,j}^{(t)}.$$

Combining the above two expressions, we have

$$\frac{\exp(H_{1,1}^{(T_2)})}{(N+m-1)\exp(\frac{9}{N-1}H_{1,1}^{(T_2)}) + \exp(H_{1,1}^{(T_2)})} \leqslant 1 - \sqrt{\frac{\epsilon}{2m}},$$

which gives

$$H_{1,1}^{(T_2)} \leqslant \frac{1}{1 - \frac{9}{N-1}} \log\left((N+m-1)\left(\sqrt{\frac{2m}{\epsilon}} - 1\right)\right) \leqslant 3 \log\left((N+m-1)\left(\sqrt{\frac{2m}{\epsilon}} - 1\right)\right).$$

By (68a), we have

$$H_{1,1}^{(T_2)} - H_{1,1}^{(T_1)} \geqslant \frac{\eta\epsilon(T_2 - T_1)}{4mS}.$$

Combining the above two expressions, we have

$$T_2 - T_1 \leqslant \frac{12mS}{\eta\epsilon} \log\left((N+m-1)\left(\sqrt{\frac{2m}{\epsilon}} - 1\right)\right).$$

### C.1.10 Proof of Lemma 10

For notation simplicity, we let $n_{-1} = n_{-1}^{(k)}$. From (49a), it gives that

$$\Delta_x^{(t,k)} = \frac{1}{2S} \sum_{j \in [S]} \mathbb{E}\left[\left(\frac{1}{2} \sum_{\substack{l \in [S] \\ l \notin \{n_1, j, s(j)\}}} (\alpha_{l,j}^{(t,k)} + \alpha_{s(l),j}^{(t,k)})^2 + (1 - \alpha_{j,j}^{(t,k)} - \alpha_{s(j),j}^{(t,k)})^2 + (\alpha_{0,j}^{(t,k)})^2\right)\bigg| n_{-1} = j\right]$$

$$\leqslant \frac{1}{2S} \sum_{j \in [S]} \mathbb{E}\left[\left(\max_{\substack{0 \leqslant i \leqslant S, \\ i \neq j}} \alpha_{i,j}^{(t,k)} \left(\sum_{\substack{0 \leqslant i \leqslant S, \\ l \notin \{n_1, j, s(j)\}}} \alpha_{i,j}^{(t,k)}\right) + (1 - \alpha_{j,j}^{(t,k)} - \alpha_{s(j),j}^{(t,k)})^2\right)\bigg| n_{-1} = j\right]$$

$$\leqslant \frac{1}{S} \sum_{j \in [S]} \mathbb{E}\left[\left(1 - \alpha_{j,j}^{(t,k)}\right)^2 \bigg| n_{-1} = j\right],$$

and similarly, (49b) gives that

$$\Delta_y^{(t,k)} = \frac{1}{2S} \sum_{j \in [S]} \mathbb{E}\left[\left(\sum_{\substack{l \in [S], \\ l \neq j}} \left(\alpha_{l,j}^{(t,k)}\right)^2 + \left(1 - \alpha_{j,j}^{(t,k)}\right)^2\right)\bigg| n_{-1} = j\right]$$

$$\leqslant \frac{1}{S} \sum_{j \in [S]} \mathbb{E}\left[\left(1 - \alpha_{j,j}^{(t,k)}\right)^2 \bigg| n_{-1} = j\right].$$

Thus we have when $t \geqslant T_2 + 1$:

$$\mathcal{L}_{\text{train}}(\theta^{(t)}) = \sum_{k=1}^{m} \left(\Delta_x^{(t,k)} + \Delta_y^{(t,k)}\right) \leqslant \frac{2}{S} \sum_{k=1}^{m} \sum_{j \in [S]} \mathbb{E}\left[\left(1 - \alpha_{j,j}^{(t,k)}\right)^2 \bigg| n_{-1} = j\right] \leqslant \epsilon,$$

where the last inequality follows from our choice of $T_2$ (see (65)).

Since $H_{j,j}^{(t)}$ is strictly increasing after iteration $T_2$, we have

$$\forall t \geqslant T_2 + 1: \quad \frac{\exp(H_{j,j}^{(t)})}{\exp(H_{j,j}^{(t)}) + 1} \geqslant \alpha_{j,j}^{(t,k)} \geqslant 1 - \sqrt{\frac{\epsilon}{2m}},$$

from which we deduce (76).

## C.2 Proof of Theorem 5

For any non-leaf node $i$, we let $c_1(i)$ and $c_2(i)$ denote the left child and right child of node $i$, respectively. Let $\widetilde{c}_1(i) = c_1(i)$ for any $i$ that's not a leaf node, and we define $\widetilde{c}_1(g) = \widetilde{c}_1(n_{2^m}) = 0$. We also define $\widetilde{p}(i) = p(i)$ for any $i$ that's not the root, and $\widetilde{p}(r) = \widetilde{p}(n_1) = 0$.

Throughout the proof, we suppose Assumption 3 and Assumption 5 hold, $N$ is a sufficiently large constant, $\eta \lesssim \frac{1}{Nm}$, and $\epsilon \lesssim 1/\mathsf{poly}(N)$ is sufficiently small.

**Loss simplification and gradient computation.** Letting

$$x^{(k)} := o^{(k)}(1 : d_1), \quad y^{(k)} := o^{(k)}(d_1 + 1 : 2d_1), \quad z^{(k)} := o^{(k)}(2d_1 + 1 : 2d_1 + d_2),$$

$$\widehat{x}^{(k)} := \widehat{o}^{(k)}(1 : d_1), \quad \widehat{y}^{(k)} := \widehat{o}^{(k)}(d_1 + 1 : 2d_1), \quad \widehat{z}^{(k)} := \widehat{o}^{(k)}(2d_1 + 1 : 2d_1 + d_2),$$

and

$$\Delta_x^{(k)} := \frac{1}{2}\mathbb{E}_{\mathcal{T} \sim P_{\text{train}}} \left\| x^{(k)} - \widehat{x}^{(k)} \right\|_2^2, \quad \Delta_y^{(k)} := \frac{1}{2}\mathbb{E}_{\mathcal{T} \sim P_{\text{train}}} \left\| y^{(k)} - \widehat{y}^{(k)} \right\|_2^2, \quad \Delta_z^{(k)} := \frac{1}{2}\mathbb{E}_{\mathcal{T} \sim P_{\text{train}}} \left\| z^{(k)} - \widehat{z}^{(k)} \right\|_2^2,$$
(134)

we can rewrite the loss function as

$$\mathcal{L}_{\text{train}} = \sum_{k=1}^{2m} \left( \Delta_x^{(k)} + \Delta_y^{(k)} + \Delta_z^{(k)} \right). \tag{135}$$

Recalling $\widetilde{A} := (a_0, a_1, \ldots, a_S) \in \mathbb{R}^{d_1 \times (S+1)}$ and $\widetilde{S} = (s_f, s_b)$, we further define the following matrices:

$$U_l := \widetilde{A}^\top B_l \widetilde{A} \in \mathbb{R}^{(S+1) \times (S+1)}, \quad V_l := \widetilde{A}^\top C_l \widetilde{A} \in \mathbb{R}^{(S+1) \times (S+1)}, \quad l \in \{1, 2\},$$
$$U_3 := \widetilde{S}^\top B_3 \widetilde{S} \in \mathbb{R}^{2 \times 2}, \qquad V_3 := \widetilde{S}^\top C_3 \widetilde{S} \in \mathbb{R}^{2 \times 2},$$
(136)

whose entries are denoted by $U_{l,i,j}$ and $V_{l,i,j}$, respectively. Define the following gradients for all $k \in [2m]$ and $l \in \{1, 2, 3\}$ as

$$\delta x_{l,i,j}^{(k)} := \frac{\partial \Delta_x^{(k)}}{\partial U_{l,i,j}}, \qquad \delta y_{l,i,j}^{(t,k)} := \frac{\partial \Delta_y^{(k)}}{\partial U_{l,i,j}}, \qquad \delta z_{l,i,j}^{(k)} := \frac{\partial \Delta_z^{(k)}}{\partial V_{l,i,j}}, \tag{137}$$

since $\Delta_x^{(k)}$ and $\Delta_y^{(k)}$ depend only on $U_l$, and $\Delta_z^{(k)}$ depends only on $V_l$.

We define the following attention weights:

$$\alpha_{(p,q,u),(j,v)}^{(k)} := \frac{N_{p,q,u}^{(k-1)} \exp\left(a_p^\top B_1 a_j + a_q^\top B_2 a_j + s_u^\top B_3 s_v\right)}{\sum_{p',q',u'} N_{p',q',u'}^{(k-1)} \exp\left(a_{p'}^\top B_1 a_j + a_{q'}^\top B_2 a_j + s_{u'}^\top B_3 s_v\right)}, \tag{138a}$$

$$\beta_{(p,q,u),(j,v)}^{(k)} := \frac{N_{p,q,u}^{(k-1)} \exp\left(a_p^\top C_1 a_j + a_q^\top C_2 a_j + s_u^\top C_3 s_v\right)}{\sum_{p',q',u'} N_{p',q',u'}^{(k-1)} \exp\left(a_{p'}^\top C_1 a_j + a_{q'}^\top C_2 a_j + s_{u'}^\top C_3 s_v\right)}, \tag{138b}$$

where $u, v \in \{0, 1\}$, $p, q, j \in \{0, 1, \ldots, S\}$, and $N_{p,q,u}^{(k)} = N_{p,q,u}^{(k)}(\mathcal{T})$ denotes the number of times $(a_p, a_q, s_u)^\top$ appearing in the columns of $E^{(k-1)} = E^{(k-1)}(\mathcal{T})$ (set $N_{p,q,u}^{(k)} = 0$ if the combination $(a_p, a_q, s_u)^\top$ does not exist in $E^{(k-1)}$). Then we have

$$\forall k \in [2m]: \quad N_{p,q,u}^{(k-1)} \in \{0, 1, 2\},$$

and

$$\sum_{p,q,u} \alpha^{(k)}_{(p,q,u),(j,v)} = 1, \quad \sum_{p,q,u} \beta^{(k)}_{(p,q,u),(j,v)} = 1. \tag{139}$$

We simplify the loss function as in the following lemma, whose proof is given in Appendix C.2.4.

**Lemma 11** (Loss simplification). *Suppose Assumption 3 and Assumption 5 hold. For $\Delta_x^{(k)}$, we have*

$$\forall k \in [m]: \quad \Delta_x^{(k)} = \frac{1}{2} \sum_{j=1}^{S} \mathbb{E}\bigg[ \mathbb{1}\{n_{-1}^{(k)} = j\}\bigg\{ \Big(1 - \alpha^{(k)}_{(p(j),j,0),(j,1)} - \alpha^{(k)}_{(\tilde{c}_1(j),j,1),(j,1)}\Big)^2$$
$$+ \sum_{q\in[S]\backslash\{j\}} \Big(\alpha^{(k)}_{(p(q),q,0),(j,1)} + \alpha^{(k)}_{(\tilde{c}_1(q),q,1),(j,1)}\Big)^2 \bigg\}\bigg], \tag{140a}$$

$$k = m+1: \quad \Delta_x^{(m+1)} = \frac{1}{2} \sum_{j=1}^{S} \mathbb{E}\bigg[ \mathbb{1}\{n_1 = j\}\bigg\{ \Big(1 - \alpha^{(k)}_{(0,j,0),(j,1)} - \alpha^{(k)}_{(0,j,1),(j,1)}\Big)^2$$
$$+ \sum_{q\in[S]\backslash\{j\}} \Big(\alpha^{(k)}_{(p(q),q,0),(j,1)} + \alpha^{(k)}_{(\tilde{c}_1(q),q,1),(j,1)}\Big)^2 \bigg\}\bigg], \tag{140b}$$

$$m+2 \leqslant k \leqslant 2m: \quad \Delta_x^{(k)} = \frac{1}{2} \sum_{j=1}^{S} \mathbb{E}\bigg[ \mathbb{1}\{n_{-1}^{(k)} = j\}\bigg\{ \Big(1 - \alpha^{(k)}_{(p(j),j,0),(j,0)} - \alpha^{(k)}_{(\tilde{c}_1(j),j,1),(j,0)}\Big)^2$$
$$+ \sum_{q\in[S]\backslash\{j\}} \Big(\alpha^{(k)}_{(p(q),q,0),(j,0)} + \alpha^{(k)}_{(\tilde{c}_1(q),q,1),(j,0)}\Big)^2 \bigg\}\bigg]. \tag{140c}$$

*For $\Delta_y^{(k)}$, we have*

$$\forall k \in [m]: \Delta_y^{(k)} = \frac{1}{2} \sum_{j=1}^{S} \mathbb{E}\bigg[ \mathbb{1}\{n_{-1}^{(k)} = j\}\bigg\{ \Big(1 - \alpha^{(k)}_{(p(j),j,0),(j,1)} - \alpha^{(k)}_{(p(j),s(j),0),(j,1)}\Big)^2$$
$$+ \Big(\alpha^{(k)}_{(0,n_1,0),(n_{-1},1)} + \alpha^{(k)}_{(0,n_{2^m},1),(n_{-1},1)}\Big)^2$$
$$+ \sum_{\substack{q\in[S] \\ q\neq p(j)}} \Big(\alpha^{(k)}_{(q,c_1(q),0),(j,1)} + \alpha^{(k)}_{(q,c_2(q),0),(j,1)} + \alpha^{(k)}_{(q,p(q),1),(j,1)}\Big)^2 \bigg\}\bigg], \tag{141a}$$

$$k = m+1: \Delta_y^{(m+1)} = \frac{1}{2} \sum_{j=1}^{S} \mathbb{E}\bigg[ \mathbb{1}\{n_1 = j\}\bigg\{ \Big(1 - \alpha^{(k)}_{(n_2,c_1(n_2),0),(j,1)} - \alpha^{(k)}_{(n_2,c_2(n_2),0),(j,1)} - \alpha^{(k)}_{(n_2,j,1),(j,1)}\Big)^2$$
$$+ \Big(\alpha^{(k)}_{(0,j,0),(j,1)} + \alpha^{(k)}_{(0,n_{2^m},1),(j,1)}\Big)^2$$
$$+ \sum_{\substack{q\in[S] \\ q\neq n_2}} \Big(\alpha^{(k)}_{(q,c_1(q),0),(j,1)} + \alpha^{(k)}_{(q,c_2(q),0),(j,1)} + \alpha^{(k)}_{(q,p(q),1),(j,1)}\Big)^2 \bigg\}\bigg], \tag{141b}$$

$$m+2 \leqslant k \leqslant 2m: \Delta_y^{(k)} = \frac{1}{2} \sum_{i,j\in[S]} \mathbb{E}\bigg[ \mathbb{1}\{n_{-1}^{(k)} = j, c_1(j) = i\}\bigg\{ \Big(1 - \alpha^{(k)}_{(i,c_1(i),0),(j,0)} - \alpha^{(k)}_{(i,c_2(i),0),(j,0)} - \alpha^{(k)}_{(i,j,1),(j,0)}\Big)^2$$
$$+ \Big(\alpha^{(k)}_{(0,n_1,0),(j,0)} + \alpha^{(k)}_{(0,n_{2^m},1),(j,0)}\Big)^2$$
$$+ \sum_{\substack{q\in[S] \\ q\neq i}} \Big(\alpha^{(k)}_{(q,c_1(q),0),(j,0)} + \alpha^{(k)}_{(q,c_2(q),0),(j,0)} + \alpha^{(k)}_{(q,p(q),1),(j,0)}\Big)^2 \bigg\}\bigg]. \tag{141c}$$

*For $\Delta_z^{(k)}$, we have*

$$k \in [m]: \; \Delta_z^{(k)} = \frac{1}{2} \sum_{j=1}^{S} \mathbb{E}\left[ \mathbb{1}\{n_{-1}^{(k)} = j\}\left\{ \left( 1 - \sum_{q \in [S]} \beta_{(\tilde{c}_1(q),q,1),(j,1)}^{(k)} \right)^2 + \left( \sum_{q \in [S]} \beta_{(\tilde{p}(q),q,0),(j,1)}^{(k)} \right)^2 \right\} \right],$$

$$\text{(142a)}$$

$$k = m + 1: \; \Delta_z^{(m+1)} = \frac{1}{2} \sum_{j=1}^{S} \mathbb{E}\left[ \mathbb{1}\{n_1 = j\}\left\{ \left( 1 - \sum_{q \in [S]} \beta_{(\tilde{p}(q),q,0),(j,1)}^{(k)} \right)^2 + \left( \sum_{q \in [S]} \beta_{(\tilde{c}_1(q),q,1),(j,1)}^{(k)} \right)^2 \right\} \right],$$

$$\text{(142b)}$$

$$m + 2 \leqslant k \leqslant 2m: \; \Delta_z^{(k)} = \frac{1}{2} \sum_{j=1}^{S} \mathbb{E}\left[ \mathbb{1}\{n_1 = j\}\left\{ \left( 1 - \sum_{q \in [S]} \beta_{(\tilde{p}(q),q,0),(j,0)}^{(k)} \right)^2 + \left( \sum_{q \in [S]} \beta_{(\tilde{c}_1(q),q,1),(j,0)}^{(k)} \right)^2 \right\} \right].$$

$$\text{(142c)}$$

*Here, $n_{-1}^{(k)}$ denotes the node index of the embedding $x_{-1}^{(k-1)}$.*

We also provide a simplified expression of $\delta x_{l,i,j}^{(k)}$, $\delta y_{l,i,j}^{(k)}$, $\delta z_{l,i,j}^{(k)}$ in the following lemma, whose proof is given in Appendix C.2.5.

**Lemma 12** (Gradient expression). *We have for all $i \in \{0, \ldots, S\}$, $j \in [S]$, $k \in [2m]$, $\xi \in \{x, y\}$:*

$$\delta\xi_{1,i,j}^{(k)} = \frac{\partial\Delta_\xi^{(k)}}{\partial U_{1,i,j}} = \sum_{\substack{(p,q,u,v) \\ p \neq i}} \frac{\partial\Delta_\xi^{(k)}}{\partial\alpha_{(p,q,u),(j,v)}^{(k)}}\left( -\alpha_{(p,q,u),(j,v)}^{(k)} \sum_{\substack{s \in [S] \\ w \in \{0,1\}}} \alpha_{(i,s,w),(j,v)}^{(k)} \right)$$

$$+ \sum_{(q,u,v)} \frac{\partial\Delta_\xi^{(k)}}{\partial\alpha_{(i,q,u),(j,v)}^{(k)}}\alpha_{(i,q,u),(j,v)}^{(k)}\left( 1 - \sum_{\substack{s \in [S] \\ w \in \{0,1\}}} \alpha_{(i,s,w),(j,v)}^{(k)} \right), \qquad \text{(143a)}$$

$$\delta\xi_{2,i,j}^{(k)} = \frac{\partial\Delta_\xi^{(k)}}{\partial U_{2,i,j}} = \sum_{\substack{(p,q,u,v) \\ q \neq i}} \frac{\partial\Delta_\xi^{(k)}}{\partial\alpha_{(p,q,u),(j,v)}^{(k)}}\left( -\alpha_{(p,q,u),(j,v)}^{(k)} \sum_{\substack{r \in \{0,\ldots,S\} \\ w \in \{0,1\}}} \alpha_{(r,i,w),(j,v)}^{(k)} \right)$$

$$+ \sum_{(p,u,v)} \frac{\partial\Delta_\xi^{(k)}}{\partial\alpha_{(p,i,u),(j,v)}^{(k)}}\alpha_{(p,i,u),(j,v)}^{(k)}\left( 1 - \sum_{\substack{r \in \{0,\ldots,S\} \\ w \in \{0,1\}}} \alpha_{(r,i,w),(j,v)}^{(k)} \right), \qquad \text{(143b)}$$

*and for any $u, v \in \{0, 1\}$, we have*

$$\delta\xi_{3,u,v}^{(k)} = \frac{\partial\Delta_\xi^{(k)}}{\partial U_{3,u,v}} = \sum_{(p,q,j)} \frac{\partial\Delta_\xi^{(k)}}{\partial\alpha_{(p,q,u^c),(j,v)}^{(k)}}\left( -\alpha_{(p,q,u^c),(j,v)}^{(k)} \sum_{\substack{r \in \{0,\ldots,S\} \\ s \in [S]}} \alpha_{(r,s,u^c),(j,v)}^{(k)} \right)$$

$$+ \sum_{(p,q,j)} \frac{\partial\Delta_\xi^{(k)}}{\partial\alpha_{(p,q,u),(j,v)}^{(k)}}\alpha_{(p,q,u),(j,v)}^{(k)}\left( 1 - \sum_{\substack{r \in \{0,\ldots,S\} \\ s \in [S]}} \alpha_{(r,s,u),(j,v)}^{(k)} \right), \qquad \text{(143c)}$$

*where $u_c = 1 - u$. Similarly, we have*

$$\delta z_{1,i,j}^{(k)} = \frac{\partial\Delta_z^{(k)}}{\partial V_{1,i,j}} = \sum_{\substack{(p,q,u,v) \\ p \neq i}} \frac{\partial\Delta_z^{(k)}}{\partial\beta_{(p,q,u),(j,v)}^{(k)}}\left( -\beta_{(p,q,u),(j,v)}^{(k)} \sum_{\substack{s \in [S] \\ w \in \{0,1\}}} \beta_{(i,s,w),(j,v)}^{(k)} \right)$$

$$+ \sum_{(q,u,v)} \frac{\partial\Delta_z^{(k)}}{\partial\beta_{(i,q,u),(j,v)}^{(k)}}\beta_{(i,q,u),(j,v)}^{(k)}\left( 1 - \sum_{\substack{s \in [S] \\ w \in \{0,1\}}} \beta_{(i,s,w),(j,v)}^{(k)} \right), \qquad \text{(144a)}$$

$$\delta z_{2,i,j}^{(k)} = \frac{\partial\Delta_z^{(k)}}{\partial V_{2,i,j}} = \sum_{\substack{(p,q,u,v) \\ q \neq i}} \frac{\partial\Delta_z^{(k)}}{\partial\beta_{(p,q,u),(j,v)}^{(k)}}\left( -\beta_{(p,q,u),(j,v)}^{(k)} \sum_{\substack{r \in \{0,\ldots,S\} \\ w \in \{0,1\}}} \beta_{(r,i,w),(j,v)}^{(k)} \right)$$

$$+ \sum_{(p,u,v)} \frac{\partial \Delta_z^{(k)}}{\partial \beta_{(p,i,u),(j,v)}^{(k)}} \beta_{(p,i,u),(j,v)}^{(k)} \left( 1 - \sum_{\substack{r \in \{0,\dots,S\} \\ w \in \{0,1\}}} \beta_{(r,i,w),(j,v)}^{(k)} \right), \quad (144b)$$

*and for any $u, v \in \{0,1\}$, we have*

$$\delta z_{3,u,v}^{(k)} = \frac{\partial \Delta_z^{(k)}}{\partial V_{3,u,v}} = \sum_{(p,q,j)} \frac{\partial \Delta_z^{(k)}}{\partial \beta_{(p,q,u^c),(j,v)}^{(k)}} \left( - \beta_{(p,q,u^c),(j,v)}^{(k)} \sum_{\substack{r \in \{0,\dots,S\} \\ s \in [S]}} \beta_{(r,s,u^c),(j,v)}^{(k)} \right)$$

$$+ \sum_{(p,q,j)} \frac{\partial \Delta_z^{(k)}}{\partial \beta_{(p,q,u),(j,v)}^{(k)}} \beta_{(p,q,u),(j,v)}^{(k)} \left( 1 - \sum_{\substack{r \in \{0,\dots,S\} \\ s \in [S]}} \beta_{(r,s,u),(j,v)}^{(k)} \right). \quad (144c)$$

Define

$$w_0^{(k)} := \begin{cases} \sum_{q \in [S]} \beta_{(\widetilde{p}(q),q,0),(n_{-1},1)}^{(k)}, & k \in [m+1] \\ \sum_{q \in [S]} \beta_{(\widetilde{p}(q),q,0),(n_{-1},0)}^{(k)}, & k \in \{m+2,\dots,2m\} \end{cases}, \quad (145)$$

$$w_1^{(k)} := \begin{cases} \sum_{q \in [S]} \beta_{(\widetilde{c}_1(q),q,1),(n_{-1},1)}^{(k)}, & k \in [m+1] \\ \sum_{q \in [S]} \beta_{(\widetilde{c}_1(q),q,1),(n_{-1},0)}^{(k)}, & k \in \{m+2,\dots,2m\} \end{cases}, \quad (146)$$

which represent the attention put on the tokens with stage embedding $s_f$ and $s_b$ respectively at the $k$-th reasoning step. We have

$$\forall k \in [2m]: \quad w_0^{(k)} + w_1^{(k)} = 1. \quad (147)$$

We also define

$$p_0^{(k)} := \begin{cases} \beta_{(0,n_1,0),(n_{-1},1)}^{(k)}, & k \in [m+1] \\ \beta_{(0,n_1,0),(n_{-1},0)}^{(k)}, & k \in \{m+2,\dots,2m\} \end{cases}, \quad q_0^{(k)} := \begin{cases} \beta_{(0,n_{2m},1),(n_{-1},1)}^{(k)}, & k \in [m+1] \\ \beta_{(0,n_{2m},1),(n_{-1},0)}^{(k)}, & k \in \{m+2,\dots,2m\} \end{cases},$$
$$(148)$$

for all $i \in \{0,\dots,S\}$, we define

$$p_{1,i}^{(k)} := \begin{cases} \beta_{(i,\widetilde{c}_1(i),0),(n_{-1},1)}^{(k)}, & k \in [m+1] \\ \beta_{(i,\widetilde{c}_1(i),0),(n_{-1},0)}^{(k)}, & k \in \{m+2,\dots,2m\} \end{cases}, \quad p_{2,i}^{(k)} := \begin{cases} \beta_{(i,c_2(i),0),(n_{-1},1)}^{(k)}, & k \in [m+1] \\ \beta_{(i,c_2(i),0),(n_{-1},0)}^{(k)}, & k \in \{m+2,\dots,2m\} \end{cases},$$
$$(149)$$

and for any $i \in [S]$, we define

$$p_i^{(k)} := \begin{cases} \beta_{(\widetilde{p}(i),i,0),(n_{-1},1)}^{(k)}, & k \in [m+1] \\ \beta_{(\widetilde{p}(i),i,0),(n_{-1},0)}^{(k)}, & k \in \{m+2,\dots,2m\} \end{cases}, \quad q_i^{(k)} := \begin{cases} \beta_{(\widetilde{c}_1(i),i,1),(n_{-1},1)}^{(k)}, & k \in [m+1] \\ \beta_{(\widetilde{c}_1(i),i,1),(n_{-1},0)}^{(k)}, & k \in \{m+2,\dots,2m\} \end{cases}.$$
$$(150)$$

We further simplify $\delta z_l^{(k)}$ ($l \in \{1,2,3\}$) in the following lemma, whose proof is given in Appendix C.2.6.

**Lemma 13.** *We have*

$$\forall j \in [S]: \quad \delta z_{1,0,j}^{(k)} = \begin{cases} \frac{2}{S} \mathbb{E}\left[ \left(w_0^{(k)}\right)^2 w_1^{(k)} \left( -\frac{q_0^{(k)}}{w_1^{(k)}} + \frac{p_0^{(k)}}{w_0^{(k)}} \right) \Big| n_{-1} = j \right], & k \in [m] \\ \frac{2}{S} \mathbb{E}\left[ \left(w_1^{(k)}\right)^2 w_0^{(k)} \left( -\frac{p_0^{(k)}}{w_0^{(k)}} + \frac{q_0^{(k)}}{w_1^{(k)}} \right) \Big| n_{-1} = j \right], & k \in \{m+1,\dots,2m\} \end{cases};$$
$$(151)$$

$$\forall i,j \in [S]: \quad \delta z_{1,i,j}^{(k)} = \begin{cases} \frac{2}{S} \mathbb{E}\left[ \left(w_0^{(k)}\right)^2 w_1^{(k)} \left( -\frac{q_{p(i)}^{(k)}}{w_1^{(k)}} + \frac{p_{1,i}^{(k)}+p_{2,i}^{(k)}}{w_0^{(k)}} \right) \Big| n_{-1} = j \right], & k \in [m] \\ \frac{2}{S} \mathbb{E}\left[ \left(w_1^{(k)}\right)^2 w_0^{(k)} \left( -\frac{p_{1,i}^{(k)}+p_{2,i}^{(k)}}{w_0^{(k)}} + \frac{q_{p(i)}^{(k)}}{w_1^{(k)}} \right) \Big| n_{-1} = j \right], & k \in \{m+1,\dots,2m\} \end{cases};$$
$$(152)$$

$$\forall i,j \in [S]: \quad \delta z_{2,i,j}^{(k)} = \begin{cases} \frac{2}{S}\mathbb{E}\left[\left(w_0^{(k)}\right)^2 w_1^{(k)}\left(-\frac{q_i^{(k)}}{w_1^{(k)}} + \frac{p_i^{(k)}}{w_0^{(k)}}\right)\Big| n_{-1} = j\right], \quad k \in [m] \\ \frac{2}{S}\mathbb{E}\left[\left(w_1^{(k)}\right)^2 w_0^{(k)}\left(-\frac{p_i^{(k)}}{w_0^{(k)}} + \frac{q_i^{(k)}}{w_1^{(k)}}\right)\Big| n_{-1} = j\right], \quad k \in \{m+1,\ldots,2m\} \end{cases};$$

$$(153)$$

*and*

$$\delta z_{3,0,1}^{(k)} = -\delta z_{3,1,1}^{(k)} = \begin{cases} 2\mathbb{E}\left[\left(w_0^{(k)}\right)^2 w_1^{(k)}\right], & k \in [m] \\ 2\mathbb{E}\left[-\left(w_1^{(k)}\right)^2 w_0^{(k)}\right], & k = m+1 \\ 0, & k \in \{m+2,\ldots,2m\} \end{cases}, \qquad (154)$$

$$\delta z_{3,0,0}^{(k)} = -\delta z_{3,1,0}^{(k)} = \begin{cases} 0, & k \in [m+1] \\ 2\mathbb{E}\left[-\left(w_1^{(k)}\right)^2 w_0^{(k)}\right], & k \in \{m+2,\ldots,2m\} \end{cases}. \qquad (155)$$

*We also have*

$$\forall i,j \in \{0,\ldots,S\}: \quad \delta z_{1,i,0}^{(k)} = \delta z_{2,0,j}^{(k)} = 0. \qquad (156)$$

We next give a lemma that simplifies the gradient expressions of $\delta x_l^{(k)}$ and $\delta y_l^{(k)}$, for $l = 1,2,3$.

**Lemma 14.** *For any $j \in [S]$, we have*

$$\delta x_{1,0,j}^{(1)} = \frac{1}{S}\mathbb{E}\Big[ -\left(1 - \alpha_{(p(j),j,0),(j,1)}^{(1)} - \alpha_{(0,j,1),(j,1)}^{(1)}\right)\alpha_{(0,j,1),(j,1)}^{(1)}\left(1 - \alpha_{(0,n_1,0),(j,1)}^{(1)} - \alpha_{(0,j,1),(j,1)}^{(1)}\right)$$

$$+ \left(\alpha_{(0,n_1,0),(j,1)}^{(1)}\right)^2\left(1 - \alpha_{(0,n_1,0),(j,1)}^{(1)} - \alpha_{(0,j,1),(j,1)}^{(1)}\right)$$

$$+ \left(1 - \alpha_{(p(j),j,0),(j,1)}^{(1)} - \alpha_{(0,j,1),(j,1)}^{(1)}\right)\alpha_{(p(j),j,0),(j,1)}^{(1)}\left(\alpha_{(0,n_1,0),(j,1)}^{(1)} + \alpha_{(0,j,1),(j,1)}^{(1)}\right)$$

$$- \sum_{q\in[S]\setminus\{j,n_1\}}\left(\alpha_{(p(q),q,0),(j,1)}^{(1)} + \alpha_{(\tilde{c}_1(q),q,1),(j,1)}^{(1)}\right)^2\left(\alpha_{(0,n_1,0),(j,1)}^{(1)} + \alpha_{(0,j,1),(j,1)}^{(1)}\right)\Big| n_{2m} = j\Big]. \qquad (157)$$

*For $k \in \{2,\ldots,m\}$, we have*

$$\delta x_{1,0,j}^{(k)} = \frac{1}{S}\mathbb{E}\Big[ \left(\alpha_{(0,n_1,1),(j,1)}^{(k)}\right)^2\left(1 - \alpha_{(0,n_1,1),(j,1)}^{(k)} - \alpha_{(0,n_{2m},1),(j,1)}^{(k)}\right)$$

$$+ \left(\alpha_{(p(n_{2m}),n_{2m},1),(j,1)}^{(k)} + \alpha_{(0,n_{2m},1),(j,1)}^{(k)}\right)\alpha_{(p(n_{2m}),n_{2m},1),(j,1)}^{(k)}\left(1 - \alpha_{(0,n_1,0),(j,1)}^{(k)} - \alpha_{(0,n_{2m},1),(j,1)}^{(k)}\right)$$

$$- \left(\alpha_{(p(n_{2m}),n_{2m},0),(j,1)}^{(k)} + \alpha_{(0,n_{2m},1),(j,1)}^{(k)}\right)\alpha_{(p(n_{2m}),n_{2m},0),(j,1)}^{(k)}\left(\alpha_{(0,n_1,0),(j,1)}^{(k)} + \alpha_{(0,n_{2m},1),(j,1)}^{(k)}\right)$$

$$+ \left(1 - \alpha_{(p(j),j,0),(j,1)}^{(k)} - \alpha_{(\tilde{c}_1(j),j,1),(j,1)}^{(k)}\right)\left(\alpha_{(p(j),j,0),(j,1)}^{(k)} + \alpha_{(\tilde{c}_1(j),j,1),(j,1)}^{(k)}\right)\left(\alpha_{(0,n_1,0),(j,1)}^{(k)} + \alpha_{(0,n_{2m},1),(j,1)}^{(k)}\right)$$

$$- \sum_{\substack{q\in[S]\\q\neq n_1,n_{2m},j}}\left(\alpha_{(p(q),q,0),(j,1)}^{(k)} + \alpha_{(\tilde{c}_1(q),q,1),(j,1)}^{(k)}\right)^2\left(\alpha_{(0,n_1,0),(j,1)}^{(k)} + \alpha_{(0,n_{2m},1),(j,1)}^{(k)}\right)\Big| n_{-1} = j\Big], \qquad (158)$$

*for $k = m+1$, we have*

$$\delta x_{1,0,j}^{(m+1)} = \frac{1}{S}\mathbb{E}\Big[ -\left(1 - \alpha_{(0,j,0),(j,1)}^{(m+1)} - \alpha_{(\tilde{c}_1(j),j,1),(j,1)}^{(m+1)}\right)\alpha_{(0,j,0),(j,1)}^{(m+1)}\left(1 - \alpha_{(0,j,0),(j,1)}^{(m+1)} - \alpha_{(0,n_{2m},1),(j,1)}^{(m+1)}\right)$$

$$+ \left(\alpha_{(p(n_{2m}),n_{2m},0),(j,1)}^{(m+1)} + \alpha_{(0,n_{2m},1),(j,1)}^{(m+1)}\right)\alpha_{(p(n_{2m}),n_{2m},0),(j,1)}^{(m+1)}\left(1 - \alpha_{(0,j,0),(j,1)}^{(m+1)} - \alpha_{(0,n_{2m},1),(j,1)}^{(m+1)}\right)$$

$$+ \left(1 - \alpha_{(0,j,0),(j,1)}^{(m+1)} - \alpha_{(\tilde{c}_1(j),j,1),(j,1)}^{(m+1)}\right)\alpha_{(\tilde{c}_1(j),j,1),(j,1)}^{(m+1)}\left(\alpha_{(0,j,0),(j,1)}^{(m+1)} + \alpha_{(0,n_{2m},1),(j,1)}^{(m+1)}\right)$$

$$- \left(\alpha_{(p(n_{2m}),n_{2m},0),(j,1)}^{(m+1)} + \alpha_{(0,n_{2m},1),(j,1)}^{(m+1)}\right)\alpha_{(p(n_{2m}),n_{2m},0),(j,1)}^{(m+1)}\left(\alpha_{(0,j,0),(j,1)}^{(m+1)} + \alpha_{(0,n_{2m},1),(j,1)}^{(m+1)}\right)$$

$$- \sum_{\substack{q\in[S]\\q\neq n_{2m},j}}\left(\alpha_{(p(q),q,0),(j,1)}^{(m+1)} + \alpha_{(\tilde{c}_1(q),q,1),(j,1)}^{(m+1)}\right)^2\left(\alpha_{(0,j,0),(j,1)}^{(m+1)} + \alpha_{(0,n_{2m},1),(j,1)}^{(m+1)}\right)\Big| n_{-1} = j\Big], \qquad (159)$$

*for $k \in \{m+2, \ldots, 2m\}$, we have*

$$\delta x_{1,0,j}^{(k)} = \frac{1}{S}\mathbb{E}\Bigg[ \left( \alpha_{(0,n_1,0),(j,0)}^{(k)} + \alpha_{(n_2,n_1,1),(j,0)}^{(k)} \right) \alpha_{(0,n_1,0),(j,0)}^{(k)} \left( 1 - \alpha_{(0,n_1,0),(j,0)}^{(k)} - \alpha_{(0,n_{2m},1),(j,0)}^{(k)} \right)$$

$$+ \left( \alpha_{(0,n_{2m},1),(j,0)}^{(k)} + \alpha_{(p(n_{2m}),n_{2m},0),(j,0)}^{(k)} \right) \alpha_{(0,n_{2m},1),(j,0)}^{(k)} \left( 1 - \alpha_{(0,n_1,0),(j,0)}^{(k)} - \alpha_{(0,n_{2m},1),(j,0)}^{(k)} \right)$$

$$+ \left( 1 - \alpha_{(p(j),j,0),(j,0)}^{(k)} - \alpha_{(\tilde{c}_1(j),j,1),(j,0)}^{(k)} \right) \left( \alpha_{(p(j),j,0),(j,0)}^{(k)} + \alpha_{(\tilde{c}_1(j),j,1),(j,0)}^{(k)} \right) \left( \alpha_{(0,n_1,0),(j,0)}^{(k)} + \alpha_{(0,n_{2m},1),(j,0)}^{(k)} \right)$$

$$- \left( \alpha_{(0,n_1,0),(j,0)}^{(k)} + \alpha_{(n_2,n_1,1),(j,0)}^{(k)} \right) \alpha_{(n_2,n_1,1),(j,0)}^{(k)} \left( \alpha_{(0,n_1,0),(j,0)}^{(k)} + \alpha_{(0,n_{2m},1),(j,0)}^{(k)} \right)$$

$$- \left( \alpha_{(0,n_{2m},1),(j,0)}^{(k)} + \alpha_{(p(n_{2m}),n_{2m},0),(j,0)}^{(k)} \right) \alpha_{(p(n_{2m}),n_{2m},0),(j,0)}^{(k)} \left( \alpha_{(0,n_1,0),(j,0)}^{(k)} + \alpha_{(0,n_{2m},1),(j,0)}^{(k)} \right)$$

$$- \sum_{\substack{q \in [S] \\ q \neq n_1, n_{2m}, j}} \left( \alpha_{(p(q),q,0),(j,0)}^{(k)} + \alpha_{(\tilde{c}_1(q),q,1),(j,0)}^{(k)} \right)^2 \left( \alpha_{(0,n_1,0),(j,0)}^{(k)} + \alpha_{(0,n_{2m},1),(j,0)}^{(k)} \right) \Bigg| n_{-1} = j \Bigg].$$

(160)

*For all $k \in [m]$, we have*

$$\delta y_{1,0,j}^{(k)} = \frac{1}{S}\mathbb{E}\Bigg[ \left( \alpha_{(0,n_1,0),(j,1)}^{(k)} + \alpha_{(0,n_{2m},1),(j,1)}^{(k)} \right)^2 \left( 1 - \alpha_{(0,n_1,0),(j,1)}^{(k)} - \alpha_{(0,n_{2m},1),(j,1)}^{(k)} \right)$$

$$+ \left( 1 - \alpha_{(p(j),j,0),(j,1)}^{(k)} - \alpha_{(p(j),s(j),0),(j,1)}^{(k)} \right) \left( \alpha_{(p(j),j,0),(j,1)}^{(k)} + \alpha_{(p(j),s(j),0),(j,1)}^{(k)} \right) \left( \alpha_{(0,n_1,0),(j,1)}^{(k)} + \alpha_{(0,n_{2m},1),(j,1)}^{(k)} \right)$$

$$- \sum_{\substack{q \in [S] \\ q \neq p(j)}} \left( \alpha_{(q,c_1(q),0),(j,1)}^{(k)} + \alpha_{(q,\tilde{c}_2(q),0),(j,1)}^{(k)} + \alpha_{(q,p(q),1),(j,1)}^{(k)} \right)^2 \left( \alpha_{(0,n_1,0),(j,1)}^{(k)} + \alpha_{(0,n_{2m},1),(j,1)}^{(k)} \right) \Bigg| n_{-1} = j \Bigg],$$

(161)

*for $k = m + 1$, we have*

$$\delta y_{1,0,j}^{(m+1)} = \frac{1}{S}\mathbb{E}\Bigg[ \left( \alpha_{(0,j,0),(j,1)}^{(m+1)} + \alpha_{(0,n_{2m},1),(j,1)}^{(m+1)} \right)^2 \left( 1 - \alpha_{(0,j,0),(j,1)}^{(m+1)} - \alpha_{(0,n_{2m},1),(j,1)}^{(m+1)} \right)$$

$$+ \left( 1 - \alpha_{(n_2,c_1(n_2),0),(j,1)}^{(m+1)} - \alpha_{(n_2,c_1(n_2),0),(j,1)}^{(m+1)} - \alpha_{(n_2,j,1),(j,1)}^{(m+1)} \right)$$

$$\cdot \left( \alpha_{(n_2,c_1(n_2),0),(j,1)}^{(m+1)} + \alpha_{(n_2,c_2(n_2),0),(j,1)}^{(m+1)} + \alpha_{(n_2,j,1),(j,1)}^{(m+1)} \right) \left( \alpha_{(0,j,0),(j,1)}^{(m+1)} + \alpha_{(0,n_{2m},1),(j,1)}^{(m+1)} \right)$$

$$- \sum_{\substack{q \in [S] \\ q \neq n_2}} \left( \alpha_{(q,c_1(q),0),(j,1)}^{(m+1)} + \alpha_{(q,c_2(q),0),(j,1)}^{(m+1)} + \alpha_{(q,p(q),1),(j,1)}^{(m+1)} \right)^2 \left( \alpha_{(0,j,0),(j,1)}^{(m+1)} + \alpha_{(0,n_{2m},1),(j,1)}^{(m+1)} \right) \Bigg| n_{-1} = j \Bigg],$$

(162)

*and for all $k \in \{m+2, \cdots, 2m\}$, we have*

$$\delta y_{1,0,j}^{(k)} = \frac{1}{S}\sum_{i \in [S]}\mathbb{E}\Bigg[ \mathbb{1}\{n_{2^{k-m}} = i\}\bigg\{ \left( \alpha_{(0,n_1,0),(j,0)}^{(k)} + \alpha_{(0,n_{2m},1),(j,0)}^{(k)} \right)^2 \left( 1 - \alpha_{(0,n_1,0),(j,0)}^{(k)} - \alpha_{(0,n_{2m},1),(j,0)}^{(k)} \right)$$

$$+ \left( 1 - \alpha_{(i,c_1(i),0),(j,0)}^{(k)} - \alpha_{(i,c_2(i),0),(j,0)}^{(k)} - \alpha_{(i,j,1),(j,0)}^{(k)} \right)$$

$$\cdot \left( \alpha_{(i,c_1(i),0),(j,0)}^{(k)} + \alpha_{(i,c_2(i),0),(j,0)}^{(k)} + \alpha_{(i,j,1),(j,0)}^{(k)} \right) \left( \alpha_{(0,n_1,0),(j,0)}^{(k)} + \alpha_{(0,n_{2m},1),(j,0)}^{(k)} \right)$$

$$- \sum_{\substack{q \in [S] \\ q \neq n_1, n_{2m}}} \left( \alpha_{(q,c_1(q),0),(j,0)}^{(k)} + \alpha_{(q,c_2(q),0),(j,0)}^{(k)} + \alpha_{(q,p(q),1),(j,0)}^{(k)} \right)^2 \left( \alpha_{(0,n_1,0),(j,0)}^{(k)} + \alpha_{(0,n_{2m},1),(j,0)}^{(k)} \right) \bigg\} \Bigg| n_{-1} = j \Bigg].$$

(163)

*For all $k \in [m]$, we have*

$$\delta x_{2,j,j}^{(k)} = \frac{1}{S}\mathbb{E}\Bigg[ - \left( 1 - \alpha_{(p(j),j,0),(j,1)}^{(k)} - \alpha_{(\tilde{c}_1(j),j,1),(j,1)}^{(k)} \right)^2 \left( \alpha_{(p(j),j,0),(j,1)}^{(k)} + \alpha_{(\tilde{c}_1(j),j,1),(j,1)}^{(k)} \right)$$

$$- \sum_{\substack{q \in [S] \\ q \neq j}} \left( \alpha_{(p(q),q,0),(j,1)}^{(k)} + \alpha_{(\tilde{c}_1(q),q,1),(j,1)}^{(k)} \right)^2 \left( \alpha_{(p(j),j,0),(j,1)}^{(k)} + \alpha_{(\tilde{c}_1(j),j,1),(j,1)}^{(k)} \right) \Bigg| n_{-1} = j \Bigg],$$

(164)

*for $k = m + 1$, we have*

$$\delta x_{2,j,j}^{(m+1)} = \frac{1}{S} \mathbb{E}\Bigg[ - \left( 1 - \alpha_{(0,j,0),(j,1)}^{(m+1)} - \alpha_{(\tilde{c}_1(j),j,1),(j,1)}^{(m+1)} \right)^2 \left( \alpha_{(0,j,0),(j,1)}^{(m+1)} + \alpha_{(\tilde{c}_1(j),j,1),(j,1)}^{(m+1)} \right)$$

$$- \sum_{\substack{q \in [S] \\ q \neq j}} \left( \alpha_{(p(q),q,0),(j,1)}^{(m+1)} + \alpha_{(\tilde{c}_1(q),q,1),(j,1)}^{(m+1)} \right)^2 \left( \alpha_{(0,j,0),(j,1)}^{(m+1)} + \alpha_{(\tilde{c}_1(j),j,1),(j,1)}^{(m+1)} \right) \Bigg| n_{-1} = j \Bigg],$$

(165)

*and for all $k \in \{m + 2, \cdots, 2m\}$, we have*

$$\delta x_{2,j,j}^{(k)} = \frac{1}{S} \mathbb{E}\Bigg[ - \left( 1 - \alpha_{(p(j),j,0),(j,0)}^{(k)} - \alpha_{(\tilde{c}_1(j),j,1),(j,0)}^{(k)} \right)^2 \left( \alpha_{(p(j),j,0),(j,0)}^{(k)} + \alpha_{(\tilde{c}_1(j),j,1),(j,0)}^{(k)} \right)$$

$$- \sum_{\substack{q \in [S] \\ q \neq j}} \left( \alpha_{(p(q),q,0),(j,0)}^{(k)} + \alpha_{(\tilde{c}_1(q),q,1),(j,0)}^{(k)} \right)^2 \left( \alpha_{(p(j),j,0),(j,0)}^{(k)} + \alpha_{(\tilde{c}_1(j),j,1),(j,0)}^{(k)} \right) \Bigg| n_{-1} = j \Bigg].$$

(166)

*For $k = 1$, we have*

$$\delta y_{2,j,j}^{(1)} = \frac{1}{S} \mathbb{E}\Bigg[ - \left( 1 - \alpha_{(p(j),j,0),(j,1)}^{(1)} - \alpha_{(p(j),s(j),1),(j,1)}^{(1)} \right) \alpha_{(p(j),j,0),(j,1)}^{(1)} \left( 1 - \alpha_{(p(j),j,0),(j,1)}^{(1)} - \alpha_{(0,j,1),(j,1)}^{(1)} \right)$$

$$+ \left( \alpha_{(0,n_1,0),(j,1)}^{(1)} + \alpha_{(0,j,1),(j,1)}^{(1)} \right) \alpha_{(0,j,0),(j,1)}^{(1)} \left( 1 - \alpha_{(p(j),j,0),(j,1)}^{(1)} - \alpha_{(0,j,1),(j,1)}^{(1)} \right)$$

$$+ \left( 1 - \alpha_{(p(j),j,0),(j,1)}^{(1)} - \alpha_{(p(j),s(j),0),(j,1)}^{(1)} \right) \alpha_{(p(j),j,0),(j,1)}^{(1)} \left( \alpha_{(p(j),j,0),(j,1)}^{(1)} + \alpha_{(0,j,1),(j,1)}^{(1)} \right)$$

$$- \left( \alpha_{(0,n_1,0),(j,1)}^{(1)} + \alpha_{(0,j,1),(j,1)}^{(1)} \right) \alpha_{(0,n_1,0),(j,1)}^{(1)} \left( \alpha_{(p(j),j,0),(j,1)}^{(1)} + \alpha_{(0,j,1),(j,1)}^{(1)} \right)$$

$$- \sum_{\substack{q \in [S] \\ q \neq p(j)}} \left( \alpha_{(q,c_1(q),0),(j,1)}^{(1)} + \alpha_{(q,c_2(q),0),(j,1)}^{(1)} \right)^2 \left( \alpha_{(p(j),j,0),(j,1)}^{(1)} + \alpha_{(0,j,1),(j,1)}^{(1)} \right) \Bigg| n_{-1} = j \Bigg],$$

(167)

*for all $k \in \{2, \cdots, m\}$, we have*

$$\delta y_{2,j,j}^{(k)} = \frac{1}{S} \mathbb{E}\Bigg[ - \left( 1 - \alpha_{(p(j),j,0),(j,1)}^{(k)} - \alpha_{(p(j),s(j),0),(j,1)}^{(k)} \right) \alpha_{(p(j),j,0),(j,1)}^{(k)} \left( 1 - \alpha_{(p(j),j,0),(j,1)}^{(k)} - \alpha_{(\tilde{c}_1(j),j,1),(j,1)}^{(k)} \right)$$

$$+ \left( \alpha_{(c_1(j),j,1),(j,1)}^{(k)} + \alpha_{(c_1(j),c_1^2(j),0),(j,1)}^{(k)} + \alpha_{(c_1(j),c_2(c_1(j)),0),(j,1)}^{(k)} \right)$$

$$\cdot \alpha_{(c_1(j),j,1),(j,1)}^{(k)} \left( 1 - \alpha_{(p(j),j,0),(j,1)}^{(k)} - \alpha_{(\tilde{c}_1(j),j,1),(j,1)}^{(k)} \right)$$

$$+ \left( 1 - \alpha_{(p(j),j,0),(j,1)}^{(k)} - \alpha_{(p(j),s(j),0),(j,1)}^{(k)} \right) \alpha_{(p(j),s(j),0),(j,1)}^{(k)} \left( \alpha_{(p(j),j,0),(j,1)}^{(k)} + \alpha_{(\tilde{c}_1(j),j,1),(j,1)}^{(k)} \right)$$

$$- \left( \alpha_{(0,n_1,0),(j,1)}^{(k)} + \alpha_{(0,n_{2m},1),(j,1)}^{(k)} \right)^2 \left( \alpha_{(p(j),j,0),(j,1)}^{(k)} + \alpha_{(\tilde{c}_1(j),j,1),(j,1)}^{(k)} \right)$$

$$- \left( \alpha_{(c_1(j),j,1),(j,1)}^{(k)} + \alpha_{(c_1(j),c_1^2(j),0),(j,1)}^{(k)} + \alpha_{(c_1(j),c_2(c_1(j)),0),(j,1)}^{(k)} \right)$$

$$\cdot \left( \alpha_{(c_1(j),c_1^2(j),0),(j,1)}^{(k)} + \alpha_{(c_1(j),c_2(c_1(j)),0),(j,1)}^{(k)} \right) \left( \alpha_{(p(j),j,0),(j,1)}^{(k)} + \alpha_{(\tilde{c}_1(j),j,1),(j,1)}^{(k)} \right)$$

$$- \sum_{\substack{q \in [S] \\ q \neq p(j), c_1(j)}} \left( \alpha_{(q,c_1(q),0),(j,1)}^{(k)} + \alpha_{(q,c_2(q),0),(j,1)}^{(k)} + \alpha_{(q,p(q),1),(j,1)}^{(k)} \right)^2$$

$$\cdot \left( \alpha_{(p(j),j,0),(j,1)}^{(k)} + \alpha_{(\tilde{c}_1(j),j,1),(j,1)}^{(k)} \right) \Bigg| n_{-1} = j \Bigg],$$

(168)

*for $k = m + 1$, we have*

$$\delta y_{2,j,j}^{(m+1)} = \frac{1}{S} \mathbb{E}\Bigg[ - \left( 1 - \alpha_{(n_2,c_1(n_2),0),(j,1)}^{(m+1)} - \alpha_{(n_2,c_2(n_2),0),(j,1)}^{(m+1)} - \alpha_{(n_2,j,1),(j,1)}^{(m+1)} \right) \alpha_{(n_2,j,1),(j,1)}^{(m+1)}$$

$$\cdot \left(1 - \alpha_{(0,n_2,0),(j,1)}^{(m+1)} - \alpha_{(n_2,j,1),(j,1)}^{(m+1)}\right)$$

$$+ \left(\alpha_{(0,j,0),(j,1)}^{(m+1)} + \alpha_{(0,n_2^m,1),(j,1)}^{(m+1)}\right) \alpha_{(0,j,0),(j,1)}^{(m+1)} \left(1 - \alpha_{(0,n_2,0),(j,1)}^{(m+1)} - \alpha_{(n_2,j,1),(j,1)}^{(m+1)}\right)$$

$$+ \left(1 - \alpha_{(n_2,c_1(n_2),0),(j,1)}^{(m+1)} - \alpha_{(n_2,c_2(n_2),0),(j,1)}^{(m+1)} - \alpha_{(n_2,j,1),(j,1)}^{(m+1)}\right) \left(\alpha_{(n_2,c_1(n_2),0),(j,1)}^{(m+1)} + \alpha_{(n_2,c_2(n_2),0),(j,1)}^{(m+1)}\right)$$

$$\cdot \left(\alpha_{(0,j,0),(j,1)}^{(m+1)} + \alpha_{(n_2,j,1),(j,1)}^{(m+1)}\right)$$

$$- \left(\alpha_{(0,j,0),(j,1)}^{(m+1)} + \alpha_{(0,n_2^m,1),(j,1)}^{(m+1)}\right) \alpha_{(0,n_2^m,1),(j,1)}^{(m+1)} \left(\alpha_{(0,j,0),(j,1)}^{(m+1)} + \alpha_{(n_2,j,1),(j,1)}^{(m+1)}\right)$$

$$- \sum_{\substack{q\in[S]\\ q\neq n_2}} \left(\alpha_{(q,c_1(q),0),(j,1)}^{(m+1)} + \alpha_{(q,c_2(q),0),(j,1)}^{(m+1)} + \alpha_{(q,p(q),1),(j,1)}^{(m+1)}\right)^2 \left(\alpha_{(0,j,0),(j,1)}^{(m+1)} + \alpha_{(n_2,j,1),(j,1)}^{(m+1)}\right) \Bigg| n_{-1} = j \Bigg],$$

$$\tag{169}$$

and for $k \in \{m+2, \cdots, 2m\}$, we have

$$\delta y_{2,j,j}^{(k)}$$

$$= \frac{1}{S} \sum_{i\in[S]} \mathbb{E}\Bigg[ \mathbb{1}\{n_{2k-m} = i\}\Bigg\{ - \left(1 - \alpha_{(i,c_1(i),0),(j,0)}^{(k)} - \alpha_{(i,c_2(i),0),(j,0)}^{(k)} - \alpha_{(i,j,1),(j,0)}^{(k)}\right) \alpha_{(i,j,1),(j,0)}^{(k)}$$

$$\cdot \left(1 - \alpha_{(i,j,1),(j,0)}^{(k)} - \alpha_{(p(j),j,0),(j,0)}^{(k)}\right)$$

$$+ \left(\alpha_{(p(j),j,0),(j,0)}^{(k)} + \alpha_{(p(j),s(j),0),(j,0)}^{(k)} + \alpha_{(p(j),p^2(j),1),(j,0)}^{(k)}\right) \alpha_{(p(j),j,0),(j,0)}^{(k)} \left(1 - \alpha_{(i,j,1),(j,0)}^{(k)} - \alpha_{(p(j),j,1),(j,0)}^{(k)}\right)$$

$$+ \left(1 - \alpha_{(i,c_1(i),0),(j,0)}^{(k)} - \alpha_{(i,c_2(i),0),(j,0)}^{(k)} - \alpha_{(i,j,1),(j,0)}^{(k)}\right) \left(\alpha_{(i,c_1(i),0),(j,0)}^{(k)} + \alpha_{(i,c_2(i),0),(j,0)}^{(k)}\right)$$

$$\cdot \left(\alpha_{(i,j,1),(j,0)}^{(k)} + \alpha_{(p(j),j,0),(j,0)}^{(k)}\right)$$

$$- \left(\alpha_{(0,n_1,0),(j,0)}^{(k)} + \alpha_{(0,n_2^m,1),(j,0)}^{(k)}\right)^2 \left(\alpha_{(i,j,1),(j,0)}^{(k)} + \alpha_{(p(j),j,0),(j,0)}^{(k)}\right)$$

$$- \left(\alpha_{(p(j),j,0),(j,0)}^{(k)} + \alpha_{(p(j),s(j),0),(j,0)}^{(k)} + \alpha_{(p(j),p^2(j),1),(j,0)}^{(k)}\right) \left(\alpha_{(i,j,1),(j,0)}^{(k)} + \alpha_{(p(j),j,0),(j,0)}^{(k)}\right)$$

$$- \sum_{\substack{q\in[S]\\ q\neq p(j)}} \left(\alpha_{(q,c_1(q),0),(j,0)}^{(k)} + \alpha_{(q,c_2(q),0),(j,0)}^{(k)} + \alpha_{(q,p(q),1),(j,0)}^{(k)}\right)^2 \left(\alpha_{(i,j,1),(j,0)}^{(k)} + \alpha_{(p(j),j,0),(j,0)}^{(k)}\right) \Bigg\}\Bigg| n_{-1} = j \Bigg].$$

$$\tag{170}$$

For $k \in [m]$, we have

$$\delta x_{3,0,0}^{(k)} = -\delta x_{3,1,0}^{(k)} = 0, \tag{171}$$

and

$$\delta x_{3,0,1}^{(k)} = -\delta x_{3,1,1}^{(k)}$$

$$= \frac{1}{S} \mathbb{E}\Bigg[ - \left(1 - \alpha_{(p(j),j,0),(j,1)}^{(k)} - \alpha_{(c_1(j),j,1),(j,1)}^{(k)}\right) \alpha_{(p(j),j,0),(j,1)}^{(k)} \left(1 - \sum_{q\in[S]} \alpha_{(\tilde{p}(q),q,0),(j,1)}^{(k)}\right)$$

$$+ \sum_{q\in[S]\setminus\{j\}} \left(\alpha_{(\tilde{p}(q),q,0),(j,1)}^{(k)} + \alpha_{(\tilde{c}_1(q),q,1),(j,1)}^{(k)}\right) \alpha_{(\tilde{p}(q),q,0),(j,1)}^{(k)} \alpha_{(\tilde{c}_1(q),q,1),(j,1)}^{(k)} \left(1 - \sum_{q\in[S]} \alpha_{(\tilde{p}(q),q,0),(j,1)}^{(k)}\right)$$

$$+ \left(1 - \alpha_{(p(j),j,0),(j,1)}^{(k)} - \alpha_{(c_1(j),j,1),(j,1)}^{(k)}\right) \alpha_{(c_1(j),j,1),(j,1)}^{(k)} \left(\sum_{q\in[S]} \alpha_{(\tilde{p}(q),q,0),(j,1)}^{(k)}\right)$$

$$- \sum_{q\in[S]\setminus\{j\}} \left(\alpha_{(p(q),q,0),(j,1)}^{(k)} + \alpha_{(c_1(q),q,1),(j,1)}^{(k)}\right) \alpha_{(c_1(j),j,1),(j,1)}^{(k)} \left(\sum_{q\in[S]} \alpha_{(\tilde{p}(q),q,0),(j,1)}^{(k)}\right) \Bigg| n_{-1} = j \Bigg].$$

$$\tag{172}$$

For $k = m+1$, we have

$$\delta x_{3,0,0}^{(m+1)} = -\delta x_{3,1,0}^{(m+1)} = 0, \tag{173}$$

*and*

$$\delta x_{3,0,1}^{(m+1)} = -\delta x_{3,1,1}^{(m+1)}$$

$$= \frac{1}{S}\mathbb{E}\Bigg[ -\left(1 - \alpha_{(0,j,0),(j,1)}^{(m+1)} - \alpha_{(c_1(j),j,1),(j,1)}^{(m+1)}\right)\alpha_{(0,j,0),(j,1)}^{(m+1)}\left(1 - \sum_{q\in[S]}\alpha_{(\widetilde{p}(q),q,0),(j,1)}^{(m+1)}\right)$$

$$+ \sum_{q\in[S]\setminus\{j\}}\left(\alpha_{(\widetilde{p}(q),q,0),(j,1)}^{(m+1)} + \alpha_{(\widetilde{c}_1(q),q,1),(j,1)}^{(m+1)}\right)\alpha_{(\widetilde{p}(q),q,0),(j,1)}^{(m+1)}\alpha_{(\widetilde{c}_1(q),q,1),(j,1)}^{(m+1)}\left(1 - \sum_{q\in[S]}\alpha_{(\widetilde{p}(q),q,0),(j,1)}^{(m+1)}\right)$$

$$+ \left(1 - \alpha_{(0,j,0),(j,1)}^{(m+1)} - \alpha_{(c_1(j),j,1),(j,1)}^{(m+1)}\right)\alpha_{(c_1(j),j,1),(j,1)}^{(m+1)}\left(\sum_{q\in[S]}\alpha_{(\widetilde{p}(q),q,0),(j,1)}^{(m+1)}\right)$$

$$- \sum_{q\in[S]\setminus\{j\}}\left(\alpha_{(p(q),q,0),(j,1)}^{(m+1)} + \alpha_{(c_1(q),q,1),(j,1)}^{(m+1)}\right)\alpha_{(c_1(q),q,1),(j,1)}^{(m+1)}\left(\sum_{q\in[S]}\alpha_{(\widetilde{p}(q),q,0),(j,1)}^{(m+1)}\right)\Bigg|n_{-1} = j\Bigg]. \tag{174}$$

*For $k \in \{m+2, \cdots, 2m\}$, we have*

$$\delta x_{3,0,1}^{(k)} = -\delta x_{3,1,1}^{(k)} = 0, \tag{175}$$

*and*

$$\delta x_{3,0,0}^{(k)} = -\delta x_{3,1,0}^{(k)}$$

$$= \frac{1}{S}\mathbb{E}\Bigg[ -\left(1 - \alpha_{(p(j),j,0),(j,0)}^{(k)} - \alpha_{(c_1(j),j,1),(j,0)}^{(k)}\right)\alpha_{(p(j),j,0),(j,0)}^{(k)}\left(1 - \sum_{q\in[S]}\alpha_{(\widetilde{p}(q),q,0),(j,0)}^{(k)}\right)$$

$$+ \sum_{q\in[S]\setminus\{j\}}\left(\alpha_{(\widetilde{p}(q),q,0),(j,0)}^{(k)} + \alpha_{(\widetilde{c}_1(q),q,1),(j,0)}^{(k)}\right)\alpha_{(\widetilde{p}(q),q,0),(j,0)}^{(k)}\alpha_{(\widetilde{c}_1(q),q,1),(j,0)}^{(k)}\left(1 - \sum_{q\in[S]}\alpha_{(\widetilde{p}(q),q,0),(j,0)}^{(k)}\right)$$

$$+ \left(1 - \alpha_{(p(j),j,0),(j,0)}^{(k)} - \alpha_{(c_1(j),j,1),(j,0)}^{(k)}\right)\alpha_{(c_1(j),j,1),(j,0)}^{(k)}\left(\sum_{q\in[S]}\alpha_{(\widetilde{p}(q),q,0),(j,0)}^{(k)}\right)$$

$$- \sum_{q\in[S]\setminus\{j\}}\left(\alpha_{(p(q),q,0),(j,0)}^{(k)} + \alpha_{(c_1(q),q,1),(j,0)}^{(k)}\right)\alpha_{(c_1(q),q,1),(j,0)}^{(k)}\left(\sum_{q\in[S]}\alpha_{(\widetilde{p}(q),q,0),(j,0)}^{(k)}\right)\Bigg|n_{-1} = j\Bigg]. \tag{176}$$

*For $k \in [m]$, we have*

$$\delta y_{3,0,0}^{(k)} = -\delta y_{3,1,0}^{(k)} = 0, \tag{177}$$

*and*

$$\delta y_{3,0,1}^{(k)} = -\delta y_{3,1,1}^{(k)}$$

$$= \frac{1}{S}\mathbb{E}\Bigg[ -\left(1 - \alpha_{(p(j),j,0),(j,1)}^{(k)} - \alpha_{(p(j),s(j),0),(j,1)}^{(k)}\right)\left(\alpha_{(p(j),j,0),(j,1)}^{(k)} + \alpha_{(p(j),s(j),0),(j,1)}^{(k)}\right)\left(1 - \sum_{q\in[S]}\alpha_{(\widetilde{p}(q),q,0),(j,1)}^{(k)}\right)$$

$$+ \left(\alpha_{(0,n_1,0),(j,1)}^{(k)} + \alpha_{(0,n_2m,1),(j,1)}^{(k)}\right)\alpha_{(0,n_1,1),(j,1)}^{(k)}\left(1 - \sum_{q\in[S]}\alpha_{(\widetilde{p}(q),q,0),(j,1)}^{(k)}\right)$$

$$+ \sum_{q\in[S]\setminus\{p(j)\}}\left(\alpha_{(q,c_1(q),0),(j,1)}^{(k)} + \alpha_{(q,c_2(q),0),(j,1)}^{(k)} + \alpha_{(q,p(q),1),(j,1)}^{(k)}\right)\left(\alpha_{(q,c_1(q),0),(j,1)}^{(k)} + \alpha_{(q,c_2(q),0),(j,1)}^{(k)}\right)$$

$$\cdot \left(1 - \sum_{q\in[S]}\alpha_{(\widetilde{p}(q),q,0),(j,1)}^{(k)}\right)$$

$$- \left(\alpha_{(0,n_1,0),(j,1)}^{(k)} + \alpha_{(0,n_2m,1),(j,1)}^{(k)}\right)\alpha_{(0,n_2m,1),(j,1)}^{(k)}\left(\sum_{q\in[S]}\alpha_{(\widetilde{p}(q),q,0),(j,1)}^{(k)}\right)$$

$$- \sum_{q \in [S] \setminus \{p(j)\}} \left( \alpha^{(k)}_{(q,c_1(q),0),(j,1)} + \alpha^{(k)}_{(q,c_2(q),0),(j,1)} + \alpha^{(k)}_{(q,p(q),1),(j,1)} \right) \alpha^{(k)}_{(q,p(q),1),(j,1)} \left( \sum_{q \in [S]} \alpha^{(k)}_{(\tilde{p}(q),q,0),(j,1)} \right) \Bigg| n_{-1} = j \Bigg].$$

(178)

*For $k = m + 1$, we have*

$$\delta y^{(m+1)}_{3,0,0} = -\delta y^{(m+1)}_{3,1,0} = 0,$$

(179)

*and*

$$\delta y^{(m+1)}_{3,0,1} = -\delta y^{(m+1)}_{3,1,1}$$

$$= \frac{1}{S} \mathbb{E} \Bigg[ - \left( 1 - \alpha^{(m+1)}_{(n_2,j,1),(j,1)} - \alpha^{(m+1)}_{(n_2,c_1(n_2),0),(j,1)} - \alpha^{(m+1)}_{(n_2,c_2(n_2),0),(j,1)} \right)$$

$$\cdot \left( \alpha^{(m+1)}_{(n_2,c_1(n_2),0),(j,1)} + \alpha^{(m+1)}_{(n_2,c_2(n_2),0),(j,1)} \right) \left( 1 - \sum_{q \in [S]} \alpha^{(m+1)}_{(\tilde{p}(q),q,0),(j,1)} \right)$$

$$+ \left( \alpha^{(m+1)}_{(0,j,0),(j,1)} + \alpha^{(m+1)}_{(0,n_{2^m},1),(j,1)} \right) \alpha^{(m+1)}_{(0,j,0),(j,1)} \left( 1 - \sum_{q \in [S]} \alpha^{(m+1)}_{(\tilde{p}(q),q,0),(j,1)} \right)$$

$$+ \sum_{q \in [S] \setminus \{n_2\}} \left( \alpha^{(m+1)}_{(q,c_1(q),0),(j,1)} + \alpha^{(m+1)}_{(q,c_2(q),0),(j,1)} + \alpha^{(m+1)}_{(q,p(q),1),(j,1)} \right) \left( \alpha^{(m+1)}_{(q,c_1(q),0),(j,1)} + \alpha^{(m+1)}_{(q,c_2(q),0),(j,1)} \right)$$

$$\cdot \left( 1 - \sum_{q \in [S]} \alpha^{(m+1)}_{(\tilde{p}(q),q,0),(j,1)} \right)$$

$$+ \left( 1 - \alpha^{(m+1)}_{(n_2,j,1),(j,1)} - \alpha^{(m+1)}_{(n_2,c_1(n_2),0),(j,1)} - \alpha^{(m+1)}_{(n_2,c_2(n_2),0),(j,1)} \right) \alpha^{(m+1)}_{(n_2,j,1),(j,1)} \left( \sum_{q \in [S]} \alpha^{(m+1)}_{(\tilde{p}(q),q,0),(j,1)} \right)$$

$$- \sum_{q \in [S] \setminus \{n_2\}} \left( \alpha^{(m+1)}_{(q,c_1(q),0),(j,1)} + \alpha^{(m+1)}_{(q,c_2(q),0),(j,1)} + \alpha^{(m+1)}_{(q,p(q),1),(j,1)} \right) \alpha^{(m+1)}_{(q,p(q),1),(j,1)} \left( \sum_{q \in [S]} \alpha^{(m+1)}_{(\tilde{p}(q),q,0),(j,1)} \right) \Bigg| n_{-1} = j \Bigg].$$

(180)

*For $k \in \{m + 2, \cdots, 2^m\}$, we have*

$$\delta y^{(k)}_{3,0,1} = -\delta y^{(k)}_{3,1,1} = 0,$$

(181)

*and*

$$\delta y^{(k)}_{3,0,0} = -\delta y^{(k)}_{3,1,0}$$

$$= \frac{1}{S} \sum_{i \in [S]} \mathbb{E} \Bigg[ \mathbb{1}\{n_{2^{k-m}} = i\} \Bigg\{ - \left( 1 - \alpha^{(k)}_{(i,c_1(i),0),(j,0)} - \alpha^{(k)}_{(i,c_2(i),0),(j,0)} - \alpha^{(k)}_{(i,j,1),(j,0)} \right)$$

$$\cdot \left( \alpha^{(k)}_{(i,c_1(i),0),(j,0)} + \alpha^{(k)}_{(i,c_2(i),0),(j,0)} \right) \left( 1 - \sum_{q \in [S]} \alpha^{(k)}_{(\tilde{p}(q),q,0),(j,0)} \right)$$

$$+ \left( \alpha^{(k)}_{(0,n_1,0),(j,0)} - \alpha^{(k)}_{(0,n_{2^m},1),(j,0)} \right) \alpha^{(k)}_{(0,n_1,0),(j,0)} \left( 1 - \sum_{q \in [S]} \alpha^{(k)}_{(\tilde{p}(q),q,0),(j,0)} \right)$$

$$+ \sum_{q \in [S] \setminus \{i\}} \left( \alpha^{(k)}_{(q,c_1(q),0),(j,0)} + \alpha^{(k)}_{(q,c_2(q),0),(j,0)} + \alpha^{(k)}_{(q,p(q),1),(j,0)} \right)$$

$$\cdot \left( \alpha^{(k)}_{(q,c_1(q),0),(j,0)} + \alpha^{(k)}_{(q,c_2(q),0),(j,0)} \right) \left( 1 - \sum_{q \in [S]} \alpha^{(k)}_{(\tilde{p}(q),q,0),(j,0)} \right)$$

$$+ \left( 1 - \alpha^{(k)}_{(i,c_1(i),0),(j,0)} - \alpha^{(k)}_{(i,c_2(i),0),(j,0)} - \alpha^{(k)}_{(i,j,1),(j,0)} \right) \alpha^{(k)}_{(i,j,1),(j,0)} \left( \sum_{q \in [S]} \alpha^{(k)}_{(\tilde{p}(q),q,0),(j,0)} \right)$$

$$- \left( \alpha^{(k)}_{(0,n_1,0),(j,0)} - \alpha^{(k)}_{(0,n_2m,1),(j,0)} \right) \alpha^{(k)}_{(0,n_2m,1),(j,0)} \left( \sum_{q \in [S]} \alpha^{(k)}_{(\widetilde{p}(q),q,0),(j,0)} \right)$$

$$- \sum_{q \in [S] \backslash \{i\}} \left( \alpha^{(k)}_{(q,c_1(q),0),(j,0)} + \alpha^{(k)}_{(q,c_2(q),0),(j,0)} + \alpha^{(k)}_{(q,p(q),1),(j,0)} \right) \alpha^{(k)}_{(q,p(q),1),(j,0)} \left( \sum_{q \in [S]} \alpha^{(k)}_{(\widetilde{p}(q),q,0),(j,0)} \right) \Biggr\} \Bigg|_{n_{-1}=j} \Biggr].$$

(182)

Lemma 14 can be verified by straightforward calculation, whose proof is thus omitted for conciseness.

### C.2.1 Proof outline

Let the iteration-varying versions of (136) be

$$U_1^{(t)} := \widetilde{A}^\top B_1^{(t)} A \in \mathbb{R}^{(S+1) \times S}, \ \ U_2^{(t)} := A^\top \widetilde{B}_2^{(t)} A \in \mathbb{R}^{S \times S}, \ \ U_3^{(t)} := \widetilde{S}^\top \widetilde{B}_3^{(t)} \widetilde{S} \in \mathbb{R}^{2 \times 2}, \quad \text{(183a)}$$

$$V_1^{(t)} := \widetilde{A}^\top C_1^{(t)} A \in \mathbb{R}^{(S+1) \times 2}, \ \ V_2^{(t)} := A^\top C_2^{(t)} A \in \mathbb{R}^{S \times 2}, \ \ V_3^{(t)} := \widetilde{S}^\top C_3^{(t)} \widetilde{S} \in \mathbb{R}^{2 \times 2}, \quad \text{(183b)}$$

and the index starts from 0 for $U_l^{(t)}$ and $V_l^{(t)}$ when $l = 1, 2$ (i.e., $U_{l,0,j} = a_0^\top B_l^{(t)} a_j$, $V_{l,0,j} = a_0^\top C_l^{(t)} a_j$). Then we have for all $l \in [3]$:

$$\forall i \in \{0, \dots, S\}, j \in [S]: \quad U_{l,i,j}^{(t+1)} = U_{l,i,j}^{(t)} - \eta \sum_{k=1}^{2m} \left( \delta x_{l,i,j}^{(t,k)} + \delta y_{l,i,j}^{(t,k)} \right), \quad \text{(184a)}$$

$$V_{l,i,j}^{(t+1)} = V_{l,i,j}^{(t)} - \eta \sum_{k=1}^{2m} \delta z_{l,i,j}^{(t,k)}. \quad \text{(184b)}$$

Therefore, it is possible to analyze the dynamics of $U_l^{(t)}$ and $V_l^{(t)}$ separately.

Similar as the backward case, by symmetry of the training distribution, we have for any $t \in \mathbb{N}$, $i, j \in [S]$, $i \neq j$, $U_{l,i,j}^{(t)}$ ($l = 1, 2$) are equal to each other, $U_{l,j,j}^{(t)}$ ($l = 1, 2$) are equal to each other, and $U_{l,0,j}^{(t)}$ are equal to each other. Note that $a_0$ only appears in $X^{(k)}$ but never appears in $Y^{(k)}$, thus $U_{1,j,0}^{(t)} = U_{2,j,0}^{(t)} = U_{2,0,j}^{(t)} = 0$ for all $t \in \mathbb{N}$. The same holds for $V_{l,i,j}^{(t)}$ ($l = 1, 2$). Therefore, we could define the following pattern:

$$U_1^{(t)} = \begin{pmatrix} 0 & -a^{(t)}\mu_1^{(t)} & -a^{(t)}\mu_1^{(t)} & \cdots & -a^{(t)}\mu_1^{(t)} \\ 0 & \nu_1^{(t)} & \nu_{1,1}^{(t)} & \cdots & \nu_{1,1}^{(t)} \\ 0 & \nu_{1,1}^{(t)} & \nu_1^{(t)} & \cdots & \nu_{1,1}^{(t)} \\ \vdots & \vdots & \vdots & \ddots & \vdots \\ 0 & \nu_{1,1}^{(t)} & \nu_{1,1}^{(t)} & \cdots & \nu_1^{(t)} \end{pmatrix}, \quad U_2^{(t)} = \begin{pmatrix} 0 & 0 & 0 & \cdots & 0 \\ 0 & \mu_1^{(t)} & \nu_{1,2}^{(t)} & \cdots & \nu_{1,2}^{(t)} \\ 0 & \nu_{1,2}^{(t)} & \mu_1^{(t)} & \cdots & \nu_{1,2}^{(t)} \\ \vdots & \vdots & \vdots & \ddots & \vdots \\ 0 & \nu_{1,2}^{(t)} & \nu_{1,2}^{(t)} & \cdots & \mu_1^{(t)} \end{pmatrix},$$

(185)

and

$$V_1^{(t)} = \begin{pmatrix} 0 & \mu_2^{(t)} & \mu_2^{(t)} & \cdots & \mu_2^{(t)} \\ 0 & \nu_2^{(t)} & \nu_{2,1}^{(t)} & \cdots & \nu_{2,1}^{(t)} \\ 0 & \nu_{2,1}^{(t)} & \nu_2^{(t)} & \cdots & \nu_{2,1}^{(t)} \\ \vdots & \vdots & \vdots & \ddots & \vdots \\ 0 & \nu_{2,1}^{(t)} & \nu_{2,1}^{(t)} & \cdots & \nu_2^{(t)} \end{pmatrix}, \quad V_2^{(t)} = \begin{pmatrix} 0 & 0 & 0 & \cdots & 0 \\ 0 & b^{(t)}\mu_2^{(t)} & \nu_{2,2}^{(t)} & \cdots & \nu_{2,2}^{(t)} \\ 0 & \nu_{2,2}^{(t)} & b^{(t)}\mu_2^{(t)} & \cdots & \nu_{2,2}^{(t)} \\ \vdots & \vdots & \vdots & \ddots & \vdots \\ 0 & \nu_{2,2}^{(t)} & \nu_{2,2}^{(t)} & \cdots & b^{(t)}\mu_2^{(t)} \end{pmatrix}, \quad \text{(186)}$$

for some $a^{(t)}, b^{(t)}, \mu_1^{(t)}, \mu_2^{(t)}, \nu_1^{(t)}, \nu_2^{(t)}, \nu_{1,1}^{(t)}, \nu_{1,2}^{(t)}, \nu_{2,1}^{(t)}, \nu_{2,2}^{(t)} \in \mathbb{R}$. We set $a^{(0)} = b^{(0)} = 1$.

Regarding $U_3^{(t)}$ and $V_3^{(t)}$, by Lemma 13 and Lemma 14, it follows for $\xi \in \{x, y, z\}$

$$\forall t \in \mathbb{N}: \quad \delta\xi_{3,0,1}^{(t)} = -\delta\xi_{3,1,1}^{(t)}, \quad \text{and} \quad \delta\xi_{3,0,0}^{(t)} = -\delta\xi_{3,1,0}^{(t)}.$$

Thus by our initialization $(U_3, V_3 = 0)$, we have

$$\forall t \in \mathbb{N}: \quad U_{3,0,1}^{(t)} = -U_{3,1,1}^{(0)}, \quad \text{and} \quad U_{3,0,0}^{(t)} = -U_{3,1,0}^{(0)},$$
$$V_{3,0,1}^{(t)} = -V_{3,1,1}^{(0)}, \quad \text{and} \quad V_{3,0,0}^{(t)} = -V_{3,1,0}^{(0)}.$$

Therefore, we could define

$$U_3^{(t)} = \mu_1^{(t)} \begin{pmatrix} -b_1^{(t)} & b_2^{(t)} \\ b_1^{(t)} & -b_2^{(t)} \end{pmatrix}, \quad V_3^{(t)} = \mu_2^{(t)} \begin{pmatrix} c_1^{(t)} & -c_2^{(t)} \\ -c_1^{(t)} & c_2^{(t)} \end{pmatrix} \tag{187}$$

for some $b_1^{(t)}, b_2^{(t)}, c_1^{(t)}, c_2^{(t)} \in \mathbb{R}$, where $\mu_1^{(t)}, \mu_2^{(t)}$ is defined in (185) and (186).

### C.2.2 Training dynamics of $C_1, C_2, C_3$

**Training dynamics of Phase I.1-C.** Define

$$T_1^C := \max \left\{ t \in \mathbb{N} : \exp(\mu_2^{(t)}) \leqslant \sqrt{2N} \right\}. \tag{188}$$

We show that the following key relations hold during Phase I.1-C.

**Lemma 15.** *Assume the learning rate satisfies $\eta \lesssim \frac{1}{Nm}$. When $t \in [T_1^C]$, we have $\mu_2^{(t)}$ monotonically increases, and*

$$\mu_2^{(t)} \gtrsim \frac{t\eta}{SN^{3/2}}, \tag{189a}$$

$$b^{(t)} \mu_2^{(t)} = \left( 1 \pm \mathcal{O}\left( \frac{\log N}{N} \right) \right) \mu_2^{(t)}, \tag{189b}$$

$$|\nu_{2,1}^{(t)}| = \mathcal{O}\left( \frac{\log N}{N} \right) \mu_2^{(t)}, \quad |\nu_2^{(t)}| = \mathcal{O}\left( \frac{\log N}{N} \right) \mu_2^{(t)}, \quad |\nu_{2,2}^{(t)}| = \mathcal{O}\left( \frac{\log N}{N} \right) \mu_2^{(t)}, \tag{189c}$$

$$c_2^{(t)} \geqslant 1/2. \tag{189d}$$

The proof of Lemma 15 is given in Appendix C.2.7. By (189a) and (188), we know that

$$T_1^C \lesssim \frac{SN^{3/2} \log N}{\eta}. \tag{190}$$

**Training dynamics of Phase I.2-C.** We let

$$T_2^C := \max \left\{ t \in \mathbb{N} : \exp(\mu_2^{(t)}) \leqslant N \right\}. \tag{191}$$

We will show the following relations hold during Phase I.2-C.

**Lemma 16.** *Assume the learning rate satisfies $\eta \lesssim \frac{1}{Nm}$. When $t \in \{T_1^C, \cdots, T_2^C\}$, we have $\mu_2^{(t)}$ monotonically increases, and*

$$\mu_2^{(t)} - \mu_2^{(T_1^C)} \gtrsim \frac{\eta(t - T_1^C)}{SN}, \tag{192a}$$

$$b^{(t)} \mu_2^{(t)} = \left( 1 \pm \mathcal{O}\left( \frac{\log^2 N}{N} \right) \right) \mu_2^{(t)}, \tag{192b}$$

$$|\nu_{2,1}^{(t)}| = \mathcal{O}\left( \frac{\log N}{N} \right) \mu_2^{(t)}, \quad |\nu_2^{(t)}| = \mathcal{O}\left( \frac{\log N}{N} \right) \mu_2^{(t)}, \quad |\nu_{2,2}^{(t)}| = \mathcal{O}\left( \frac{\log N}{N} \right) \mu_2^{(t)}, \tag{192c}$$

$$c_2^{(t)} \geqslant 1/4. \tag{192d}$$

*At the end of Phase I.2-C, we have $c_2^{(T_2^C)} \in [0.25, 0.26]$, and $c_1^{(T_2^C)} \geqslant 0.48$.*

The proof of Lemma 16 is given in Appendix C.2.8. By (192a), we know that

$$T_2^C - T_1^C \lesssim \frac{SN \log N}{\eta}. \tag{193}$$

**Training dynamics of Phase II-C.** We set

$$T_3^C := \max \left\{ t \in \mathbb{N} : \frac{N}{\exp\left(0.47 \mu_2^{(t)}\right) + N} \geqslant \sqrt{\frac{\epsilon}{6m}} \right\}. \tag{194}$$

We show the following relations hold during Phase II-C.

**Lemma 17.** *Assume $\epsilon_0 > 0$ is small enough obeying $\epsilon_0 \lesssim 1/\mathbf{poly}(N)$ and $\eta \lesssim \frac{1}{Nm}$. When $t \in \{T_2^C, \cdots, T_3^C\}$, for any $\epsilon \in (0, \epsilon_0]$, we have $\mu_2^{(t)}$ monotonically increases, and*

$$\mu_2^{(t)} - \mu_2^{(T_2^C)} \gtrsim \frac{\left(t - T_2^C\right)\eta}{S} \left(\frac{\epsilon}{m}\right)^{3/2}, \tag{195a}$$

$$b^{(t)} \mu_2^{(t)} = \left(1 \pm \tilde{\mathcal{O}}\left(\frac{1}{N}\right)\right) \mu_2^{(t)}, \tag{195b}$$

$$\left|\nu_{2,1}^{(t)}\right| = \tilde{\mathcal{O}}\left(\frac{1}{N}\right) \mu_2^{(t)}, \quad \left|\nu_2^{(t)}\right| = \tilde{\mathcal{O}}\left(\frac{1}{N}\right) \mu_2^{(t)}, \quad \left|\nu_{2,2}^{(t)}\right| = \tilde{\mathcal{O}}\left(\frac{1}{N}\right) \mu_2^{(t)}, \tag{195c}$$

$$c_2^{(t)} \in [0.24, 0.26], \quad c_1^{(t+1)} \geqslant c_2^{(t+1)} - \tilde{\mathcal{O}}\left(\frac{1}{N}\right). \tag{195d}$$

The proof of Lemma 17 is given in Appendix C.2.9. By (194) and (195a), we have

$$T_3^C - T_2^C \lesssim \frac{S}{\eta}\left(\frac{m}{\epsilon}\right)^{3/2} \log\left(\frac{N}{\epsilon}\right). \tag{196}$$

It's easy to see from the proof of Lemma 17 that $\mu_2^{(t)}$ keeps increasing after $T_3^C$ and (195b), (195c) and (195d) hold for all $t \geqslant T_3^C$. We give the following lemma to mark the convergence of $\sum_{k=1}^{2m} \Delta_z^{(k)}$, whose proof is given in Appendix C.2.10.

**Lemma 18.** *After $t \geqslant T_3^C + 1$, we have*

$$\sum_{k=1}^{2m} \Delta_z^{(k)} \leqslant \frac{\epsilon}{3}. \tag{197}$$

### C.2.3 Training dynamics of $B_1, B_2, B_3$

**Training dynamics of Phase I.1-B.** We define

$$T_1^B := \max\left\{t \in \mathbb{N} : \exp\left(\mu_1^{(t)}\right) \leqslant N\right\}, \tag{198}$$

and call the period $t \leqslant T_1^B$ as Phase I.1-B. The following lemma gives the training dynamics of $U_1, U_2, U_3$ in Phase I.1-B.

**Lemma 19.** *When $t \in \{0, 1, \cdots, T_1^B\}$, we have $\mu_1^{(t)}, b_1^{(t)} \mu_1^{(t)}$ monotonically increase, $b_2^{(t)} \mu_1^{(t)}$ first decreases and then increases, and*

$$\mu_1^{(t)} \gtrsim \frac{\eta t}{NS}, \tag{199a}$$

$$b_2^{(t)} \leqslant \frac{1}{2}, \quad \text{and} \quad \exp\left((1 - 2b_2^{(t)})\mu_1^{(t)}\right) \lesssim \frac{N}{m} \quad \text{when } b_2^{(t)} \leqslant 0, \tag{199b}$$

$$\nu_1^{(t)} = \tilde{\mathcal{O}}\left(\frac{\mu_1^{(t)}}{N}\right), \left|\nu_{1,1}^{(t)}\right| = \tilde{\mathcal{O}}\left(\frac{\mu_1^{(t)}}{N}\right), \left|\nu_{1,2}^{(t)}\right| = \tilde{\mathcal{O}}\left(\frac{\mu_1^{(t)}}{N}\right), \tag{199c}$$

$$\exp\left((1 - a^{(t)})\mu_1^{(t)}\right) \lesssim N. \tag{199d}$$

The proof of Lemma 19 is given in Appendix C.2.12. By (199b) we know that $b_2^{(T_1^B)} \geqslant 0$, and by (199d) we know that $a^{(T_1^B)} \geqslant 0$. Further, by (199a) we know that

$$T_1^B \lesssim \frac{NS \log N}{\eta}. \tag{200}$$

**Training dynamics of Phase I.2-B.** Let

$$T_2^B := \max \left\{ t \geqslant T_1^B : \exp \left( (1 - 2b_2^{(t)})\mu_1^{(t)} \right) \geqslant C_0 N \right\}, \tag{201}$$

where $C_0$ is a sufficiently large constant. Without loss of generality, we assume $T_2^B > T_1^B$. We call the period $t \in \{T_1^B, T_1^B + 2, \cdots, T_2^B\}$ as Phase I.2-B. The following lemma formally describes the training dynamics of $U_1, U_2, U_3$ in Phase I.2-B, whose proof is given in Appendix C.2.13.

**Lemma 20.** *When $t \in \{T_1^B, T_1^B + 2, \cdots, T_2^B\}$, we have $\mu_1^{(t)}, b_1^{(t)}\mu_1^{(t)}$ monotonically increase, and*

$$\mu_1^{(t)} - \mu_1^{(T_1^B)} \gtrsim \frac{\eta \left( t - T_1^B \right)}{NS}, \tag{202a}$$

$$\exp \left( (1 - a^{(t)})\mu_1^{(t)} \right) \lesssim N, \tag{202b}$$

$$0 \leqslant b_2^{(t)} \leqslant \frac{1}{2}, \tag{202c}$$

$$\nu_1^{(t)} = \tilde{\mathcal{O}} \left( \frac{\mu_1^{(t)}}{N} \right), \left| \nu_{1,1}^{(t)} \right| = \tilde{\mathcal{O}} \left( \frac{\mu_1^{(t)}}{N} \right), \left| \nu_{1,2}^{(t)} \right| = \tilde{\mathcal{O}} \left( \frac{\mu_1^{(t)}}{N} \right). \tag{202d}$$

*At the end of Phase I.2-B, we have*

$$\exp \left( (1 - a^{(T_2^B)})\mu_1^{(T_2^B)} \right) \asymp N, \tag{203a}$$

$$b_2^{(T_2^B)} = \left( 1/4 \pm \tilde{\mathcal{O}}(\frac{1}{N}) \right) a^{(T_2^B)}. \tag{203b}$$

By (202a) and (203b) we know that

$$T_2^B - T_1^B \lesssim \frac{NS \log N}{\eta}. \tag{204}$$

**Training dynamics of Phase II-B.** We define

$$T_3^B := \max \left\{ t \in \mathbb{N} : \frac{N + 2m - 1}{\exp \left( 0.46\mu_1^{(t)} \right) + N + 2m - 1} \geqslant \sqrt{\frac{\epsilon}{6m}} \right\}, \tag{205}$$

and call the period $t \in \{T_2^B, T_2^B + 1, \cdots, T_3^B\}$ as Phase II-B. We'll show that at the start of the next phase (i.e., shortly after $T_2^B$), $\exp \left( (1 - a^{(t)}) \mu_1^{(t)} \right)$ increases to, and stays at $\mathcal{O}(N)$, indicating that when training long enough, $a^{(t)}$ will approach 1, $b_2^{(t)}$ will approach and stay at 0.25, and $b_1^{(t)}$ is larger than $b_2^{(t)}$. The training dynamics of $U_1, U_2, U_3$ in Phase II-B is formally given in the following lemma, whose proof is given in Appendix C.2.14.

**Lemma 21.** *When $t \in \{T_2^B, T_2^B + 1, \cdots, T_3^B\}$, we have $\mu_1^{(t)}, b_1^{(t)}\mu_1^{(t)}$ monotonically increase, and*

$$\mu_1^{(t)} - \mu_1^{(T_2^B)} \gtrsim \frac{\left( t - T_2^C \right) \eta}{S} \left( \frac{\epsilon}{m} \right)^{3/2}, \tag{206a}$$

$$\exp \left( (1 - a^{(t)})\mu_1^{(t)} \right) \asymp N, \tag{206b}$$

$$b_2^{(t)} \in [0.24, 0.26] \, a^{(t)}, \tag{206c}$$

$$\nu_1^{(t)} = \tilde{\mathcal{O}} \left( \frac{\mu_1^{(t)}}{N} \right), \left| \nu_{1,1}^{(t)} \right| = \tilde{\mathcal{O}} \left( \frac{\mu_1^{(t)}}{N} \right), \left| \nu_{1,2}^{(t)} \right| = \tilde{\mathcal{O}} \left( \frac{\mu_1^{(t)}}{N} \right). \tag{206d}$$

*At the end of Phase II-B, we have*

$$a^{(T_3^B)} \in [0.99, 1), \tag{207a}$$

$$b_1^{(T_3^B)} \geq b_2^{(T_3^B)} + \tilde{\mathcal{O}}\left(\frac{1}{N}\right). \tag{207b}$$

By (206a) we know that

$$T_3^B - T_2^B \lesssim \frac{S}{\eta}\left(\frac{m}{\epsilon}\right)^{3/2} \log\left(\frac{N}{\epsilon}\right). \tag{208}$$

**After Phase II-B.** It's easy to see from our proof that after Phase II-B, $\mu_1^{(t)}$ keeps increasing (but slowly), (206d) and (207) of Lemma 21 still hold. After Phase II-B, the loss reaches $\epsilon$ as indicated by the following lemma, whose proof is given in Appendix C.2.15.

**Lemma 22.** *After* $t \geq T_3^B + 1$, *we have*

$$\sum_{k=1}^{2m}\left(\Delta_x^{(t,k)} + \Delta_y^{(t,k)}\right) \leq \frac{2\epsilon}{3}. \tag{209}$$

### C.2.4 Proof of Lemma 11

Note that when $k \in [m]$ we have $n_{-1}^{(k)} = n_{2^{m-k+1}}$, and hence by (9), when $k \in [m]$, we have

$$
\begin{aligned}
\Delta_x^{(k)} &= \frac{1}{2}\mathbb{E}\bigg\| a_{n_{-1}} - \sum_{i=1}^{2^m-1}\left(\alpha_{(i,2i,0),(n_{-1},1)}^{(k)} + \alpha_{(i,2i+1,0),(n_{-1},1)}^{(k)}\right)a_{2i+1} \\
&\qquad - \alpha_{(0,n_1,0),(n_{-1},1)}^{(k)}a_{n_1} - \alpha_{(0,n_{2^m},1),(n_{-1},1)}^{(k)}a_{n_{2^m}} - \sum_{l=m-k+2}^{m}\alpha_{(n_{2^l},n_{2^{l-1}},1),(n_{-1},1)}^{(k)}a_{n_{2^{l-1}}}\bigg\|_2^2 \\
&= \frac{1}{2}\sum_{j=1}^{S}\mathbb{E}\bigg\| \mathbb{1}\{n_{-1} = j\}\bigg\{\left(1 - \alpha_{(p(j),j,0),(j,1)}^{(k)} - \alpha_{(\tilde{c}_1(j),j,0),(j,1)}^{(k)}\right)a_j \\
&\qquad - \sum_{q \in [S]\backslash\{j\}}\left(\alpha_{(p(q),j,0),(j,1)}^{(k)} + \alpha_{(\tilde{c}_1(q),j,0),(j,1)}^{(k)}\right)a_q\bigg\}\bigg\|_2^2.
\end{aligned}
\tag{210}
$$

By Assumption 5, we have $a_0, a_1, \cdots, a_S$ are orthonormal, and hence

$$
\begin{aligned}
\Delta_x^{(k)} &= \frac{1}{2}\sum_{j=1}^{S}\mathbb{E}\bigg[ \mathbb{1}\{n_{-1} = j\}\bigg\{\left(1 - \alpha_{(p(j),j,0),(j,1)}^{(k)} - \alpha_{(\tilde{c}_1(j),j,0),(j,1)}^{(k)}\right)^2 \\
&\qquad + \sum_{q \in [S]\backslash\{j\}}\left(\alpha_{(p(q),q,0),(j,1)}^{(k)} + \alpha_{(\tilde{c}_1(q),q,1),(j,1)}^{(k)}\right)^2\bigg\}\bigg].
\end{aligned}
\tag{211}
$$

This gives (140a). The relations (140b) and (140c) can be verified in a similar way.

For $\Delta_y^{(k)}$, when $k \in [m]$, $x_{-1}^{(k-1)} = a_{n_{2^{m-k}}} = a_{p(n_{-1})}$, and hence by (9), we have

$$
\begin{aligned}
\Delta_y^{(k)} &= \frac{1}{2}\mathbb{E}\bigg\| a_{p(n_{-1})} - \sum_{i=1}^{2^m-1}\left(\alpha_{(i,2i,0),(n_{-1},1)}^{(k)} + \alpha_{(i,2i+1,0),(n_{-1},1)}^{(k)}\right)a_i \\
&\qquad - \left(\alpha_{(0,n_1,0),(n_{-1},1)}^{(k)} + \alpha_{(0,n_{2^m},1),(n_{-1},1)}^{(k)}\right)a_0 - \sum_{l=m-k+2}^{m}\alpha_{(n_{2^l},n_{2^{l-1}},1),(n_{-1},1)}^{(k)}a_{n_{2^l}}\bigg\|_2^2 \\
&= \frac{1}{2}\sum_{j=1}^{S}\mathbb{E}\bigg\| \mathbb{1}\{n_{-1} = j\}\bigg\{\left(1 - \alpha_{(p(j),j,0),(j,1)}^{(k)} - \alpha_{(p(j),s(j),0),(j,1)}^{(k)}\right)a_j - \left(\alpha_{(0,n_1,0),(j,1)}^{(k)} + \alpha_{(0,n_{2^m},1),(j,1)}^{(k)}\right)a_0
\end{aligned}
$$

$$
- \sum_{q \in [S] \setminus \{j\}} \left( \alpha^{(k)}_{(q,c_1(q),0),(j,1)} + \alpha^{(k)}_{(q,c_2(q),0),(j,1)} + \alpha^{(k)}_{(q,p(q),1),(j,1)} \right) a_q \right\} \Bigg\|_2^2 .
$$

(212)

Since $a_0, a_1, \cdots, a_S$ are orthonormal, we have

$$
\Delta_y^{(k)} = \frac{1}{2} \sum_{j=1}^{S} \mathbb{E} \Bigg[ \mathbb{1}\{n_{-1} = j\} \Bigg\{ \left( 1 - \alpha^{(k)}_{(p(j),j,0),(j,1)} - \alpha^{(k)}_{(p(j),s(j),0),(j,1)} \right)^2 + \left( \alpha^{(k)}_{(0,n_1,0),(j,1)} + \alpha^{(k)}_{(0,n_{2^m},1),(j,1)} \right)^2
$$

$$
+ \sum_{q \in [S] \setminus \{j\}} \left( \alpha^{(k)}_{(q,c_1(q),0),(j,1)} + \alpha^{(k)}_{(q,c_2(q),0),(j,1)} + \alpha^{(k)}_{(q,p(q),1),(j,1)} \right)^2 \Bigg\} \Bigg] . \quad (213)
$$

This gives (141a). The relations (141b) and (141c) can be computed analogously.

Finally, for $\Delta_z^{(k)}$, we have when $k \in [m]$,

$$
\Delta_z^{(k)} = \frac{1}{2} \mathbb{E} \Bigg\| s_b - \sum_{i=1}^{2^m - 1} \left( \beta^{(k)}_{(i,2i,0),(n_{-1},1)} + \beta^{(k)}_{(i,2i+1,0),(n_{-1},1)} \right) s_f
$$

$$
- \beta^{(k)}_{(0,n_1,0),(n_{-1},1)} s_b - \beta^{(k)}_{(0,n_{2^m},1),(n_{-1},1)} s_f - \sum_{l=m-k+2}^{m} \beta^{(k)}_{(n_{2^l},n_{2^l-1},1),(n_{-1},1)} s_f \Bigg\|_2^2
$$

$$
= \frac{1}{2} \sum_{j=1}^{S} \mathbb{E} \Bigg\| \mathbb{1}\{n_{-1} = j\} \Bigg\{ \left( 1 - \sum_{q \in [S]} \beta^{(k)}_{(\tilde{c}_1(q),q,1),(j,1)} \right)^2 + \left( \sum_{q \in [S]} \beta^{(k)}_{(\tilde{p}(q),q,0),(j,1)} \right)^2 \Bigg\} \Bigg\|_2^2 .
$$

(214)

This gives (142a). The relations (142b) and (142c) can be computed analogously.

### C.2.5 Proof of Lemma 12

It's easy to verify by straightforward computation that for any $b \in \{0, 1, \ldots, S\}$, $c \in [S]$, $j \in [S]$, $u, v \in \{0, 1\}$, $k \in [2m]$, we have

$$
\frac{\partial \alpha^{(k)}_{(p,q,u),(j,v)}}{\partial a_b^\top B_1^{(t)} a_c} = \begin{cases} \alpha^{(k)}_{(p,q,u),(j,v)} \left( 1 - \sum_{\substack{s \in [S] \\ z \in \{0,1\}}} \alpha^{(k)}_{(p,s,z),(j,v)} \right), & \text{if } b = p, c = j, \\ -\alpha^{(k)}_{(p,q,u),(j,v)} \sum_{\substack{s \in [S] \\ z \in \{0,1\}}} \alpha^{(k)}_{(b,s,z),(j,v)}, & \text{if } b \neq p, c = j, \\ 0, & \text{otherwise,} \end{cases} \quad (215)
$$

$$
\frac{\partial \alpha^{(k)}_{(p,q,u),(j,v)}}{\partial a_b^\top B_2^{(t)} a_c} = \begin{cases} \alpha^{(k)}_{(p,q,u),(j,v)} \left( 1 - \sum_{\substack{r \in [S] \\ z \in \{0,1\}}} \alpha^{(k)}_{(r,q,z),(j,v)} \right), & \text{if } b = q, c = j, \\ -\alpha^{(k)}_{(p,q,u),(j,v)} \sum_{\substack{r \in [S] \\ z \in \{0,1\}}} \alpha^{(k)}_{(r,q,z),(j,v)}, & \text{if } b \neq q, c = j, \\ 0, & \text{otherwise,} \end{cases} \quad (216)
$$

and for any $b, c \in \{0, 1\}$, we have

$$
\frac{\partial \alpha^{(k)}_{(p,q,u),(j,v)}}{\partial s_b^\top B_3^{(t)} s_c} = \begin{cases} \alpha^{(k)}_{(p,q,u),(j,v)} \left( 1 - \sum_{\substack{r \in \{0,\ldots,S\} \\ s \in [S]}} \alpha^{(k)}_{(r,s,z),(j,v)} \right), & \text{if } b = u, c = v, \\ -\alpha^{(k)}_{(p,q,u),(j,v)} \sum_{\substack{r \in \{0,\ldots,S\} \\ s \in [S]}} \alpha^{(k)}_{(r,s,b),(j,v)}, & \text{if } b \neq u, c = v, \\ 0, & \text{otherwise.} \end{cases} \quad (217)
$$

Then by the chain rule and the above three expressions, we have (143a), (143b) and (143c). Also, (144a), (144b) and (144c) can be computed analogously.

### C.2.6 Proof of Lemma 13

Using (142a) and (144a), we can compute that for any $k \in [m]$:

$$
\delta z_{1,0,j}^{(k)} = \frac{1}{S} \mathbb{E} \Bigg[ - \left( 1 - w_1^{(k)} \right) q_0^{(k)} \left( 1 - q_0^{(k)} - p_0^{(k)} \right) + w_0^{(k)} q_0^{(k)} \left( 1 - q_0^{(k)} - p_0^{(k)} \right)
$$

$$+ \left( 1 - w_1^{(k)} \right) \left( w_1^{(k)} - q_0^{(k)} \right) \left( q_0^{(k)} + p_0^{(k)} \right) - w_0^{(k)} \left( w_0^{(k)} - p_0^{(k)} \right) \left( q_0^{(k)} + p_0^{(k)} \right) \Big| n_{-1} = j \Big].$$
(218)

By using the relation $w_0^{(k)} + w_1^{(k)} = 1$, we can further simplify the above expression as

$$\forall k \in [m]: \quad \delta z_{1,0,j}^{(k)} = \frac{2}{S} \mathbb{E} \left[ \left( w_0^{(k)} \right)^2 w_1^{(k)} \left( -\frac{q_0^{(k)}}{w_1^{(k)}} + \frac{p_0^{(k)}}{w_0^{(k)}} \right) \Big| n_{-1} = j \right].$$
(219)

Similarly, we can compute that at the turning point where $k = m + 1$,

$$\delta z_{1,0,j}^{(m+1)} = \frac{2}{S} \mathbb{E} \left[ \left( w_1^{(m+1)} \right)^2 w_0^{(m+1)} \left( -\frac{p_0^{(m+1)}}{w_0^{(m+1)}} + \frac{q_0^{(m+1)}}{w_1^{(m+1)}} \right) \Big| n_{-1} = j \right],$$
(220)

and for any $k \in \{m + 2, \ldots, 2m\}$,

$$\delta z_{1,0,j}^{(k)} = \frac{2}{S} \mathbb{E} \left[ \left( w_1^{(k)} \right)^2 w_0^{(k)} \left( -\frac{p_0^{(k)}}{w_0^{(k)}} + \frac{q_0^{(k)}}{w_1^{(k)}} \right) \Big| n_{-1} = j \right].$$
(221)

Combining (219), (220) and (221), we have (151). The other expressions in Lemma 13 can be computed analogously.

### C.2.7 Proof of Lemma 15

We prove the lemma by induction. First note that (189) is true when $t = 0$ by our initialization. For induction, we assume (189) holds for all time steps $s \leqslant t$ ($0 \leqslant t \leqslant T_1^C - 1$). Below we prove they hold at time $t + 1$.

**Step 1: showing** (189a) **holds at step** $t + 1$. By (184b) and (186), we know that

$$\mu_2^{(t+1)} = \mu_2^{(t)} - \eta \sum_{k=1}^{2m} \left( -\delta z_{1,0,j}^{(t,k)} \right).$$
(222)

Thus to track the growth of $\mu_2$ and show (189a) holds at step $t + 1$, we need to estimate its the terms in the gradient expression (151).

We start from bounding the iteration-varying versions of $-\frac{q_0^{(k)}}{w_1^{(k)}} + \frac{p_0^{(k)}}{w_0^{(k)}}$ for $k \in [m]$, and $-\frac{p_0^{(k)}}{w_0^{(k)}} + \frac{q_0^{(k)}}{w_1^{(k)}}$ for $k \in \{1, \ldots, 2m\}$. Note that for $\frac{q_0^{(t,k)}}{w_1^{(t,k)}}$,

- when $k = 1$, we have

$$\frac{q_0^{(t,k)}}{w_1^{(t,k)}} = 1,$$
(223)

where $q_0^{(t,k)}$ and $w_1^{(t,k)}$ are the iteration-varying versions of (148) and (146), respectively;

- when $k \in \{2, \ldots, m + 1\}$, we have

$$\frac{q_0^{(t,k)}}{w_1^{(t,k)}} = \frac{\exp\left( (1 + c_2^{(t)})\mu_2^{(t)} + \nu_{2,2}^{(t)} \right)}{\exp\left( (1 + c_2^{(t)})\mu_2^{(t)} + \nu_{2,2}^{(t)} \right) + (k - 2)\exp\left( \nu_{2,1}^{(t)} + \nu_{2,2}^{(t)} + c_2^{(t)}\mu_2^{(t)} \right) + \exp\left( \nu_{2,1}^{(t)} + (b^{(t)} + c_2^{(t)})\mu_2^{(t)} \right)}$$

$$= \left( 1 \pm \mathcal{O}\left( \frac{\log^2 N}{N} \right) \right) \frac{\exp\left( (1 + c_2^{(t)})\mu_2^{(t)} \right)}{\exp\left( (1 + c_2^{(t)})\mu_2^{(t)} \right) + (k - 2)\exp\left( c_2^{(t)}\mu_2^{(t)} \right) + \exp\left( (1 + c_2^{(t)})\mu_2^{(t)} \right)}$$

$$= \left( 1 \pm \mathcal{O}\left( \frac{\log^2 N}{N} \right) \right) \frac{\exp\left( \mu_2^{(t)} \right)}{2\exp\left( \mu_2^{(t)} \right) + k - 2} \geqslant \left( 1 \pm \mathcal{O}\left( \frac{\log^2 N}{N} \right) \right) \frac{1}{k},$$
(224)

where in the second line we use the fact that when $t \leqslant T_1^C$, we have

$$\exp(x) = 1 + \mathcal{O}(x) = 1 + \mathcal{O}\left(\frac{\log^2 N}{N}\right),$$

where $x = \nu_{2,1}^{(t)}, \nu_{2,2}^{(t)}, (1 - b^{(t)})\mu_2^{(t)}$ (note that $\mu_2^{(t)} \lesssim \log N$ by our choice of $T_1^C$);

- when $k \in \{m + 2, \ldots, 2m\}$, similarly, we have

$$\frac{q_0^{(t,k)}}{w_1^{(t,k)}} = \left(1 \pm \mathcal{O}\left(\frac{\log^2 N}{N}\right)\right) \frac{\exp\left((1 - c_1^{(t)})\mu_2^{(t)}\right)}{\exp\left((1 - c_1^{(t)})\mu_2^{(t)}\right) + (m - 1)\exp\left(-c_1^{(t)}\mu_2^{(t)}\right) + \exp\left((1 - c_1^{(t)})\mu_2^{(t)}\right)}$$

$$= \left(1 \pm \mathcal{O}\left(\frac{\log^2 N}{N}\right)\right) \frac{\exp\left(\mu_2^{(t)}\right)}{2\exp\left(\mu_2^{(t)}\right) + m - 1}, \tag{225}$$

where the last step follows from our induction hypothesis.

Besides, turning to $\frac{p_0^{(t,k)}}{w_0^{(t,k)}}$,

- when $k \in [m]$:

$$\frac{p_0^{(t,k)}}{w_0^{(t,k)}} = \left(1 \pm \mathcal{O}\left(\frac{\log^2 N}{N}\right)\right) \frac{\exp\left((1 - c_2^{(t)})\mu_2^{(t)}\right)}{\exp\left((1 - c_2^{(t)})\mu_2^{(t)}\right) + (N - 2)\exp\left(-c_2^{(t)}\mu_2^{(t)}\right) + \exp\left((b^{(t)} - c_2^{(t)})\mu_2^{(t)}\right)}$$

$$= \left(1 \pm \mathcal{O}\left(\frac{\log^2 N}{N}\right)\right) \frac{\exp\left(\mu_2^{(t)}\right)}{2\exp\left(\mu_2^{(t)}\right) + N - 2} \lesssim \frac{1}{\sqrt{N}}, \tag{226}$$

where the last two steps follow from induction hypothesis and the choice of $T_1^C$;

- when $k = m + 1$, we have

$$\frac{p_0^{(t,m+1)}}{w_0^{(t,m+1)}} = \left(1 \pm \mathcal{O}\left(\frac{\log^2 N}{N}\right)\right) \frac{\exp\left((2 - c_2^{(t)})\mu_2^{(t)}\right)}{(N - 1)\exp\left(-c_2^{(t)}\mu_2^{(t)}\right) + \exp\left((2 - c_2^{(t)})\mu_2^{(t)}\right)}$$

$$= \left(1 \pm \mathcal{O}\left(\frac{\log^2 N}{N}\right)\right) \frac{\exp\left(2\mu_2^{(t)}\right)}{N - 1 + \exp\left(2\mu_2^{(t)}\right)}. \tag{227}$$

- when $k \in \{m + 2, \ldots, 2m\}$, we have

$$\frac{p_0^{(t,k)}}{w_0^{(t,k)}} = \left(1 \pm \mathcal{O}\left(\frac{\log^2 N}{N}\right)\right) \frac{\exp\left((1 + c_2^{(t)})\mu_2^{(t)}\right)}{(N - 4 + k - m)\exp\left(c_2^{(t)}\mu_2^{(t)}\right) + 3\exp\left((1 + c_2^{(t)})\mu_2^{(t)}\right)}$$

$$= \left(1 \pm \mathcal{O}\left(\frac{\log^2 N}{N}\right)\right) \frac{\exp\left(\mu_2^{(t)}\right)}{3\exp\left(\mu_2^{(t)}\right) + N - 4 + k - m}. \tag{228}$$

Together, (224) and (226) give that for any $k \in [m]$:

$$\frac{q_0^{(t,k)}}{w_1^{(t,k)}} - \frac{p_0^{(t,k)}}{w_0^{(t,k)}} \gtrsim \frac{1}{k} - \frac{1}{\sqrt{N}}. \tag{229}$$

Besides, from (223) and (224) we can see that when $k = 1$:

$$\frac{q_0^{(t,1)}}{w_1^{(t,1)}} - \frac{p_0^{(t,1)}}{w_0^{(t,1)}} = 1 - \left(1 \pm \mathcal{O}\left(\frac{\log^2 N}{N}\right)\right) \frac{\exp\left(\mu_2^{(t)}\right)}{2\exp\left(\mu_2^{(t)}\right) + m - 1} \asymp 1, \tag{230}$$

and when $k \in \{2, \ldots, m\}$:

$$\frac{q_0^{(t,k)}}{w_1^{(t,k)}} - \frac{q_0^{(t,m+1)}}{w_1^{(t,m+1)}} = \left(1 \pm \mathcal{O}\left(\frac{\log^2 N}{N}\right)\right) \left(\frac{\exp\left(\mu_2^{(t)}\right)}{2\exp\left(\mu_2^{(t)}\right) + k - 2} - \frac{\exp\left(\mu_2^{(t)}\right)}{2\exp\left(\mu_2^{(t)}\right) + m - 1}\right) \gtrsim \frac{1}{\sqrt{N}},$$

(231)

where the last inequality follows from the choice of $T_1^C$, which guarentees $\mu_2^{(t)} \lesssim \sqrt{N}$, and by (226) and (227), we have

$$\frac{p_0^{(t,k)}}{w_0^{(t,k)}} \leqslant \frac{p_0^{(t,m+1)}}{w_0^{(t,m+1)}}.$$

(232)

The above two inequalities imply that for any $k \in [m]$:

$$-\left(-\frac{q_0^{(t,k)}}{w_1^{(t,k)}} + \frac{p_0^{(t,k)}}{w_0^{(t,k)}}\right) \geqslant -\frac{p_0^{(t,m+1)}}{w_0^{(t,m+1)}} + \frac{q_0^{(t,m+1)}}{w_1^{(t,m+1)}} + \mathcal{O}\left(\frac{\log^2 N}{N}\right).$$

(233)

By comparing (224), (225), (226), (228), we have

$$\forall k \in [m], \ k' \in \{m+2, \ldots, 2m\} : \quad \frac{q_0^{(t,k)}}{w_1^{(t,k)}} - \frac{p_0^{(t,k)}}{w_0^{(t,k)}} \gtrsim \frac{q_0^{(t,k')}}{w_1^{(t,k')}} - \frac{p_0^{(t,k')}}{w_0^{(t,k')}} > 0.$$

(234)

We next bound the iteration-varying versions of the terms $\left(w_0^{(k)}\right)^2 w_1^{(k)}$ and $\left(w_1^{(k)}\right)^2 w_0^{(k)}$ in the expression of $\delta z_{1,0,j}^{(k)}$ (c.f. (151)) for $k \in [m]$ and $k \in \{m+1, \ldots, 2m\}$, respectively. By the induction hypothesis, we have

- When $k = 1$,

$$w_0^{(t,1)} = \left(1 \pm \mathcal{O}\left(\frac{\log^2 N}{N}\right)\right) \frac{(N-2)\exp\left(-c_2^{(t)}\mu_2^{(t)}\right) + 2\exp\left((1 - c_2^{(t)})\mu_2^{(t)}\right)}{\exp\left((2 + c_2^{(t)})\mu_2^{(t)}\right) + (N-2)\exp\left(-c_2^{(t)}\mu_2^{(t)}\right) + 2\exp\left((1 - c_2^{(t)})\mu_2^{(t)}\right)},$$

(235)

$$w_1^{(t,1)} = \left(1 \pm \mathcal{O}\left(\frac{\log^2 N}{N}\right)\right) \frac{\exp\left((2 + c_2^{(t)})\mu_2^{(t)}\right)}{\exp\left((2 + c_2^{(t)})\mu_2^{(t)}\right) + (N-2)\exp\left(-c_2^{(t)}\mu_2^{(t)}\right) + 2\exp\left((1 - c_2^{(t)})\mu_2^{(t)}\right)}.$$

(236)

- When $k \in \{2, \ldots, m\}$:

$$w_0^{(t,k)} = \left(1 \pm \mathcal{O}\left(\frac{\log^2 N}{N}\right)\right)$$
$$\cdot \frac{(N-2)\exp\left(-c_2^{(t)}\mu_2^{(t)}\right) + 2\exp\left((1 - c_2^{(t)})\mu_2^{(t)}\right)}{2\exp\left((1 + c_2^{(t)})\mu_2^{(t)}\right) + (k-2)\exp(c_2^{(t)}\mu_2^{(t)}) + (N-2)\exp\left(-c_2^{(t)}\mu_2^{(t)}\right) + 2\exp\left((1 - c_2^{(t)})\mu_2^{(t)}\right)},$$

(237)

$$w_1^{(t,k)} = \left(1 \pm \mathcal{O}\left(\frac{\log^2 N}{N}\right)\right)$$
$$\cdot \frac{2\exp\left((1 + c_2^{(t)})\mu_2^{(t)}\right) + (k-2)\exp(c_2^{(t)}\mu_2^{(t)})}{2\exp\left((1 + c_2^{(t)})\mu_2^{(t)}\right) + (k-2)\exp(c_2^{(t)}\mu_2^{(t)}) + (N-2)\exp\left(-c_2^{(t)}\mu_2^{(t)}\right) + 2\exp\left((1 - c_2^{(t)})\mu_2^{(t)}\right)}.$$

(238)

- When $k = m + 1$, we have

$$w_0^{(t,m+1)} = \left(1 \pm \mathcal{O}\left(\frac{\log^2 N}{N}\right)\right)$$

$$\cdot \frac{(N-1)\exp\left(-c_2^{(t)}\mu_2^{(t)}\right) + \exp\left((2-c_2^{(t)})\mu_2^{(t)}\right)}{2\exp\left((1+c_2^{(t)})\mu_2^{(t)}\right) + (m-1)\exp(c_2^{(t)}\mu_2^{(t)}) + (N-1)\exp\left(-c_2^{(t)}\mu_2^{(t)}\right) + \exp\left((2-c_2^{(t)})\mu_2^{(t)}\right)},$$

$$(239)$$

$$w_1^{(t,m+1)} = \left(1 \pm \mathcal{O}\left(\frac{\log^2 N}{N}\right)\right)$$

$$\cdot \frac{2\exp\left((1+c_2^{(t)})\mu_2^{(t)}\right) + (m-1)\exp(c_2^{(t)}\mu_2^{(t)})}{2\exp\left((1+c_2^{(t)})\mu_2^{(t)}\right) + (m-1)\exp(c_2^{(t)}\mu_2^{(t)}) + (N-1)\exp\left(-c_2^{(t)}\mu_2^{(t)}\right) + \exp\left((2-c_2^{(t)})\mu_2^{(t)}\right)}.$$

$$(240)$$

- When $k \in \{m+2, \ldots, 2m\}$, we have

$$w_1^{(t,k)} = \left(1 \pm \mathcal{O}\left(\frac{\log^2 N}{N}\right)\right)$$

$$\cdot \frac{2\exp\left((1-c_1^{(t)})\mu_2^{(t)}\right) + (m-1)\exp\left(-c_1^{(t)}\mu_2^{(t)}\right)}{(N-4+k-m)\exp\left(c_1^{(t)}\mu_2^{(t)}\right) + 3\exp\left((1+c_1^{(t)})\mu_2^{(t)}\right) + 2\exp\left((1-c_1^{(t)})\mu_2^{(t)}\right) + (m-1)\exp\left(-c_1^{(t)}\mu_2^{(t)}\right)}$$

$$\lesssim \max\left\{\frac{\log N}{N}, \frac{2}{N\exp\left((2c_1^{(t)}-1)\mu_2^{(t)}\right)}\right\}, \tag{241}$$

$$w_0^{(t,k)} = \left(1 \pm \mathcal{O}\left(\frac{\log^2 N}{N}\right)\right)$$

$$\cdot \frac{(N-4+k-m)\exp\left(c_1^{(t)}\mu_2^{(t)}\right) + 3\exp\left((1+c_1^{(t)})\mu_2^{(t)}\right)}{(N-4+k-m)\exp\left(c_1^{(t)}\mu_2^{(t)}\right) + 3\exp\left((1+c_1^{(t)})\mu_2^{(t)}\right) + 2\exp\left((1-c_1^{(t)})\mu_2^{(t)}\right) + (m-1)\exp\left(-c_1^{(t)}\mu_2^{(t)}\right)}.$$

$$(242)$$

By (237), (238), (239), (240) and the fact that $\exp\left(\mu_2^{(t)}\right) \lesssim \sqrt{N}$ (c.f., (188)), we can compute that for any $s \leqslant t$:

$$\forall k \in [m]: \quad \left(w_0^{(t,k)}\right)^2\left(w_1^{(t,k)}\right) \asymp \left(\frac{1}{1 + \frac{2}{N}\exp\left((1+2c_2^{(t)})\mu_2^{(t)}\right)}\right)^2\left(\frac{\frac{2}{N}\exp\left((1+2c_2^{(t)})\mu_2^{(t)}\right)}{1 + \frac{2}{N}\exp\left((1+2c_2^{(t)})\mu_2^{(t)}\right)}\right),$$

$$(243)$$

$$\left(w_1^{(t,m+1)}\right)^2\left(w_0^{(t,m+1)}\right) \asymp \left(\frac{\frac{2}{N}\exp\left((1+2c_2^{(t)})\mu_2^{(t)}\right)}{1 + \frac{2}{N}\exp\left((1+2c_2^{(t)})\mu_2^{(t)}\right)}\right)^2\left(\frac{1}{1 + \frac{2}{N}\exp\left((1+2c_2^{(t)})\mu_2^{(t)}\right)}\right),$$

$$(244)$$

which gives

$$\frac{\left(w_0^{(t,k)}\right)^2\left(w_1^{(t,k)}\right)}{\left(w_1^{(t,m+1)}\right)^2\left(w_0^{(t,m+1)}\right)} \asymp \frac{N}{\exp\left((1+2c_2^{(t)})\mu_2^{(t)}\right)}. \tag{245}$$

With the above expressions, we could derive the following lemma, whose proof can be found in Appendix C.2.11.

**Lemma 23.** *Under our induction hypothesis, there exist absolute constants $C, C' > 0$ such that when $\eta \leqslant \frac{C}{Nm}$, we have*

$$\sum_{k=1}^{m}\left(w_0^{(t,k)}\right)^2 w_1^{(t,k)} \geqslant \left(1 - \mathcal{O}\left(\frac{m}{N}\right)\right)\left(w_1^{(k,m+1)}\right)^2 w_0^{(t,m+1)}. \tag{246}$$

By Lemma 23, (233) and (151) we know that

$$\sum_{k=1}^{m}\left(-\delta z_{1,0,j}^{(k)}\right) - \delta z_{1,0,j}^{(m+1)} \gtrsim \frac{1}{S\sqrt{N}}\sum_{k=1}^{m}\mathbb{E}\left[\left(w_0^{(k)}\right)^2 w_1^{(k)}\right]. \tag{247}$$

Lemma 23 combined with (245) also implies that

$$\exp\left((1 + 2c_2^{(t)})\mu_2^{(t)}\right) \lesssim mN. \tag{248}$$

By (243), (248) and our initialization we know that for any $k \in [m]$:

$$\mathbb{E}\left[\left(w_0^{(t,k)}\right)^2 w_1^{(t,k)}\right] \gtrsim \frac{1}{N}. \tag{249}$$

Moreover, similar as how we prove Lemma 23 in Appendix C.2.11, using relations (237), (238), (241), (242), we can show by contradiction that

$$\forall k \in [m],\ k' \in \{m+2, \cdots, 2m\}: \quad \left(w_1^{(t,k')}\right)^2 w_0^{(t,k')} \lesssim \frac{\log N}{N}\left(w_0^{(t,k)}\right)^2 w_1^{(t,k)}. \tag{250}$$

Recall (151) gives

$$\forall j \in [S]: \ \delta z_{1,0,j}^{(t,k)} = \begin{cases} \frac{2}{S}\mathbb{E}\left[\left(w_0^{(t,k)}\right)^2 w_1^{(t,k)}\left(-\frac{q_0^{(t,k)}}{w_1^{(k)}} + \frac{p_0^{(t,k)}}{w_0^{(t,k)}}\right)\Big| n_{-1} = j\right], & k \in [m] \\ \frac{2}{S}\mathbb{E}\left[\left(w_1^{(t,k)}\right)^2 w_0^{(t,k)}\left(-\frac{p_0^{(t,k)}}{w_0^{(t,k)}} + \frac{q_0^{(t,k)}}{w_1^{(t,k)}}\right)\Big| n_{-1} = j\right], & k \in \{m+1,\ldots,2m\}. \end{cases}$$

Thus by (250) and (234), we have

$$\forall j \in [S]: \ |\delta z_{1,0,j}^{(t,k')}| \lesssim \frac{\log N}{N}\cdot\left(-\delta z_{1,0,j}^{(t,k)}\right), \quad \forall k \in [m],\ k' \in \{m+2,\ldots,2m\}. \tag{251}$$

Combining (247), (249) and (251), we have

$$\sum_{k=1}^{2m}\left(-\delta z_{1,0,j}^{(t,k)}\right) \gtrsim \frac{1}{S\sqrt{N}}\sum_{k=1}^{m}\mathbb{E}\left[\left(w_0^{(t,k)}\right)^2 w_1^{(t,k)}\right] \gtrsim \frac{1}{SN^{3/2}}, \tag{252}$$

and thus

$$\mu_2^{(t+1)} \overset{(222)}{=} \mu_2^{(t)} - \eta\sum_{k=1}^{2m}\left(-\delta z_{1,0,j}^{(t,k)}\right) \gtrsim \frac{(t+1)\eta}{SN^{3/2}}, \tag{253}$$

where we use the induction hypothesis in the last step. This shows (189a) is true at time $t+1$. Moreover, by (252) we know that $\mu_2^{(t+1)} > \mu_2^{(t)}$.

**Step 2: showing** (189b) **holds at time** $t+1$**.** By (151), (153) and the induction hypothesis, we have for any $s \leqslant t$:

$$\sum_{k=1}^{m+1}\left(-\delta z_{1,j,j}^{(s,k)}\right) = \left(1 \pm \mathcal{O}\left(\frac{\log N}{N}\right)\right)\sum_{k=1}^{m+1}\left(-\delta z_{1,0,j}^{(s,k)}\right), \tag{254}$$

and when $k \in \{m+2,\ldots,2m\}$,

$$-\delta z_{1,j,j}^{(s,k)} \asymp -\delta z_{1,0,j}^{(s,k)}. \tag{255}$$

By (251), we have

$$\sum_{k=1}^{2m}\left(-\delta z_{1,j,j}^{(s,k)}\right) = \left(1 \pm \mathcal{O}\left(\frac{\log N}{N}\right)\right)\sum_{k=1}^{2m}\left(-\delta z_{1,0,j}^{(s,k)}\right). \tag{256}$$

Combining the above expresion with (184a), and summing over $s = 0, 1, \ldots, t$, we have

$$b^{(t+1)}\mu^{(t+1)} = \left(1 \pm \mathcal{O}\left(\frac{\log N}{N}\right)\right)\mu^{(t+1)}. \tag{257}$$

This shows (189b) holds at time $t + 1$.

**Step 3: showing** (189c) **holds at time** $t+1$**.** For any $i, j \in [S]$ and $n_{-1}^{(k)} = j$, by definition (c.f. (150)) we have

$$\forall k \in [m+1]: \quad q_{p(i)}^{(t,k)} = \beta_{(i,p(i),1),(j,1)}^{(t,k)}.$$

Thus we have

- when $k = 1$:

$$\frac{q_{p(i)}^{(t,k)}}{w_1^{(t,k)}} = 0. \tag{258}$$

- When $k \in \{2, \ldots, m+1\}$,

$$\frac{q_{p(i)}^{(t,k)}}{w_1^{(t,k)}} = \mathbb{1}\{\exists l \in \{0, \ldots, k-2\}, n_{2^{m-l}} = i | n_{-1}^{(t,k)} = j\} \frac{q_{p(i)}^{(t,k)}}{w_1^{(t,k)}}$$

$$= \mathbb{1}\{p(i) = j | n_{-1}^{(k)} = j\} \left(1 \pm \mathcal{O}\left(\frac{\log^2 N}{N}\right)\right) \frac{\exp\left((1 + c_2^{(t)})\mu_2^{(t)}\right)}{\exp\left((1 + c_2^{(t)})\mu_2^{(t)}\right) + (k-2)\exp\left(c_2^{(t)}\mu_2^{(t)}\right) + \exp\left((1 + c_2^{(t)})\mu_2^{(t)}\right)}$$

$$+ \mathbb{1}\{\exists l \in \{0, \ldots, k-3\}, n_{2^{m-l}} = i | n_{-1}^{(k)} = j\} \cdot \left(1 \pm \mathcal{O}\left(\frac{\log^2 N}{N}\right)\right)$$

$$\cdot \frac{\exp\left(c_2^{(t)}\mu_2^{(t)}\right)}{\exp\left((1 + c_2^{(t)})\mu_2^{(t)}\right) + (k-2)\exp\left(c_2^{(t)}\mu_2^{(t)}\right) + \exp\left((1 + c_2^{(t)})\mu_2^{(t)}\right)}$$

$$\overset{(224)}{\lesssim} \mathbb{1}\{p(i) = j | n_{-1}^{(k)} = j\} \frac{q_0^{(k)}}{w_1^{(k)}} + \mathbb{1}\{\exists l \in \{0, \ldots, k-3\}, n_{2^{m-l}} = i | n_{-1}^{(k)} = j\} \frac{1}{k}. \tag{259}$$

In addition, we have

- When $k \in \{1, \ldots, m\}$:

$$\frac{p_{1,i}^{(t,k)} + p_{2,i}^{(t,k)}}{w_0^{(t,k)}} = \mathbb{1}\{i \in \mathcal{V}(\mathcal{T}), i \text{ is not a leaf} | n_{-1}^{(t,k)} = j\} \frac{p_{1,i}^{(t,k)} + p_{2,i}^{(t,k)}}{w_0^{(t,k)}}$$

$$= \mathbb{1}\{c_1(i) = j | n_{-1}^{(t,k)} = j\} \left(1 \pm \mathcal{O}\left(\frac{\log^2 N}{N}\right)\right) \frac{\exp\left((1 - c_2^{(t)})\mu_2^{(t)}\right) + \exp\left(-c_2^{(t)}\mu_2^{(t)}\right)}{(N-2)\exp\left(-c_2^{(t)}\mu_2^{(t)}\right) + 2\exp\left((1 - c_2^{(t)})\mu_2^{(t)}\right)}$$

$$+ \mathbb{1}\{i \in \mathcal{V}(\mathcal{T}), i \text{ is not a leaf}, c_1(i) \neq j | n_{-1}^{(t,k)} = j\} \left(1 \pm \mathcal{O}\left(\frac{\log^2 N}{N}\right)\right)$$

$$\cdot \frac{2\exp\left(-c_2^{(t)}\mu_2^{(t)}\right)}{(N-2)\exp\left(-c_2^{(t)}\mu_2^{(t)}\right) + 2\exp\left((1 - c_2^{(t)})\mu_2^{(t)}\right)}$$

$$\overset{(227)}{\lesssim} \mathbb{1}\{c_1(i) = j | n_{-1}^{(t,k)} = j\} \frac{p_0^{(t,k)}}{w_0^{(t,k)}} + \mathbb{1}\{i \in \mathcal{V}(\mathcal{T}), i \text{ is not a leaf}, c_1(i) \neq j | n_{-1}^{(t,k)} = j\} \frac{1}{N}, \tag{260}$$

- When $k = m+1$,

$$\frac{p_{1,i}^{(t,m+1)} + p_{2,i}^{(t,m+1)}}{w_0^{(t,m+1)}} = \mathbb{1}\{i \in \mathcal{V}(\mathcal{T}), i \text{ is not a leaf and root} | n_{-1}^{(t,m+1)} = j\}$$

$$\cdot \left(1 \pm \mathcal{O}\left(\frac{\log^2 N}{N}\right)\right) \frac{2\exp\left(-c_2^{(t)}\mu_2^{(t)}\right)}{(N-1)\exp\left(-c_2^{(t)}\mu_2^{(t)}\right) + \exp\left((2 - c_2^{(t)})\mu_2^{(t)}\right)}$$

$$\lesssim \mathbb{1}\{i \in \mathcal{V}(\mathcal{T}), i \text{ is not a leaf and root} | n_{-1}^{(t,m+1)} = j\} \frac{1}{N}. \tag{261}$$

Thus we have for all $k \in [m+1]$:

$$
\begin{aligned}
\left|\delta z_{1,i,j}^{(t,k)}\right| &\overset{(152)}{\leqslant} \frac{2}{S}\mathbb{E}\left[\left(w_0^{(t,k)}\right)^2 w_1^{(t,k)}\left(\frac{q_{p(i)}^{(t,k)}}{w_1^{(t,k)}} + \frac{p_{1,i}^{(t,k)} + p_{2,i}^{(t,k)}}{w_0^{(t,k)}}\right)\Bigg| n_{-1} = j\right] \\
&\lesssim \frac{2}{N}\cdot\frac{1}{S}\mathbb{E}\left[\left(w_0^{(t,k)}\right)^2 w_1^{(t,k)}\left(\frac{q_0^{(t,k)}}{w_1^{(t,k)}} + \frac{p_0^{(t,k)}}{w_0^{(t,k)}}\right)\Bigg| n_{-1} = j\right] + \frac{2}{N}\cdot\frac{1}{S}\mathbb{E}\left[\left(w_0^{(t,k)}\right)^2 w_1^{(t,k)}\right] \\
&\lesssim \frac{\log N}{N}\left(-\delta z_{1,0,j}^{(t,k)}\right),
\end{aligned}
$$

where the second line follows from (259), (260) and (261), and the last line follows from (151), (224) and (227). When $k \in \{m+2, \ldots, 2m\}$, similarly, we can compute that

$$
\left|\delta z_{1,i,j}^{(t,k)}\right| \lesssim \frac{\log N}{N}\left|\delta z_{1,0,j}^{(t,k)}\right| \overset{(251)}{\lesssim} \frac{\log N}{N}\cdot\frac{\log N}{N}\left(-\delta z_{1,0,j}^{(t,k')}\right), \quad \forall k' \in [m]. \tag{262}
$$

Combining the above two expressions, we have

$$
\forall i, j \in [S]: \quad \left|\sum_{k=1}^{2m}\delta z_{1,i,j}^{(t,k)}\right| \lesssim \frac{\log N}{N}\sum_{k=1}^{2m}\left(-\delta z_{1,0,j}^{(t,k)}\right). \tag{263}
$$

This combined with (186) and (184b) gives for any $i, j \in [S], i \neq j$:

$$
\left|\nu_{2,1}^{(t+1)}\right| \leqslant \left|\nu_{2,1}^{(t)}\right| + \eta\left|\sum_{k=1}^{2m}\delta z_{1,i,j}^{(t,k)}\right| \lesssim \frac{\log N}{N}\mu_2^{(t)} + \frac{\eta\log N}{N}\sum_{k=1}^{2m}\left(-\delta z_{1,0,j}^{(t,k)}\right) = \frac{\log N}{N}\mu_2^{(t+1)}, \tag{264}
$$

$$
\left|\nu_2^{(t+1)}\right| \leqslant \left|\nu_2^{(t)}\right| + \eta\left|\sum_{k=1}^{2m}\delta z_{1,j,j}^{(t,k)}\right| \lesssim \frac{\log N}{N}\mu_2^{(t)} + \frac{\eta\log N}{N}\sum_{k=1}^{2m}\left(-\delta z_{1,0,j}^{(t,k)}\right) = \frac{\log N}{N}\mu_2^{(t+1)}. \tag{265}
$$

where the second inequality follows from (263) and the induction hypothesis.

Analogously, we can show that

$$
\left|\nu_{2,2}^{(t+1)}\right| \lesssim \frac{\log N}{N}\mu_2^{(t+1)}. \tag{266}
$$

Combining (264), (265) and (266), we know that (189c) holds at time $t+1$.

**Step 4: showing (189d) holds at time $t+1$.** We show $c_2^{(t+1)} \geqslant 1/2$ by contradiction. If

$$
c_2^{(t+1)} < 1/2, \tag{267}
$$

since $\eta \lesssim \frac{1}{Nm}$, by (245) we have

$$
\frac{\sum_{k=1}^m\left(w_0^{(t,k)}\right)^2\left(w_1^{(t,k)}\right)}{\left(w_1^{(t,m+1)}\right)^2\left(w_0^{(t,m+1)}\right)} \asymp \left(1 \pm \mathcal{O}\left(\frac{1}{Nm}\right)\right)\frac{mN}{\exp\left((1+2c_2^{(t+1)})\mu_2^{(t+1)}\right)} \gtrsim \log N,
$$

which combined with (151), (154), (251) and Lemma 23 indicates that for any $j \in [S]$:

$$
-\sum_{k=1}^{2m}\delta z_{3,1,1}^{(t,k)} \gtrsim -\sum_{k=1}^{2m}S\left(-\delta z_{1,0,j}^{(t,k)}\right). \tag{268}
$$

This indicates that

$$
c_2^{(t+1)} = \frac{C_{3,1,1}^{(t)} - \sum_{k=1}^{2m}\delta z_{3,1,1}^{(t,k)}}{C_{1,0,j}^{(t)} - \sum_{k=1}^{2m}\delta z_{1,0,j}^{(t,k)}} \geqslant \frac{1}{2},
$$

where the last inequality follows from (268) and our hypothesis that $c_2^{(t)} \geqslant 1/2$. The above expression contradicts (267), thus we have $c_2^{(t+1)} \geqslant 1/2$.

### C.2.8 Proof of Lemma 16

We prove by induction that (192a), (192b), (192c), (192d) and

$$\frac{\sum_{k=m+2}^{2m} \left(w_1^{(t,k)}\right)^2 w_0^{(t,k)}}{\left(w_1^{(t,m+1)}\right)^2 w_0^{(t,m+1)}} \lesssim \frac{\log^2 N}{N}, \tag{269}$$

$$\exp\left(\left(1 + 2c_2^{(t)}\right)\mu_2^{(t)}\right) \geqslant N \tag{270}$$

hold for all $t \in [T_1^C, T_2^C]$.

**Base case.** When $t = T_1^C$, (192a), (192b), (192c) hold. By (189d) we know that (192d) and (270) hold at time $t = T_1^C$. By (189d), (239) and (240), we have

$$\left(w_1^{(T_1^C, m+1)}\right)^2 w_0^{(T_1^C, m+1)} \asymp \frac{1}{\exp\left(\left(2c_2^{(T_1^C)} - 1\right)\mu_2^{(T_1^C)}\right)}, \tag{271}$$

and by (242) and (241), we have

$$\left(w_1^{(T_1^C, k)}\right)^2 w_0^{(T_1^C, k)} \asymp \frac{1}{N^2 \exp\left(2\left(2c_1^{(T_1^C)} - 1\right)\mu_2^{(T_1^C)}\right)}. \tag{272}$$

Thus we have

$$\frac{\sum_{k=m+2}^{2m} \left(w_1^{(T_1^C, k)}\right)^2 w_0^{(T_1^C, k)}}{\left(w_1^{(T_1^C, m+1)}\right)^2 w_0^{(T_1^C, m+1)}} \asymp \frac{m \exp\left(\left(2c_2^{(T_1^C)} - 1\right)\mu_2^{(T_1^C)}\right)}{N^2 \exp\left(2\left(2c_1^{(T_1^C)} - 1\right)\mu_2^{(T_1^C)}\right)}$$

$$= \frac{m \exp\left(\left(2c_2^{(T_1^C)} + 1\right)\mu_2^{(T_1^C)}\right)}{N^2 \exp\left(4c_1^{(T_1^C)}\mu_2^{(T_1^C)}\right)} \overset{(248)}{\lesssim} \frac{\log^2 N}{N}, \tag{273}$$

where the last inequality also uses the fact that $V_{3,0,0}^{(t)} \overset{(187)}{=} c_1^{(t)}\mu_2^{(t)} > 0$ for any $t$, because by (155), we know that $V_{3,0,0}^{(t)}$ strictly increases with $t$. By (273) we know that (269) holds at time $t = T_1^C$.

**Induction.** Assume that (192a), (192b), (192c), (192d) and (269) hold for all time steps $s \in [T_1^C, t]$ ($T_1^C \leqslant t \leqslant T_2^C - 1$). Below we prove that (192a), (192b), (192c) and (269) hold for $t + 1$.

**Step 1: Showing (192a) for $t + 1$.** By (227) and (224) and the induction hypothesis, we have

$$\frac{p_0^{(t,m+1)}}{w_0^{(t,m+1)}} = \left(1 \pm \tilde{\mathcal{O}}\left(\frac{1}{N}\right)\right)\frac{\exp\left(2\mu_2^{(t)}\right)}{N - 1 + \exp\left(2\mu_2^{(t)}\right)} = \left(1 \pm \tilde{\mathcal{O}}\left(\frac{1}{N}\right)\right)\cdot\frac{2}{3},$$

$$\frac{q_0^{(t,m+1)}}{w_1^{(t,m+1)}} = \left(1 \pm \tilde{\mathcal{O}}\left(\frac{1}{N}\right)\right)\frac{\exp\left(\mu_2^{(t)}\right)}{2\exp\left(\mu_2^{(t)}\right) + m - 1} = \left(1 \pm \tilde{\mathcal{O}}\left(\frac{1}{N}\right)\right)\cdot\frac{1}{2}, \tag{274}$$

and thus by (151) and the above expression, we have

$$-\delta z_{1,0,j}^{(t,m+1)} \asymp \frac{2}{S}\mathbb{E}\left[\left(w_1^{(t,m+1)}\right)^2 w_0^{(t,m+1)}\Big| n_{-1} = j\right]. \tag{275}$$

By (224), (226) and (151), we have $-\delta z_{1,0,j}^{(t,m+1)} \geqslant 0$, and by (269) we have

$$\left|\sum_{k=m+2}^{2m} \delta z_{1,0,j}^{(t,k)}\right| \lesssim \frac{\log^2 N}{N}\left(-\delta z_{1,0,j}^{(t,m+1)}\right). \tag{276}$$

Thus we have

$$\mu_2^{(t+1)} \gtrsim \mu_2^{(t)} - \eta\delta z_{1,0,j}^{(t,m+1)}. \tag{277}$$

By (239) and (240), we have

$$\left(w_1^{(t,m+1)}\right)^2 w_0^{(t,m+1)} \asymp \begin{cases} \frac{1}{\exp\left(\left(2c_2^{(t)}-1\right)\mu_2^{(t)}\right)}, & \text{when } c_2^{(t)} \geqslant 0.5 \\ \frac{1}{\exp\left(2\left(1-2c_2^{(t)}\right)\mu_2^{(t)}\right)}, & \text{when } c_2^{(t)} \leqslant 0.5 \end{cases}, \tag{278}$$

and when $c_2^{(t)} \geqslant 0.5$, by (237) and (238), we have $\forall k \in [m]$:

$$\left(w_0^{(t,k)}\right)^2 w_1^{(t,k)} \asymp \frac{N^2}{\exp\left(2\left(2c_2^{(t)}+1\right)\mu_2^{(t)}\right)}, \text{ and } \frac{\left(w_0^{(t,k)}\right)^2 w_1^{(t,k)}}{\left(w_1^{(t,m+1)}\right)^2 w_0^{(t,m+1)}} \asymp \frac{N^2}{\exp\left(\left(2c_2^{(t)}+3\right)\mu_2^{(t)}\right)}. \tag{279}$$

Following the same argument as we show (248) in Phase 1 by leveraging (245), here by leveraging (279), we can show that when $\eta \lesssim \frac{1}{mN}$ and $c_2^{(t)} \geqslant 0.5$,

$$\exp\left(\left(2c_2^{(t)}+3\right)\mu_2^{(t)}\right) \lesssim mN^2. \tag{280}$$

This combined with (278) suggests that when $c_2^{(t)} \geqslant 0.5$, we have

$$\left(w_1^{(t,m+1)}\right)^2 w_0^{(t,m+1)} \gtrsim \frac{\exp\left(4\mu_2^{(t)}\right)}{mN^2} \gtrsim \frac{1}{m}. \tag{281}$$

Thus by (277) and the above relation, we have

$$\mu_2^{(t+1)} - \mu_2^{(T_1^C)} \gtrsim \mu_2^{(t)} - \mu_2^{(T_1^C)} + \eta\frac{1}{Sm} \gtrsim \frac{\eta(t+1-T_1^C)}{SN}, \tag{282}$$

where the last step follows from our induction hypothesis.

On the other hand, (280) suggests that

$$c_2^{(t)} \leqslant 0.5 \text{ after } \mu^{(t)} \gtrsim m^{1/4}\sqrt{N}. \tag{283}$$

After $c_2^{(t)} \leqslant 0.5$, by (278) and (192d), we have

$$\left(w_1^{(t,m+1)}\right)^2 w_0^{(t,m+1)} \gtrsim \frac{1}{\exp\left(\mu_2^{(t)}\right)} \geqslant \frac{1}{N}. \tag{284}$$

Thus by (277) and the above relation, we have

$$\mu_2^{(t+1)} - \mu_2^{(T_1^C)} \gtrsim \mu_2^{(t)} - \mu_2^{(T_1^C)} + \eta\frac{1}{N} \gtrsim \frac{\eta(t+1-T_1^C)}{SN}, \tag{285}$$

where the last step follows from our induction hypothesis. (285) and (282) suggest that (192a) holds for $t+1$.

**Step 2: Showing** (192b) **and** (192c) **hold for** $t+1$**.** Note that (269) indicates that

$$\left|\sum_{k=m+2}^{2m} \delta z_{1,0,j}^{(t,k)}\right| \lesssim \frac{\log^2 N}{N}\left(-\delta z_{1,0,j}^{(t,m+1)}\right). \tag{286}$$

Following a similar argument as we show (189b) and (189c) in Phase 1 in Appendix C.2.7 (Step 2 and Step 3), we can show that (192b) and (192c) hold for $t+1$. Here we omit the details to avoid repetition.

**Step 3: Showing** (270) **hold for** $t+1$**.** We prove this by contradiction. Assume that

$$\exp\left(\left(1+2c_2^{(t+1)}\right)\mu_2^{(t+1)}\right) < N \tag{287}$$

then when $\eta$ is small enough, we have

$$\exp\left(\left(1+2c_2^{(t)}\right)\mu_2^{(t)}\right) < 2N. \tag{288}$$

Then by (237) and (238), we have

$$\forall k \in [m]: \quad \left(w_0^{(t,k)}\right)^2 w_1^{(t,k)} \asymp \frac{\exp\left(\left(2c_2^{(t)}+1\right)\mu_2^{(t)}\right)}{N}. \tag{289}$$

By the above relation and (278), we have

$$\frac{\sum_{k=1}^m \left(w_0^{(t,k)}\right)^2 w_1^{(t,k)}}{\left(w_1^{(t,m+1)}\right)^2 w_0^{(t,m+1)}} \asymp \frac{m\exp\left(\left(3-2c_2^{(t)}\right)\mu_2^{(t)}\right)}{N} \geqslant \frac{m\exp\left(4\mu_2^{(t)}\right)}{2N^2} \gtrsim m, \tag{290}$$

where the first inequality follows from (288). Thus by (154), we have

$$\sum_{k=1}^{2m} \left(-\delta z_{3,1,1}^{(t,k)}\right) \gtrsim S \sum_{k=1}^{2m} \left(-\delta z_{1,0,j}^{(t,k)}\right), \tag{291}$$

indicating that

$$\exp\left(\left(1+2c_2^{(t+1)}\right)\mu_2^{(t+1)}\right) \gtrsim \exp\left(\left(1+2c_2^{(t)}\right)\mu_2^{(t)}\right) \gtrsim N, \tag{292}$$

where the last step follows from our induction hypothesis. This contradicts with the fact that (287).

**Step 4: Showing** (192d) **hold for** $t+1$**.** By (270), we have after $c_2^{(t)} \leqslant 0.5$,

$$\frac{\sum_{k=1}^m \left(w_0^{(t,k)}\right)^2 w_1^{(t,k)}}{\left(w_1^{(t,m+1)}\right)^2 w_0^{(t,m+1)}} \asymp \frac{mN^2}{\exp\left(8c_2^{(t)}\mu_2^{(t)}\right)}. \tag{293}$$

By a similar argument as in Step 3, we can show that when $\eta$ is small enough, $c_2^{(t+1)} \geqslant 0.25$, by leveraging the above relation.

**Step 5: Showing** (269) **hold for** $t+1$**.** We can compute that for any $k \in [m+2, 2m]$,

$$\frac{\left(w_1^{(t,k)}\right)^2 w_0^{(t,k)}}{\left(w_1^{(t,m+1)}\right)^2 w_0^{(t,m+1)}} \asymp \begin{cases} \frac{\exp\left(\left(2c_2^{(t)}+1-4c_1^{(t)}\right)\mu_2^{(t)}\right)}{N^2}, & \text{when } c_2^{(t)} \geqslant 0.5 \\ \frac{\exp\left(\left(4-4c_2^{(t)}-4c_1^{(t)}\right)\mu_2^{(t)}\right)}{N^2}, & \text{when } c_2^{(t)} \leqslant 0.5 \end{cases}. \tag{294}$$

When $c_2^{(t)} \geqslant 0.5$, by (280) and the above relation, we have

$$\frac{\sum_{k=m+2}^{2m} \left(w_1^{(t+1,k)}\right)^2 w_0^{(t+1,k)}}{\left(w_1^{(t+1,m+1)}\right)^2 w_0^{(t+1,m+1)}} \lesssim \frac{m^2 N^2}{\exp\left(2\mu_2^{(t+1)}\right) N^2} \lesssim \frac{m^2}{N}. \tag{295}$$

When $c_2^{(t)} \leqslant 0.5$, we can show (269) also holds at time $t+1$ by contradiction. First by contradiction similar to Step 3, using (293) we can show that

$$\frac{\sum_{k=m+2}^{2m} \left(w_1^{(t+1,k)}\right)^2 w_0^{(t+1,k)}}{\left(w_1^{(t+1,m+1)}\right)^2 w_0^{(t+1,m+1)}} \lesssim m. \tag{296}$$

This together with (151) and induction hypothesis implies that

$$\sum_{k=m+2}^{2m} \left(-\delta z_{1,0,j}^{(t+1,k)}\right) \lesssim m\left(-\delta z_{1,0,j}^{(t,m+1)}\right). \tag{297}$$

By the induction hypothesis, there exists an absolute constant $C$ such that

$$\frac{\sum_{k=m+2}^{2m} \left(w_1^{(s,k)}\right)^2 w_0^{(s,k)}}{\left(w_1^{(s,m+1)}\right)^2 w_0^{(s,m+1)}} \leqslant \frac{Cm^2}{N} \tag{298}$$

for all $s \in [T_1^C, t]$. We now claim

$$\frac{\sum_{k=m+2}^{2m} \left(w_1^{(t+1,k)}\right)^2 w_0^{(t+1,k)}}{\left(w_1^{(t+1,m+1)}\right)^2 w_0^{(t+1,m+1)}} \leqslant \frac{Cm^2}{N},$$

which can be shown by contradiction. If

$$\frac{\sum_{k=m+2}^{2m} \left(w_1^{(t+1,k)}\right)^2 w_0^{(t+1,k)}}{\left(w_1^{(t+1,m+1)}\right)^2 w_0^{(t+1,m+1)}} > \frac{Cm^2}{N}, \tag{299}$$

then when $\eta$ is small enough, similar as in the proof of Lemma 23 in Appendix C.2.11, we can show that when learning rate $\eta \lesssim \frac{1}{Nm}$, we have

$$\frac{\sum_{k=m+2}^{2m} \left(w_1^{(t,k)}\right)^2 w_0^{(t,k)}}{\left(w_1^{(t,m+1)}\right)^2 w_0^{(t,m+1)}} \geqslant \frac{Cm^2}{2N}. \tag{300}$$

By (151) and (155) we have

$$-\delta z_{3,0,0}^{(t,m+1)} \gtrsim \frac{m^2 S}{2N} \left(-\delta z_{1,0,j}^{(t,m+1)}\right) \overset{(297)}{\gtrsim} \frac{mS}{2N} \sum_{k=1}^{2m} \left(-\delta z_{1,0,j}^{(t,k)}\right), \tag{301}$$

indicating $4 - 4c_2^{(t)} - 4c_1^{(t)}$ decreases from step $t$ to step $t+1$, i.e.,

$$4 - 4c_2^{(t+1)} - 4c_1^{(t+1)} < 4 - 4c_2^{(t)} - 4c_1^{(t)}. \tag{302}$$

Thus by (294), we have

$$\frac{\sum_{k=m+2}^{2m} \left(w_1^{(t+1,k)}\right)^2 w_0^{(t+1,k)}}{\left(w_1^{(t+1,m+1)}\right)^2 w_0^{(t+1,m+1)}} \leqslant \frac{\sum_{k=m+2}^{2m} \left(w_1^{(t,k)}\right)^2 w_0^{(t,k)}}{\left(w_1^{(t,m+1)}\right)^2 w_0^{(t,m+1)}} \leqslant \frac{Cm^2}{N}. \tag{303}$$

This contradicts with (299). The induction is complete.

**Step 6: Showing $c_2^{(T_2^C)} \leqslant 0.26$, $c_1^{(T_2^C)} \geqslant 0.48$.** Recall in Step 1 we demonstrated that (c.f.(283))

$$c_2^{(t)} \leqslant 0.5 \text{ after } \mu^{(t)} \gtrsim m^{1/4}\sqrt{N}. \tag{304}$$

After $c_2^{(t)} \leqslant 0.5$, using a similar argument as we show (291), by (293) we know that before $c_2^{(t)} \leqslant 0.26$,

$$\sum_{k=1}^{2m} \left(-\delta z_{3,1,1}^{(t,k)}\right) \gtrsim S \sum_{k=1}^{2m} \left(-\delta z_{1,0,j}^{(t,k)}\right), \tag{305}$$

indicating that $c_2^{(t)} \mu_2^{(t)}$ decreases approximately $S$ times faster than $\mu_2$ increases, and will decrease below 0.26 before $T_2^C$.

In addition, after $c_2^{(t)} \leqslant 0.5$, by (269) and (294) we know that for any $k \in [m+2, 2m]$,

$$\frac{\exp\left(\left(4 - 4c_2^{(T_2^C)} - 4c_1^{(T_2^C)}\right) \mu_2^{(T_2^C)}\right)}{N^2} \lesssim \frac{\log N}{N}. \tag{306}$$

This together with the fact that $c_2^{(T_2^C)} \leqslant 0.26$ and $\exp\left(\mu_2^{(T_2^C)}\right) \asymp N$ implies that

$$c_1^{(T_2^C)} \geqslant 0.48. \tag{307}$$

### C.2.9 Proof of Lemma 17

We first compute some important quantities that will be used in the proof.

**Basic computations.** When $t \geqslant T_2^C$, by induction hypothesis and a similar computation as in Step 1 of the proof of Lemma 15, we have when $\epsilon$ is small enough:

$$w_0^{(t,1)} \asymp \exp\left(\pm\tilde{\mathcal{O}}\left(\frac{\mu_2^{(t)}}{N}\right)\right)\frac{1}{\exp\left((1+2c_2^{(t)})\mu_2^{(t)}\right)}, \quad w_1^{(t,1)} \asymp 1. \tag{308}$$

and for any $k \in \{2,\ldots,m\}$:

$$w_0^{(t,k)} \asymp \exp\left(\pm\tilde{\mathcal{O}}\left(\frac{\mu_2^{(t)}}{N}\right)\right)\frac{1}{\exp\left((2c_2^{(t)})\mu_2^{(t)}\right)}, \quad w_1^{(t,k)} \asymp 1. \tag{309}$$

When $k = m+1$, we have

$$w_0^{(t,m+1)} \asymp 1, \quad w_1^{(t,m+1)} \asymp \exp\left(\pm\tilde{\mathcal{O}}\left(\frac{\mu_2^{(t)}}{N}\right)\right)\frac{1}{\exp\left((1-2c_2^{(t)})\mu_2^{(t)}\right)}. \tag{310}$$

When $k \in \{m+2,\ldots,2m\}$, we have

$$w_1^{(t,k)} \asymp \exp\left(\pm\tilde{\mathcal{O}}\left(\frac{\mu_2^{(t)}}{N}\right)\right)\frac{1}{\exp\left(2c_1^{(t)}\mu_2^{(t)}\right)}, \quad w_0^{(t,k)} \asymp 1. \tag{311}$$

The above expressions give that

$$\left(w_0^{(t,k)}\right)^2 w_1^{(t,k)} = \begin{cases} \exp\left(\pm\tilde{\mathcal{O}}\left(\frac{\mu_2^{(t)}}{N}\right)\right)\frac{1}{\exp\left(2(1+2c_2^{(t)})\mu_2^{(t)}\right)}, & k = 1 \\ \exp\left(\pm\tilde{\mathcal{O}}\left(\frac{\mu_2^{(t)}}{N}\right)\right)\frac{1}{\exp\left((4c_2^{(t)})\mu_2^{(t)}\right)}, & k \in \{2,\ldots,m\}, \end{cases} \tag{312}$$

and

$$\left(w_1^{(t,m+1)}\right)^2 w_0^{(t,m+1)} = \exp\left(\pm\tilde{\mathcal{O}}\left(\frac{\mu_2^{(t)}}{N}\right)\right)\frac{1}{\exp\left(2(1-2c_2^{(t)})\mu_2^{(t)}\right)}. \tag{313}$$

and

$$\forall k \in \{m+2,\ldots,2m\}, \quad \left(w_1^{(t,k)}\right)^2 w_0^{(t,k)} = \exp\left(\pm\tilde{\mathcal{O}}\left(\frac{\mu_2^{(t)}}{N}\right)\right)\frac{1}{\exp\left(4c_1^{(t)}\mu_2^{(t)}\right)}. \tag{314}$$

Below we show Lemma 17 By Lemma 16 we know that Lemma 17 holds for $t = T_2^C$. We assume that Lemma 17 holds for all $s \leqslant t$ ($t \in [T_2^C, T_3^C - 1]$). Below we show that Lemma 17 holds for $t+1$.

**Step 1: showing** (195a) **holds for** $t+1$**.** Following the same computation as in Step 1 in the proof of Lemma 16, we could compute that (277) holds here, i.e., we have

$$\mu_2^{(t+1)} \gtrsim \mu_2^{(t)} - \eta\delta z_{1,0,j}^{(t,m+1)}, \tag{315}$$

and

$$-\delta z_{1,0,j}^{(t,m+1)} \asymp \frac{2}{S}\mathbb{E}\left[\left(w_1^{(t,m+1)}\right)^2 w_0^{(t,m+1)}\Big|n_{-1} = j\right]. \tag{316}$$

By (313) and induction hypothesis we know that when $\epsilon$ is small enough, we have

$$\left(w_1^{(t,m+1)}\right)^2 w_0^{(t,m+1)} \geqslant \frac{1}{\exp\left((2(1-0.48)+0.01)\mu_2^{(t)}\right)} = \frac{1}{\exp\left(1.05\mu_2^{(t)}\right)} \geqslant \left(\frac{\epsilon}{6m}\right)^{3/2}. \tag{317}$$

This together with (315) implies that

$$\mu_2^{(t+1)} \gtrsim \mu_2^{(t)} + \frac{\eta}{S}\left(\frac{\epsilon}{m}\right)^{3/2}, \tag{318}$$

indicating that (195a) holds for $t+1$.

**Step 2: showing** (195b)**,** (195c) **hold for** $t+1$**.** This step is almost the same as Step 2 and 3 in the proof of Lemma 15 and is omitted here.

**Step 3: showing** (195d) **holds for** $t+1$**.** Similar to Step 3 in the proof of Lemma 16, we could show by contradiction that when $\eta \lesssim \frac{1}{mN}$:

$$\frac{\sum_{k=1}^{m}\left(w_0^{(t,k)}\right)^2 w_1^{(t,k)}}{\left(w_1^{(t,m+1)}\right)^2 w_0^{(t,m+1)}} = 1 \pm \tilde{\mathcal{O}}\left(\frac{1}{N}\right), \tag{319}$$

since by (154), if

$$\sum_{k=1}^{m}\left(w_0^{(t,k)}\right)^2 w_1^{(t,k)} > \left(w_1^{(t,m+1)}\right)^2 w_0^{(t,m+1)}, \tag{320}$$

$c_2^{(t)}$ will increase, i.e., $c_2^{(t+1)} > c_2^{(t)}$ and if

$$\sum_{k=1}^{m}\left(w_0^{(t,k)}\right)^2 w_1^{(t,k)} < \left(w_1^{(t,m+1)}\right)^2 w_0^{(t,m+1)}, \tag{321}$$

$c_2^{(t)}$ will decrease, i.e., $c_2^{(t+1)} < c_2^{(t)}$. Combining (319) with (312) and (313), we have

$$\exp\left(\pm\tilde{\mathcal{O}}\left(\frac{\mu_2^{(t+1)}}{N}\right)\right)\frac{\exp\left(2(1-2c_2^{(t+1)})\mu_2^{(t+1)}\right)}{\exp\left((4c_2^{(t+1)})\mu_2^{(t+1)}\right)} \asymp 1, \tag{322}$$

indicating that $c_2^{(t+1)} \in [0.24, 0.26]$.

Next, we show by contradiction that

$$c_1^{(t+1)} \geqslant c_2^{(t+1)} - \tilde{\mathcal{O}}\left(\frac{1}{N}\right).$$

By induction hypothesis, there exists some absolute constant $C$ such that for all $s \in [T_2^C, t]$,

$$c_1^{(s)} - c_2^{(s)} \geqslant -\frac{C\mathsf{poly}(\log N)}{N},$$

where $\mathsf{poly}(\log N)$ stands for some polynomial in $\log N$. (155), (154) and (319) give that

$$-\sum_{k=1}^{2m}\delta z_{3,0,0}^{(t,k)} = 2\sum_{k=m+2}^{2m}\mathbb{E}\left[\left(w_1^{(t,k)}\right)^2 w_0^{(t,k)}\right],$$

$$-\sum_{k=1}^{2m}\delta z_{3,1,1}^{(t,k)} = \tilde{\mathcal{O}}\left(\frac{1}{N}\right)\sum_{k=1}^{m}\mathbb{E}\left[\left(w_1^{(t,k)}\right)^2 w_0^{(t,k)}\right]. \tag{323}$$

Thus if

$$c_1^{(t+1)} < c_2^{(t+1)} - \frac{C\mathsf{poly}(\log N)}{N}, \tag{324}$$

then when $\eta \lesssim \frac{1}{mN}$, we have

$$c_1^{(t)} < c_2^{(t)} - \frac{C\mathsf{poly}(\log N)}{2N}.$$

By (312), (314) and (323), when $C$ is large enough, we have

$$-\delta z_{3,0,0}^{(t,m+1)} \gg -\delta z_{3,1,1}^{(t,m+1)},$$

leading to

$$c_1^{(t+1)} - c_2^{(t+1)} \geq c_1^{(t)} - c_2^{(t)} \geq c_1^{(t+1)} - c_2^{(t+1)} - \frac{C\mathsf{poly}(\log N)}{N},$$

where the last inequality uses induction hypothesis. The above expression contradicts (324), thus we have

$$c_1^{(t+1)} \geq c_2^{(t+1)} - \tilde{\mathcal{O}}\left(\frac{1}{N}\right).$$

### C.2.10 Proof of Lemma 18

Note that by our definitions of $w_0^{(k)}$, $w_1^{(k)}$ and Lemma 11 we could rewrite $\sum_{k=1}^{2m} \Delta_z^{(k)}$ as

$$\sum_{k=1}^{2m} \Delta_z^{(k)} = \sum_{k=1}^{m} \mathbb{E}\left[\left(w_0^{(t,k)}\right)^2\right] + \sum_{k=m+1}^{2m} \mathbb{E}\left[\left(w_1^{(t,k)}\right)^2\right]. \tag{325}$$

By a similar calculation as in Step 1 of the proof of Lemma 15, Lemma 17 (and the fact that the relations in Lemma 17 still hold after time $T_3^C$), we have for all $t \geq T_3^C + 1$:

$$\forall k \in [m]: \quad w_0^{(t,k)} \leq \frac{N}{\exp\left(0.47\mu_2^{(t)}\right) + N} \leq \sqrt{\frac{\epsilon}{6m}}, \tag{326}$$

$$\forall k \in [m+1, 2m]: \quad w_1^{(t,k)} \leq \frac{m+1}{\exp\left(0.47\mu_2^{(t)}\right) + m + 1} \leq \sqrt{\frac{\epsilon}{6m}}. \tag{327}$$

Plugging the above into (325), we have

$$\sum_{k=1}^{2m} \Delta_z^{(k)} \leq \frac{\epsilon}{3}.$$

### C.2.11 Proof of Lemma 23

We prove this lemma by contradiction. Assume that there exists the first time $t_0 \leq t$ such that

$$\sum_{k=1}^{m} \left(w_0^{(t_0,k)}\right)^2 w_1^{(t_0,k)} < \left(1 - \frac{C'm}{N}\right) \left(w_1^{(t_0,m+1)}\right)^2 w_0^{(t_0,m+1)} \tag{328}$$

for some absolute constant $C'$.

First note that when $t = 0$, we have

$$w_0^{(0,k)} = \begin{cases} \frac{N}{N+k}, & k \in [m] \\ \frac{N+k-m-1}{N+k}, & k \in \{m+1, \ldots, 2m\} \end{cases}, \quad w_1^{(0,k)} = \begin{cases} \frac{k}{N+k}, & k \in [m] \\ \frac{m+1}{N+k}, & k \in \{m+1, \ldots, 2m\} \end{cases}, \tag{329}$$

which implies that (246) holds when $t = 0$, thus we have $t_0 \geq 1$.

By (155) we have for any $t \in \mathbb{N}$:

$$\sum_{k=1}^{2m} \delta z_{3,0,1}^{(t,k)} = 2\sum_{k=1}^{m} \mathbb{E}\left[\left(w_0^{(t,k)}\right)^2 w_1^{(t,k)}\right] - 2\mathbb{E}\left[\left(w_1^{(t,m+1)}\right)^2 w_0^{(t,m+1)}\right]. \tag{330}$$

This implies that

$$\left|\sum_{k=1}^{2m} \delta z_{3,0,1}^{(k)}\right| \leq \frac{m}{2},$$

and thus by (184b) we have at time $t_0 - 1$,

$$V_{3,0,1}^{(t_0-1)} = V_{3,0,1}^{(t_0)} + \eta \sum_{k=1}^{2m} \delta z_{3,0,1}^{(t_0,k)} \leqslant V_{3,0,1}^{(t_0)} + C\frac{1}{2N}. \tag{331}$$

Similarly, by (151), (153) and (184a), we have

$$U_{l,i,j}^{(t_0-1)} = U_{l,i,j}^{(t_0)} + \mathcal{O}\left(\frac{C}{NS}\right). \tag{332}$$

Thus by the expression of $w_0^{(t,k)}$ and $w_1^{(t,k)}$ (c.f.(237), (238), (239), (240), (241) and (242)), we know that

$$\forall k \in [2m]: \quad w_0^{(t_0-1,k)} = \left(1 \pm \mathcal{O}\left(\frac{C}{N}\right)\right) w_0^{(t_0,k)}, \quad w_1^{(t_0-1,k)} = \left(1 \pm \mathcal{O}\left(\frac{C}{N}\right)\right) w_1^{(t_0,k)}, \tag{333}$$

and therefore, we have

$$\frac{\sum_{k=1}^{m} \left(w_0^{(t_0-1,k)}\right)^2 w_1^{(t_0-1,k)}}{\left(w_1^{(t_0-1,m+1)}\right)^2 w_0^{(t_0-1,m+1)}} \leqslant \left(1 + \mathcal{O}\left(\frac{C}{N}\right)\right) \frac{\sum_{k=1}^{m} \left(w_0^{(t_0,k)}\right)^2 w_1^{(t_0,k)}}{\left(w_1^{(t_0,m+1)}\right)^2 w_0^{(t_0,m+1)}}. \tag{334}$$

By (328) and the above expression, we know that when $C$ is small enough, we have

$$\frac{\sum_{k=1}^{m} \left(w_0^{(t_0-1,k)}\right)^2 w_1^{(t_0-1,k)}}{\left(w_1^{(t_0-1,m+1)}\right)^2 w_0^{(t_0-1,m+1)}} \leqslant 1 - \frac{C'm}{2N}. \tag{335}$$

By the above expression and (330) we know that

$$-\sum_{k=1}^{2m} \delta z_{3,0,1}^{(t_0-1,k)} \geqslant \frac{C'm}{N} \mathbb{E}\left[\left(w_1^{(t_0-1,m+1)}\right)^2 w_0^{(t_0-1,m+1)}\right]. \tag{336}$$

Thus by (184b) we know that $V_{3,0,1}$ (recall $V_{3,0,1} = -c_2\mu_2$) will increase from $t_0 - 1$ to $t_0$. , i.e.,

$$c_2^{(t_0)}\mu_2^{(t_0)} < c_2^{(t_0-1)}\mu_2^{(t_0-1)}. \tag{337}$$

Furthermore, by (219) we know that

$$\left|\sum_{k=1}^{2m} \delta z_{1,0,1}^{(t_0-1,k)}\right| \leqslant \frac{2m}{S} \mathbb{E}\left[\left(w_1^{(t_0-1,m+1)}\right)^2 w_0^{(t_0-1,m+1)}\right] \leqslant \frac{2C'N}{S}\left(-\sum_{k=1}^{2m} \delta z_{3,0,1}^{(t_0-1,k)}\right), \tag{338}$$

suggesting we could choose the absolute constant $C'$ large enough so that

$$(1 + 2c_2^{(t_0-1)})\mu_2^{(t_0-1)} > (1 + 2c_2^{(t_0)})\mu_2^{(t_0)}. \tag{339}$$

(243), (244) and the above expression imply that we could choose the absolute constant $C'$ large enough so that

$$\left(w_0^{(t_0,k)}\right)^2 \left(w_1^{(t_0,k)}\right) > \left(w_0^{(t_0-1,k)}\right)^2 \left(w_1^{(t_0-1,k)}\right), \tag{340}$$

$$\left(w_1^{(t_0,m+1)}\right)^2 \left(w_0^{(t_0,m+1)}\right) < \left(w_1^{(t_0-1,m+1)}\right)^2 \left(w_0^{(t_0-1,m+1)}\right). \tag{341}$$

Therefore, by (328) we have

$$\sum_{k=1}^{m} \left(w_0^{(t_0-1,k)}\right)^2 w_1^{(t_0-1,k)} < \left(1 - \frac{C'm}{N}\right) \left(w_1^{(t_0-1,m+1)}\right)^2 w_0^{(t_0-1,m+1)}. \tag{342}$$

This contradicts with our choice that $t_0$ is the first time that (328) is satisfied.

### C.2.12 Proof of Lemma 19

We first define

$$r^{(k)} := \sum_{i=1}^{N+k} \exp\left( x_i^{(k-1)\top} B_1 y_{-1}^{(k-1)} + y_i^{(k-1)\top} B_2 y_{-1}^{(k-1)} + z_i^{(k-1)\top} B_3 z_{-1}^{(k-1)} \right), \tag{343}$$

where we let $x_i^{(k-1)}, y_i^{(k-1)}, z_i^{(k-1)}$ be the $i$-th column of $X^{(k-1)}, Y^{(k-1)}, Z^{(k-1)}$ respectively. $r^{(k)}$ is the denominator of the attention score at the $k$-th reasoning step.

For any $i, j \in \{0, \dots, S\}$, we also define

$$\delta_{1,i,j}^{(t)} := \sum_{k=1}^{2m} \left( \delta x_{1,i,j}^{(t,k)} + \delta y_{1,i,j}^{(t,k)} \right), \tag{344}$$

and we define $\delta_2^{(t)}, \delta_3^{(t)}$ likewise. Then by (184a) we know that

$$U_l^{(t+1)} = U_l^{(t)} - \eta \delta_l^{(t)}. \tag{345}$$

From Lemma 3 we can see that each $\delta_l^{(t)}$ consists of some positive terms and some negative terms. We denote the sum of all the positive terms as $\delta_l^{(t,+)}$ and the absolute value of the sum of all the negative terms as $\delta_l^{(t,-)}$. Then we have

$$\delta_l^{(t)} = \delta_l^{(t,+)} - \delta_l^{(t,-)}. \tag{346}$$

With the above definitions, we prove Lemma 19 by induction. In addition, we'll also show that after $b_2^{(t)} > 0$, $b_2^{(t)} \mu_1^{(t)}$ (i.e., $U_{3,0,1}^{(t)}$) monotonically increases.

By our initialization, Lemma 19 and the above claim hold when $t = 0$. We assume Lemma 19 and the above claim holds at time $t$ ($t \in \{0, 1, \cdots, T_1^B - 1\}$) and prove it at time $t + 1$.

**Step 1: Showing (199a) holds at time $t + 1$.** We break discussion into two cases.

**Case 1: $b_2^{(t)} \leqslant 0$.** When $b_2^{(t)} \leqslant 0$, by Lemma 14 and induction hypothesis we can compute that

$$
\begin{aligned}
\delta_{2,j,j}^{(t,-)} &\asymp -\delta x_{2,j,j}^{(t,m+1)} \\
&\asymp \frac{1}{S} \mathbb{E}\left[ \left( 1 - \alpha_{(0,j,0),(j,1)}^{(t,m+1)} - \alpha_{(\tilde{c}_1(j),j,1),(j,1)}^{(t,m+1)} \right)^2 \left( \alpha_{(0,j,0),(j,1)}^{(t,m+1)} + \alpha_{(\tilde{c}_1(j),j,1),(j,1)}^{(t,m+1)} \right) \middle| n_1 = j \right] \\
&\asymp \frac{N^2 \exp\left( \left( 1 + b_2^{(t)} \right) \mu_1^{(t)} \right)}{S \left( r^{(t,m+1)} \right)^3}, 
\end{aligned} \tag{347}
$$

and

$$
\begin{aligned}
\delta_{2,j,j}^{(t,+)} &\asymp \sum_{k=2}^{m} \frac{1}{S} \mathbb{E}\left[ \left( \alpha_{(c_1(j),j,1),(j,1)}^{(k)} + \alpha_{(c_1(j),c_1^2(j),0),(j,1)}^{(k)} + \alpha_{(c_1(j),c_2(c_1(j)),0),(j,1)}^{(k)} \right) \alpha_{(c_1(j),j,1),(j,1)}^{(k)} \right. \\
&\qquad\qquad \left. \cdot \left( 1 - \alpha_{(p(j),j,0),(j,1)}^{(k)} - \alpha_{(\tilde{c}_1(j),j,1),(j,1)}^{(k)} \right) \middle| n^{-1} = j \right] \\
&\asymp \frac{mN \exp\left( \left( 2 - b_2^{(t)} \right) \mu_1^{(t)} \right)}{S \left( r^{(t,k)} \right)^3}
\end{aligned} \tag{348}
$$

for any $k \in \{2, \cdots, m\}$, where the second and third line is the second term of the expression of $\delta y_{2,j,j}^{(m+1)}$ given in (168).

By induction hypothesis we also have

$$r^{(t,m+1)} = \left( 1 + \tilde{\mathcal{O}}\left( \frac{1}{N} \right) \right) \left( (N-1) \exp\left( b_2^{(t)} \mu_1^{(t)} \right) + \exp\left( (b_2^{(t)} + 1 - a^{(t)}) \mu_1^{(t)} \right) \right)$$

$$+ \exp\left((-a^{(t)} - b_2^{(t)})\mu_1^{(t)}\right) + (m-1)\exp\left(-b_2^{(t)}\mu_1^{(t)}\right) + \exp\left((1 - b_2^{(t)})\mu_1^{(t)}\right)\right)$$

$$\asymp N \exp\left(b_2^{(t)}\mu_1^{(t)}\right), \tag{349}$$

where the second relation follows from (199d) and (199b), and for all $k \in \{2, \cdots, m\}$ we have

$$r^{(t,k)} = \left(1 + \tilde{\mathcal{O}}\left(\frac{1}{N}\right)\right)\left((N-2)\exp\left(b_2^{(t)}\mu_1^{(t)}\right) + \exp\left((b_2^{(t)} + 1)\mu_1^{(t)}\right) + \exp\left((-b_2^{(t)} - a^{(t)})\mu_1^{(t)}\right)\right.$$

$$\left. + \exp\left((b_2^{(t)} - a^{(t)})\mu_1^{(t)}\right) + (k-2)\exp\left(-b_2^{(t)}\mu_1^{(t)}\right) + \exp\left((1 - b_2^{(t)})\mu_1^{(t)}\right)\right)$$

$$\asymp N \exp\left(b_2^{(t)}\mu_1^{(t)}\right). \tag{350}$$

(note that this relations holds during Phase I.1-C, even when $b_2^{(t)} > 0$.) Thus we have

$$r^{(t,m+1)} \asymp r^{(t,k)}, \quad \forall k \in \{2, \cdots, m\}. \tag{351}$$

By (199b) we know that $\delta x_{2,j,j}^{(t,-)}$ dominates $\delta x_{2,j,j}^{(t,+)}$, and thus

$$-\delta_{2,j,j}^{(t)} \asymp \delta_{2,j,j}^{(t,-)} \asymp \frac{\exp\left(\left(1 - 2b_2^{(t)}\right)\mu_1^{(t)}\right)}{SN} \gtrsim \frac{1}{SN}, \tag{352}$$

where the second relation follows from (347), (351), and the last relation follows from (199b). (352) indicates that (199a) holds at time $t+1$ if $b_2^{(t)} \leqslant 0$,

**Case 2:** $b_2^{(t)} > 0$. When $b_2^{(t)} > 0$, by Lemma 14 and induction hypothesis we have

$$-\delta_{2,j,j}^{(t)} \asymp \delta_{2,j,j}^{(t,-)}$$

$$\asymp \sum_{k=1}^m \frac{1}{S}\mathbb{E}\left[\left(1 - \alpha_{(p(j),j,0),(j,1)}^{(k)} - \alpha_{(p(j),s(j),0),(j,1)}^{(k)}\right)\alpha_{(p(j),j,0),(j,1)}^{(k)}\right.$$

$$\left. \cdot \left(1 - \alpha_{(p(j),j,0),(j,1)}^{(k)} - \alpha_{(\tilde{c}_1(j),j,1),(j,1)}^{(k)}\right)\Big| n^{-1} = j\right]$$

$$\asymp \frac{mN^2 \exp\left(\left(1 + 3b_2^{(t)}\right)\mu_1^{(t)}\right)}{S\left(r^{(t,k)}\right)^3} \asymp \frac{m\exp\left(\mu_1^{(t)}\right)}{SN} \gtrsim \frac{1}{SN}. \tag{353}$$

where the first and second line corresponds to the first term of $\delta y_{2,j,j}^{(k)}$ given in (168), and $k \in \{2, \cdots, m\}$, and the last relation uses (350). Therefore, (352) holds at time $t+1$ if $b_2^{(t)} > 0$.

**Step 2: Showing** (199b) **holds at time** $t+1$ **if** $b_2^{(t)} > 0$**, then** $U_{3,0,1}^{(t+1)} \geqslant U_{3,0,1}^{(t)}$. We first show by contradiction that if $b_2^{(t+1)} \leqslant 0$, then we have

$$\exp\left((1 - 2b_2^{(t+1)})\mu_1^{(t+1)}\right) \lesssim \frac{N}{m}.$$

Suppose for contradiction that $b_2^{(t+1)} \leqslant 0$ and

$$\exp\left((1 - 2b_2^{(t+1)})\mu_1^{(t+1)}\right) > \frac{CN}{m}. \tag{354}$$

By the induction hypothesis we have

$$b_2^{(s)} \leqslant 0, \quad \forall s \in \{0, 1, \cdots, t\},$$

and there exists some absolute constant $C > 0$ such that

$$\exp\left((1 - 2b_2^{(s)})\mu_1^{(s)}\right) \leqslant \frac{CN}{m}, \quad \forall s \in \{0, 1, \cdots, t\}. \tag{355}$$

By (354), when the learning rate $\eta \lesssim \frac{1}{mN}$, we have

$$\exp\left((1 - 2b_2^{(t)})\mu_1^{(t)}\right) \geqslant \frac{CN}{2m}. \tag{356}$$

Using the induction hypothesis and Lemma 14, we can compute that

$$\delta_{3,0,1}^{(t,+)} \asymp \frac{1}{S}\mathbb{E}\left[\left(1 - \alpha_{(n_2,j,1),(j,1)}^{(t,m+1)} - \alpha_{(n_2,c_1(n_2),0),(j,1)}^{(t,m+1)} - \alpha_{(n_2,c_2(n_2),0),(j,1)}^{(t,m+1)}\right)\alpha_{(n_2,j,1),(j,1)}^{(t,m+1)}\right.$$

$$\left.\cdot\left(\sum_{q\in[S]}\alpha_{(\widetilde{p}(q),q,0),(j,1)}^{(t,m+1)}\right)\Bigg| n_1 = j\right]$$

$$+ \frac{1}{S}\mathbb{E}\left[\left(1 - \alpha_{(0,j,0),(j,1)}^{(t,m+1)} - \alpha_{(\widetilde{c}_1(j),j,1),(j,1)}^{(t,m+1)}\right)\alpha_{(0,j,0),(j,1)}^{(t,m+1)}\left(\sum_{q\in[S]}\alpha_{(\widetilde{p}(q),q,0),(j,1)}^{(t,m+1)}\right)\Bigg| n_1 = j\right]$$

$$\asymp \frac{N^2\exp\left(\left(1 + b_2^{(t)}\right)\mu_1^{(t)}\right)}{S\left(r^{(t,m+1)}\right)^3}, \tag{357}$$

where the second and third line corresponds to the fourth term of $\delta y_{3,0,1}^{(m+1)}$ given in (180), and the fourth line corresponds to the third term of $\delta x_{3,0,1}^{(m+1)}$ given in (174). By the above expression and (356) we have

$$\delta_{2,j,j}^{(t,-)} + 2\delta_{3,0,1}^{(t,+)} \asymp \frac{N^2\exp\left(\left(1 + b_2^{(t)}\right)\mu_1^{(t)}\right)}{S\left(r^{(t,m+1)}\right)^3} \lesssim \frac{mN}{CS\left(r^{(t,m+1)}\right)^3}\exp\left((2 - b_2^{(t)})\mu_1^{(t)}\right). \tag{358}$$

By (348), (351) and the above expression we have when $C$ is large enough, $\delta_{2,j,j}^{(t,-)} + 2\delta_{3,0,1}^{(t,+)}$ is dominated by $\delta_{2,j,j}^{(t,+)}$, leading to

$$\delta_{2,j,j}^{(t,-)} - \delta_{2,j,j}^{(t,+)} - 2\left(\delta_{3,0,1}^{(t,-)} - \delta_{3,0,1}^{(t,+)}\right) \leqslant 0. \tag{359}$$

This implies that

$$\left(1 - 2b_2^{(t+1)}\right)\mu_1^{(t+1)} \leqslant \left(1 - 2b_2^{(t)}\right)\mu_1^{(t)}, \tag{360}$$

and thus by (355) we have

$$\exp\left((1 - 2b_2^{(t+1)})\mu_1^{(t+1)}\right) \leqslant \frac{CN}{m},$$

which contradicts with (354).

We next show

$$b_2^{(t+1)} \leqslant \frac{1}{2}$$

by contradiction. If $b_2^{(t+1)} > \frac{1}{2}$, then when $\eta \lesssim \frac{1}{mN}$, we have

$$b_2^{(t)} \geqslant \frac{1}{2} - \mathcal{O}\left(\frac{1}{N}\right). \tag{361}$$

We can compute by Lemma 14 and induction hypothesis that

$$\delta_{3,0,1}^{(t,-)} \asymp -\sum_{k=1}^{m}\delta y_{3,0,1}^{(k)} \asymp \frac{mN\exp\left(\left(2 + b_2^{(t)}\right)\mu_1^{(t)}\right)}{S\left(r^{(t,k)}\right)^3} \overset{(353)}{\lesssim} \mathcal{O}\left(\frac{1}{N}\right)\left(-\delta_{2,j,j}^{(t)}\right), \tag{362}$$

where the last step also uses (361). This implies that from step $t$ to step $t + 1$,

$$\mu_1^{(t+1)} - \mu_1^{(t)} \gtrsim N\left(b_2^{(t+1)}\mu_1^{(t+1)} - b_2^{(t)}\mu_1^{(t)}\right),$$

and thus

$$b_2^{(t+1)} \leqslant b_2^{(t)} \leqslant \frac{1}{2}. \tag{363}$$

This leads to a contradiction.

Lastly, if $b_2^{(t)} > 0$, by (362) and (357) we have

$$\frac{\delta_{3,0,1}^{(t,-)}}{\delta_{3,0,1}^{(t,+)}} \mu_1^{(t)} \asymp \frac{m \exp\left(\mu_1^{(t)}\right)}{N},$$

indicating that $-\delta_{3,0,1}^{(t)} = \delta_{3,0,1}^{(t,-)} - \delta_{3,0,1}^{(t,+)}$ increases with $t$ after $b_2^{(t)} \geqslant 0$. Therefore, $U_{3,0,1}^{(t+1)} \geqslant U_{3,0,1}^{(t)}$.

**Step 3: Showing** (199c) **holds at time** $t+1$. This can be shown by following a similar calculation as in Step 3 in the proof of Lemma 15, and we omit the details.

**Step 4: Showing** (199d) **holds at time** $t+1$. If $a^{(t+1)} \geqslant 0$, then by our choice of $T_1^C$ we know that (199d) holds at time $t+1$. Below we consider the case when $a^{(t+1)} < 0$. By induction hypothesis and Lemma 14, we can compute that

$$\delta_{1,0,j}^{(t,-)} \asymp \frac{1}{S} \mathbb{E}\left[ \left(1 - \alpha_{(0,j,0),(j,1)}^{(t,m+1)} - \alpha_{(\tilde{c}_1(j),j,1),(j,1)}^{(t,m+1)}\right) \alpha_{(0,j,0),(j,1)}^{(t,m+1)} \left(1 - \alpha_{(0,j,0),(j,1)}^{(t,m+1)} - \alpha_{(n_2m,j,1),(j,1)}^{(t,m+1)}\right) \bigg| n_1 = j \right]$$

$$\asymp \frac{N^2 \exp\left(\left(3b_2^{(t)} + 1 - a^{(t)}\right)\mu_1^{(t)}\right)}{S\left(r^{(t,m+1)}\right)^3}. \tag{364}$$

where the first two lines correspond to the first term of $\delta x_{1,0,j}^{(m+1)}$ given in (159).

We could also compute that

$$\delta_{1,0,j}^{(t,+)} \gtrsim \frac{1}{S} \mathbb{E}\left[ \left(\alpha_{(0,j,0),(j,1)}^{(t,m+1)} + \alpha_{(0,n_2m,1),(j,1)}^{(t,m+1)}\right)^2 \left(1 - \alpha_{(0,j,0),(j,1)}^{(t,m+1)} - \alpha_{(0,n_2m,1),(j,1)}^{(t,m+1)}\right) \bigg| n_1 = j \right]$$

$$\asymp \frac{N \exp\left(\left(3b_2^{(t)} + 2 - 2a^{(t)}\right)\mu_1^{(t)}\right)}{S\left(r^{(t,m+1)}\right)^3}. \tag{365}$$

By comparing the above expressions and using a contradiction argument similar as in Step 2 in the proof of Lemma 20, we can show that

$$\exp\left((1 - a^{(t)})\mu_1^{(t)}\right) \lesssim N. \tag{366}$$

**Step 5: Showing** $U_{1,0,j}^{(t+1)} > U_{1,0,j}^{(t)}$.

By Lemma 14 and induction hypothesis, we can compute that for all $k \in \{m + 2, \cdots, m + 1 + 2^m\}$,

$$\delta x_{3,0,0}^{(k)} \asymp \frac{1}{S} \mathbb{E}\left[ \left(1 - \alpha_{(p(j),j,0),(j,0)}^{(k)} - \alpha_{(c_1(j),j,1),(j,0)}^{(k)}\right) \alpha_{(c_1(j),j,1),(j,0)}^{(k)} \left(\sum_{q \in [S]} \alpha_{(\tilde{p}(q),q,0),(j,0)}^{(k)}\right) \bigg| n_{-1} = j \right]$$

$$\asymp \frac{(2\exp(2\mu_1) + N\exp(\mu_1))(N\exp(-b_1\mu_1) + (m-1)\exp(b_1\mu_1))}{S\left(r^{(k)}\right)^3}, \tag{367}$$

and

$$\delta y_{3,0,0}^{(k)} \asymp \frac{1}{S} \mathbb{E}\left[ \left(1 - \alpha_{(i,c_1(i),0),(j,0)}^{(k)} - \alpha_{(i,c_2(i),0),(j,0)}^{(k)} - \alpha_{(i,j,1),(j,0)}^{(k)}\right) \alpha_{(i,j,1),(j,0)}^{(k)} \left(\sum_{q \in [S]} \alpha_{(\tilde{p}(q),q,0),(j,0)}^{(k)}\right) \bigg| n_{-1} = j \right]$$

$$\asymp \frac{(N\exp((1 - b_1)\mu_1) + 2\exp((2 - b_1)\mu_1) + \exp((1 + b_1)\mu_1))(N + 2\exp(\mu_1))}{S\left(r^{(k)}\right)^3}. \tag{368}$$

Recall that Lemma 14 gives that for all $k \in [m+1]$:

$$\delta x_{3,0,0}^{(k+m+1)} = \delta y_{3,1,0}^{(k)} = 0. \tag{369}$$

Thus by the above expressions we can see that

$$\delta_{3,0,0}^{(t)} = \sum_{k=m+2}^{m+1+2^m} \delta x_{3,0,0}^{(t,k)} + \sum_{k=1}^{m} \delta y_{3,0,0}^{(t,k)}$$

is strictly positive, indicating that $U_{3,0,0}^{(t+1)} < U_{3,0,0}^{(t)}$.

### C.2.13  Proof of Lemma 20

We follow the same notation as in the proof of Lemma 19. We first show (202) holds by induction.

From Lemma 19 we know that when $t = T_1^B$, (202) holds. We assume that (202) holds for all $s \in \{T_1^B, T_1^B + 2, \cdots, t\}$ ($T_1^B \leqslant t < T_2^B$). We next show that (202) also holds for $t+1$.

**Step 1: Showing $\mu_1^{(t+1)} > \mu_1^{(t)}$ and (202a) holds for $t+1$.** Similar to the calculation in Step 1 of Lemma 19, we can compute that (note $\exp(\mu_1^{(t)}) \gtrsim N$ by induction hypothesis, $\exp\left((1 - 2b_2^{(t)})\mu_1^{(t)}\right) \lesssim N$ by our choice of $T_1^B$)

$$\forall k \in [m]: \quad \frac{r^{(t,m+1)}}{r^{(t,k)}} \asymp \frac{N}{\exp\left(\mu_1^{(t)}\right)}, \tag{370}$$

since

$$r^{(t,m+1)} \asymp N \exp(b_2^{(t)}\mu_1^{(t)}), \quad r^{(t,k)} \asymp \exp\left((1 + b_2^{(t)})\mu_1^{(t)}\right), \quad \forall k \in [m]. \tag{371}$$

And we can compute that

$$\delta_{2,j,j}^{(t,+)} \asymp \frac{mN \exp\left(\left(2 - b_2^{(t)}\right)\mu_1^{(t)}\right)}{S\left(r^{(t,k)}\right)^3}, \tag{372}$$

$$\delta_{2,j,j}^{(t,-)} \asymp \frac{N^2 \exp\left(\left(1 + b_2^{(t)}\right)\mu_1^{(t)}\right)}{S\left(r^{(t,m+1)}\right)^3} + \frac{mN^2 \exp\left(\left(1 + 3b_2^{(t)}\right)\mu_1^{(t)}\right)}{S\left(r^{(t,k)}\right)^3}. \tag{373}$$

By (370) and the above two expressions we can see that $\delta_{2,j,j}^{(t,-)}$ dominates $\delta_{2,j,j}^{(t,+)}$, and thus

$$-\delta_{2,j,j}^{(t)} = \delta_{2,j,j}^{(t,-)} \gtrsim \frac{N^2 \exp\left(\left(1 + b_2^{(t)}\right)\mu_1^{(t)}\right)}{S\left(r^{(t,m+1)}\right)^3} \asymp \frac{\exp\left(\left(1 - 2b_2^{(t)}\right)\mu_1^{(t)}\right)}{SN} \geqslant \frac{1}{SN}, \tag{374}$$

where the last inequality follows from the induction hypothesis that $b_2^{(t)} \leqslant \frac{1}{2}$. The above expression implies that (202a) holds for $t+1$, and $\mu_1^{(t+1)} > \mu_1^{(t)}$.

**Step 2: Showing (202b) holds for $t+1$.** By Lemma 14 and induction hypothesis, we can compute that

$$\delta_{1,0,j}^{(t,+)} \asymp \delta y_{1,0,j}^{(t,m+1)}$$

$$\asymp \frac{1}{S} \sum_{i \in [S]} \mathbb{E}\Bigg[ \mathbb{1}\{n_{2^{k-m}} = i\} \left(\alpha_{(0,j,0),(j,0)}^{(t,k)} + \alpha_{(0,n_{2^m},1),(j,0)}^{(t,k)}\right)^2 \left(1 - \alpha_{(0,j,0),(j,0)}^{(t,k)} - \alpha_{(0,n_{2^m},1),(j,0)}^{(t,k)}\right)$$

$$+ \left(1 - \alpha_{(i,c_1(i),0),(j,0)}^{(t,k)} - \alpha_{(i,c_2(i),0),(j,0)}^{(t,k)} - \alpha_{(i,j,1),(j,0)}^{(t,k)}\right) \left(\alpha_{(i,c_1(i),0),(j,0)}^{(t,k)} + \alpha_{(i,c_2(i),0),(j,0)}^{(t,k)} + \alpha_{(i,j,1),(j,0)}^{(t,k)}\right)$$

$$\cdot \left(\alpha_{(0,j,0),(j,0)}^{(t,k)} + \alpha_{(0,n_{2^m},1),(j,0)}^{(t,k)}\right) \Bigg| n_1 = j \Bigg]$$

$$\asymp \underbrace{\frac{N \exp\left(\left(3b_2^{(t)} + 2 - 2a^{(t)}\right)\mu_1^{(t)}\right)}{S\left(r^{(t,m+1)}\right)^3}}_{(a)} + \underbrace{\frac{\exp\left(\left(b_2^{(t)} + 2 - a^{(t)}\right)\mu_1^{(t)}\right)\left(N + \exp\left(\left(1 - a^{(t)}\right)\mu_1^{(t)}\right)\right)}{S\left(r^{(t,m+1)}\right)^3}}_{(b)}.$$

(375)

And same as (364), here we also have

$$\delta_{1,0,j}^{(t,-)} \asymp \frac{N^2 \exp\left(\left(3b_2^{(t)} + 1 - a^{(t)}\right)\mu_1^{(t)}\right)}{S\left(r^{(t,m+1)}\right)^3}.$$

(376)

We show by contradiction that

$$\exp\left((1 - a^{(t)})\mu_1^{(t)}\right) \lesssim N.$$

(377)

By the induction hypothesis, there exists some absolute constant $C > 0$ such that

$$\exp\left((1 - a^{(s)})\mu_1^{(s)}\right) \leqslant CN, \quad \forall s \in \{T_1^B, T_1^B + 2, \cdots, t\}.$$

(378)

If at time $t + 1$, we can compute that

$$\exp\left((1 - a^{(t)})\mu_1^{(t)}\right) > CN,$$

(379)

then when $\eta \lesssim \frac{1}{mN}$, we have

$$\exp\left((1 - a^{(t)})\mu_1^{(t)}\right) > \frac{CN}{2}.$$

(380)

When $C$ is large enough, by (375) and (376) we have the term (a) is larger than $\delta_{1,0,j}^{(t,-)}$, and thus

$$\delta_{1,0,j}^{(t)} \asymp \delta_{1,0,j}^{(t,+)}.$$

(381)

Moreover, we have

$$r^{(t,m+1)} \asymp \exp\left(\left(1 + b_2^{(t)} - a^{(t)}\right)\mu_1^{(t)}\right), \quad r^{(t,k)} \asymp \exp\left(\left(1 + b_2^{(t)}\right)\mu_1^{(t)}\right), \quad \forall k \in [m], \quad (382)$$

and in Step 1 we have computed that

$$-\delta_{2,j,j}^{(t)} \asymp \underbrace{\frac{N^2 \exp\left(\left(1 + b_2^{(t)}\right)\mu_1^{(t)}\right)}{S\left(r^{(t,m+1)}\right)^3}}_{(c)} + \underbrace{\frac{mN^2 \exp\left(\left(1 + 3b_2^{(t)}\right)\mu_1^{(t)}\right)}{S\left(r^{(t,k)}\right)^3}}_{(d)}.$$

(383)

By (375) we have that

$$(b) \gtrsim \frac{N \exp\left(\left(b_2^{(t)} + 2 - a^{(t)}\right)\mu_1^{(t)}\right)}{S\left(r^{(t,m+1)}\right)^3} \gtrsim \frac{CN^2 \exp\left(\left(1 + b_2^{(t)} - a^{(t)}\right)\mu_1^{(t)}\right)}{S\left(r^{(t,m+1)}\right)^3} \gtrsim C \times (c), \quad (384)$$

where the last step uses (380). By (375), (383) and (382) we have that

$$(d) = \frac{mN^2 \exp\left(\left(1 + 3b_2^{(t)}\right)\mu_1^{(t)}\right)}{S\left(r^{(t,m+1)}\right)^3}\frac{\left(r^{(t,m+1)}\right)^3}{\left(r^{(t,k)}\right)^3}$$

$$\overset{(382)}{\asymp} \frac{mN^2 \exp\left(\left(1 + 3b_2^{(t)} - 3a^{(t)}\right)\mu_1^{(t)}\right)}{S\left(r^{(t,m+1)}\right)^3} = \frac{mN^2 \exp\left(\left(1 + 3b_2^{(t)} - 3a^{(t)}\right)\mu_1^{(t)}\right)}{S\left(r^{(t,m+1)}\right)^3}. \quad (385)$$

where $k \in [m]$ is arbitrary. On the other hand, by (375) and (380) we have that

$$(a) \gtrsim \frac{CN^2 \exp\left(\left(3b_2^{(t)} + 1 - a^{(t)}\right)\mu_1^{(t)}\right)}{S\left(r^{(t,m+1)}\right)^3} \gtrsim C \times (d).$$

(386)

By (384) and (386) we have

$$\delta_{1,0,j}^{(t)} \gtrsim -C\delta_{2,j,j}^{(t)}. \tag{387}$$

This indicates that we could choose $C$ large enough such that

$$\left(1 - a^{(t+1)}\right)\mu_1^{(t+1)} < \left(1 - a^{(t)}\right)\mu_1^{(t)}. \tag{388}$$

Combining this with (378) we have that

$$\exp\left((1 - a^{(t+1)})\mu_1^{(t+1)}\right) \leqslant CN. \tag{389}$$

This contradicts with (379).

**Step 3: Showing** (202c) **holds for** $t + 1$**.** Same as (362), (357) given in the proof of Lemma 19, here we also have

$$\delta_{3,0,1}^{(t,-)} \asymp -\sum_{k=1}^{m} \delta y_{3,0,1}^{(k)} \asymp \frac{mN \exp\left(\left(2 + b_2^{(t)}\right)\mu_1^{(t)}\right)}{S\left(r^{(t,k)}\right)^3} \tag{390}$$

and

$$\delta_{3,0,1}^{(t,+)} \asymp \frac{N^2 \exp\left(\left(1 + b_2^{(t)}\right)\mu_1^{(t)}\right)}{S\left(r^{(t,m+1)}\right)^3} \gtrsim \frac{N^2 \exp\left(\left(1 + b_2^{(t)}\right)\mu_1^{(t)}\right)}{S\left(r^{(t,k)}\right)^3} \tag{391}$$

where the last line uses (370). Thus following the same contradiction argument as in Step 2 in the proof of Lemma 19, we can show that $b_2^{(t+1)} \leqslant 1/2$.

On the other hand, by our choice of $T_2^B$ we have that

$$r^{(t,m+1)} \gtrsim \exp\left(\left(1 - b_2^{(t)}\right)\mu_1^{(t)}\right). \tag{392}$$

This together with (371), (391) yields

$$
\begin{aligned}
\delta_{3,0,1}^{(t,+)} &\lesssim \frac{N^2 \exp\left(\left(1 + b_2^{(t)}\right)\mu_1^{(t)}\right)}{S\left(r^{(t,k)}\right)^3} \cdot \exp\left(6b_2^{(t)}\mu_1^{(t)}\right) \\
&= \frac{N^2 \exp\left(\left(1 + 7b_2^{(t)}\right)\mu_1^{(t)}\right)}{S\left(r^{(t,k)}\right)^3} \lesssim \frac{N \exp\left(\left(2 + 7b_2^{(t)}\right)\mu_1^{(t)}\right)}{S\left(r^{(t,k)}\right)^3}.
\end{aligned}
\tag{393}
$$

By comparing the above expression with (390) we have that if $b_2^{(t)} \leqslant 0$, $\delta_{3,0,1}^{(t,-)}$ dominates $\delta_{3,0,1}^{(t,+)}$, and thus it can also be shown by contradiction that $b_2^{(t+1)} \geqslant 0$.

**Step 4: Showing** (202d) **holds for** $t + 1$**, and** $U_{3,1,0}^{(t+1)} > U_{3,1,0}^{(t)}$**.** (202d) can be shown by following a similar calculation as in Step 3 in the proof of Lemma 15, and $U_{3,1,0}^{(t+1)} > U_{3,1,0}^{(t)}$ can be shown by following the same calculation as in Step 5 in the proof of Lemma 19. We omit the details here. The induction is complete.

We now move on to show (203).

**Step 5: Showing** (203a) **holds.** After

$$\exp\left((1 - 2b_2^{(T_2^B)})\mu_1^{(T_2^B)}\right) \asymp N, \quad \exp\left(\left(1 - a^{(T_2^B)}\right)\mu_1^{(T_2^B)}\right) \asymp N, \tag{394}$$

we can compute that

$$r^{(T_2^B,m+1)} \asymp \exp\left((1 - b_2^{(T_2^B)})\mu_1^{(T_2^B)}\right), \quad r^{(T_2^B,k)} \asymp \exp\left((1 + b_2^{(T_2^B)})\mu_1^{(T_2^B)}\right), \quad \forall k \in [m], \tag{395}$$

and

$$\delta_{3,0,1}^{(t,-)} \asymp - \sum_{k=1}^{m} \delta y_{3,0,1}^{(t,k)} \asymp \frac{mN \exp\left(\left(3 - b_2^{(t)}\right)\mu_1^{(t)}\right)}{S\left(r^{(t,k)}\right)^3} \asymp \frac{mN \exp\left(\left(3 - 7b_2^{(t)}\right)\mu_1^{(t)}\right)}{S\left(r^{(t,m+1)}\right)^3}, \quad (396)$$

$$\delta_{3,0,1}^{(t,+)} \asymp \delta y_{3,0,1}^{(t,m+1)} \asymp \frac{N^2 \exp\left(\left(3 - 2a^{(t)} + b_2^{(t)}\right)\mu_1^{(t)}\right)}{S\left(r^{(t,m+1)}\right)^3}. \quad (397)$$

Note that when $b_2^{(t)} = 1/4$, the above two terms are equal in magnitude. If $\delta_{3,0,1}^{(t,-)}$ (resp. $\delta_{3,0,1}^{(t,+)}$) dominates $\delta_{3,0,1}^{(t,+)}$ (resp. $\delta_{3,0,1}^{(t,-)}$), then $b_2^{(t+1)}$ will increase (resp. decrease) to near $1/4$. Thus by contradiction similar as in Step 3 we can show that when $C_0$ in (201) is large enough, there exists $s \leqslant T_2^B$ such that

$$\forall t \in \{s, s+1, \cdots, T_2^B\}: \quad b_2^{(t)} = \left(\frac{1}{4} \pm \tilde{\mathcal{O}}\left(\frac{1}{N}\right)\right) a^{(t)}. \quad (398)$$

**Step 6: Showing** (203a) **holds.** When (394) is satisfied, by Lemma 14 we can compute that

$$\delta_{1,0,j}^{(t,-)} \asymp \frac{1}{S}\mathbb{E}\left[ -\left(1 - \alpha_{(0,j,0),(j,1)}^{(t,m+1)} - \alpha_{(\tilde{c}_1(j),j,1),(j,1)}^{(t,m+1)}\right)\alpha_{(0,j,0),(j,1)}^{(t,m+1)}\left(1 - \alpha_{(0,j,0),(j,1)}^{(t,m+1)} - \alpha_{(0,n_{2^m},1),(j,1)}^{(t,m+1)}\right)\Big| n_1 = j\right]$$

$$\asymp \frac{N \exp\left(\left(b_2^{(t)} + 2 - a^{(t)}\right)\mu_1^{(T_2^B)}\right)}{S\left(r^{(t,m+1)}\right)^3} \gtrsim \delta_{1,0,j}^{(t,+)}. \quad (399)$$

In addition,

$$\delta_{1,0,j}^{(t,+)} \asymp \delta y_{1,0,j}^{(t,m+1)}$$

$$\asymp \frac{1}{S}\sum_{i \in [S]}\mathbb{E}\Bigg[ \mathbb{1}\{n_{2^{k-m}} = i\}\left(\alpha_{(0,j,0),(j,0)}^{(t,k)} + \alpha_{(0,n_{2^m},1),(j,0)}^{(t,k)}\right)^2\left(1 - \alpha_{(0,j,0),(j,0)}^{(t,k)} - \alpha_{(0,n_{2^m},1),(j,0)}^{(t,k)}\right)$$

$$+ \left(1 - \alpha_{(i,c_1(i),0),(j,0)}^{(t,k)} - \alpha_{(i,c_2(i),0),(j,0)}^{(t,k)} - \alpha_{(i,j,1),(j,0)}^{(t,k)}\right)\left(\alpha_{(i,c_1(i),0),(j,0)}^{(t,k)} + \alpha_{(i,c_2(i),0),(j,0)}^{(t,k)} + \alpha_{(i,j,1),(j,0)}^{(t,k)}\right)$$

$$\cdot \left(\alpha_{(0,j,0),(j,0)}^{(t,k)} + \alpha_{(0,n_{2^m},1),(j,0)}^{(t,k)}\right)\Big| n_1 = j\Bigg]$$

$$\asymp \underbrace{\frac{N \exp\left(\left(3b_2^{(t)} + 2 - 2a^{(t)}\right)\mu_1^{(t)}\right)}{S\left(r^{(t,m+1)}\right)^3}}_{(a)} + \underbrace{\frac{\exp\left(\left(b_2^{(t)} + 2 - a^{(t)}\right)\mu_1^{(t)}\right)\left(N + \exp\left(\left(1 - a^{(t)}\right)\mu_1^{(t)}\right)\right)}{S\left(r^{(t,m+1)}\right)^3}}_{(b)}.$$

$$(400)$$

By comparing the above two expressions we can see that if

$$\left(1 - a^{(t)}\right)\mu_1^{(t)} \lesssim CN$$

for some small enough constant $C > 0$, then $\delta_{1,0,j}^{(t,-)}$ dominates $\delta_{1,0,j}^{(t,+)}$, causing $a^{(t+1)}\mu_1^{(t+1)} < a^{(t)}\mu_1^{(t)}$. And since $\mu_1^{(t+1)} > \mu_1^{(t)}$, $(1 - a^{(t+1)})\mu_1^{(t+1)}$ increases faster than $\mu_1^{(t)}$ and could reacch $\mathcal{O}(N)$ before $t$ reaches $T_2^B$.

### C.2.14 Proof of Lemma 21

**Proof of** (206). We use the same notation as in the proof of Lemma 19, and prove (206) by induction. By Lemma 20 we know that that (206) holds for $t = T_2^B$. We assume that (206) holds for $s \in \{T_2^B, T_2^B + 1, \cdots, t\}$, and will show that it continues to hold for $t + 1$. In fact, the relations (206b), (206c) and (206d) hold for $t + 1$, as well as that $b_1^{(t+1)}\mu_1^{(t+1)} > b_1^{(t)}\mu_1^{(t)}$ can be shown by

following a similar calculation as in the proof of Lemma 20. We omit the details here and only compute that (206a) holds for $t + 1$.

Similar as in the proof of Lemma 20, we can compute that

$$
r^{(t,k)} = \begin{cases} \exp\left(\tilde{\mathcal{O}}\left(\pm\frac{\mu_1^{(t)}}{N}\right)\right)\exp\left((1 + b_2^{(t)})\mu_1^{(t)}\right), & k \in [m], \\ \exp\left(\tilde{\mathcal{O}}\left(\pm\frac{\mu_1^{(t)}}{N}\right)\right)\exp\left((1 - b_2^{(t)})\mu_1^{(t)}\right), & k = m + 1, \\ \exp\left(\tilde{\mathcal{O}}\left(\pm\frac{\mu_1^{(t)}}{N}\right)\right)\exp\left((1 + b_1^{(t)})\mu_1^{(t)}\right), & k \in \{m + 2, \cdots, 2m\}. \end{cases} \tag{401}
$$

Then following a similar calculation as in the proof of Lemma 20, we can compute that

$$
\begin{aligned}
\delta_{2,j,j}^{(t)} &= \exp\left(\pm\tilde{\mathcal{O}}\left(\frac{\mu_1^{(t)}}{N}\right)\right)\frac{N^2\exp\left(\left(1 + b_2^{(t)}\right)\mu_1^{(t)}\right)}{S\left(r^{(t,m+1)}\right)^3} \\
&= \exp\left(\pm\tilde{\mathcal{O}}\left(\frac{\mu_1^{(t)}}{N}\right)\right)\frac{N^2\exp\left(\left(4b_2^{(t)} - 2\right)\mu_1^{(t)}\right)}{S} \\
&\geqslant \frac{\exp\left((4\times 0.24 - 2 - 0.01)\mu_1^{(T_3^B)}\right)}{S} \overset{(205)}{\geqslant} \frac{1}{S}\left(\frac{\epsilon}{6m}\right)^{3/2}
\end{aligned} \tag{402}
$$

when $\epsilon$ is small enough, where the last step uses the induction hypothesis, and thus (206a) holds for $t + 1$.

**Proof of** (207). By (206b) we know that when $\epsilon$ is small enough, (207a) holds. For (207b), by computation we could verify that (367), (368) is still valid here, and by which we deduce that when $\mu_1^{(t)}$ is large enough:

$$
\begin{aligned}
\delta_{3,0,0}^{(t)} &= \exp\left(\pm\tilde{\mathcal{O}}\left(\frac{\mu_1^{(t)}}{N}\right)\right)\left(\sum_{k=m+1}^{2m}\frac{\left(2\exp\left(2\mu_1^{(t)}\right) + N\exp\left(\mu_1^{(t)}\right)\right)\left(N\exp\left(-b_1^{(t)}\mu_1^{(t)}\right) + (m-1)\exp(b_1^{(t)}\mu_1^{(t)})\right)}{S\left(r^{(t,k)}\right)^3}\right. \\
&\quad \left. + \frac{\left(N\exp\left(\left(1 - b_1^{(t)}\right)\mu_1^{(t)}\right) + 2\exp\left(\left(2 - b_1^{(t)}\right)\mu_1^{(t)}\right) + \exp\left((1 + b_1^{(t)})\mu_1^{(t)}\right)\right)\left(N + 2\exp(\mu_1^{(t)})\right)}{S\left(r^{(t,k)}\right)^3}\right) \\
&\overset{(401)}{=} \exp\left(\pm\tilde{\mathcal{O}}\left(\frac{\mu_1^{(t)}}{N}\right)\right)\frac{\exp\left(\left(3 - b_1^{(t)}\right)\mu_1^{(t)}\right) + \exp\left(\left(2 + b_1^{(t)}\right)\mu_1^{(t)}\right)}{S\left(\left(3 + 3b_1^{(t)}\right)\mu_1^{(t)}\right)^3}.
\end{aligned} \tag{403}
$$

On the other hand, we can verify by straightforward calculation that (396) and (397) are still valid, by which we have

$$
-\delta_{3,0,1}^{(t)} \lesssim \exp\left(\pm\tilde{\mathcal{O}}\left(\frac{\mu_1^{(t)}}{N}\right)\right)\frac{\exp\left(\left(3 - b_2^{(t)}\right)\mu_1^{(t)}\right)}{S\exp\left(\left(3 + 3b_2^{(t)}\right)\mu_1^{(t)}\right)}. \tag{404}
$$

By comparing the above two expressions and using contradiction similar as, for example, Step 2 in the proof of Lemma 20, we can show that

$$
b_1^{(t)} \geqslant b_2^{(t)} - \tilde{\mathcal{O}}\left(\frac{1}{N}\right).
$$

### C.2.15 Proof of Lemma 22

By Lemma 11 we know that for all $k \in [m]$, we have

$$
\Delta_x^{(t,k)} = \frac{1}{2}\sum_{j=1}^{S}\mathbb{E}\left[\mathbb{1}\{n_{-1} = j\}\left\{\left(1 - \alpha_{(p(j),j,0),(j,1)}^{(t,k)} - \alpha_{(\tilde{c}_1(j),j,1),(j,1)}^{(t,k)}\right)^2\right.\right.
$$

$$+ \sum_{q \in [S] \setminus \{j\}} \left( \alpha^{(t,k)}_{(p(q),q,0),(j,1)} + \alpha^{(t,k)}_{(\tilde{c}_1(q),q,1),(j,1)} \right)^2 \Bigg\} \Bigg]$$

$$\leqslant \sum_{j=1}^{S} \mathbb{E} \left[ \mathbb{1}\{n_{-1} = j\} \left\{ \left( 1 - \alpha^{(t,k)}_{(p(j),j,0),(j,1)} - \alpha^{(t,k)}_{(\tilde{c}_1(j),j,1),(j,1)} \right)^2 \right\} \right]$$

$$\leqslant \exp\left( \pm \tilde{\mathcal{O}}\left( \frac{\mu_1^{(t)}}{N} \right) \right) \left( \frac{(N + 2m - 1) \exp\left( b_2^{(t)} \mu_1^{(t)} \right)}{\exp\left( \left(1 + b_2^{(t)}\right) \mu_1^{(t)} \right) + (N + 2m - 1) \exp\left( b_2^{(t)} \mu_1^{(t)} \right)} \right)^2 .$$

(405)

where the second inequality follows from that fact that

$$(u_1 + \ldots + u_n)^2 \geqslant \sum_{i=1}^{n} u_i^2, \quad \forall u_i \geqslant 0,$$

and the last relation uses Lemma 21 (and the fact that the relations other than (206a) in Lemma 21 still hold after time $T_3^B$).

Following the same computation we can obtain that when $k = m + 1$,

$$\Delta_x^{(t,m+1)} = \exp\left( \pm \tilde{\mathcal{O}}\left( \frac{\mu_1^{(t)}}{N} \right) \right) \left( \frac{(N + 2m - 1) \exp\left( b_2^{(t)} \mu_1^{(t)} \right)}{\exp\left( \left(1 - b_2^{(t)}\right) \mu_1^{(t)} \right) + (N + 2m - 1) \exp\left( b_2^{(t)} \mu_1^{(t)} \right)} \right)^2 ,$$

(406)

when $k \in \{m + 2, \cdots, 2m\}$,

$$\Delta_x^{(t,k)} = \exp\left( \pm \tilde{\mathcal{O}}\left( \frac{\mu_1^{(t)}}{N} \right) \right) \left( \frac{(N + 2m - 1) \exp\left( b_2^{(t)} \mu_1^{(t)} \right)}{\exp\left( \left(1 + b_1^{(t)}\right) \mu_1^{(t)} \right) + (N + 2m - 1) \exp\left( b_2^{(t)} \mu_1^{(t)} \right)} \right)^2 ,$$

(407)

and for $\Delta_y^{(t,k)}$, we have for all $k \in [m]$,

$$\Delta_y^{(t,k)} = \exp\left( \pm \tilde{\mathcal{O}}\left( \frac{\mu_1^{(t)}}{N} \right) \right) \left( \frac{(N + 2m - 1) \exp\left( 1 - b_2^{(t)} \mu_1^{(t)} \right)}{\exp\left( \left(1 + b_2^{(t)}\right) \mu_1^{(t)} \right) + (N + 2m - 1) \exp\left( 1 - b_2^{(t)} \mu_1^{(t)} \right)} \right)^2 ,$$

(408)

$$\Delta_y^{(t,m+1)} = \exp\left( \pm \tilde{\mathcal{O}}\left( \frac{\mu_1^{(t)}}{N} \right) \right) \left( \frac{(N + 2m - 1) \exp\left( 1 - b_2^{(t)} \mu_1^{(t)} \right)}{\exp\left( \left(1 + b_2^{(t)}\right) \mu_1^{(t)} \right) + (N + 2m - 1) \exp\left( 1 - b_2^{(t)} \mu_1^{(t)} \right)} \right)^2 ,$$

(409)

and for $k \in \{m + 2, \cdots, 2m\}$,

$$\Delta_y^{(t,k)} = \exp\left( \pm \tilde{\mathcal{O}}\left( \frac{\mu_1^{(t)}}{N} \right) \right) \left( \frac{(N + 2m - 1) \exp\left( \max\{1 - b_1^{(t)}, b_1^{(t)}\} \mu_1^{(t)} \right)}{\exp\left( \left(1 + b_1^{(t)}\right) \mu_1^{(t)} \right) + (N + 2m - 1) \exp\left( \max\{1 - b_1^{(t)}, b_1^{(t)}\} \mu_1^{(t)} \right)} \right)^2 ,$$

(410)

and by Lemma 21, all $\Delta_\xi^{(t,k)}$ ($\xi \in \{x, y\}$, $k \in [2m]$) can be upper bounded by the following:

$$\Delta_\xi^{(t,k)} \leqslant \left( \frac{(N + 2m - 1) \exp\left( 1 - b_2^{(t)} \mu_1^{(t)} \right)}{\exp\left( \left(1 + b_2^{(t)} - 0.01\right) \mu_1^{(t)} \right) + (N + 2m - 1) \exp\left( 1 - b_2^{(t)} \mu_1^{(t)} \right)} \right)^2$$

$$\leq \left( \frac{N + 2m - 1}{\exp\left((0.24 \times 2 \times 0.99 - 0.01)\mu_1^{(t)}\right) + N + 2m - 1} \right)^2$$

$$= \left( \frac{N + 2m - 1}{\exp\left(0.46\mu_1^{(t)}\right) + N + 2m - 1} \right)^2 \leq \frac{\epsilon}{6m}, \tag{411}$$

as long as $\epsilon$ is small enough, where the last inequality follows from (205). This gives (209).

# D  Generalization Analysis

## D.1  Proof of Theorem 4

**Notation.** Throughout the proof, we suppose Assumptions 3 and 4 hold, and the model is trained for $t \geq T$ steps under the training distribution, where $T$ is the same as in Theorem 3. Let

$$E^{(k-1)} = (E, \widehat{O}^{(k-1)})$$

be the input at the $k$-th reasoning step. Define

$$x_i := E^{(\widetilde{m})}(1 : d_1, i), \quad y_i := E^{(\widetilde{m})}(d_1 + 1 : 2d_1, i), \quad \forall i \in [\widetilde{N} + \widetilde{m} + 1],$$

and

$$\widehat{x}^{(k)} := \widehat{o}^{(k)}(1 : d_1), \quad \widehat{y}^{(k)} := \widehat{o}^{(k)}(d_1 + 1 : 2d_1),$$
$$x^{(k)} := o^{(k)}(1 : d_1), \quad y^{(k)} := o^{(k)}(d_1 + 1 : 2d_1).$$

Furthermore, let $r = r(\mathcal{T})$ and $g = g(\mathcal{T})$ denote the root and goal node of $\mathcal{T}$, respectively. Let $p(i)$ be the parent of $i$ in $\mathcal{T}$, with $p(r) = 0$. Let $c(i) = c(i; \mathcal{T})$ be the child set of $i$, with $c(0) = \{r\}$. We also let

$$\mu^{(t)} := H_{1,1}^{(t)}, \quad \nu^{(t)} := H_{1,2}^{(t)}. \tag{412}$$

Then for all $i, j \in [S]$, $i \neq j$, we have $H_{j,j}^{(t)} = \mu^{(t)}$ and $H_{i,j}^{(t)} = \nu^{(t)}$.

We first give the following lemma, whose proof is provided in Appendix D.1.1.

**Lemma 24.** *For any $q \in [\widetilde{N} + 1 + \widetilde{m}]$, there exists $\{\beta_i^{(q)}\}_{i=1}^S$ such that*

$$\begin{pmatrix} x_q \\ y_q \end{pmatrix} = \sum_{i=0}^S \beta_i^{(q)} \begin{pmatrix} a_{p(i)} \\ a_i \end{pmatrix}, \quad and \quad \sum_{i=0}^S \beta_i^{(q)} = 1, \ \beta_i^{(q)} \geq 0, \ \forall 0 \leq i \leq S. \tag{413}$$

**Main proof.** We begin with reformulating the test-time loss. Define

$$\alpha^{(1)} = \frac{\exp(\mu)}{\exp(\mu) + \left(\widetilde{N} - 1\right)\exp(\nu) + 1} = \beta_g^{(\widetilde{N}+2)}, \tag{414}$$

which is the attention weight of $x_{-1}^{(0)} = y^{(1)} = a_g$ on $a_g$. Further, define

$$\forall k \in [\widetilde{m}] : \quad \alpha^{(k)} := \beta_{p^{k-1}(g)}^{(\widetilde{N}+1+k)}. \tag{415}$$

Then $\alpha^{(k)}$ is the proportion of $o^{(k)}$ in $\widehat{o}^{(k)}$. By our definition, we have for all $k \in [\widetilde{m}]$,

$$\widehat{o}^{(k)} = \alpha^{(k)}o^{(k)} + \sum_{\substack{i \in \mathcal{V}(\mathcal{T}) \\ i \neq p^{k-1}(g)}} \beta_i^{(k)} \begin{pmatrix} a_{p(i)} \\ a_i \end{pmatrix}, \quad o^{(k)} = \begin{pmatrix} a_{p^k(g)} \\ a_{p^{k-1}(g)} \end{pmatrix}, \tag{416}$$

and

$$\alpha^{(k)} + \sum_{\substack{i \in \mathcal{V}(\mathcal{T}) \\ i \neq p^{k-1}(g)}} \beta_i^{(k)} = 1. \tag{417}$$

Thus we have

$$
\mathcal{L}_{\text{test}}(\mathcal{T}; \theta^{(t)}) = \frac{1}{2} \sum_{k=1}^{\widetilde{m}} \left\| \widehat{o}^{(k)} - o^{(k)} \right\|_2^2
$$

$$
= \frac{1}{2} \sum_{k=1}^{\widetilde{m}} \left\| \sum_{\substack{i \in \mathcal{V}(\mathcal{T}) \\ i \neq p^{k-1}(g)}} \beta_i^{(k)} \begin{pmatrix} a_{p(i)} \\ a_i \end{pmatrix} - \left(1 - \alpha^{(k)}\right) \begin{pmatrix} a_{p^k(g)} \\ a_{p^{k-1}(g)} \end{pmatrix} \right\|_2^2
$$

$$
= \frac{1}{2} \sum_{k=1}^{\widetilde{m}} \left\| \sum_{\substack{i \in \mathcal{V}(\mathcal{T}) \\ i \neq p^{k-1}(g)}} \beta_i^{(k)} a_{p(i)} - \left(1 - \alpha^{(k)}\right) a_{p^k(g)} \right\|_2^2 + \frac{1}{2} \sum_{k=1}^{\widetilde{m}} \left\| \sum_{\substack{i \in \mathcal{V}(\mathcal{T}) \\ i \neq p^{k-1}(g)}} \beta_i^{(k)} a_i - \left(1 - \alpha^{(k)}\right) a_{p^{k-1}(g)} \right\|_2^2 .
$$

$$
\tag{418}
$$

The first term can be bounded via

$$
\left\| \sum_{\substack{i \in \mathcal{V}(\mathcal{T}) \\ i \neq p^{k-1}(g)}} \beta_i^{(k)} a_{p(i)} - \left(1 - \alpha^{(k)}\right) a_{p^k(g)} \right\|_2^2 = \sum_{\substack{i \in \mathcal{V}(\mathcal{T}) \\ i \neq p^{k-1}(g)}} \left( \beta_i^{(k)} \right)^2 + \left(1 - \alpha^{(k)}\right)^2
$$

$$
= \max_{\substack{i \in \mathcal{V}(\mathcal{T}) \\ i \neq p^{k-1}(g)}} |\beta_i^{(k)}| \cdot \sum_{\substack{i \in \mathcal{V}(\mathcal{T}) \\ i \neq p^{k-1}(g)}} \beta_i^{(k)} + \left(1 - \alpha^{(k)}\right)^2 \overset{(417)}{\leqslant} 2(1 - \alpha^{(k)})^2 .
$$

Similarly, we have

$$
\left\| \sum_{\substack{i \in \mathcal{V}(\mathcal{T}) \\ i \neq p^{k-1}(g)}} \beta_i^{(k)} a_i - \left(1 - \alpha^{(k)}\right) a_{p^{k-1}(g)} \right\|_2^2 \leqslant 2(1 - \alpha^{(k)})^2 .
$$

Substituting the above two inequalities into (418), we have

$$
\mathcal{L}_{\text{test}}(\mathcal{T}; \theta^{(t)}) \leqslant 2\widetilde{m}(\underbrace{1 - \alpha^{(k)}}_{:=\delta^{(k)}})^2 . \tag{419}
$$

Therefore, we can bound the test-time loss by bounding

$$
\delta^{(k)} := 1 - \alpha^{(k)} . \tag{420}
$$

We give the following lemma, where $T_2$ is defined in (65). The proof is deferred to Appendix D.1.2.

**Lemma 25** (bounding $\delta^{(k)}$). *For all $k \in [\widetilde{m}]$, we have*

$$
\delta^{(k)} \leqslant \max \left\{ 1, \frac{\widetilde{N} + \widetilde{m} - 1}{N + m - 2} \right\} \cdot 2\delta, \tag{421}
$$

*where*

$$
\delta := 1 - \frac{\exp(\mu^{(t)})}{(N + m - 2)\exp(\nu^{(t)}) + \exp(\mu^{(t)}) + 1} \overset{(65)}{\leqslant} \sqrt{\frac{\epsilon}{2m}} \tag{422}
$$

*for $t > T_2$.*

By Lemma 25 and (419), we have

$$
\mathcal{L}_{\text{test}}(\mathcal{T}; \theta^{(t)}) \leqslant \max \left\{ 1, \left( \frac{\widetilde{N} + \widetilde{m} - 1}{N + m - 2} \right)^2 \right\} \cdot 4\frac{\widetilde{m}}{m}\epsilon . \tag{423}
$$

### D.1.1  Proof of Lemma 24

We prove this lemma by induction on $k$. First, (413) holds trivially for $1 \leqslant q \leqslant \widetilde{N} + 1$ ($(x_q, y_q)^\top$ is in $E^{(0)}$). Assume that (413) holds for $q \leqslant s - 1$ ($\widetilde{N} + 2 \leqslant s \leqslant \widetilde{N} + 1 + \widetilde{m}$). We prove that (413) holds for $s$.

For notational convenience, we let $k = s - 1 - \widetilde{N}$, and let

$$\gamma^{(k)} = (\gamma_1^{(k)}, \ldots, \gamma_{s-1}^{(k)}) = \mathsf{softmax}\left(Y^{(k-1)\top} B \widehat{x}^{(k-1)}\right). \tag{424}$$

Then by (5), we have

$$\begin{pmatrix} x_s \\ y_s \end{pmatrix} = \begin{pmatrix} \widehat{x}^{(k)} \\ \widehat{y}^{(k)} \end{pmatrix} = \sum_{i=1}^{s-1} \gamma_i^{(k)} \begin{pmatrix} x_i \\ y_i \end{pmatrix} = \sum_{j=0}^{S} \left(\sum_{i=1}^{s-1} \gamma_i^{(k)} \beta_j^{(i)}\right) \begin{pmatrix} a_{p(j)} \\ a_j \end{pmatrix},$$

where the second equality uses the induction hypothesis. Let $\beta_j^{(s)} = \sum_{i=1}^{s-1} \gamma_i^{(k)} \beta_j^{(i)}$, then we have

$$\begin{pmatrix} x_s \\ y_s \end{pmatrix} = \sum_{j=0}^{S} \beta_j^{(s)} \begin{pmatrix} a_{p(j)} \\ a_j \end{pmatrix}, \tag{425}$$

and by the induction hypothesis, we have $\beta_j^{(s)} \geqslant 0$, $\forall j \in [S]$ and

$$\sum_{j=0}^{S} \beta_j^{(s)} = \sum_{j=0}^{S} \left(\sum_{i=1}^{s-1} \gamma_i^{(k)}\right) \beta_j^{(i)} \overset{(424)}{=} \sum_{j=0}^{S} \beta_j^{(i)} = 1. \tag{426}$$

This completes the induction.

### D.1.2  Proof of Lemma 25

We prove by induction. When $k = 1$, by (422) and (414) we immediately know that if $N + m - 2 \geqslant \widetilde{N} - 1$, we have $\delta^{(1)} \leqslant \delta$. If $N + m - 2 < \widetilde{N} - 1$, we have

$$\frac{\delta^{(1)}}{\delta} = \underbrace{\frac{\left(\widetilde{N} - 1\right)\exp(\nu^{(t)}) + 1}{(N + m - 2)\exp(\nu^{(t)}) + 1}}_{\leqslant \frac{\widetilde{N} - 1}{N + m - 2}} \cdot \underbrace{\frac{\exp(\mu^{(t)}) + (N + m - 2)\exp(\nu^{(t)}) + 1}{\exp(\mu^{(t)}) + \left(\widetilde{N} - 1\right)\exp(\nu^{(t)}) + 1}}_{\leqslant 1} \leqslant \frac{\widetilde{N} - 1}{N + m - 2}. \tag{427}$$

Thus in either case, we have (421) holds for $k = 1$.

Assume that (421) holds for all $s \in [k]$ ($1 \leqslant k \leqslant \widetilde{m} - 1$). We aim to show that (421) holds for $k + 1$. Let $B = B^{(t)}$, where $t \geqslant T_2$. Then $a_{p^k(g)} = x^{(k)} = y^{(k+1)}$. And note that $\beta_r^{(k)}$ is the proportion of $(0, a_r)^\top$ in $\widehat{o}^{(k)}$, thus we have

$$a_{p^k(g)}^\top B \widehat{x}^{(k)} = \alpha^{(k)}\mu + (1 - \alpha^{(k)} - \beta_r^{(k)})\nu. \tag{428}$$

For all $i \in \mathcal{V}(\mathcal{T}) \backslash \{p^k(g)\}$, we have

$$a_i^\top B \widehat{x}^{(k)} \leqslant (1 - \alpha^{(k)} - \beta_r^{(k)})\mu + \alpha^{(k)}\nu. \tag{429}$$

We define $p^{-1}(g) = 0$ for all $s \in [k]$, and recall $c(0) = \{r\}$, we have

$$\widehat{y}^{(s)\top} B \widehat{x}^{(k)} = \left(\alpha^{(s)} a_{p^{s-1}(g)} + \sum_{\substack{i \in \mathcal{V}(\mathcal{T}) \cup \{0\} \\ i \neq p^{s-1}(g)}} \beta_i^{(s)} a_i\right)^\top B \left(\alpha^{(k)} a_{p^k(g)} + \sum_{\substack{i \in \mathcal{V}(\mathcal{T}) \cup \{0\} \\ i \neq p^{k-1}(g)}} \beta_i^{(i)} a_{p(i)}\right)$$

$$= \underbrace{\left(\alpha^{(s)} \beta_{p^{s-2}} + \beta_{p^k(g)}^{(s)} \alpha^{(k)} + \sum_{\substack{i \in \mathcal{V}(\mathcal{T}) \cup \{0\} \\ i \neq p^{s-1}(g), p^k(g)}} \beta_i^{(s)} \sum_{j \in c(i)} \beta_j^{(k)}\right) \mu}_{\triangle}$$

$$+\left((1-\beta_r^{(k)})(1-\beta_0^{(s)}) - \left(\alpha^{(s)}\beta_{p^{s-2}} + \beta_{p^k(g)}^{(s)}\alpha^{(k)} + \underbrace{\sum_{\substack{i\in\mathcal{V}(\mathcal{T})\cup\{0\}\\ i\neq p^{s-1}(g),p^k(g)}} \beta_i^{(s)}\sum_{j\in c(i)}\beta_j^{(k)}}_{\triangle}\right)\right)\nu.$$

$$(430)$$

Note that

$$\triangle \leqslant \alpha^{(s)}(1-\alpha^{(k)}-\beta_r^{(k)}) + \alpha^{(k)}(1-\alpha^{(s)}-\beta_0^{(s)}) + (1-\alpha^{(s)}-\beta_0^{(s)})(1-\alpha^{(k)}-\beta_r^{(k)})$$
$$= (1-\beta_r^{(k)})(1-\beta_0^{(s)}) - \alpha^{(s)}\alpha^{(k)}, \qquad (431)$$

and by Theorem 3, we have $\mu > \nu$ and $\mu > 0$. Thus we have

$$\widehat{y}^{(s)\top}B\widehat{x}^{(k)} \leqslant \left((1-\beta_r^{(k)})(1-\beta_0^{(s)}) - \alpha^{(s)}\alpha^{(k)}\right)\mu + \alpha^{(s)}\alpha^{(k)}\nu$$
$$\leqslant \left(1 - \beta_r^{(k)} - \alpha^{(s)}\alpha^{(k)}\right)\mu + \alpha^{(s)}\alpha^{(k)}\nu. \qquad (432)$$

Combining (428), (429), and (432), and denote $\widetilde{\alpha}^{(k)} = \min_{s\in[k]}\{\alpha^{(s)}\}$, we have

$$\alpha^{(k+1)} = \frac{\exp\left(a_{p^k(g)}^\top B\widehat{x}^{(k)}\right)}{\exp\left(a_{p^k(g)}^\top B\widehat{x}^{(k)}\right) + \sum_{\substack{i\in\mathcal{V}(\mathcal{T})\\ i\neq p^k(g)}}\exp\left(a_i^\top B\widehat{x}^{(k)}\right) + \sum_{s=1}^k\exp\left(\widehat{y}^{(s)\top}B\widehat{x}^{(k)}\right) + 1}$$

$$\geqslant \frac{\exp\left(\widetilde{\alpha}^{(k)}\mu + (1-\widetilde{\alpha}^{(k)}-\beta_r^{(k)})\nu\right)}{\exp\left(\widetilde{\alpha}^{(k)}\mu + (1-\widetilde{\alpha}^{(k)}-\beta_r^{(k)})\nu\right) + (\widetilde{N}-2+\widetilde{m})\exp\left(\left(1-\beta_r^{(k)}-\left(\widetilde{\alpha}^{(k)}\right)^2\right)\mu + \left(\widetilde{\alpha}^{(k)}\right)^2\nu\right) + 1}$$

$$\geqslant \frac{\exp\left(\left(\widetilde{\alpha}^{(k)} + \left(\widetilde{\alpha}^{(k)}\right)^2 - 1 + \beta_r^{(k)}\right)(\mu-\nu)\right)}{\exp\left(\left(\widetilde{\alpha}^{(k)} + \left(\widetilde{\alpha}^{(k)}\right)^2 - 1 + \beta_r^{(k)}\right)(\mu-\nu)\right) + \widetilde{N}-2+\widetilde{m} + \exp\left(-\left(\widetilde{\alpha}^{(k)}\right)^2\nu\right)}$$

$$\geqslant \frac{\exp\left(\left(\widetilde{\alpha}^{(k)} + \left(\widetilde{\alpha}^{(k)}\right)^2 - 1\right)(\mu-\nu)\right)}{\exp\left(\left(\widetilde{\alpha}^{(k)} + \left(\widetilde{\alpha}^{(k)}\right)^2 - 1\right)(\mu-\nu)\right) + \widetilde{N}-2+\widetilde{m} + \exp\left(-\left(\widetilde{\alpha}^{(k)}\right)^2\nu\right)}, \qquad (433)$$

where the second line uses the following fact:

$$(1-\alpha^{(k)}-\beta_r^{(k)})\mu + \alpha^{(k)}\nu \leqslant \left(1 - \beta_r^{(k)} - \alpha^{(s)}\alpha^{(k)}\right)\mu + \alpha^{(s)}\alpha^{(k)}\nu$$
$$\leqslant \left(1 - \beta_r^{(k)} - \left(\widetilde{\alpha}^{(k)}\right)^2\right)\mu + \left(\widetilde{\alpha}^{(k)}\right)^2\nu. \qquad (434)$$

Define

$$\widetilde{\delta}^{(k)} = 1 - \widetilde{\alpha}^{(k)} = \max_{s\in[k]}\{\delta^{(s)}\},$$

then we have

$$1 - \left(\widetilde{\alpha}^{(k)} + \left(\widetilde{\alpha}^{(k)}\right)^2 - 1\right) = \left(2 + \widetilde{\alpha}^{(k)}\right)\left(1 - \widetilde{\alpha}^{(k)}\right) \leqslant 3\widetilde{\delta}^{(k)}. \qquad (435)$$

Combining the above inequality with (433), we have

$$\alpha^{(k+1)} \geqslant \frac{\exp\left(\left(1 - 3\widetilde{\delta}^{(k)}\right)(\mu-\nu)\right)}{\exp\left(\left(1 - 3\widetilde{\delta}^{(k)}\right)(\mu-\nu)\right) + \widetilde{N}-2+\widetilde{m} + \exp\left(-\left(\widetilde{\alpha}^{(k)}\right)^2\nu\right)}, \qquad (436)$$

which gives

$$\delta^{(k+1)} = 1 - \alpha^{(k+1)} \leqslant \frac{\widetilde{N}-2+\widetilde{m} + \exp\left(-\left(\widetilde{\alpha}^{(k)}\right)^2\nu\right)}{\exp\left(\left(1 - 3\widetilde{\delta}^{(k)}\right)(\mu-\nu)\right) + \widetilde{N}-2+\widetilde{m} + \exp\left(-\left(\widetilde{\alpha}^{(k)}\right)^2\nu\right)}. \qquad (437)$$

On the other hand, (422) gives

$$\mu - \nu = \log\left((N + m - 2 + \exp(-\nu))\left(\frac{1}{\delta} - 1\right)\right). \tag{438}$$

Thus if $\exp(-\nu) \leqslant \tilde{N} - 2 + \tilde{m}$, we have

$$\mu - \nu \leqslant \log\left(2(N + m - 2)\left(\frac{1}{\delta} - 1\right)\right). \tag{439}$$

If $\exp(-\nu) \geqslant \tilde{N} - 2 + \tilde{m}$, we have

$$\mu - \nu \leqslant \log\left(2\exp(-\nu)\left(\frac{1}{\delta} - 1\right)\right) \leqslant \log\left(2\left(\frac{1}{\delta} - 1\right)\right) - \nu. \tag{440}$$

By (74) and Lemma 10, we know that there exists $\epsilon_1(m) > 0$ such that for all $\epsilon \in (0, \epsilon_1(m)]$, after $t > T_2$, we have

$$-\nu \leqslant \frac{5}{S}\mu \leqslant \frac{5}{S}(\mu - \nu), \tag{441}$$

where the second inequality uses the fact that $\nu \leqslant 0$, since $\exp(-\nu) \geqslant \tilde{N} - 2 + \tilde{m}$. Combining the above two inequalities, we have

$$\mu - \nu \leqslant 2\log\left(2\left(\frac{1}{\delta} - 1\right)\right). \tag{442}$$

Combining (439) and (442), we have when $\epsilon \in (0, \epsilon_1(m)]$,

$$\mu - \nu \leqslant 2\log\left(2(N + m - 2)\left(\frac{1}{\delta} - 1\right)\right). \tag{443}$$

Note that

$$\delta\log\left(2(N + m - 2)\left(\frac{1}{\delta} - 1\right)\right) \to 0 \quad \text{as } \delta \to 0 + . \tag{444}$$

Thus we could choose $\epsilon_0(m, \tilde{N}, \tilde{m}) \leqslant \epsilon_1(m)$ small enough such that for all $\epsilon \in (0, \epsilon_0(m, \tilde{N}, \tilde{m})]$, we have

$$6\frac{\tilde{N} + \tilde{m} - 1}{N + m - 2}\delta(\mu - \nu) \leqslant \log 2, \tag{445}$$

which gives

$$\exp\left(3\bar{\delta}^{(k)}(\mu - \nu)\right) \leqslant \exp\left(6\max\left\{1, \frac{\tilde{N} + \tilde{m} - 1}{N + m - 2}\right\}\delta(\mu - \nu)\right) \leqslant 2, \tag{446}$$

where the first inequality uses induction hypothesis.

Define

$$a := 2\max\left\{1, \frac{\tilde{N} + \tilde{m} - 1}{N + m - 2}\right\}. \tag{447}$$

Then

- if $\tilde{N} + \tilde{m} - 2 < N + m - 2$, we have $a = 2$ and

$$\frac{N + m - 2 + \exp(-\nu)}{\tilde{N} + \tilde{m} - 2 + \exp\left(-\left(\tilde{\alpha}^{(k)}\right)^2\nu\right)} \geqslant \begin{cases} \frac{N + m - 2}{\tilde{N} + \tilde{m} - 1} \geqslant 1 & \text{if } \nu \geqslant 0 \\ \frac{N + m - 2 + \exp(-\nu)}{\tilde{N} + \tilde{m} - 2 + \exp(-\nu)} \geqslant 1 & \text{if } \nu < 0 \end{cases}, \tag{448}$$

which gives

$$\frac{1}{a\delta} - 1 = \frac{1}{2\delta} - 1 \leqslant \frac{1}{2}\left(\frac{1}{\delta} - 1\right) \leqslant \frac{\left(\frac{1}{\delta} - 1\right)(N + m - 2 + \exp(-\nu))}{2\left(\tilde{N} + \tilde{m} - 2 + \exp\left(-\left(\tilde{\alpha}^{(k)}\right)^2\nu\right)\right)}. \tag{449}$$

• if $\widetilde{N} + \widetilde{m} - 2 \geqslant N + m - 2$, we have $a = 2 \cdot \frac{\widetilde{N}+\widetilde{m}-1}{N+m-2} \geqslant 1$, and

$$\frac{2}{a}\left(\frac{1}{\delta} - a\right) = \frac{N+m-2}{\widetilde{N}+\widetilde{m}-1}\left(\frac{1}{\delta} - a\right) \leqslant \frac{N+m-2}{\widetilde{N}+\widetilde{m}-1}\left(\frac{1}{\delta} - 1\right). \tag{450}$$

Moreover, we have

$$\frac{N+m-2+\exp(-\nu)}{\widetilde{N}+\widetilde{m}-2+\exp\left(-\left(\widetilde{\alpha}^{(k)}\right)^2\nu\right)} \geqslant \begin{cases} \frac{N+m-2+\exp(-\nu)}{\widetilde{N}+\widetilde{m}-2+\exp(-\nu)} \geqslant \frac{N+m-2}{\widetilde{N}+\widetilde{m}-1} & \text{if } \nu < 0 \\ \frac{N+m-2}{\widetilde{N}+\widetilde{m}-1} & \text{if } \nu \geqslant 0. \end{cases} \tag{451}$$

Combined with (450), we have when $\widetilde{N} + \widetilde{m} - 2 \geqslant N + m - 2$,

$$\frac{1}{a\delta} - 1 \leqslant \frac{1}{2} \cdot \frac{N+m-2}{\widetilde{N}+\widetilde{m}-1}\left(\frac{1}{\delta} - 1\right) \leqslant \frac{\left(\frac{1}{\delta}-1\right)(N+m-2+\exp(-\nu))}{2\left(\widetilde{N}+\widetilde{m}-2+\exp\left(-\left(\widetilde{\alpha}^{(k)}\right)^2\nu\right)\right)}. \tag{452}$$

By (449) and (452), we have the above relation holds for both cases. Furthermore, from this relation we derive that

$$\exp(\mu - \nu) \overset{(438)}{=} (N+m-2+\exp(-\nu))\left(\frac{1}{\delta} - 1\right)$$

$$\overset{(452)}{\geqslant} 2\left(\widetilde{N}+\widetilde{m}-2+\exp\left(-\left(\widetilde{\alpha}^{(k)}\right)^2\nu\right)\right)\left(\frac{1}{a\delta} - 1\right)$$

$$\overset{(446)}{\geqslant} \exp\left(3\overline{\delta}^{(k)}(\mu-\nu)\right)\left(\widetilde{N}+\widetilde{m}-2+\exp\left(-\left(\widetilde{\alpha}^{(k)}\right)^2\nu\right)\right)\left(\frac{1}{a\delta} - 1\right). \tag{453}$$

Thus we have

$$\exp\left((1-3\overline{\delta}^{(k)})(\mu-\nu)\right) \geqslant \left(\widetilde{N}+\widetilde{m}-2+\exp\left(-\left(\widetilde{\alpha}^{(k)}\right)^2\nu\right)\right)\left(\frac{1}{a\delta} - 1\right)$$

$$\Leftrightarrow \frac{\exp\left((1-3\overline{\delta}^{(k)})(\mu-\nu)\right)}{\widetilde{N}+\widetilde{m}-2+\exp\left(-\left(\widetilde{\alpha}^{(k)}\right)^2\nu\right)} \geqslant \frac{1}{a\delta} - 1$$

$$\Leftrightarrow \frac{\widetilde{N}+\widetilde{m}-2+\exp\left(-\left(\widetilde{\alpha}^{(k)}\right)^2\nu\right)}{\widetilde{N}+\widetilde{m}-2+\exp\left(-\left(\widetilde{\alpha}^{(k)}\right)^2\nu\right)+\exp\left((1-3\overline{\delta}^{(k)})(\mu-\nu)\right)} \leqslant a\delta. \tag{454}$$

Combining the above with (437), we have

$$\delta^{(k+1)} \leqslant a\delta. \tag{455}$$

The induction is complete.

## D.2 Proof of Theorem 6

Assume the tree $\mathcal{T}$ at test time is $\mathcal{T}$ with $\widetilde{N}$ distinct nodes chosen from $[S]$ and a path length $\widetilde{m}$, and Assumption 3, 4 hold, and the model is trained for $t \geqslant \max\{T_3^B, T_3^C\} + 1$ steps with the trianing distribution, where $T_3^B$ and $T_3^C$ are defined in (205) and (194). We let

$$E^{(k-1)} = (E, \widehat{O}^{(k-1)})$$

be the input at the $k$-th reasoning step, and define

$$x_i := E^{(\widetilde{m})}(1:d_1, i), \quad y_i := E^{(\widetilde{m})}(d_1+1:2d_1, i), \quad \forall i \in [\widetilde{N}+\widetilde{m}+1]. \tag{456}$$

We let $\widehat{o}^{(k)} = \widehat{o}^{(k)}(\mathcal{T}) \in \mathbb{R}^{2d_1+d_2}$ denote the output of the $k$-th reasoning step of the model at test time for all $k \in [2\widetilde{m}]$. We define

$$\widehat{x}^{(k)} := \widehat{o}^{(k)}(1:d_1), \quad \widehat{y}^{(k)} := \widehat{o}^{(k)}(d_1+1:2d_1), \quad \widehat{z}^{(k)} := \widehat{o}^{(k)}(2d_1+1:2d_1+d_2), \tag{457}$$

and let
$$x^{(k)} := o^{(k)}(1:d_1), \quad y^{(k)} := o^{(k)}(d_1+1:2d_1), \quad z^{(k)} := o^{(k)}(2d_1+1:2d_1+d_2) \quad (458)$$

denote the label of the $k$-th reasoning step.

By the same argument as Lemma 24, we know that at each reasoning step $k$, there exist $\left\{\beta_i^{(k)}\right\}_{i=1}^{\widetilde{N}+k}$, $\left\{\gamma_i^{(k)}\right\}_{i=1}^{\widetilde{N}+k}$ such that

$$\left(\widehat{x}^{(k)\top}, \widehat{y}^{(k)\top}\right)^\top = \sum_{i=1}^{\widetilde{N}+k} \beta_i^{(k)} \left(x_i^\top, y_i^\top\right)^\top, \quad \widehat{z}^{(k)} = \sum_{i=1}^{\widetilde{N}+k} \gamma_i^{(k)} z_i,$$

$$\sum_{i=1}^{\widetilde{N}+k} \beta_i^{(k)} = \sum_{i=1}^{\widetilde{N}+k} \gamma_i^{(k)} = 1, \quad \beta_i^{(k)}, \gamma_i^{(k)} \geqslant 0. \quad (459)$$

Especially, we let $\beta_\star^{(k)}$ denote the proportions of $(x^{(k)\top}, y^{(k)\top})^\top$ in $(\widehat{x}^{(k)\top}, \widehat{y}^{(k)\top})^\top$, and let $\gamma_\star^{(k)}$ denote the proportions of $z^{(k)}$ in $\widehat{z}^{(k)}$. We let

$$\alpha^{(k)} := \min\left\{\beta_\star^{(k)}, \gamma_\star^{(k)}\right\}, \quad \delta^{(k)} := \max\left\{1 - \beta_\star^{(k)}, 1 - \gamma_\star^{(k)}\right\}. \quad (460)$$

We define

$$\mu_i^{(k)} := x_i^\top B_1 y^{(k-1)} + y_i^\top B_2 x^{(k-1)} + z_i^\top B_3 z^{(k-1)}, \quad (461\text{a})$$

$$\mu_\star^{(k)} := x^{(k)\top} B_1 y^{(k-1)} + y^{(k)\top} B_2 x^{(k-1)} + z^{(k)\top} B_3 z^{(k-1)}, \quad (461\text{b})$$

$$\nu_i^{(k)} := x_i^\top C_1 y^{(k-1)} + y_i^\top C_2 x^{(k-1)} + z_i^\top C_3 z^{(k-1)}, \quad (461\text{c})$$

$$\nu_\star^{(k)} := x^{(k)\top} C_1 y^{(k-1)} + y^{(k)\top} C_2 x^{(k-1)} + z^{(k)\top} C_3 z^{(k-1)}. \quad (461\text{d})$$

We also define for some absolute constant $C > 0$,

$$v := C(\mu_1 + \mu_2), \quad (462)$$

where $\mu_1, \mu_2$ are defined in (185) and (186). By Lemma 17 and Lemma 20 we can see that there exists $C > 0$ such that for any $i, j, k, l \in [S]$, and any $p, q \in \{0, 1\}$,

$$\left|a_i^\top B_1 a_j + a_k^\top B_2 a_l + s_p^\top B_3 s_q\right| \leqslant v^{(k)}, \quad (463)$$

$$\left|a_i^\top C_1 a_j + a_k^\top C_2 a_l + s_p^\top C_3 s_q\right| \leqslant v^{(k)}. \quad (464)$$

We set $C$ to make the above relations hold. We define

$$\forall k \in [2\widetilde{m}], \quad \delta_1^{(k)} = 1 - \beta_\star^{(k)}, \quad \delta_2^{(k)} = 1 - \gamma_\star^{(k)}, \quad (465)$$

$$\widetilde{\delta}_1^{(k)} := \max_{s \in [k]}\left\{\delta_1^{(s)}\right\}, \quad \widetilde{\delta}_2^{(k)} := \max_{s \in [k]}\left\{\delta_2^{(s)}\right\}, \quad (466)$$

$$\widetilde{\beta}_\star^{(k)} := \min_{s \in [k]}\left\{\beta_\star^{(s)}\right\}, \quad \widetilde{\gamma}_\star^{(k)} := \min_{s \in [k]}\left\{\gamma_\star^{(s)}\right\}, \quad (467)$$

and

$$\delta_1 := \frac{N + 2m - 1}{\exp\left(0.46\mu_1^{(t)}\right) + N + 2m - 1} \leqslant \sqrt{\frac{\epsilon}{6m}}, \quad \delta_2 := \frac{N}{\exp\left(0.47\mu_2\right) + N} \leqslant \sqrt{\frac{\epsilon}{6m}}, \quad (468)$$

where the inequalities follow from (205), (194) and that $t \geqslant \max\{T_3^B, T_3^C\} + 1$. We first prove by induction that

$$\forall k \in [2\widetilde{m}], \quad \widetilde{\delta}_1^{(k)} \leqslant 2\max\left\{\frac{\widetilde{N} + 2\widetilde{m} - 1}{N + 2m - 1}, 1\right\}\delta_1, \quad \widetilde{\delta}_2^{(k)} \leqslant 2\max\left\{\frac{\widetilde{N}}{N}, 1\right\}\delta_2. \quad (469)$$

We define

$$a := 2\max\left\{\frac{\widetilde{N} + 2\widetilde{m} - 1}{N + 2m - 1}, 1\right\}. \quad (470)$$

When $k = 1$, we have

$$\delta_1^{(1)} \leqslant 1 - \frac{\exp\left(\mu_\star^{(1)}\right)}{\exp\left(\mu_\star^{(1)}\right) + \sum_{\substack{i=1 \\ i \neq \star}}^{\widetilde{N}+1} \exp\left(\mu_i^{(1)}\right)} \leqslant \frac{\widetilde{N}}{\widetilde{N} + \exp\left(0.46\mu_1\right)} \overset{(468)}{\leqslant} \max\left\{\frac{\widetilde{N}}{N + 2m - 1}, 1\right\}\delta_1.$$

(471)

where we use $\star$ to denote the index of $\mu_\star^{(1)}$, and the second inequality follows from Lemma 21. Similarly, we have

$$\delta_2^{(1)} \leqslant \max\left\{\frac{\widetilde{N}}{N}, 1\right\}\delta_2.$$

(472)

Thus (469) holds for $k = 1$. Now we assume (469) holds for $k$ ($1 \leqslant k \leqslant 2\widetilde{m} - 1$), and prove it for $k + 1$.

We have

$$\beta_\star^{(k+1)} \geqslant \frac{\exp\left(\left(\widetilde{\beta}_\star^{(k)}\right)^2 \mu_\star^{(k)} - \left(1 - \left(\widetilde{\beta}_\star^{(k)}\right)^2\right)v\right)}{\exp\left(\left(\widetilde{\beta}_\star^{(k)}\right)^2 \mu_\star^{(k)} - \left(1 - \left(\widetilde{\beta}_\star^{(k)}\right)^2\right)v\right) + \sum_{\substack{i=1 \\ i \neq \star}}^{\widetilde{N}+k} \exp\left(\left(\widetilde{\beta}_\star^{(k)}\right)^2 \mu_i^{(k)} + \left(1 - \left(\widetilde{\beta}_\star^{(k)}\right)^2\right)v\right)},$$

(473)

indicating

$$\delta_1^{(k+1)} \leqslant \frac{\widetilde{N} + k - 1}{\widetilde{N} + k - 1 + \exp\left(0.46\left(\widetilde{\beta}_\star^{(k)}\right)^2 \mu_1 - 2\left(1 - \left(\widetilde{\beta}_\star^{(k)}\right)^2\right)v\right)}.$$

(474)

Note that

$$1 - \left(\widetilde{\beta}_\star^{(k)}\right)^2 = \left(1 - \widetilde{\beta}_\star^{(k)}\right)\left(1 + \widetilde{\beta}_\star^{(k)}\right) \leqslant 2\widetilde{\delta}_\star^{(k)} \leqslant 2a\delta_1,$$

(475)

where the last inequality follows from induction hypothesis and the definition of $a$ (c.f. (470)). Plugging this into (474) we have

$$\delta_1^{(k+1)} \leqslant \frac{\widetilde{N} + k - 1}{\widetilde{N} + k - 1 + \exp\left(0.46\mu_1 - \underbrace{2a\delta_1\left(2v + 0.46\mu_1\right)}_{(a)}\right)},$$

(476)

On the other hand, by the definition of $\delta_1$ and $\delta_2$ we have

$$\mu_1 = \frac{1}{0.46}\log\left(\frac{1}{\delta_1} - 1\right), \quad \mu_2 = \frac{1}{0.47}\log\left(\frac{1}{\delta_2} - 1\right).$$

(477)

and

$$\delta_1 \leqslant \sqrt{\frac{\epsilon}{6m}}, \quad \delta_2 \leqslant \sqrt{\frac{\epsilon}{6m}}.$$

(478)

Also observe that $x \log x \to 0_+$ as $x \to 0_+$. Thus we could set $\epsilon_1 = \epsilon_1(m, \widetilde{N}, \widetilde{m})$ small enough such that (a) in (476) is smaller than $\log 2$ for any $\epsilon \in (0, \epsilon_1]$. Thus by (476) we have

$$\delta_1^{(k+1)} \leqslant \frac{\widetilde{N} + k - 1}{\widetilde{N} + k - 1 + \frac{1}{2}\exp\left(0.46\mu_1\right)} \leqslant 2\frac{\widetilde{N} + k - 1}{\widetilde{N} + k - 1 + \exp\left(0.46\mu_1\right)} \leqslant 2\max\left\{\frac{\widetilde{N} + k - 1}{N + 2m - 1}, 1\right\}\delta_1.$$

(479)

Similarly, there exists $\epsilon_2 = \epsilon_2(m, \widetilde{N}, \widetilde{m})$ small enough such that

$$\delta_2^{(k+1)} \leqslant 2\max\left\{\frac{\widetilde{N}}{N}, 1\right\}\delta_2.$$

(480)

We let $\epsilon_0 = \min\{\epsilon_1, \epsilon_2\}$, then (469) holds for $k + 1$.

Following a similar calculation as in the proof of Lemma 22 and Lemma 18, we can show that

$$\Delta_x^{(k)} \leqslant \left(\delta_1^{(k)}\right)^2, \quad \Delta_y^{(k)} \leqslant \left(\delta_1^{(k)}\right)^2, \quad \Delta_z^{(k)} \leqslant \left(\delta_2^{(k)}\right)^2, \tag{481}$$

and thus

$$\mathcal{L}_{\text{test}}(\mathcal{T}; \theta) \leqslant 4 \max\left\{\left(\frac{\widetilde{N} + 2\widetilde{m} - 1}{N + 2m - 1}\right)^2, \left(\frac{\widetilde{N}}{N}\right)^2, 1\right\} \epsilon. \tag{482}$$

