# OpenReview forum: "Multi-head Transformers Provably Learn Symbolic Multi-step Reasoning via Gradient Descent"
_NeurIPS.cc/2025/Conference — NeurIPS 2025 poster_

### Official Review · Reviewer_WYm4 · 2025-06-05

**Clarity:** 3
**Significance:** 2
**Originality:** 2
**Rating:** 4
**Confidence:** 3

**Summary:**

The paper studies multi-step reasoning with chain-of-thought (CoT) in a tree search setting. The tasks considered are the backward path finding task where a path from the goal leaf back to the root must be chained together, and the forward path finding task where the backward path must be found and reversed. A one-layer two-head transformer construction is shown to achieve both tasks by using appropriately long CoT. Moreover, it is shown that GD learns such solutions from zero initialization and the trained model can generalize to unseen tree combinations.

**Questions:**

* Are only the token embeddings being learned from a learning theoretic perspective? The paper mentions that "our results can be easily extended to other types of trees", how can the tree structure and/or distribution be made more general?
* Can the authors provide a high level summary of how the CoT gradient dynamics is made tractable? What are the straightforward or nontrivial parts of the analysis? I am personally interested in seeing what insights can be extended to other CoT setups.

**Ethical Concerns:**

["NO or VERY MINOR ethics concerns only"]

**Final Justification:**

While the learning-theoretic implications of the results are somewhat unclear, the studied task is very relevant to understanding reasoning of CoT models (for which we have little existing theoretical results on) and the analysis techniques appear novel enough to merit acceptance.

**Limitations:**

Yes

**Quality:**

3

**Strengths And Weaknesses:**

**Strengths**
* The considered CoT path-finding problem is interesting and very relevant to the current test-time scaling paradigm, which lacks a strong theoretical foundation.
* The main results effectively show that CoT allows transformer models to implicitly think backwards from the goal to find the solution, which otherwise would require brute-force forward search over the tree to find.
* The constructions are paired with a generalization guarantee and dynamical analyses which show well behaved solutions may be realistically found with GD.

**Weaknesses**
* It is not clear what or why the model needs to learn from a learning-theoretic perspective, since the desired algorithm and training distribution is fixed (and the model benefits from knowing this target mechanism/construction, e.g., as we only train $\theta=(B_i,C_i)$). It seems that the token embeddings are arbitrary and the model can learn the appropriate weight matrices, but this is a rather artificial learning goal compared to learning (say) the tree structure, which is assumed to be fixed to uniform over perfect binary trees.
* The GD analysis is quite complicated and would benefit from a section summarizing the proof ideas and overall strategy.

---

> ### Author Rebuttal · Authors · 2025-07-31
>
> ## Response to Reviewer WYm4
>
> Thank you for your time reviewing our paper and your positive feedback! If our responses resolve your questions, we'd appreciate your consideration in raising the score.
>
> >**W1/ Q1 (a): regarding what is learned by the model**
>
> Thank you for your insightful question.
>
> **What is learned: the transformer's internal algorithm, not token embeddings**: First, to address the core question of what is being learned: the model learns the transformer's internal weights ($\theta=${$B_i$, $C_i$}), which collectively implement a multi-step reasoning algorithm, instead of memorizing the fixed token embeddings or the tree structure.
>
>
> - Our training distribution is fixed to uniform over perfect binary trees, since the number and order of the nodes are random, this forces the model to learn the attention weights to perform abstract rules of:
>     - finding the parent node: implement a "lookup" operation to find the parent of a given node in the edge list,
>     - chain operations: autoregressively apply this lookup to traverse the tree,
>     - identify termination/turning Points: recognize the root node to stop the backward chain and begin the forward one.
>
>     Our convergence results show that the transformer achieves this by learning to specialize heads for different subtasks (path traversal vs. stage control) and coordinating their actions, which is highly non-trivial. In contrast, simply memorizing structure or token embeddings could not lead the training loss to 0, since any two embeddings could be a parent-child pair. This is also supported by our generalization results (Theorem 4 and Theorem 8), which demonstrate that a model trained only on perfect binary trees successfully generalizes to any tree at test time.
>
> - Some parts of the weight matrices are fixed (in line 178-179 and 212-214), which is a standard practice in the literature for theoretical analysis of transformer training dynamics (see, e.g., [Huang et al., 2023],[Nichain et al., 2024]). This allows us to clearly trace how the attention patterns are formed without loss of optimality, i.e., a global minimum of the training loss can be achieved within this simplified parameter space, without cluttering the analysis with excess details that do not bring new insights.
>
> >**Q1 (b): how can the tree structure and/or distribution be made more general?**
>
> The assumption of using a uniform distribution over perfect binary trees during training (Assumption 3) provides a balanced and symmetric data distribution, which simplifies the analysis of the training dynamics under gradient descent. However, this restricted setting is not a necessary condition for the transformer to learn the underlying path-finding algorithm. The crucial requirement for the model to learn the generalizable rule—rather than merely memorizing training instances—is that the training distribution has sufficient diversity. Specifically, the transformer can be forced to learn the algorithm as long as any pair of token embeddings has a strictly positive probability of appearing as a parent-child edge in a training sample. This ensures the model learns the rule of path traversal instead of overfitting to the specific paths seen during training.
>
> It's possible to extend our analysis to handle more varied and less-structured trees with a proper handling of the imbalance of data. For example, our technique may combine with [Huang et al. 2023], which contains the convergence analysis of transformers when the feature distribution in in-context learning prompts is imbalanced. They show that transformers can successfully learn even with such imbalanced data, first learning from the dominant features and later from the under-represented ones.
>
> >**W2/Q2 (a): regarding summary of our convergence proof**
>
> Thank you for your valuable feedback! We will include the following proof outline for Theorem 3:
>
> - Our goal to prove that with gradient descent, the training loss converges to zero. This is achieved by showing that the learned matrix $H^{(t)} := A^\top B^{(t)} A$ converges to a diagonal matrix $\alpha I_S$. Specifically, we show there exists $T_1=\widetilde{O}\left(\frac{SN}{m\eta}\right)$ and $T_2=T_1+\widetilde{O}\left(\frac{mS}{\eta\epsilon}\right)$ that satisfy the following properties.
>
> - At Phase I $(t = \{0, 1, ..., T_1\})$, our Lemma 5-7 guarantee that the diagonal of $H^{(t)}$, $H_{j,j}^{(t)}$ ($j\in[S]$) are strictly increasing, and the off-diagonal elements satisfy $|H_{i,j}^{(t)}\lesssim \log N/N|$ for all $i\in[S]\setminus\{j\}$.
>
> - At Phase II $(t = \{T_1+1, ..., T_2\})$, we show that $H_{j,j}^{(t)}$ keeps on strictly increasing, and the
> off-diagonal elements $H_{i,j}$ are strictly decreasing and satisfy
>     $$-\frac{H_{j,j}^{(t)}}{S}\lesssim H_{i,j}^{(t)}\lesssim \frac{\log N}{N}.$$
>
> - After Phase II ($t\geq T_2+1$): $H_{j,j}^{(t)}$ (resp. $H_{i,j}^{(t)}$) keeps on increasing (resp. decreasing), and
>     $$H_{j,j}^{(t)}\gtrsim\log\left(\frac{m}{\epsilon}\right).$$
>    Furthermore, the training loss $L_{\text{train}}(\theta^{(t)})\leq \epsilon$.
>
> Similarly, we'll also add the following proof outline for Theorem 5 in our revised paper.
>
> - We first define $U_l$, $V_l$ ($l=1,2,3$) as we do now in Eq.(179). Then we have for all $i,j$: $U_{l,i,j}^{(t+1)}=U_{l,i,j}^{(t)}-\eta\sum_{k=1}^{2m}\left(\delta x_{l,i,j}^{(t,k)}+\delta y_{l,i,j}^{(t,k)}\right),$ and $V_{l,i,j}^{(t+1)}=V_{l,i,j}^{(t)}-\eta\sum_{k=1}^{2m}\delta z_{l,i,j}^{(t,k)},$ where the gradients $\delta x, \delta y, \delta z$ are defined in (178) and are computed in Lemma 12. This allows us to analyze the dynamics of $U_l^{(t)}$ and $V_l^{(t)}$ separately. Moreover, the symmetry/special structure of the training distribution allows us to rewrite $U_1,U_2,V_1,V_2$ to the matrix forms given in (182) and (183) (note that there shouldn't be subscripts $i,j$ in (182) and (183), we'll fix this typo). Regarding $U_3^{(t)}$ and $V_3^{(t)}$, Lemmas 18 and 13 guarantee that $U_{3}$, $V_3$ can be written into the matrix forms given in (261) and (213), respectively.
>
> - In Section C.3.2 we analyze the training dynamics of $C_1,C_2,C_3$ with three phases: Phase I.1-C ($t\in[T_1^C]$), Phase I.2-C ($t\in[T_1^C,T_2^C]$) and Phase II-C ($t\in[T_2^C,T_3^C]$), by tracking changes of the scalars defined in the matrix forms of $U_l, V_l$ ($l\in[3]$). The training dynamics of the three phases are respectively summarized in Lemmas 14, 15, and 16. At the end of this section, we give Lemma 17 to prove the convergence of $\sum_{k=1}^{2m}\Delta_z^{(k)}$, i.e., we show that after $t>T_3^C$, $\sum_{k=1}^{2m}\Delta_z^{(k)}\leq \epsilon/3$.
>
> - In section C.3.3, we analyze the training dynamics of $B_1,B_2,B_3$ with three phases: Phase I.1-B ($t\in[T_1^B]$), Phase I.2-B ($t\in[T_1^B,T_2^B]$) and Phase II-B ($t\in[T_2^B,T_3^B]$), where the change of key scalars are concluded in Lemmas 19, 20 and 21, respectively. At the end of this secion, we give Lemma 22 that shows $\sum_{k=1}^{2m}\left(\Delta_x^{(t,k)}+\Delta_y^{(t,k)}\right)\leq \frac{2\epsilon}{3}.$
>
> - Theorem 5 is obtained by combining Lemma 17 and Lemma 22.
>
>
> >**Q2 (b): regarding nontrivial parts of the analysis and extension to other CoT setups.**
>
> - **Nontrivial parts of the analysis:** The primary novelty and non-triviality of our analysis lie in tackling the complexity of multi-stage reasoning and the coordination it requires, which has not been addressed in prior theoretical work on CoT training dynamics.
>
>     - *Two-stage reasoning in a single autoregressive pass*: To our knowledge, this is the first theoretical analysis of how a transformer, using a single set of parameters, learns to execute two distinct, sequential subtasks within a single generation. This goes beyond existing analyses of single-algorithm emulation (e.g., parity checking or linear regression). The model must not only learn the algorithm for each subtask but also learn to autonomously identify the "turning point" (line 224) and switch its behavior accordingly.
>
>     - *Head specialization and coordination*: our work provides a provable mechanism for how multi-head attention enables this complex reasoning. We show how two attention heads learn to specialize: one head becomes a path-traversal head, while the other becomes a stage-control head. Analyzing their coordination is highly non-trivial, as their parameters are trained simultaneously and their outputs are coupled.
>
>     - *Complex coupled training dynamics*: the forward reasoning construction involves six trainable matrix blocks ($B_i$, $C_i$ for $i=1,2,3$). Analyzing their simultaneous convergence is a significant technical challenge. The gradient for each block is influenced by the state of all other blocks, creating a complex, coupled system. To manage this, we developed a careful multi-phase analysis (as outlined in our proof sketch for Theorem 5) to decouple the dynamics and track the intricate evolution of the key parameter matrices ($U_i$, $V_i$ defined in Eq. (16) and (17)) towards their target configurations.
>
> - **Generalizable insights for other CoT setups:** The core principle our work reveals is that a shallow, multi-head transformer can learn to implement a multi-stage algorithms in a single autoregressive pass. This can be extended to other COT tasks that can be decomposed into simpler, sequential subtasksm such as dynamic programming, logical deduction or fact verification.
>
> ---
>
> References:
>
> [Huang et al., 2023] Y Huang, Y Cheng, Y Liang. In-context convergence of transformers.
>
> [Nichain et al., 2024] E Nichani, A Damian, JD Lee. How transformers learn causal structure with gradient descent.
>
> [Ahn et al., 2023] Transformers learn to implement preconditioned gradient descent for in-context learning.
>
> [Yang et al., 2024] T Yang, Y Huang, Y Liang, Y Chi. In-context learning with representations: Contextual generalization of trained transformers.

---

> > ### Comment · Reviewer_WYm4 · 2025-08-04
> >
> > To be more precise, my main concern is that the algorithms to be "learned" (finding the parent node, chain operations, etc. as mentioned by the authors) are fixed, thus the problem is not "hard" from a purely learning-theoretic perspective. In other words, the hypothesis/target class size is 1 so there is no lower bound - the model could just have been initialized with the correct solution specific to the problem setup (given in the construction results for example). This is in contrast to (say) linear regression, where the goal is to learn a target vector $\theta^\star$ from the hypothesis space $\mathbb{R}^d$, and the challenge is to devise a learning scheme that performs well for any $\theta^\star$. Thus one could argue that the problem is somewhat artificial theoretically, even though the setup is relevant to reasoning tasks.
> >
> > Instead, the authors need to argue that it is natural to cripple the model by starting from zero initialization (which I agree with), and also that the resulting training dynamics is meaningful and interesting enough to analyze on its own. Here, I think this also holds especially considering their response to Q2, which is why I am still leaning towards recommending acceptance.

---

> > > ### Author Response · Authors · 2025-08-05
> > >
> > > Thank you for your comments. We appreciate your clarification of the concern. In our setting, the goal is to learn a mechanism that can iteratively trace the parent of the current node—starting from the goal node—to reconstruct the correct path to the root, based on a given prompt that encodes the tree structure. This mechanism aims to generalize to arbitrary trees, including those that are unseen during training and may have different depths or structures.
> > >
> > > As reviewer noted, this learning objective is different from that of linear regression, which needs to handle uncertainty due to noise and makes accurate decisions in R^d space. The two types of tasks differ in nature: our task focuses on learning structured traversal logic across varying symbolic structures, rather than estimating a continuous model vector under stochasticity.
> > >
> > > Thank you again for helping us clarify this important distinction.

---

### Official Review · Reviewer_sUWo · 2025-06-22

**Clarity:** 3
**Significance:** 3
**Originality:** 3
**Rating:** 5
**Confidence:** 3

**Summary:**

This paper investigates how one-layer transformers handle the multi-step reasoning tasks. The authors design synthetic tasks and theoretical analyze the potential solutions of the model parameters. Besides, the authors discuss that the model parameters are guided to the potential solutions after gradient descent.

**Questions:**

See weakness

**Ethical Concerns:**

["NO or VERY MINOR ethics concerns only"]

**Final Justification:**

Thanks for the explanation. I think I don't have severe concern now. Thus, I'll keep my positive score.

Personally speaking, I like the idea and the research question of this work. Even if there might be some limitations, a large gap between tiny 1-layer model and large pretrained models, and the idea might not be able to be extended to general cases, the exploration towards the model abilitiy analysis and interpretation is still meaningful and necessary. Afterall, I understand that one paper may not answer such complex research question completely. Good luck!

**Limitations:**

yes

**Quality:**

3

**Strengths And Weaknesses:**

**Strength:**

1. This work theoretical analyze how single-layer transformers handle the multi-step tasks with attention heads. The analysis is inspiring for understanding how existing pretrained models perform multi-step tasks.
2. The paper is well written, and the presentation logic is clear. The authors design synthetic multi-step tasks and present potential solutions of model parameters, then they verify with gradient descent the model will converge to these solutions.


**Weakness:**

1. Even if I understand that under the simplified setting, the authors can present the analysis in a more solid way, it seems a bit difficult to further extend the analysis for more practical transformers, i.e., pretrained models. First, some important modules (such as layer-normalization) are ignored. Besides, the training (i.e., optimizer and initial weights) has some assumptions. Thus, I’m wondering if such analysis can only be limited to the over-simplified setting. Do the authors have any idea if this conclusion is possible to extend to more practical transformers’ architecture or even pretrained models?
2. This is relevant to the first point. The authors analyze the gradient descent manually. While there are many assumptions about the model weights. Even if we cannot deny the possibility that they might hold in practice, will the training results be like those with optimizer (e.g., Adam) and randomly initialized weights?

---

> ### Author Rebuttal · Authors · 2025-07-31
>
> Response to Reviewer sUWo(5)
>
> Thank you very much for your valuable feedback and positive evaluation to our paper! The two comments of yours in the Weaknesses regarding how to extend our analysis to more practical scenarios are highly relevant and thus we address both points together in the following.
>
> We believe our conclusions offer a strong foundation that can be extended to more practical scenarios, albeit with increasing analytical complexity:
>
> - **More complex components (e.g., layer normalization):** Incorporating modules like LayerNorm would certainly make the gradient expressions more complex, as it introduces dependencies on the statistics of the activations. However, the fundamental analytical technique of tracking the evolution of key quantities (like $H:=A^\top B A$ (line 295) for our backward reasoning task and $U_i$, $V_i$ defined in Eq.(16), (17) for our forward reasoning task) through gradient dynamics would still apply. It is a tractable, though more involved, next step.
>
> - **Analysis with optimizers:** Our analysis of vanilla gradient descent serves as the theoretical baseline. Extending this to optimizers like Adam is a well-defined research direction. It would require tracking not just the gradients but also their first and second moments. Theoretical analyses of Adam exist (e.g., [Chen et al., 2018]), and integrating them with our framework would be a valuable, though non-trivial, endeavor to understand how adaptive methods might accelerate or alter the learning phases we identified.
>
> - **Initialization**:
>     - Fixed Weights: Fixing parts of the weight matrices is a standard and principled approach in the literature on transformer training dynamics (e.g., [Huang et al., 2023], [Nichani et al., 2024]). This simplification is crucial for isolating and clearly tracing how specific mechanisms, like attention patterns, are learned. It allows us to prove that a global minimum can be reached within a well-defined parameter space. Relaxing this assumption would lead to a more complex, coupled system of matrix dynamics, which is a direction for future work.
>     - Random/Pre-trained Start: Analyzing training from a random initialization is a common extension in theoretical studies (see, for example, [Yang et al., 2024], [Du et al., 2019]) and could be integrated with our work. Furthermore, starting from a well-chosen pre-trained state may actually simplify the analysis by allowing the model to bypass the initial learning phases, providing a more direct path to the specialized solution.
>
> While a full, end-to-end analysis of a large, pre-trained model remains beyond the current reach of tractable theory for the entire field, our work pushes the frontier of what can be formally understood by being the first theoretical analysis to demonstrate how a multi-head transformer can learn to execute distinct, sequential subtasks within a single autoregressive pass.
>
> ---
>
> [Yang et al., 2024] T Yang, Y Huang, Y Liang, Y Chi. In-context learning with representations: Contextual generalization of trained transformers.
>
> [Chen et al., 2018] X Chen, S Liu, R Sun, M Hong. On the convergence of a class of adam-type algorithms for non-convex optimization.
>
> [Du et al., 2019] S Du, J Lee, H Li, L Wang. Gradient Descent Finds Global Minima of Deep Neural Networks.

---

### Official Review · Reviewer_LK9v · 2025-06-25

**Clarity:** 2
**Significance:** 2
**Originality:** 3
**Rating:** 4
**Confidence:** 3

**Summary:**

This paper aims to show that a multi-head transformer can learn symbolic multi-step reasoning during training. Specifically, the paper shows that a single-layer transformer can solve the two-stage path-finding in trees problem (finding the path from a goal node to the root, which requires the model to first find the path from the root to the goal, then reverse it) within a single autoregressive pass. The author attributes the capability of a single-layer transformer in this task to the multi-head mechanism, which has one head that identifies the next node and the other reverses the path. The authors prove mathematically that a Transformer can learn this behavior through training. They also run experiments to show that the trained model can solve new trees it hasn't seen before.

**Questions:**

1. To support Theorem 3, I would like to see experiments that compare the results of single-head and multi-head transformers (with comparable parameters) on the forward reasoning tree path-finding tasks. Can we observe that only the multi-head transformer can finish the task and generalize to unseen trees?
2. Since the experiments are focused only on a single-layer transformer, is it possible that the capability of complex multi-step reasoning also comes from the multi-layer?

**Ethical Concerns:**

["NO or VERY MINOR ethics concerns only"]

**Final Justification:**

The author has addressed my concerns in the rebuttal. I updated the final score to 4.

**Limitations:**

yes

**Quality:**

2

**Strengths And Weaknesses:**

**Strength**
1. The paper is generally well-written.
2. The paper provides an interesting viewpoint on understanding the multi-head mechanism of the transformer.

**Weakness**
1. The paper claims that the multi-head mechanism allows the shallow transformer to perform complex reasoning tasks since different attention heads can specialize and coordinate autonomously. However, this raises a question.
2. The experimental settings and the tasks are weak; they only show that a single-layer multi-head transformer can converge on the tree path-finding tasks, and there is no ablation study or any other evidence to support the paper’s claim that different attention heads learn to specialize and coordinate autonomously.
3. The paper only discusses the situation of single-layer transformers; however, this may not apply to multi-layer transformers. For example, can we hypothesize that for a two-layer transformer, the two layers will also learn to specialize and coordinate autonomously to solve the two subtasks?

---

> ### Author Rebuttal · Authors · 2025-07-31
>
> ## Response to Reviewer LK9v
>
> Thank you for your time reviewing and your valuable feedback. If our responses resolve your questions, we'd appreciate your consideration in raising the score. Please don't hesitate to request any further clarification.
>
> >**W1: The paper claims that the multi-head mechanism allows the shallow transformer to perform complex reasoning tasks since different attention heads can specialize and coordinate autonomously. However, this raises a question.**
>
> Thank you for the comment. It seems the question regarding the multi-head mechanism is incomplete. We would be more than happy to respond once the full question is clarified.
>
> Based on the partial question, we would like to clarify that the multi-head mechanism primarily facilitates the automation of task switching between backward and forward reasoning. However, it is the **chain-of-thought** mechanism that enables the shallow transformer to perform multi-step reasoning. Specifically, each step in the chain decodes one additional node in the path toward the root during backward reasoning, or one additional node in the reverse path toward the goal node during forward reasoning.
>
> >**W2: the experimental settings and the tasks are weak...**
>
> - **On the "weakness" of the task and setting:**
>
>     - Our goal is not to achieve state-of-the-art performance on a complex benchmark, but to create a controlled and clean setting where we can provably analyze the learning dynamics of a fundamental reasoning pattern. As we state in our introduction (lines 25), our work is motivated by the "emergence of reasoning ability" where extending intermediate reasoning steps (i.e., CoT length) improves performance.
>
>     - Our key research question, inspired by empirical findings in works like [Brinkmann et al. 2024], is whether a shallow model can solve a complex, multi-stage task if provided with sufficient CoT "scratchpad" space. The forward reasoning task, which requires two distinct sub-steps (backward path-finding then reversal), is an ideal testbed for this. Showing that a one-layer transformer can solve this task—a task that might otherwise require deeper architectures (see [Brinkmann et al. 2024] for details)—is precisely the central, and we believe surprising, result of our paper (lines 92-102, 251-254).
>
> - **On the lack of evidence for head specialization and coordination:**
>
>     - Empirical validation: While we did not conduct an explicit ablation study, our current experimental results presented in Appendix E (Figures 8 and 9) have already provided strong evidence supporting the claim that different attention heads learn to specialize and coordinate autonomously. This is illustrated particularly in **Figure 9 (page 90)**, where we plot the training dynamics of the projection matrices $U_i$ and $V_i$ for $i = 1, 2$, corresponding to attention heads 1 and 2. As shown in the figure, the entries of these matrices converge toward values that align with those theoretically constructed and proved for solving the backward and forward reasoning steps (as explained further below). This demonstrates that the two heads learn to specialize in distinct sub-tasks and coordinate effectively, where head 1 handles the prediction of next node on the path, head 2 identifies the task switch from backward to forward. Importantly, this behavior emerges naturally from training via standard gradient descent, without any hard-coding enforcing such specialization. This provides direct, quantitative evidence that the heads learn their distinct, coordinated roles autonomously during training.
>
>     - Construction (existence): In Section 3.2, we provide an explicit construction for a two-head transformer that solves the forward reasoning task. As detailed in Theorem 2 and the subsequent discussion (lines 237-250), we demonstrate precisely how the heads must specialize: head 1 (path traversal) is responsible for recursively finding the next node in the path, and head 2 (stage controller) manages the reasoning stage, tracking whether the model is in the initial "goal-to-root" phase or the subsequent "root-to-goal" phase, and triggers the transition.
>
>     - Optimization (learnability): In Section 4.2, we further prove that these specialized roles are not just hypothetically possible, but are learnable via gradient descent. As mentioned in line 317, our convergence analysis and Theorem 5 show that the model parameters provably converge to the specialized matrices specified in our construction Eq.(10), (11). This is a highly non-trivial result that forms the technical core of our paper. As far as we know, we are the first to theoretically analyze how a transformer use the same parameters to do two different subtasks.
>
> >**W3: no discussion on multi-layer transformers**
>
> Thank you for your comment.
>
> - As we articulate in our introduction, instead of focusing on the scaling of model depth, we focus on the scaling of CoT length. The reviewer's question about a two-layer model is insightful, but it implicitly favors the "scaling depth" paradigm. Our paper's contribution is to show, provably, that the "scaling CoT" paradigm is a viable alternative for complex reasoning, even in shallow models. Therefore, extending our analysis to deep models would address a different research aspect and depart from our current central message. Thus, we will leave this direction for future exploration.
>
> - A rigorous theoretical analysis of the training dynamics of deep, non-linear transformers is a notoriously difficult, and largely open, problem in the field. Our work pushes the frontier of what is theoretically tractable. By analyzing a one-layer, multi-head model, we are able to provide the first theoretical analysis of how a transformer can learn to execute two distinct subtasks sequentially using the same set of parameters within a single autoregressive pass. This is a significant and non-trivial contribution.
>
> >**Q1: regarding experiments that compare the results of single-head and multi-head transformers**
>
> Thank you for this question. We speculate that you mean Theorem 5, not theorem 3.
>
> - We did not include experiments comparing single-head and multi-head transformer performance because it is not straightforward to implement a single-head transformer that can autonomously switch between backward and forward reasoning. In our attempts, such experiments consistently failed to converge. This is not surprising. At a high level, it is highly non-trivial for a single attention head with fixed parameters to reliably perform the two distinct and concurrent subtasks of path traversal and stage control. Without any structural guidance, it is unclear how a single head could coordinate these subtasks effectively. In contrast, our results demonstrate that a two-head architecture provides a natural and learnable decomposition of the problem, with each head specializing in one of the subtasks. This specialization emerges purely from training dynamics, and highlights the power of multi-head attention in enabling implicit task coordination.
>
> - As mentioned in our response to Weakness 1, our experiments in App. E directly supports our Theorem 5: Figure 8 confirms that the training loss for the forward reasoning task converges to zero, showing the two-head model successfully learns to solve the task; Figure 9 provides the crucial mechanistic evidence. By plotting the learned parameter entries over time, it shows that the attention heads' parameters indeed converge to the specialized matrix structures we constructed and analyzed. This empirically validates our theory that the heads learn their distinct, coordinated roles during training.
>
> >**Q2: Since the experiments are focused only on a single-layer transformer, is it possible that the capability of complex multi-step reasoning also comes from the multi-layer?**
>
> Thank you for your question. You are absolutely right to point out that architectural depth is a key mechanism for enabling complex reasoning in transformers. Your question highlights a crucial dichotomy in how transformers can achieve such capabilities: either by leveraging architectural depth or by utilizing longer intermediate reasoning steps (i.e., Chain-of-Thought length). Our work is specifically designed to investigate the latter paradigm, complementary to the empirical study [Brinkmann et al. 2024], which show that deeper models can indeed learn to perform multi-step path-finding implicitly across their layers.
>
>
> ---
>
> Reference:
>
> [Brinkmann et al. 2024] J Brinkmann et al. A mechanistic analysis of a transformer trained on a symbolic multi-step reasoning task.

---

> > ### Author Response · Authors · 2025-08-04
> >
> > Dear Reviewer LK9v,
> >
> > As the author-reviewer discussion period will end soon, we would like to check whether our responses have properly addressed your concerns? If so, could you please kindly consider increasing your initial score accordingly? Certainly, we are more than happy to answer your further questions.
> >
> > Thank you for your time and effort in reviewing our work!
> >
> > Best Regards, Authors

---

> > ### Comment · Reviewer_LK9v · 2025-08-04
> >
> > Thank the reviewer for the explanation. While some of my concerns are addresses, I still have the questions below:
> >
> > - The statement “ the multi-head mechanism primarily facilitates the automation of task switching between backward and forward reasoning.” — does this imply that without multi-head attention, the model cannot perform the backward–forward reasoning switch?
> >
> > - I am not convinced by “in our attempts, such experiments consistently failed to converge... it is highly non-trivial for a single head to perform both path traversal and stage control”. I doubt that the model would suddenly fail with one head and then suddenly work with two heads. Multi-head attention was originally introduced for efficiency. If we increase the hidden size of the single-head model so that it has the same parameter count as the two-head variant, wouldn’t it have greater capacity? In that case, would the two-head model converge but the single-head model still fail? Do you have concrete experimental results?

---

> ### Author Response · Authors · 2025-08-05
>
> Thank you very much for your follow-up question.
>
> Our central argument is that the benefit of the multi-head mechanism in this specific, complex reasoning task stems from its ability to support **architectural specialization**, rather than simply an increase in parameter count. The transition from one head to two is not a matter of the model "suddenly" working due to more parameters, but rather a qualitative architectural change that enables different heads to learn distinct, complementary roles—in this case, path traversal and stage control. A single attention mechanism is tasked with the highly non-trivial challenge of learning both functions simultaneously, which appears to be significantly more difficult for gradient-based optimization.
>
> To provide concrete evidence for this claim, we conducted a comparison experiment as you suggested. To ensure this model had sufficient capacity and was not unfairly constrained, we made its entire $W^{KQ}$ matrix and its $W^V$ matrix (they are both $(2d_1+d_2)$-by-$(2d_1+d_2)$ matrices) fully learnable, giving it more trainable parameters and flexibility than our structured two-head model. All other experimental settings were kept identical to those described in Appendix E.
>
> The training loss progression across different training steps is as follows:
>
> | Model\training step | 1 | 5 | 10 | 100 | 500 | 1000 | 1500 |
> | :--- | :---: | :---: | :---: | :---: | :---: | :---: | :---: |
> | **Our Two-Head Transformer** | 15.1991 | 10.4123 | 8.4025 | 2.4336 | 0.3237 | 0.1361 | 0.0849 |
> | **Single-Head Transformer** | 15.9122 | 15.6750 | 15.0179 | 10.1325 | 10.6257 | 10.0823 | 10.3812 |
>
> As the results clearly demonstrate, our training loss converges to 0 while the single-head transformer's loss stops decreasing around a high value 10.
>
> Thank you again for your response! If our response has satisfactorily addressed your concerns, we would be sincerely grateful if you could consider updating your rating to reflect that. Of course, we are more than happy to address any further questions you may have.

---

> > ### Comment · Reviewer_LK9v · 2025-08-05
> >
> > Thank you for addressing my questions and providing detailed experimental results. I have updated my rating to 4.

---

### Official Review · Reviewer_WaDb · 2025-07-01

**Clarity:** 1
**Significance:** 2
**Originality:** 3
**Rating:** 3
**Confidence:** 4

**Summary:**

This paper claims to formally prove that shallow transformers are able to learn multi-step reasoning on trees via gradient descent. Specifically, the authors consider the task of learning backward and forward reasoning on a tree-shaped graph, i.e. given a tree and a goal node that is a leaf node of the tree find a path from the root of the tree to that goal node. They start by first claiming that a parametrisation of a shallow attention block exists that can perform both backwards and forwards reasoning under a certain type of node embedding when applied repeatedly in a chain-of-thought-like fashion. The authors then continue to claim that gradient descent actually converges to this constructed parametrisation when using a training set with specific properties.

**Questions:**

Please find my questions intertwined with the weaknesses above.

**Ethical Concerns:**

["NO or VERY MINOR ethics concerns only"]

**Final Justification:**

The authors have shown that their can provide more clear arguments that are convincing, but the paper itself still has many more examples of unclear or vague arguments that make it almost impossible to validate all their theoretical statements. I hence increase my score slightly and have increase the scores for significance and quality, but can not accept.

**Limitations:**

No.

**Paper Formatting Concerns:**

No concerns.

**Quality:**

2

**Strengths And Weaknesses:**

#### Strengths
1. The abstract and introduction are well-written and also motivate the problem well: we want to see if one can prove that transformers, i.e. attention blocks, are able to learn reasoning in practical circumstances.
2. Figure 2 gives a nice visual abstraction of what behaviour you would like the attention block to have.
#### Weaknesses
I have severe concerns about this paper. In particular, I suspect a significant portion is produced by generative AI. Concretely, none of the theoretical results claimed in the paper are actually correctly proven as the so-called "proofs" in the appendix are filled with mistakes or seem to (try) to prove unrelated statements. In short, the paper really does not have anything of any substance. Please see a couple of precise examples below.

1. The authors claim to prove the existence of a parametrisation that makes transformers exactly perform backward reasoning (Lines 190-191). However, the statement of Theorem 1 only says that a construction exists that converges to backward reasoning under a certain limit to infinity that is not attainable in practice. While such a result could still be interesting, there is no proof given, not even in the appendix, of this statement. The appendix contains a "Lemma 1" that seems to somewhat use the same notation, but is apart from that not showing anything as stated.
2. Theorem 2 claims a similarly plagued statement for forward reasoning, which again is only a convergence claim for an impossible limit to infinity. While now the appendix does (seem) to contain a (trial) of a proof for Theorem 2, unfortunately the proof is filled with mistakes. First, Theorem 2 claims to show the existence of parametrisations that adhere to certain constraints (Equation 10 and 11), but the proof simply assumes these parametrisations exist in the first place. Even then, the proof claims a series of equalities (Equations 27-33) that should show that in the limit the softmax goes to a unit vector in the infinite limit. However, since Equation 29, 31 express equality to some positive constant multiplied with $\alpha_1$, that means that the softmax would go to a uniform vector when $\alpha_1 \to +\infty$. Hence, this proof is false.
3. Given that Theorems 3 and 4 bases itself on the constructions of Theorem 1 (Lines 295-299) and Theorem 1 was not proven, it follows that Theorems 3 and 4 are also unproven. However, I still want to put attention to the provided "proofs" in the appendix. From Line 526 onwards, we get page after page of mathematical derivations that do not go anywhere. This sort of exposition is a strong indication that modern generative AI was used to construct those "proofs".
#### Verdict
It is highly likely that this paper was mostly produced with generative AI. Even if we do not assume that it is, the claims of the paper are not proven, neither theoretically or practically. The paper tries to emulate formality by generating page after page of mathematical gibberish in the appendix, which does not hold up to close scrutiny. Clear reject.

---

> ### Author Rebuttal · Authors · 2025-07-31
>
> ## Response to Reviewer WaDb
>
> Thank you for your detailed review. We want to state firmly that all theorems and proofs in this work are our own original contributions. We hope our detailed responses below, including the one-paragraph missing proof of Theorem 1 (which was regrettably omitted) and clarify the other points, will resolve these concerns. If our response rectifies your primary concerns, we would be grateful if you would consider re-evaluating the paper and raising your score. We're happy to answer your further questions if any.
>
> >**1.No proof is given for Theorem 1**
>
> We sincerely apologize for our accidental omission of the proof of Theorem 1 in our submitted version. Because the proof is short, we originally included it in the main text but then, due to page limit, decided to move it to the appendix, but apprarently neglected doing so. The proof follows directly by applying Lemma 1 and a brief further argument. For the reviewer’s reference, we present the proof below and will include it in the revised version. We appreciate your careful review.
>
> >When Assumption 1 holds, the existence of B satisfying (6) follows from Lemma 1. Letting $\alpha\rightarrow +\infty$, it follows that
> $$softmax(A^\top B A)=I_S$$ converges to an identity matrix. By (4), we know that at the $k$-th reasoning step, softmax($Y^{(k-1)\top}Bx_{-1}^{(k-1)}$) becomes a one-hot vector with the $j$-th entry being 1 if $y_j^{(k-1)}=x_{-1}^{(k-1)}$ and 0 otherwise (note that $x^{(k-1)}_{-1}$ will appear exactly once in $Y^{(k-1)}$).
>
> >Therefore, the model's output $\hat{o}^{(k)}$ will be the embedding of the edge that connects the current node $x_{-1}^{(k-1)}$ with its parent $x_j^{(k-1)\top}$. Thus by induction, $\widehat{o}^{(k)} = o^{(k)}$ for all $k\in[m]$.
>
> ---
>
> >**2. regarding the existence of parametrisations and limit of the softmax in Theorem 2**
>
> Thanks for the question. The proof of Theorem 2 is correct. We clarify the reviewer's concerns as follows.
>
> Regarding the first point, the existence of the parameter matrices {$B_i$, $C_i$} that satisfy the constraints in (10) and (11) is guaranteed by Lemma 1, given the linear independence of embeddings from Assumption 2. We will add a sentence at the beginning of the proof to explicitly reference Lemma 1. Thank you for raising this.
>
> Regarding the softmax under the limit, the reviewer seems to have a misunderstanding about the behavior of the softmax function in the limit. The core of our proof relies on the relative growth rates of the logits, not just the fact that they go to infinity. Specifically:
>
> - **softmax principle:** given any vector $x=(x_1,...x_n)^\top\in R^n$, assuming $x_1>x_i$ for any $i\in\{2,...,n\}$, then softmax($\alpha x$)$\to (1,0,...0)^\top:=e_1$ when $\alpha\to+\infty$.
>
> - **Application to our proof:** In our construction (Eq. 27-33), all logits for $\mu^{(0)}$ are scaled by $\alpha_1$. The coefficients of $\alpha_1$ are designed to have a unique maximum. Specifically, the coefficient for the correct parent edge is $1 + b_2$ (from Eq. (27)), which, under our stated constraints, is strictly greater than the coefficients for any other token (e.g., $b_2$ from Eq. (29)). Because the logit for the correct edge grows strictly faster than all others, the softmax output correctly converges to a one-hot vector. The same principle applies to $\nu^{(0)}$ converging to $e_{l+2}$. Thus, the proof holds.
>
> >**3. mathematical derivations of Theorems 3 and 4 do not go anywhere...strong indication that modern generative AI was used to construct those "proofs"**
>
> We respectfully disagree with the reviewer’s assessment of the proofs of Theorems 3 and 4. These are rigorous, original proofs derived by the authors without AI assistance.
>
> First, as clarified in our responses above, Theorems 1 and 2 are both valid and can be rigorously proven. Therefore, Theorems 3 and 4 (among other) are built on a solid theoretical foundation.
>
> The proofs beginning at Line 526 are correct and lead to the conclusions stated in Theorems 3 and 5. To help clarify any confusion, we provide below an outline of the analytical roadmap that underpins our approach.
>
> - **On the Structure of the Proofs for Theorems 3 and 5:** We acknowledge that the detailed derivations for the training dynamics in the appendix are dense. The lengthy details are necessary to ensure full mathematical rigor. The derivations follow a carefully developed multi-phase analysis to track the evolution of the model's parameters from a zero initialization to a configuration that solves the task. Such multi-phase analysis is necessary due to the complex, nonconvex dynamic of the optimization problem.
>
> - To make the logic transparent, we'll include the following proof outlines in Appendix B to summarize the key steps for Theorem 3:
>
>
>     - Goal: to prove that with gradient descent, the training loss converges to zero. This is achieved by showing that the learned matrix $H^{(t)} := A^\top B^{(t)} A$ converges to a diagonal matrix $\alpha I_S$. Specifically, we show there exists $T_1=\widetilde{O}\left(\frac{SN}{m\eta}\right)$ and $T_2=T_1+\widetilde{O}\left(\frac{mS}{\eta\epsilon}\right)$ that satisfy the following properties.
>
>     - At Phase I $(t = \{0, 1, ..., T_1\})$, our Lemma 5-7 guarantee that the diagonal of $H^{(t)}$, $H_{j,j}^{(t)}$ ($j\in[S]$) are strictly increasing, and the off-diagonal elements satisfy $|H_{i,j}^{(t)}\lesssim \log N/N|$ for all $i\in[S]\setminus\{j\}$.
>
>     - At Phase II $(t = \{T_1+1, ..., T_2\})$, our Lemma 8,9 guarantee that $H_{j,j}^{(t)}$ keeps on strictly increasing, and the
> off-diagonal elements $H_{i,j}$ are strictly decreasing and satisfy
>     $$-\frac{H_{j,j}^{(t)}}{S}\lesssim H_{i,j}^{(t)}\lesssim \frac{\log N}{N}.$$
>
>    - After Phase II ($t\geq T_2+1$): $H_{j,j}^{(t)}$ (resp. $H_{i,j}^{(t)}$) keeps on increasing (resp. decreasing), and
>     $$H_{j,j}^{(t)}\gtrsim\log\left(\frac{m}{\epsilon}\right).$$ Furthermore, the training loss $L_{\text{train}}(\theta^{(t)})\leq \epsilon$.
>
> - Similarly, we'll also add the following proof outline for Theorem 5 in our revised paper.
>
>     - We first define $U_l$, $V_l$ ($l=1,2,3$) as we do now in Eq.(179). Then we have for all $i,j$: $U_{l,i,j}^{(t+1)}=U_{l,i,j}^{(t)}-\eta\sum_{k=1}^{2m}\left(\delta x_{l,i,j}^{(t,k)}+\delta y_{l,i,j}^{(t,k)}\right),$ and $V_{l,i,j}^{(t+1)}=V_{l,i,j}^{(t)}-\eta\sum_{k=1}^{2m}\delta z_{l,i,j}^{(t,k)},$ where the gradients $\delta x, \delta y, \delta z$ are defined in (178) and are computed in Lemma 12. This allows us to analyze the dynamics of $U_l^{(t)}$ and $V_l^{(t)}$ separately. Moreover, the symmetry/special structure of the training distribution allows us to rewrite $U_1,U_2,V_1,V_2$ to the matrix forms given in (182) and (183) (note that there shouldn't be subscripts $i,j$ in (182) and (183), we'll fix this typo). Regarding $U_3^{(t)}$ and $V_3^{(t)}$, Lemmas 18 and 13 guarantee that $U_3$, $V_3$ can be written into the matrix forms given in (261) and (213), respectively.
>
>     - In Section C.3.2 we analyze the training dynamics of $C_1,C_2,C_3$ with three phases: Phase I.1-C ($t\in[T_1^C]$), Phase I.2-C ($t\in[T_1^C,T_2^C]$) and Phase II-C ($t\in[T_2^C,T_3^C]$), by tracking changes of the scalars defined in the matrix forms of $U_l, V_l$ ($l\in[3]$). The training dynamics of the three phases are respectively summarized in Lemmas 14, 15, and 16. At the end of this section, we give Lemma 17 to prove the convergence of $\sum_{k=1}^{2m}\Delta_z^{(k)}$, i.e., we show that after $t>T_3^C$, $\sum_{k=1}^{2m}\Delta_z^{(k)}\leq \epsilon/3$.
>
>     - In section C.3.3, we analyze the training dynamics of $B_1,B_2,B_3$ with three phases: Phase I.1-B ($t\in[T_1^B]$), Phase I.2-B ($t\in[T_1^B,T_2^B]$) and Phase II-B ($t\in[T_2^B,T_3^B]$), where the change of key scalars are concluded in Lemmas 19, 20 and 21, respectively. At the end of this secion, we give Lemma 22 that shows $\sum_{k=1}^{2m}\left(\Delta_x^{(t,k)}+\Delta_y^{(t,k)}\right)\leq \frac{2\epsilon}{3}.$
>
>     - Theorem 5 is obtained by combining Lemma 17 and Lemma 22.
>
> We sincerely thank the reviewer once again for your time and effort. We hope that our responses above have assured you the originality and rigor of our proofs. If so, we would greatly appreciate it if you would consider re-evaluating the paper and possibly raising your score. Of course, we would be more than happy to address any further questions or concerns you may have.

---

> > ### Author Response · Authors · 2025-08-04
> >
> > Dear Reviewer WaDb,
> >
> > As the author-reviewer discussion period will end soon, we would like to check whether our responses have properly addressed your concerns? If so, could you please kindly consider increasing your initial score accordingly? Certainly, we are more than happy to answer your further questions.
> >
> > Thank you for your time and effort in reviewing our work!
> >
> > Best Regards, Authors

---

> > > ### Comment · Reviewer_WaDb · 2025-08-05
> > >
> > > I sincerely thank the authors for their extensive clarification. I admit my review might have been on the harsher side, but as a theoretical paper I feel that the proofs should be waterproof and, most of all, clear.
> > >
> > > **Theorem 1**
> > >
> > > The complete omission of the proof of even the first central theorem (Theorem 1) is quite a large oversight. I do see how the first statement (Equation 6) does indeed follow from the provided Lemma 1 and how that is used together with Equation 5 to give the result. Thank you for this clarification and please add this explanation to the main paper or the appendix.
> > >
> > > **Theorem 2**
> > >
> > > I had not considered the specific limit behaviour of the softmax when it is know that one of the logits is consistently larger. However, this option is also not mentioned in the proof. Simply stating "By far we can see [...] that [...]" is not an argument. Please provide the exact explanation provided in the answer in the proof as well. It is the author's responsibility to provide understandable and formal arguments for every step in their proofs. It is exactly this vagueness that triggered my harsh response; the lack of explanations *why* an argument holds.
> > >
> > > While my raised issue here is resolved, just looking a number of lines further in the proof I already encounter a similar case in line 467 "Thus the input at the $p$-th reasoning step satisfies [...]". *Why* is that the case? *Where* did you exactly use the induction hypothesis to get the equalities in Equation 36? There are a lot of different notations appearing in the subsequent lines as well that are simply "put out there". Please do not underestimate the effort it takes to communicate arguments clearly; a series of dozens of equations with hardly any explanation does not suffice. **Given the theoretical nature of the paper, I consider this general lack of explanations in the proofs as a severe weakness.**
> > >
> > > **Theorems 3 and 5**
> > >
> > > I completely understand the need for such dense mathematics; any analysis in such non-convex training dynamics is indeed hard. Hence, I thank the authors for providing a rough outline of the proof that gives some overview. Such additions go into the right direction and do alleviate my concerns slightly.
> > >
> > > **Summary**
> > >
> > > While I am overall satisfied with the explanations that the authors have given me for my specifically raised issues, there are more examples of arguments in the proofs that are too vague and lack further explanation. **I completely retract my harsh allegations about GenAI usage**, but I am still critical about the way some proofs are written. In their current form, given again the mainly theoretical contributions of the paper, I can not be certain about their complete correctness. The authors have shown in their answers that they can communicate their arguments more clearly and convincingly, and are willing to provide such explanations. Hence, I slightly increase my score, but not to acceptance as I can not validate whether the authors will treat all of their arguments with the same clarity.

---

> > > > ### Author Response · Authors · 2025-08-05
> > > >
> > > > ## Re: Reviewer WaDb
> > > >
> > > > We sincerely appreciate the reviewer’s thoughtful feedback and are glad to hear that our previous clarifications helped address the specific concerns raised. We fully understand the importance of clear and rigorous exposition, especially given the theoretical nature of our contributions.
> > > >
> > > > We take the reviewer’s comments seriously and are committed to revising the paper to ensure that all arguments and proofs are presented with the same level of clarity and completeness demonstrated in our responses. In particular, we will:
> > > >
> > > > - Expand key steps in each proof to avoid ambiguity or jumps in logic;
> > > >
> > > > - Add intuitive explanations alongside technical arguments to improve readability;
> > > >
> > > > - Highlight non-trivial transitions with additional lemmas or detailed derivations where appropriate;
> > > >
> > > > - Include a proof outline where needed, to guide the reader through the structure of longer arguments.
> > > >
> > > > While we regret that we cannot share a revised version during the discussion phase, we assure the reviewer that the same level of care shown in our rebuttal will be applied consistently throughout the paper.
> > > >
> > > > Below we answer the reviewer's questions in the last response as examples to demonstrate how we will explain the proof steps.
> > > >
> > > > >**Q1: In Line 467, the paper has "Thus the input at the $p$-th reasoning step satisfies [...]". Why is that the case? Where did you exactly use the induction hypothesis to get the equalities in Equation 36?**
> > > >
> > > > The overall goal, stated in Lines 454-455, is to prove by induction that Equation (26) holds, i.e., for any $k\in[2m]$, $\hat{o}^{(k)}\rightarrow o^{(k)}$ as $\alpha_1,\alpha_2\rightarrow +\infty$. We first prove this is true when $k=1$ (see Lines 456-464). And then we make our induction hypothesis in Line 465: we assume (26) is true for all steps from $k = 2$ up to $p-1$. Then (36) is a direct consequence of applying our induction hypothesis to the definition of the autoregressive input at step $p$ (which consists of the output from step $p-1$). Again, this relation follows directly by concatenating the multi-reasoning step using our notation.
> > > >
> > > > >**Q2: There are a lot of different notations appearing in the subsequent lines as well following line 467 that are simply "put out there".**
> > > >
> > > > The notation related to $\overline X$, $\overline Y$, $\overline Z$ introduced in Equation (36) is to deconstruct the large input matrix $E^{(p-1)}$ into its constituent parts, which are then used in the attention mechanism calculations. And same as for the basic case, $\bar\mu$, $\bar\nu$ in (39) and (40) represent the attention weights for the first and second attention head at the $p$-th reasoning step, resp.  We'll add more intuitive explanation to our notation. Thank you for your comment.
> > > >
> > > > We also want to provide some broader context for our notational choices:
> > > >
> > > > - For the most notation-heavy parts of our analysis, namely the convergence and generalization proofs, we included dedicated notation sections at the beginning of Appendices C.2 and C.3 to orient the reader. We will, of course, enhance these sections with more intuitive explanations as well.
> > > >
> > > > - We also want to point out that the symbolic reasoning task on trees that we analyze is inherently abstract and complex. Describing the multi-step, multi-stage process precisely in mathematical language necessitated a significant amount of notation, arguably more than is typical for most other learning theory topics. This complexity makes the proofs highly challenging. We fully recognize that this poses a challenge for readability, and we accept the responsibility to make our arguments as clear and accessible as possible.
> > > >
> > > > ---
> > > >
> > > > We thank the reviewer for pointing out these readability issues. We are committed to thoroughly revising the paper to ensure every step is well-motivated and transparent. If there are other specific areas where the reviewer still feels the explanation is too vague or difficult to follow, we would be very grateful for the opportunity to address them further. We are fully committed to strengthening the presentation and ensuring the correctness and transparency of our results.
> > > >
> > > > We hope that with this commitment in mind, the reviewer might consider further re-evaluating the paper. In any case, we fully understand, and we sincerely thank you again for your time and constructive engagement.

---

### Official Review · Reviewer_Avae · 2025-07-03

**Clarity:** 3
**Significance:** 3
**Originality:** 3
**Rating:** 5
**Confidence:** 3

**Summary:**

This paper considers a symbolic multi-step reasoning problem: for a given reasoning tree, the model should first conduct backward reasoning for the root, and then forward reasoning for a complete reasoning path. The authors provide a mechanism on how a single-layer softmax transformer solves this problem, accompanied by training dynamics analysis. This work provides insights for understanding CoT reasoning in LLMs.

**Questions:**

I think this is a good work. If any question, I think in the experiments in the appendix, the loss curve for backward reasoning seems not to converge in 1000 epochs, as the loss is still very high. Could you provide the full training loss curve?

**Ethical Concerns:**

["NO or VERY MINOR ethics concerns only"]

**Limitations:**

Please refer to the Weakness part in the previous section.

**Quality:**

3

**Strengths And Weaknesses:**

*Strengths*

1. This work proposes a symbolic multi-step reasoning task, which mimics the reasoning procedure for real-world LLMs.
2. This work provides a detailed mechanistic explanation on how a single-layer transformer can solve this task, which incorporates a detailed designed transition point between two attention heads, which is both interesting and insightful.
3. This work provides training dynamics analysis, accompanied by experiments to validate their findings.
4. The mechanistic interpretation, together with additional theoretical analysis for the forward process, takes an important step for theoretically understanding multi-step reasoning processes in transformers.

*Weaknesses*
- The backward pass process constructs a transformer adapting copying mechanism, which is well studied in many theoretical analysis papers [1][2].
- In the construction for forward reasoning, each position has two nodes and a flag, which seems too artificial.

[1] Huang, Yu, Yuan Cheng, and Yingbin Liang. "In-context convergence of transformers." arXiv preprint arXiv:2310.05249 (2023).

[2] Nichani, Eshaan, Alex Damian, and Jason D. Lee. "How transformers learn causal structure with gradient descent." arXiv preprint arXiv:2402.14735 (2024).

---

> ### Author Rebuttal · Authors · 2025-07-31
>
> ## Response to Reviewer Avae
>
> Thank you very much for your insightful comments and your appreciation to our work!
>
> >**W1: The backward pass process constructs a transformer adapting copying mechanism, which is well studied in many theoretical analysis papers [1][2].**
>
> Thank you for your insightful feedback. We agree that the core mechanism in the backward reasoning task—sequentially copying parent nodes—is conceptually related to well-studied copying or induction head mechanisms [1, 2]. This analysis provides a clear, simplified setting to introduce our problem formulation and the basic mechanics of single-head reasoning on trees, making the readers easier to understand our paper.
>
> >**W2: In the construction for forward reasoning, each position has two nodes and a flag, which seems too artificial.**
>
> Thank you for this insightful comment. Our structured output, which includes a stage flag, is a deliberate design choice. It allows us to mechanistically prove how even a one-layer transformer can learn to autonomously switch between subtasks. While a deeper model like the one in [Brinkmann et al., 2024] might manage this implicitly across layers, our work shows how this control can be explicitly learned as a specialized head function within a shallow architecture, providing a clear illustration of multi-stage reasoning.
>
>
> >**Q1: I think this is a good work. If any question, I think in the experiments in the appendix, the loss curve for backward reasoning seems not to converge in 1000 epochs, as the loss is still very high. Could you provide the full training loss curve?**
>
> Thank you for this excellent point. You are correct that the current plot only shows the initial training phase. We will update this figure in our revision to show the complete loss curve until convergence to 0.
>
> ---
>
> Reference:
>
> [Brinkmann et al. 2024] J Brinkmann et al. A mechanistic analysis of a transformer trained on a symbolic multi-step reasoning task.

---

### Decision · Program_Chairs · 2025-09-17

**Decision:**

Accept (poster)

**Comment:**

The paper provides a theoretical analysis of how one-layer, multi-head transformers can provably learn symbolic multi-step reasoning tasks through gradient descent. The paper shows that for path-finding on trees different attention heads learn to specialize and coordinate on the distinct subtasks within a single autoregressive pass. The motivation here is that this task requires a two-stage process of backward reasoning followed by forward reasoning. Overall, this work aims to provide a mechanistic explanation for the emergence reasoning in transformers.

On the positive side, the reviewers found that the paper addresses an interesting and important problem, providing valuable theoretical insights into how transformers perform complex reasoning. Reviewers noted that the mechanistic explanation of how attention heads specialize to solve different subtasks is a significant contribution. The theoretical analysis, which includes training dynamics and generalization guarantees, is solid. The reviewers found that it is a clear step forward in the theoretical understanding of multi-step reasoning in transformers, which is a long standing and important research area.

The reviewers identified areas of improvement, primarily concerning the clarity of the theoretical proofs and the simplicity of the experimental setup. Several reviewers found the proofs to be dense and complex, suggesting that adding high-level outlines and more intuitive explanations would significantly improve readability and accessibility. Some reviewers also noted that the problem setting, while useful for a controlled theoretical analysis, felt artificial and simplified. This included the use of a one-layer transformer, the specific structure of the model's output, and the idealized training conditions (e.g., using vanilla gradient descent without components like LayerNorm). Reviewers also initially requested more direct evidence for the claim of head specialization, which the authors later provided in the rebuttal. By incorporating this feedback to enhance the exposition of the theoretical arguments and contextualize the experimental design choices, the authors will produce a more polished and high-quality paper.

Based on these reviews, I recommend accepting this paper. The reviewers were excited about the paper's novel approach to providing a provable, mechanistic explanation for multi-step reasoning in transformers. They highlighted that the work offers an interesting and insightful viewpoint on the role of the multi-head mechanism and that the analysis is relevant to understanding the theoretical foundations of chain-of-thought, an area that currently lacks strong theory. While there were initial concerns, the authors addressed most of them effectively during the discussion period. I expect the authors to incorporate the reviewers' feedback into the final version of the paper, particularly by improving the readability of the theoretical proofs and perhaps adding more experiments.